# Attenuated growth factor signaling during cell death initiation sensitizes membranes towards peroxidation

André Gollowitzer [1,24], Helmut Pein[2,24], Zhigang Rao [1], Lorenz Waltl[1], Leonhard Bereuter [1,3], Konstantin Loeser[2], Tobias Meyer[4,5], Vajiheh Jafari[2], Finja Witt[1], René Winkler [6,7], Fengting Su [1,3], Silke Große [8], Maria Thürmer [2], Julia Grander [1], Madlen Hotze [9], Sönke Harder [10], Lilia Espada[11], Alexander Magnutzki[12], Ronald Gstir[12], Christina Weinigel[13], Silke Rummler[13], Günther Bonn[12], Johanna Pachmayr[14], Maria Ermolaeva [11], Takeshi Harayama[15], Hartmut Schlüter [10], Christian Kosan [6], Regine Heller [8], Kathrin Thedieck [9,16,17,18,19], Michael Schmitt [4], Takao Shimizu[20,21], Jürgen Popp[4,5], Hideo Shindou[22,23], Marcel Kwiatkowski [9] & Andreas Koeberle [1,2,3] ✉

Cell death programs such as apoptosis and ferroptosis are associated with aberrant redox homeostasis linked to lipid metabolism and membrane function. Evidence for cross-talk between these programs is emerging. Here, we show that cytotoxic stress channels polyunsaturated fatty acids via lysophospholipid acyltransferase 12 into phospholipids that become susceptible to peroxidation under additional redox stress. This reprogramming is associated with altered acyl-CoA synthetase isoenzyme expression and caused by a decrease in growth factor receptor tyrosine kinase (RTK)-phosphatidylinositol-3-kinase signaling, resulting in suppressed fatty acid biosynthesis, for specific stressors via impaired Akt-SREBP1 activation. The reduced availability of de novo synthesized fatty acids favors the channeling of polyunsaturated fatty acids into phospholipids. Growth factor withdrawal by serum starvation mimics this phenotype, whereas RTK ligands counteract it. We conclude that attenuated RTK signaling during cell death initiation increases cells' susceptibility to oxidative membrane damage at the interface of apoptosis and alternative cell death programs.

Programmed cell death regulates tissue homeostasis and, when dysregulated, causes neurodegenerative disorders, autoimmune diseases, and cancer[1–4]. The initial steps that induce cell death are diverse and include death-receptor signaling, disruption of ion homeostasis, interference with gene expression, blockade of cell cycle progression, energy depletion, and metabolic stress[5]. Signal transduction initiated by these triggers activates the caspase cascade, which executes apoptotic cell death[1,5,6]. Alternative cell death pathways rely on membrane peroxidation, necrosome assembly, autophagy, or gasdermin-dependent pore formation[1,7]. Cross-talk between cell death programs is emerging[8] and converging in redox and membrane(-adaptive) responses[9,10].

Cell death results from an imbalance between survival and pro-death pathways and is kept in check by tissue-specific growth factors[11–14]. As ligands of receptor tyrosine kinases (RTKs), growth factors trigger manifold signaling pathways, such as the

phosphatidylinositol-3-kinase (PI3K)/Akt axis, which promotes cell proliferation, counteracts cell death induction, and regulates cell metabolism, among many other functions[15,16]. The ubiquitously expressed serine/threonine kinase Akt (also known as AKT1 and protein kinase B) binds to phosphatidylserine (PS)[17] and phosphatidylinositol-3,4,5-trisphosphate (PI(3,4,5)P₃)[18,19] that is formed upon receptor activation by scaffolded phosphoinositide kinases at the plasma membrane[20]. Co-recruited kinases activate Akt through phosphorylation at serine (S)473 and threonine (T)308. Rab5-positive endosomal membranes anchoring the isoenzyme Akt2 through PI(3,4,5)P₃ or phosphatidylinositol-3,4-bisphosphate (PI(3,4)P₂) are another important site for Akt activation[21–24].

The RTK-PI3K cascade is central to the regulation of fatty acid and phospholipid metabolism[21] and activates sterol regulatory element-binding protein (SREBP)1c, a major transcription factor that drives anabolic lipid metabolism[25]. It is generally accepted that lipid metabolism defines the susceptibility of cells to membrane peroxidation[26–29] and shapes resistance to (cancer) cell death[25,30–32]. Specifically, PUFA uptake, biosynthesis, and membrane incorporation sensitize cells to oxidative membrane damage[33,34], whereas exogenous monounsaturated fatty acids (MUFAs) and an enhanced expression of MUFA biosynthetic enzymes, e.g., stearoyl-CoA desaturase (SCD)1, are membrane-protective[35,36]. Resistance to oxidative cell death is also dictated by glucose metabolism, i.e., glycolysis, the pentose phosphate pathway, pyruvate oxygenation, and the tricarboxylic acid (TCA) cycle, which are regulated by PI3K and provide substrates for fatty acid and phospholipid biosynthesis as well as redox metabolism[26,37,38]. Together, the RTK-PI3K axis is recognized as a central pathway promoting lipid anabolism and cell survival, but systematic studies of its role in controlling membrane homeostasis during initial cell death are lacking.

Here, we report on an early cell death-induced decrease in RTK signaling that preferentially limits PI3K-derived phosphoinositide formation and coordinately attenuates de novo fatty acid biosynthesis. Under these conditions, cells preferentially channel exogenous or intracellularly released fatty acids into continuously remodeled phospholipids, which raises the membrane PUFA density, particularly in phosphatidylcholine (PC), and increases their susceptibility to peroxidation and ferroptotic cell death under oxidative membrane stress. In response to specific cytotoxic stressors, including serum deprivation, the increase in the membrane PUFA ratio is mediated by impaired Akt/SREBP1 signaling, requires lysophospholipid acyltransferase (LPLAT) 12/LPCAT3, and is associated with a switch in the expression of long-chain acyl-CoA synthetase (ACSL) isoenzymes. This cell death-fueling mechanism is driven by apoptosis induction and may be supported by cell cycle arrest and decreased cell confluency, and is common to cytotoxic inducers with different modes of action. Together, we here provide strong evidence for a general caspase-independent signaling cascade that sensitizes stressed cells to oxidative membrane damage at the crossroads of apoptotic and necrotic cell death programs.

## Results

### Different cell death conditions increase PUFA ratios in PC

Programmed cell death was induced in fibroblasts via the intrinsic apoptotic pathway by cytotoxic agents covering a broad mechanistic range (Supplementary Fig. 1a). We investigated i) the disruption of cellular K⁺ homeostasis by valinomycin (VAL)[39], which has also been reported to interfere with protein biosynthesis[40] ii) the induction of mitochondrial protein aggregation and impairment of mitochondrial functions by binding of myrtucommulone A (MC) to heat-shock protein (HSP)60[41], iii) pan-kinase inhibition by staurosporine (STS)[42], iv) the blockade of protein translation by cycloheximide (CHX)[43], v) induction of DNA strand breaks by the topoisomerase inhibitor etoposide (ETO)[44], and vi) the depletion of endoplasmic reticulum (ER) Ca²⁺-stores and induction of ER stress by thapsigargin (TPG)[45]. The concentrations of cytotoxic stressors were selected based on the first

appearance of morphological changes such as blebbing, nuclear shrinkage, confluence and/or cell detachment. How these different cytotoxic conditions affect cell proliferation, cell morphology, metabolic activity, membrane integrity, PS externalization, and poly (ADP-ribose) polymerase (PARP) cleavage is shown in Supplementary Fig. 1 and 2, and described in detail in Supplementary Note 1.

Of the six cell death-inducing stress conditions, VAL significantly and three others (MC, STS, TPG) tendentially reduced the cellular content of glycerophospholipids (Fig. 1a), and all consistently increased the availability of free fatty acids (FFA) within 48 h (Fig. 1b). Notably, the cellular content of cardiolipin (CL) increased, as exemplarily shown for VAL (Supplementary Fig. 3a). Supplementary Fig. 3b summarizes the heterogenous kinetic changes in the cellular content of individual lipid subclasses, i.e., PC, phosphatidylethanolamine (PE), PS, phosphatidylglycerol (PG), and FFA. Phosphatidylinositol (PI) levels were hardly affected by the cytotoxic stressors applied here, as reported previously[46].

To identify phospholipid species that are comparably regulated in cell death across cytotoxic settings, we created a co-regulated phospholipid network, in which positively correlated phospholipids are interconnected by lines and placed in close proximity (Fig. 1c). Distant phospholipids from separate clusters are either not or negatively co-regulated. The clusters of the network combine species with different phospholipid headgroups but similar fatty acid composition (Fig. 1c). The lower left cluster is characterized by phospholipids containing saturated fatty acids (SFA) and MUFAs, partially in combination with PUFAs, whereas the remainder is enriched in SFA/PUFA-containing phospholipids.

Principal component analysis was used to identify phospholipids regulated in cell death in the majority of cytotoxic settings. Clearly separated from the bulk of detected lipids, and thus prone to intense regulation, were various PC, several PI and few PE and PS species with preferentially saturated SFAs at the sn-1 position, i.e., palmitic acid (16:0) and stearic acid (18:0), and PUFAs in the sn-2 position, i.e., eicosatrienoic acid (20:3), arachidonic acid (20:4), docosapentaenoic acid (22:5), and docosahexaenoic acid (22:6) (Supplementary Fig. 3c). The cellular fraction of these PUFA-containing phospholipid species increased, whereas species with de novo-synthesized fatty acids, i.e., 16:0, palmitoleic acid (16:1), 18:0, and oleic acid (18:1), but also linoleic acid (18:2) were depleted (Fig. 1d, Supplementary Fig. 4-6). As a result of these partially compensatory effects, the total amount of PC and PE was either barely affected or partially reduced compared to the vehicle control (Supplementary Fig. 3b), mainly due to the decrease of SFA- and MUFA-containing species that dominate the total phospholipids (Fig. 1e and Supplementary Fig. 7a, b). The PUFA-PC and PUFA-PE pools were instead less affected or even slightly increased (Fig. 1e and Supplementary Fig. 7c).

In combination, these effects caused a strong increase in the cellular proportion of PUFA-PC throughout the cytotoxic mechanisms (Fig. 1f), specifically of the PC subclasses containing PUFAs with more than two double bonds (Supplementary Fig. 6). Similar but less pronounced effects were found for PE, PS, and PI (Fig. 1f, Supplementary Fig. 3b, Supplementary Fig. 5a, and Supplementary Fig. 7b–d), whereas the amount of CL increased (Supplementary Fig. 3a), both within SFA- and unsaturated fatty acid (UFA)-containing fractions (Supplementary Fig. 5b and Supplementary Fig. 7e).

For membrane properties, the relative composition of phospholipids is more relevant than the total amount of cellular phospholipids, which largely reflects membrane abundance. Therefore, our focus in the following is on the proportions of cellular phospholipids. Independent of the cytotoxic stimulus and concomitant with the detection of first apoptotic markers (Supplementary Fig. 2), the proportion of PUFA-PC increased within 24 to 48 h (Fig. 1f). The cellular proportion of non-esterified PUFAs was instead not persistently elevated[46], which rather excludes an enhanced uptake or biosynthesis of PUFAs as a driver for the sustained increase of the PUFA-PC ratio in programmed

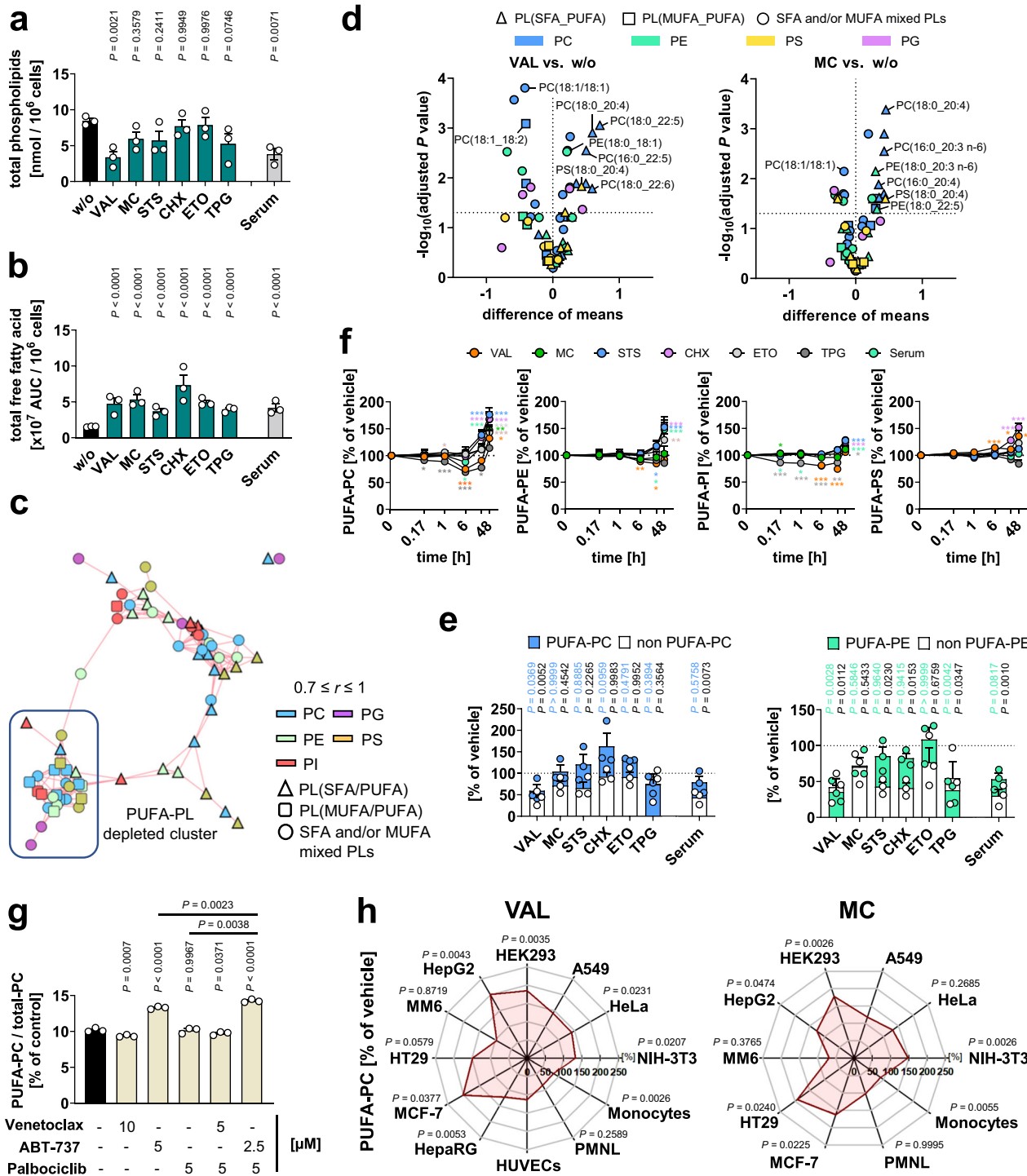

**Fig. 1 | Cytotoxic stress elevates the proportion of PUFAs in phospholipids.**
**a**–**f** NIH-3T3 cells were treated with vehicle, VAL (10 μM), MC (10 μM), STS (0.3 μM), CHX (20 μg/ml), ETO (10 μM), or TPG (2 μM), or were serum starved (Serum) for 48 h unless otherwise indicated. **a** Cellular phospholipid (PC, PE, PS, PG) content. **b** Changes in the free fatty acid content. **c** Co-regulated phospholipid (PL) network (mean percentage changes of phospholipid ratios from three independent experiments) visualizing positive lipid-lipid correlations (correlation factor ($r$) ≥ 0.7). For PI, previously published data were reanalyzed[46]. **d** Volcano plots showing the comparisons of the mean difference of log-transformed phospholipid proportions and the negative log$_{10}$(adjusted $P$ value) in VAL- and MC-treated cells from three (PE, PG, PS) or four (PC) independent experiments. Adjusted $P$ values are given vs. vehicle control; two-tailed multiple unpaired student $t$-tests from log-transformed data with correction for multiple comparisons (false discovery rate 5%). **e** Overlaid changes in the total cellular content of PC and PE that either contain

or are free of PUFAs. **f** Time-dependent changes of the cellular proportion of PUFA-PC, PUFA-PE, PUFA-PI, and PUFA-PS. $P$ values are listed in the source data. For PUFA-PI, previously published data were reanalyzed[46]. **g** PUFA-PC ratio of NIH-3T3 cells after treatment with vehicle, the Bcl2-family inhibitors Venetoclax or ABT-737, or the CDK-inhibitor Palbociclib for 48 h. **h** Cancer cell lines, immortalized cell lines, and primary cells were treated with vehicle, VAL (10 μM) or MC (10 μM) for 8 h (PMNL) or 48 h (all other cell lines/types). Radar charts indicate the percentage changes of PUFA-PC ratios in VAL- or MC-treated cells relative to vehicle control. Mean (**d**, **h**), or mean + s.e.m. (**f**) and single data (**a**, **b**, **e**, **g**) from $n$ = 2-5 independent experiments. Repeated measures one-way or two-way ANOVA of log-transformed data + Dunnett's post hoc test or two-tailed paired student $t$-test of log-transformed data. The exact number of experiments, details on statistical tests and exact $P$ values of panels **b** and **e**–**g** are given in the source data, which are provided with this paper.

cell death. The consistent upregulation of PUFA-PC starting at 24-48 h was preceded by a moderate decrease at 6 h for TPG and VAL (Fig. 1f), which likely depends on the activation of phospholipases A$_2$ releasing PUFAs from phospholipids[47].

The consequences of VAL on intracellular PC pools were investigated by quantitative lipidomics after subcellular fractionation (Supplementary Fig. 8a, b). The PUFA-PC ratio was substantially increased in VAL-treated (peri)nuclear fractions as well as in non-nuclear membranes (Supplementary Fig. 8b), with the strongest effects on PC species containing sn-1-SFAs and sn-2-PUFAs (Supplementary Fig. 8c). Together, PUFA-PC is enriched in major (intra)cellular membranes in fibroblasts during cell death.

The cytotoxic increase in the PUFA-PC ratio may be due to a lower cell confluence at which PUFAs in the culture medium are consumed less rapidly, thereby increasing the fraction of PUFAs available for PC incorporation. Although treatment with VAL is significantly more effective, mimicking lower cell densities by seeding fewer cells does indeed result in markedly increased PUFA-PC ratios (Supplementary Fig. 9a). To exclude that this mechanism dominates the cytotoxic increase in PUFA-PC ratios, we changed the culture medium after 24 h before harvesting the fibroblasts at 48 h. Consistent with our hypothesis that cytotoxic stress manipulates fatty acid or phospholipid metabolism, the change in culture medium slightly increased basal PUFA-PC levels but hardly affected the ability of VAL to upregulate the proportion of PUFA-PC (Supplementary Fig. 9b). Our findings strongly suggest that the cytotoxic decrease in cell density contributes to the increase in PUFA-PC ratios, but seemingly not by limiting the access to exogenous substrates.

Cell cycle arrest, often preceding programmed cell death, has recently been described to coincide with sensitization to ferroptosis by increasing the proportion of PUFAs in membrane phospholipids[48]. To obtain a first notion of whether cell cycle arrest or apoptosis induction is more relevant for the increase in PUFA-PC ratios, we performed kinetic studies. The increase in PUFA-PC ratios upon treatment with VAL (starting at 48 h) corresponded to that of apoptosis induction (Supplementary Fig. 1c and 26c, d) and was preceded by an intermediate G1 cell cycle arrest at 24 h, which disappeared again at 48 h (Supplementary Fig. 9c). Other cytotoxic stressors also increased both the PUFA-PC ratio (Fig. 1f) and induced apoptosis (determined as PARP cleavage) between 24-48 h (Supplementary Fig. 2a), but there are discrepancies for individual stressors. For example, MC induced a marked PARP cleavage already at 24 h, while it only slightly increased the PUFA-PC ratio at this time point. This heterogeneity suggests that both pro-apoptotic events (e.g., cell cycle arrest) and apoptosis can contribute to an increase in the PUFA-PC ratio. To better distinguish between the effects of cell cycle arrest and apoptosis induction, we compared the efficacy of the CDK4/6 inhibitor palbociclib (inducing cell cycle arrest) with the BCL-2 family member inhibitor ABT-737 and the selective BCL-2 inhibitor venetoclax (inducing apoptosis). While pan-inhibition of BCL-2 family members increased the PUFA-PC ratio as expected, selective inhibition of BCL-2 or CDK4/6 had no effect, even in combination (Fig. 1g and Supplementary Fig. 9d). These data indicate that apoptosis induction rather than cell cycle arrest dominates metabolic reprogramming under our experimental settings, although the mechanism of induction must meet specific criteria to be effective.

Using the optimized settings for NIH-3T3 fibroblasts, we treated seven human cancer and immortalized cell lines and four primary human cell types with VAL or MC. Despite varying susceptibility, many cancer cell lines responded to the cytotoxic stressors by increasing the PUFA ratio of membrane phospholipids (Fig. 1h and Supplementary Fig. 10). The cellular proportion of PUFA-PC was significantly increased in A549 lung carcinoma cells, HEK-293 embryonic kidney cells, HepG2 hepatocarcinoma cells, HT-29 colon adenocarcinoma cells, MCF-7 breast adenocarcinoma cells, HeLa cervical carcinoma cells, and differentiated HepaRG hepatocarcinoma cells. Moreover, we observed a trend towards PUFA-PC enrichment in human umbilical vein

endothelial cells (HUVECs) (Fig. 1h and Supplementary Fig. 10). Primary human immune cells, i.e., monocytes and polymorphonuclear leukocytes (PMNL), were not markedly affected.

## Cell death-induced switch in fatty acid metabolism

The decrease in phospholipids during cell death (Fig. 1a) is mainly due to SFA-, MUFA-, and 18:2-containing species and is partially compensated by elevated levels of diverse 20:4-, 22:5- and 22:6-containing phospholipids (Fig. 1d and Supplementary Fig. 4). In combination, these two effects strongly increase the proportion of PUFA-containing species across phospholipid subclasses (Fig. 1f). Mechanistically, we speculated that cytotoxic stress reduces SFA/MUFA biosynthesis and depletes phospholipids of de novo-biosynthesized fatty acids by favoring the competing incorporation of exogenous fatty acids (with a higher proportion of PUFAs). In contrast to SFAs and MUFAs, which are both biosynthesized and taken up for further cellular metabolism, PUFAs are predominantly supplied by the cell culture medium[49]. Our hypothesis is supported by recent studies on breast cancer and endothelial cells, in which the ratio of PUFA-containing phospholipids was increased when fatty acid biosynthesis was blocked with the acetyl-CoA carboxylase (ACC) inhibitor soraphen A[50,51].

We first determined the effect of different cytotoxic settings on fatty acid biosynthesis and degradation[52]. De novo fatty acid biosynthesis was moderately to severely reduced for all six cell death conditions after 48 h (Fig. 2a), whereas the rate of β-oxidation (Fig. 2b), as well as cellular levels of the β-oxidation intermediate butyryl-CoA, were markedly increased (Fig. 2c). Together, various cytotoxic settings induce a metabolic switch from fatty acid biosynthesis to catabolism.

To investigate whether the decrease in fatty acid biosynthesis induced by cell death triggers the increase in PUFA-PC ratio, we treated fibroblasts with the selective ACC inhibitors soraphen A or 5-(tetradecyloxy)-2-furoic acid (TOFA) or the selective SCD1 inhibitor CAY10566. SCD1 and ACC inhibition decreased the cell number, with the latter having less effect (Fig. 2d). ACC inhibition additionally reduced the cellular content of PC within 24 to 48 h (Fig. 2d), and all three compounds strongly increased the PUFA-PC ratio (Fig. 2e). Comparable results were obtained when ACC1 was silenced by siRNA (Fig. 2f and Supplementary Fig. 11). Next, we investigated the role of ACC in upregulating the PUFA-PC ratio under cytotoxic conditions. When ACC was blocked, our panel of cell death inducers was comparably or slightly less efficient in increasing the proportion of PUFA-PC (Fig. 2g and Supplementary Fig. 12a) and decreasing cell number (Supplementary Fig. 12b).

The cellular PUFA-PC ratio further increased synergistically with the combined treatment of VAL and TOFA when exogenous PUFAs were added (Fig. 2h). Conclusively, our data suggest that cytotoxic stimuli increase the proportion of PUFA-PC, in particular when the capacity for de novo fatty acid biosynthesis is low and exogenous PUFAs are available. Note that fatty acid transporters, ACSLs, and LPLATs are required to utilize systemic fatty acids for phospholipid remodeling[53,54].

To elucidate the mechanism by which ACC inhibition induces the shift to PUFA-PC, we performed metabolic flux studies to determine whether PUFA biosynthesis is increased or the incorporation of exogenous PUFAs into phospholipids is stimulated. TOFA, and even more strongly VAL, increased the incorporation of deuterium-labeled 20:4 (20:4-d$_8$) into PC, with the two compounds acting synergistically (Fig. 2i and Supplementary Fig. 12c). On the other hand, neither TOFA nor VAL substantially channeled deuterium-labeled acetate (acetate-d$_3$) into 20:4 biosynthesis from 18:2 or linolenic acid (18:3), and also further elongation to adrenic acid (22:4), 22:5, and 22:6 was barely evident (Supplementary Fig. 12d). The combination of TOFA and VAL, instead, led to a marked incorporation of d$_3$-acetate into 22:5 and 22:6 and less into other PUFAs, albeit at a low rate (Supplementary Fig. 12d). While ACC does not appear to be rate-limiting for PUFA biosynthesis under basal conditions, the enzyme is essential for the elongation of

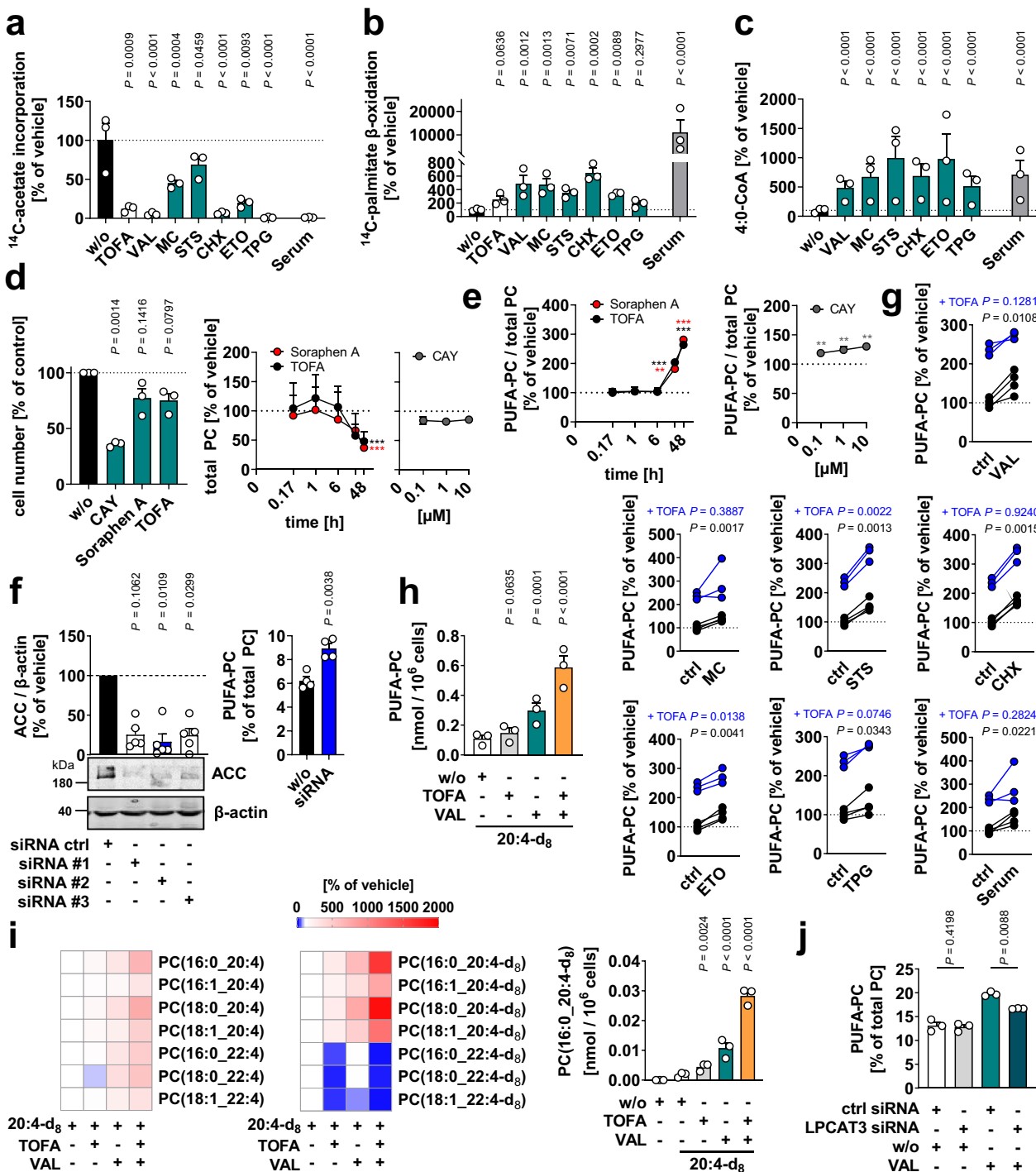

**Fig. 2 | Cytotoxic increase in phospholipid PUFA incorporation. a–c** Fibroblasts were cultured under cytotoxic stress conditions for 48 h. **a** Incorporation of [1-$^{14}$C]-acetate into cellular lipids during the last 4 h of cytotoxic treatment. **b** Degradation of [1-$^{14}$C]-palmitate during the last 3 h of cytotoxic treatment. **c** Cellular butyryl (4:0)-CoA level analyzed by UPLC-MS/MS. **d, e** Viable cell numbers (**d**, left panel), time and concentration-dependent changes in the PC content (**d**, right panels) and PUFA-PC proportion (**e**) of fibroblasts treated with TOFA (5 μM), soraphen A (100 nM) or CAY10566 (CAY; 3 μM if not indicated otherwise). *P* values are listed in the source data. **f** ACC1 was silenced using three different siRNA sequences (#1-3) (left panel) or siRNA #2 (right panel). Non-targeting siRNA was transfected as control (ctrl). Left panel: Protein expression of ACC. Western Blots are representative of five independent experiments. Right panel: Cellular PUFA-PC proportion. **g** Proportion of PUFA-containing diacyl-PC of fibroblasts exposed to cytotoxic stress in presence and absence of TOFA. Interconnected lines indicate data from

the same independent experiment. **h, i** PUFA-PC ratio of fibroblasts that were supplemented with deuterium-labeled 20:4 (1 μM) before being treated with vehicle, TOFA, and/or VAL for 48 h. **h** Cellular PUFA-PC content. **i** Heatmaps showing percentage changes in the cellular content of PUFA-PC species (left panel) and their deuterated analogs (right panel) versus vehicle control. The bar chart depicts the cellular PC(16:0_20:4-d8) content. **j** Fibroblasts with silenced LPLAT12/LPCAT3 were treated with vehicle or VAL, and the cellular proportion of PUFA-PC was determined after 48 h. Non-targeting siRNA was transfected as control (ctrl). Mean (**i**), mean + s.e.m. (**d**, right panel; **e**) and single data (**a–c; d**, left panel; **f, h; i**, right panel; **j**) or paired data (**g**) from n = 3-5 independent experiments. Repeated measures one-way ANOVA of log-transformed data + Dunnett's post hoc test or two-tailed paired student *t*-test of log-transformed data. The exact number of experiments, details on statistical tests and exact *P* values of panels **a–c, e, h,** and **i** (right panel) are given in the source data, which are provided with this paper.

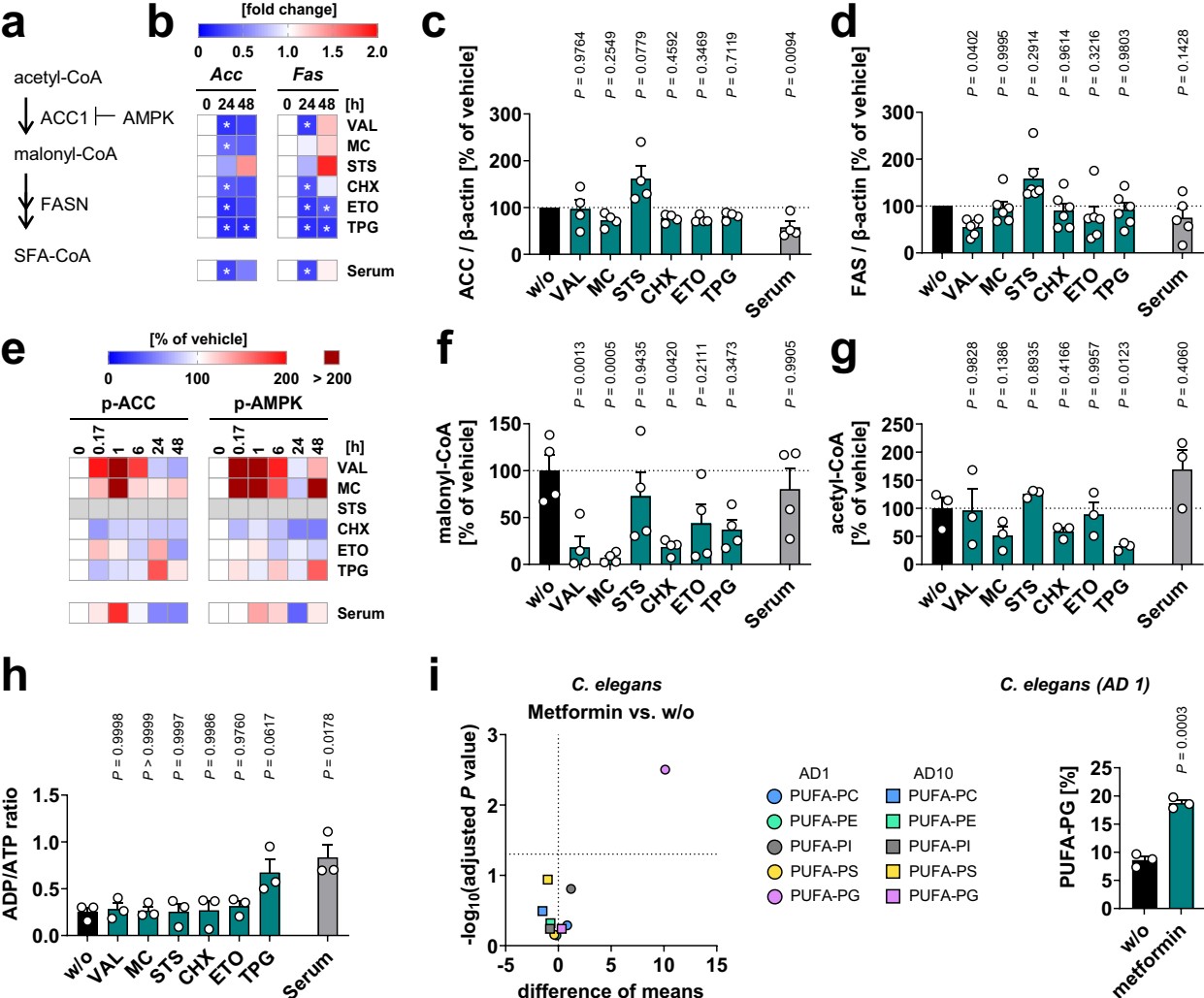

**Fig. 3 | Cytotoxic stress induces a switch from SFA/MUFA biosynthesis to β-oxidation through multifaceted mechanisms. a** Schematic overview about de novo fatty acid biosynthesis. **b–h** Fibroblasts were cultivated under diverse cytotoxic conditions for 48 h (**c**, **d**, **f–h**) or as indicated (**b**, **e**). **b** Heatmaps showing the time-dependent effect on *Acc* and *Fas* mRNA levels (normalized to total cellular RNA amounts). **c**, **d** ACC (**c**) and FAS (**d**) protein expression (representative Western blots: Supplementary Fig. 14a, b). **e** Heatmaps showing the time-dependent changes of cellular p-ACC and p-AMPK normalized to β-actin or GAPDH (representative Western blots: Supplementary Fig. 14a, c). **f**, **g** Cellular level of malonyl-CoA (**f**) and acetyl-CoA (**g**) analyzed by UPLC-MS/MS. **h** Cellular ADP/ATP ratio. Mean (**b**, **e**) or mean + s.e.m. and single data (**c**, **d**, **f–h**) from n = 1–6 independent experiments. The exact number of experiments and exact P values (**b** and **h**) are given in the source data, which are provided with this paper. *P < 0.05 (**b**) or P values vs. vehicle control; mixed-effects model (REML) (**b**, Acc 48 h, **d**) or repeated measures one-way

ANOVA of log-transformed data (**b**, Acc 24 h, Fas 24/48 h, **c**, **f–h**) + Dunnett's post hoc test. **i** Young (adulthood day 1, AD1) and old (AD10) *C. elegans* (≥ 700 animals) were treated with vehicle or metformin (50 mM) for 24 h, and the PUFA proportion in phospholipid subclasses was analyzed on organismal level. For PC and PE, previously published data were reanalyzed[58]. Volcano plots indicate phospholipid subclasses, whose PUFA proportions are strongly and significantly altered by metformin. Comparisons of the indicated treatment groups show the mean difference of the PUFA-containing phospholipid proportions, as indicated, and the negative log$_{10}$(adjusted P value) from three independent experiments. Adjusted P values given vs. vehicle control; two-tailed multiple unpaired student t-tests with correction for multiple comparisons using a two-stage linear step-up procedure by Benjamini, Krieger, and Yekutieli (false discovery rate 5%). Mean + s.e.m. and single data from the bar chart (**i**) were compared using a two-tailed unpaired student t-test.

exogenous PUFAs, as indicated by the greatly reduced conversion of deuterium-labeled 20:4 to 22:4 by TOFA (Fig. 2i and Supplementary Fig. 12e). VAL was less efficient in inhibiting this elongation step, suggesting that ACC is either only partially suppressed by cytotoxic stress or only after interception of free deuterium-labeled 20:4, e.g., by channeling the fatty acid into esterified cellular lipid pools[53]. Together, our data suggest that the reduced availability of SFAs and MUFAs upon ACC inhibition decreases the competition between acyl-CoA species in favor of PUFA-CoAs, which are efficiently incorporated into the overall depleting membrane phospholipids (Fig. 1a). Accordingly, increasing the availability of MUFAs by 18:1 supplementation prevented the rise in PUFA-PC and PUFA-PE ratios (Supplementary Fig. 13a). Knockdown

studies in VAL-treated fibroblasts indicate that LPLAT12/LPCAT3 is involved in membrane PUFA incorporation (Fig. 2j and Supplementary Fig. 13b). This finding is consistent with the substrate specificity of LPLAT12/LPCAT3, which preferentially accepts PUFA-CoA and acylates lyso-PC, lyso-PE, and lyso-PS at the *sn*-2 position[53]. Together, our data indicate that VAL-treated cells have an increased capacity to incorporate exogenous PUFAs into phospholipids.

We then addressed the individual mechanisms by which cell death inducers reduce de novo SFA and MUFA biosynthesis. The generation of SFAs from acetyl-CoA depends on the concerted action of ACC, the rate-limiting enzyme of fatty acid biosynthesis, and fatty acid synthase (FAS) (Fig. 3a), both of which are subject to transcriptional regulation[52].

The mRNA levels of *Acc* or *Fas* were reduced after 24 and/or 48 h under the cytotoxic conditions (except for STS; Fig. 3b), but this decrease was only partially translated into lower ACC (Fig. 3c) and FAS (Fig. 3d) protein levels, reaching significance for serum depletion (ACC) and VAL (FAS). Note that *β-act*in and *Gapdh* mRNA expression (unlike protein expression, Supplementary Fig. 14) was variable over time and between treatments[46]. Therefore, we normalized ACC and FAS expression to total RNA content. In addition, VAL, MC, TPG, and serum depletion inhibited ACC by phosphorylation at S79 (Fig. 3e) for a limited time between 10 min and 24 h, a critical period that determines PUFA-PC levels at later stages. One of the major kinases that phosphorylates ACC at S79 is AMP-activated protein kinase (AMPK), a cellular energy sensor that controls cell growth, cell metabolism, and autophagy[55] (Fig. 3a). Phosphorylation and thus activation of AMPK partially correlated with the cellular levels of inactivated p-ACC for VAL, MC, and serum depletion (Fig. 3e). The next step in fatty acid biosynthesis, the desaturation of SFAs to MUFAs by SCD1, was more consistently suppressed in all cytotoxic settings after 48 h, as recently shown by us[46], even when the culture medium was replaced after 24 h (Supplementary Fig. 14d). Overall, the mechanisms by which cytotoxic stress interferes with de novo fatty acid biosynthesis are heterogeneous. VAL, MC, CHX, TPG, and serum depletion significantly interfere with at least one of the steps investigated, while ETO tends to do so and is also the least active stressor in upregulating PUFA-PC ratios between 6 and 48 h (Fig. 1f). The mechanisms by which STS increases PUFA-PC ratios remain unclear.

The levels of the ACC product, malonyl-CoA, were efficiently decreased by VAL, MC, and CHX and by trend by ETO and TPG (Fig. 3f). Given the limited ACC repression (Fig. 3c) and inhibitory phosphorylation (Fig. 3e), additional mechanisms may contribute to the depletion of malonyl-CoA, such as the reduced availability of the ACC substrate acetyl-CoA under some of the cytotoxic conditions (Fig. 3g). Notably, malonyl-CoA inhibits the rate-limiting step in β-oxidation, i.e., the mitochondrial fatty acid uptake via the carnitine acyltransferase transporter[52].

Energy-consuming fatty acid biosynthesis might also be compromised by cellular energy depletion. Therefore, we investigated whether our set of cell death conditions affects cellular ADP and ATP levels. Serum depletion and, by trend, TPG, but not other cytotoxic agents, increased the ADP/ATP ratio (Fig. 3h), indicating a low cellular energy status. To investigate whether energy depletion together with AMPK activation increases the cellular PUFA ratio in phospholipids at an organismal level, we treated *Caenorhabditis elegans* with metformin, which inhibits mitochondrial respiration, activates AMPK, and mediates its insulin-sensitizing effect through both AMPK-dependent and independent mechanisms[56,57]. As expected, metformin substantially decreased fatty acid biosynthesis[58] and increased the phospholipid PUFA fraction specifically in PG, favorably in young adult worms (Fig. 3i).

Together, fatty acid biosynthesis was severely reduced for many cytotoxic settings with varying kinetics and concomitant with different processes including i) reduction of ACC and FAS expression, ii) inhibition of ACC by post-translational modification, iii) limitation of ACC substrate supply, and/or iv) energy depletion. Several of the cytotoxic settings affect multiple pathways, but none affect all.

To unravel the metabolic network behind the cytotoxic enrichment of PUFA-containing phospholipids, we quantitatively analyzed the proteome of fibroblasts challenged with either VAL or MC (Supplementary Data 1 and 2)[46]. Supplementary Fig. 15 lists the 49 biological processes among the top 100 enriched for differentially regulated proteins significantly depleted by both VAL and MC, with the order reflecting the overall effect of the two stressors (Supplementary Data 3). With few exceptions, these differentially regulated processes were related to energy metabolism (e.g., aerobic respiration), energy-consuming biosynthetic pathways (e.g., nucleotide biosynthesis), or gene expression (e.g., RNA splicing) and included fatty acid

metabolism, consistent with the metabolic shift observed in the lipidomic profiles. These metabolic adaptions in VAL-treated cells were not only compensated but even counter-regulated by co-treatment with the MUFA 18:1 (Supplementary Fig. 15), which prevented the accumulation of PUFA-PC (Supplementary Fig. 13a). In contrast, 18:1 had little or no effect on non-stressed cells (Supplementary Fig. 15). Inhibition of fatty acid biosynthesis by the SCD1 inhibitor CAY10566 mimicked the metabolic adaptations to a limited extent, seemingly by depleting PI(18:1/18:1) (Supplementary Fig. 15). These results emphasize that changes in fatty acid metabolism are critical for global metabolic alterations in stressed cells, but also show that additional mechanisms beyond fatty acid biosynthesis are implicated.

To narrow down the proteomic data to the key metabolic stress responses, we focused on proteins that were similarly regulated by both cytotoxic stressors. VAL and MC interfered with glycolysis, the TCA cycle, and oxidative phosphorylation (Fig. 4a, b, Supplementary Fig. 16 and 17). Both cytotoxic stressors also repressed enzymes involved in fatty acid biosynthesis, turnover, and degradation as well as de novo phospholipid biosynthesis (Fig. 4a, b, Supplementary Fig. 18 and 19a). Notably, the mitochondrial outer membrane isoenzyme ACSL5 was upregulated 2.0 to 4.5-fold by VAL, MC, and TPG, but less by other cytotoxic stressors (Fig. 4c, d), and both ACC (negatively) and ACSL5 levels (positively) strongly correlate with the cellular PUFA-PC ratio (Supplementary Fig. 19b). ACSL5 converts a variety of medium- to long-chain fatty acids, including SFAs, MUFAs, and PUFAs, into the corresponding CoA-esters[59] and has been reported to channel exogenous rather than endogenously synthesized fatty acids towards lipid biosynthesis[60]. While ACSL5, together with other ACSLs, may help to sustain PUFA incorporation under cytotoxic stress, it does not drive the increase in PUFA-PC. On the contrary, ACSL5 knockdown diminishes the incorporation of SFAs/MUFAs into PC, thereby increasing the PUFA-PC ratio in vehicle-treated but hardly in VAL-treated cells, in which ACSL5 is upregulated (Fig. 4e).

We also observed elevated levels of cytosolic phospholipase A$_{2\alpha}$ (PLA2G4A) (Fig. 4a, b, f and Supplementary Fig. 18), a major Ca$^{2+}$-regulated phospholipase that releases *sn*-2 fatty acids from phospholipids[47]. This aligns with the decreased cellular phospholipid content (Fig. 1a) and increased FFA levels (Fig. 1b) upon cell death induction. The level of lysophospholipids, the second PLA2 cleavage products, was not consistently increased (Supplementary Fig. 19c, d), possibly due to a rapid turnover under steady-state conditions[46]. Together, cell death induction disrupts the ability of cells to synthesize fatty acids, including PUFAs, while inducing a switch in ACSL isoenzyme expression that may maintain efficient activation of diverse fatty acids. These fatty acyl-CoA esters are then either incorporated into phospholipids or degraded. While cytotoxic stress enhanced fatty acid oxidation (Fig. 2b, c), we observed in the proteome decreased levels of mitochondrial β-oxidation enzymes, especially those with specificity for unsaturated (DECR1, ECI1) and medium- and long-chain fatty acids (HADH, HADHB, ACADL, ACADM, ACAT1, ACAA2) (Fig. 4a, b and Supplementary Fig. 18). It is therefore tempting to speculate that very long-chain mitochondrial or peroxisomal fatty acid catabolism is gaining importance, as suggested by the maintained expression of ACADVL (Supplementary Data 1).

In addition, cytotoxic conditions induced a shift from hexokinase (HK)1 to HK2 (Fig. 4g, h and Supplementary Fig. 16). The former isoenzyme is associated with glycolysis, whereas the latter preferentially channels glucose to anabolic pathways and the pentose phosphate shunt[61]. Thus, HK2 induction combined with HK1 repression may be a strategy of stressed cells to meet the demand for NADPH, e.g., to detoxify membrane hydroperoxides. Consistent with this hypothesis, the oxidative phase of the pentose phosphate pathway to glucono-1,5-lactone-6-phosphate via glucose-6-phosphate isomerase (GPI1) and glucose-6-phosphate dehydrogenase (G6PDX) was hardly affected by cytotoxic stress (Supplementary Data 1).

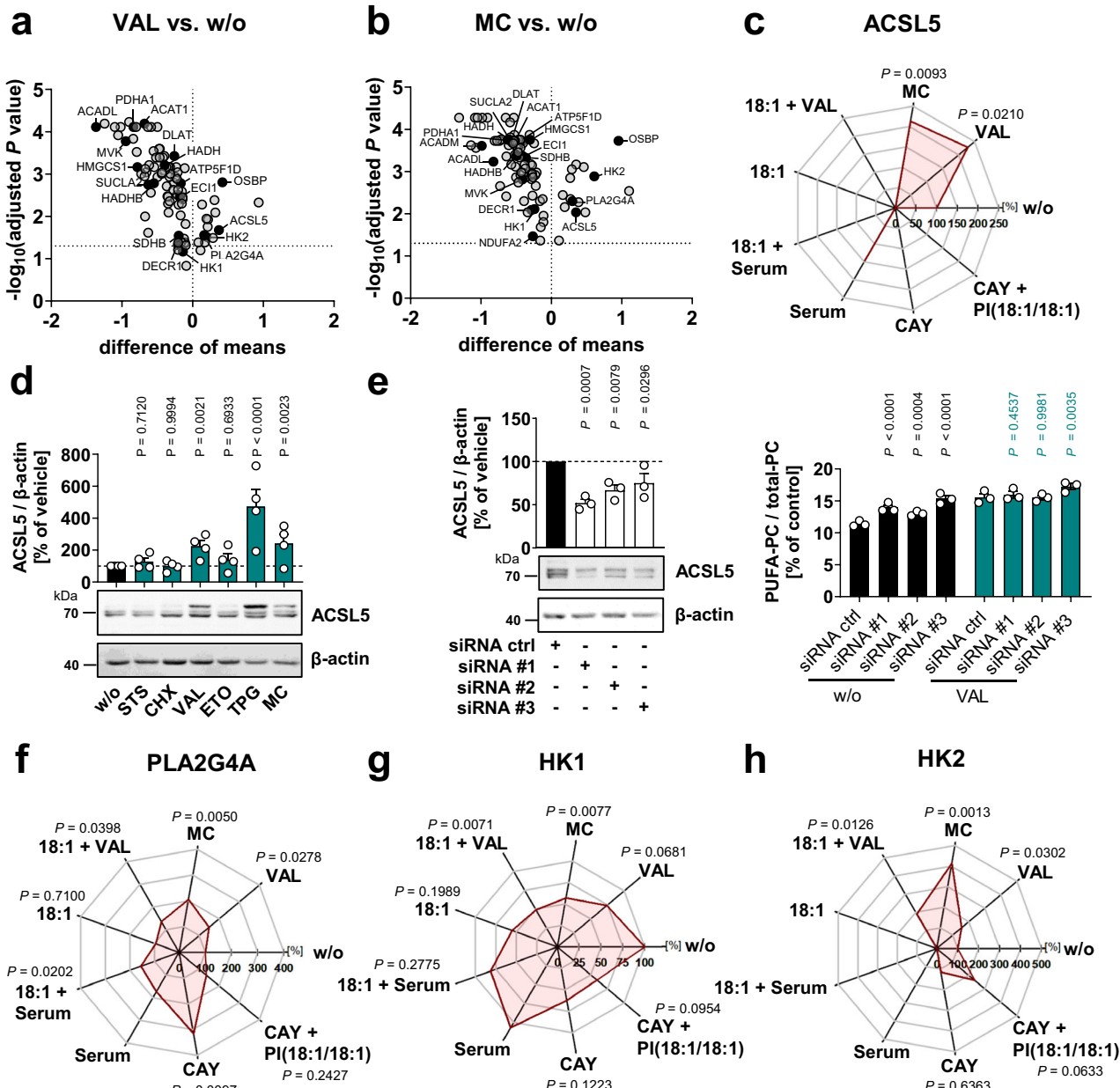

**Fig. 4 | Mechanistic insights into metabolic stress adaption from quantitative proteomics.** Focus: proteins in glycolysis, gluconeogenesis, the TCA cycle, fatty acid biosynthesis, uptake and degradation, (phospho)lipid metabolism, and peroxisome biogenesis and metabolism that are up-/downregulated by VAL and MC in the same direction ($\geq$ 20%). **a, b** Fibroblasts were exposed to cytotoxic stress (48 h). Volcano plots showing the difference of mean absolute intensities of $\log_{10}$ data and the negative $\log_{10}$(adjusted $P$ value) versus vehicle control in VAL- and MC-treated cells. **c, f–h** Fibroblasts were treated with vehicle, VAL (10 µM), MC (10 µM), 18:1 (100 µM), VAL (10 µM) plus 18:1 (100 µM), CAY10566 (CAY, 3 µM), CAY (3 µM) plus PI(18:1/18:1) (50 µM), STS (0.3 µM), CHX (20 µg/ml), ETO (10 µM), TPG (2 µM), or were serum starved (serum) in presence or absence of 18:1 (100 µM) for 48 h. Radar charts indicate the percentage changes of cellular ACSL5 (**c**), PLA2G4A (**f**), HK1 (**g**), and HK2 (**h**) relative to vehicle control. **d** ACSL5 expression normalized to β-actin versus vehicle control. **e** ACSL5 was silenced using three different siRNA sequences (#1–3) and non-targeting siRNA was transfected as control (ctrl). Transfected cells were treated with vehicle or VAL (10 µM) for 48 h. Left panel: ACSL5 protein expression. Western Blots are representative of three independent experiments. Right panel: Cellular PUFA-PC proportion. Mean (**a–c**, **f–h**) or mean + s.e.m and single data (**d**, **e**) from $n$ = 3 (**a–c**, **f–h**, except $n$ = 2 for Serum; **e**) or $n$ = 4 (**d**) independent experiments. Mean from $n$ = 3 (except $n$ = 2 for serum) independent experiments. Adjusted $P$ values given vs. vehicle control (**a–c**, **f–h**); two-tailed multiple unpaired student $t$-tests from log-transformed data with correction for multiple comparisons (false discovery rate 5%) (**a–c**, **f–h**). $P$ values vs. vehicle control (**d**) or respective non-targeting siRNA (**e**); repeated measures one-way ANOVA of log transformed data + Dunnett's post hoc test (**d**, **e**). The exact $P$ values of panel **d**, **e** (right panel) are listed in the source data, which are provided with this paper.

## Cytotoxic stress attenuates insulin and growth factor signaling

Next, we asked whether cytotoxic stress influences superordinate signaling pathways that regulate fatty acid metabolism and trigger the metabolic shift from SFA- and MUFA-containing phospholipids to PUFA-containing PC and other phospholipids. Again, only proteomic changes that were equally evident in VAL- and MC-treated fibroblasts were considered. Cytotoxic stress decreased the cellular levels of growth factor RTKs (IGF1R, FGFR1), receptor-stabilizing proteins (CDH2), adapter proteins (PTPN11, NCK1, FRS2) as well as downstream signaling components, including ribosomal protein S6 kinase alpha-5 (RPS6kA5) and GSK-3β (Fig. 5a–g and Supplementary Data 1). In parallel, VAL, and MC induced the expression of insulin-like growth factor-binding protein 6 (IGFBP6) (Fig. 5a–c, h), which decreases the availability of the RTK ligand IGF2 and has anti-proliferative and pro-

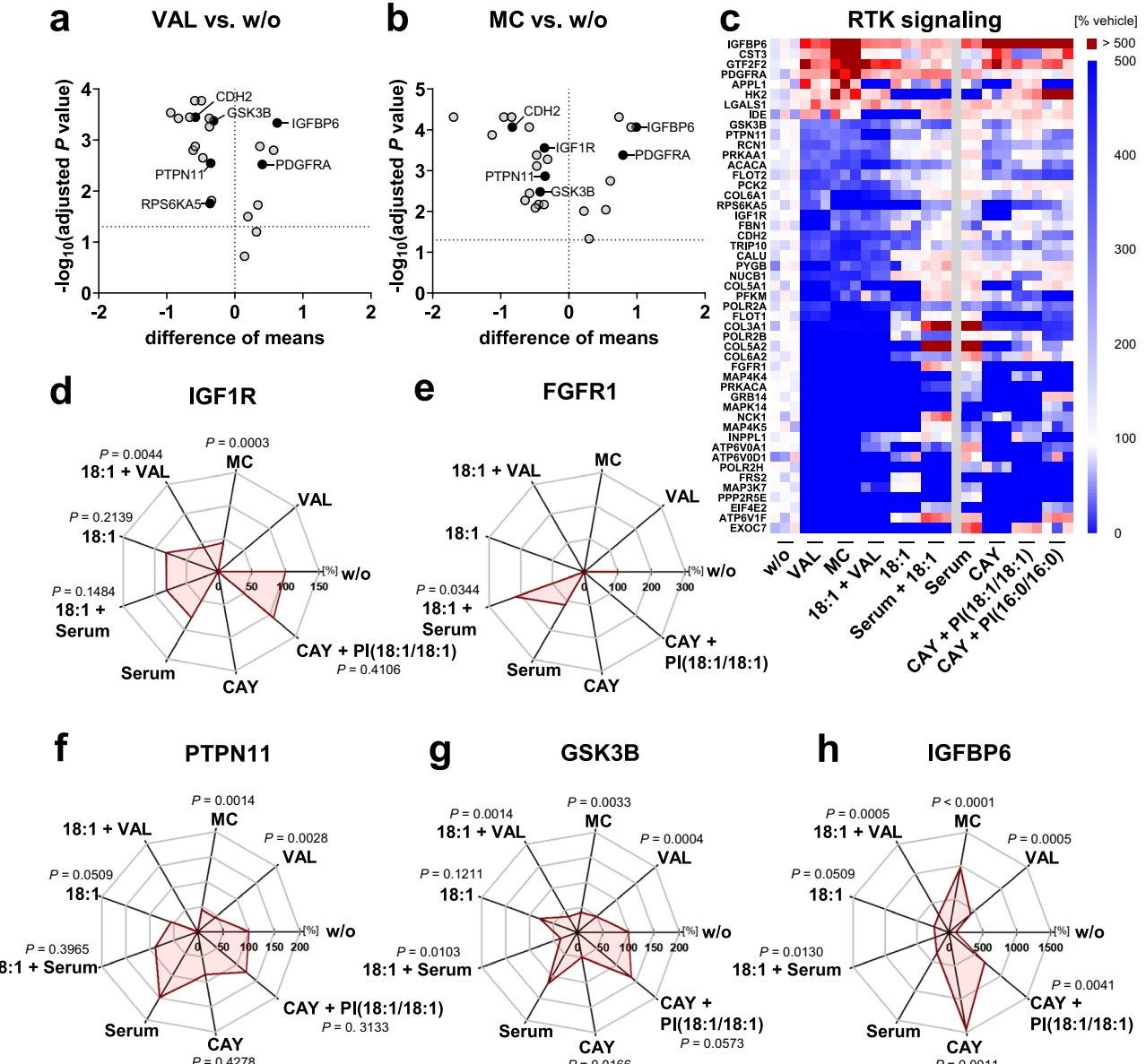

**Fig. 5 | Cytotoxic stress impairs growth factor signaling on proteome level.** Fibroblasts were treated with vehicle, VAL (10 μM), MC (10 μM), 18:1 (100 μM), VAL (10 μM) plus 18:1 (100 μM), CAY10566 (CAY, 3 μM), or CAY (3 μM) plus PI(18:1/18:1) (50 μM) or the saturated control PI(16:0/16:0) (50 μM), or were serum starved (serum) in presence or absence of 18:1 (100 μM) for 48 h. Focus is placed on proteins that participate in insulin and growth factor (IGF1, PDGF, and FGF) signaling and are up- or downregulated by VAL and MC in the same direction (≥ 50%). **a**, **b** Volcano plots highlight proteins that are regulated by VAL (**a**) or MC (**b**) compared to vehicle control. Comparisons of the indicated treatment groups show the difference of mean absolute intensities of log$_{10}$ data and the negative log$_{10}$(adjusted *P* value). **c** Heatmap showing relative changes in protein levels. Single data of independent experiments (*n* = 3, except *n* = 2 for serum depletion) were calculated as the percentage of vehicle control. **d**–**h** Radar charts indicate the percentage changes of cellular IGF1R (**d**), FGFR1 (**e**), PTPN11 (**f**), GSK3B (**g**), and IGFBP6 (**h**) relative to vehicle control. Mean from *n* = 3 (except *n* = 2 for serum depletion) independent experiments. Adjusted *P* values given vs. vehicle control; two-tailed multiple unpaired student *t*-tests from log-transformed data with correction for multiple comparisons using a two-stage linear step-up procedure by Benjamini, Krieger, and Yekutieli (false discovery rate 5%). The exact *P* values of panel **h** are listed in the source data. Source data are provided with this paper.

apoptotic functions[62]. RTKs drive anabolic lipid metabolism through the PI3K/Akt-axis, both through mTORC1-dependent and -independent mechanisms[63], and are critically involved in survival signaling and stress adaption[38,64–66]. Their stringent downregulation upon initiation of cell death might contribute to its progression and provides a rational for the switch from fatty acid biosynthesis to degradation.

We further explored whether these stress-induced changes in the proteome link with impaired fatty acid (18:1) availability and/or the inhibition of MUFA biosynthesis (i.e., by the selective SCD1 inhibitor CAY10566). Several factors at key nodes in glucose (PFKM, PYGB), energy (NDUFS1), and lipid (PLA2G4A, ESYT1/2) metabolism, as well as

insulin/growth factor signaling (IGF1R, IGFBP6, IDE) were at least partially up- or downregulated following the decrease in fatty acid availability, and some of the effects (PLA2G4A, NDUFS1, IGF1R, IGFBP6) were mediated by the depletion of the SCD1-derived lipokine PI(18:1/18:1)[46] (Fig. 4f, Fig. 5c, d, g, h, Supplementary Figs. 16, 18 and 19a). Our data indicate that the cell death-induced decrease in fatty acid biosynthesis shapes the proteome towards a more severe cytotoxic response.

Quantitative proteomics suggests that insulin- and growth-factor-signaling are major lipid anabolic pathways that are downregulated upon cell death induction in fibroblasts. Accordingly, the cytotoxic stressors reduced cellular levels of PI(3,4,5)P$_3$ (Fig. 6a), and three of the

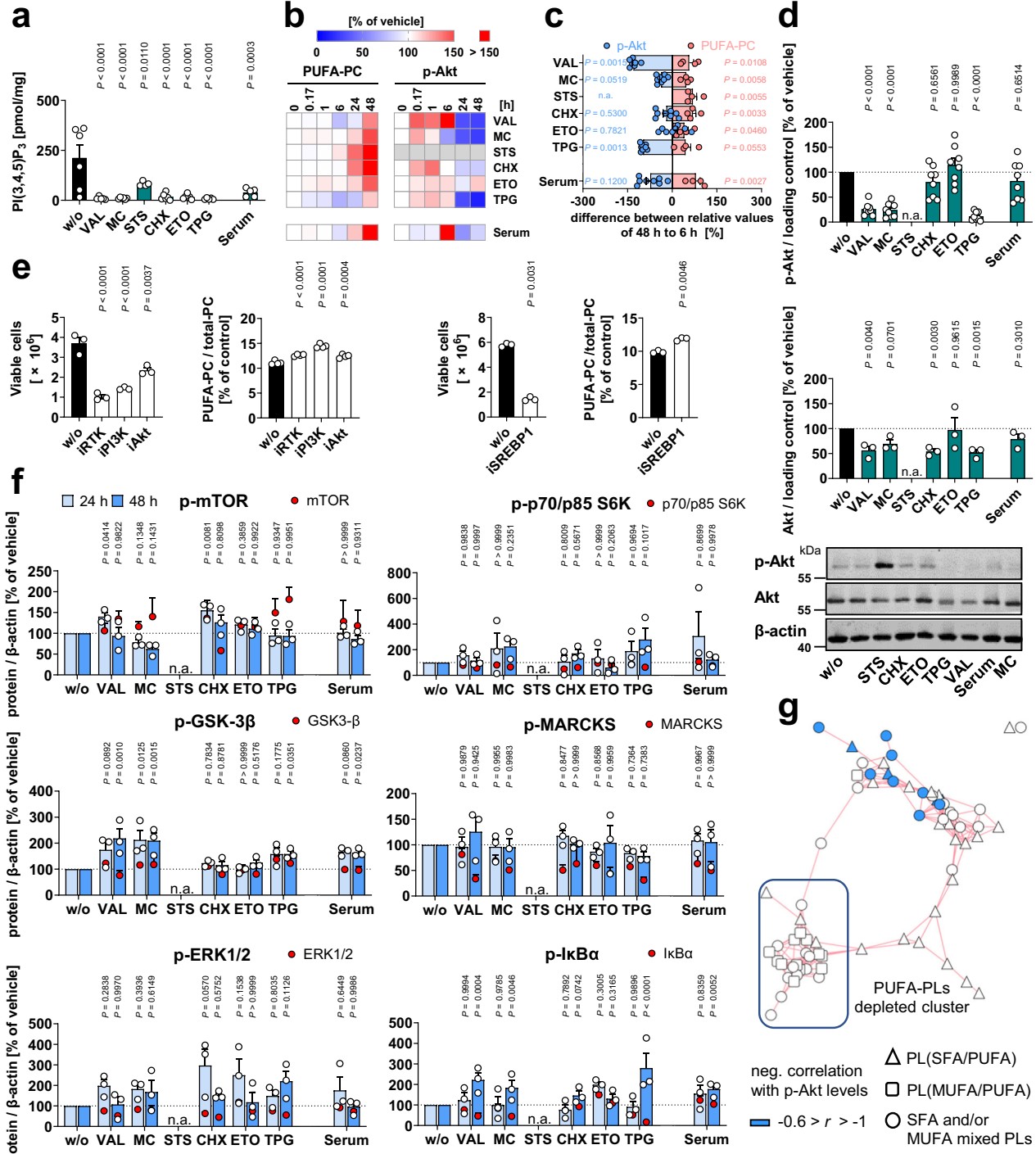

**Fig. 6 | Cytotoxic stress suppresses PI3K/Akt signaling.** Fibroblasts were culti-vated under cytotoxic conditions for 48 h or as indicated. **a** Cellular PI(3,4,5)P$_3$ levels measured by ELISA. **b** Heatmaps showing the time-dependent changes of cellular p-Akt (normalized to β-actin or GAPDH) and the cellular proportion of PUFA-PC. Data for PUFA-PC is also shown in Fig. 1f. **c** Regulation of cellular p-Akt levels and the proportion of PUFA-PC within 6 to 48 h; n.a., not analyzed. Data were calculated by subtracting the average percentage of control for the treatment group at 6 h from the values at 48 h. **d** Changes in the cellular levels of p-Akt and total Akt normalized to β-actin or GAPDH relative to vehicle control; n.a., not analyzed. **e** Viable cell numbers and PUFA-PC ratio after treatment with LY294002 (iPI3K, 10 μM), picropodophyllin (iRTK, 10 μM), ipatasertib (iAkt, 10 μM), or fatos-tatin (iSREBP1, 20 μM) for 48 h. **f** Phosphorylation of kinases or kinase substrates and total protein expression (red dots) was determined after 24 and 48 h and

normalized to β-actin; n.a., not analyzed. Representative Western blots are shown in Supplementary Fig. 23. Mean (**b**) or mean + s.e.m. and single data (**a**, **c**–**f**) from $n$ = 2-8 independent experiments. The exact number of experiments is given in the source data. $P$ values vs. vehicle control; mixed-effects model (REML) (**a**) or repe-ated measures one-way ANOVA of log-transformed data (**d**; **e**, left panels; **f**) + Dunnett´s post hoc test or two-tailed unpaired student $t$-test of log-transformed data (**c**, **e** right panels). **g** Negative correlations (−0.6 > $r$ > −1) between cellular p-Akt (Ser472) levels and the cellular proportion of phospholipid (PL) species are shown for the co-regulated lipid network described in Fig. 1c. Pearson correlations were calculated for the mean of p-Akt levels from eight independent experiments. The exact $P$ values of panels **a**, **d** (for p-Akt), and **e** (left panels) and **f** are listed in the source data. Source data are provided with this paper.

six stressors (plus serum depletion by trend) also diminished Akt phosphorylation at Ser473 within 6 to 48 h (Fig. 6b–d, Supplementary Fig. 20a, b). Total Akt levels were also decreased, but the effect was less pronounced (Fig. 6d). STS-induced changes in kinase phosphorylation, including the increase in Akt phosphorylation (Supplementary Fig. 20b), were not considered further because they are not necessarily indicative of kinase activation, as STS is a pan-kinase inhibitor[42].

As a central driver of SREBP1 proteolytic processing, nuclear transport, and activation, the RTK-PI3K-Akt axis enhances the expression of lipogenic SREBP1 target genes, such as ACC, SCD1, and FAS[67]. Accordingly, inhibition of IGFR by picropodophyllin, PI3K by LY294002, Akt by ipatasertib, and SREBP1 by fatostatin increased the PUFA-PC ratio (Fig. 6e). IGF1R and Akt inhibition also induced the expression of ACSLs, preferentially of ACSL5, and PI3K inhibition tended to suppress SCD1 expression (Supplementary Fig. 21a), similar to specific cytotoxic stressors (Fig. 4c, d)[46]. Immunofluorescence staining of SREBP1 revealed that VAL increased the nuclear localization of SREBP1 (Supplementary Fig. 21b), which follows SREBP1 maturation and is required for SREBP1-dependent gene expression. We speculated that VAL may interfere with the biosynthesis of SREBP1 target proteins or attenuate SREBP1 activity by post-translational modification[67]. Screening of small molecule inhibitors/activators of SREBP1-regulatory pathways suggests that p53 (pifithrin-β) and AMPK (BAY3827), which is activated by VAL (Fig. 3e), may contribute to a limited extent to the VAL-induced increase in PUFA-PC ratios (Supplementary Fig. 21c). Together, cytotoxic stress, at least when induced by VAL, represses RTK signaling towards SREBP1 (and possibly other pathways), thereby increasing the PUFA-PC ratio.

Regulatory phosphorylation or substrate turnover of other insulin/growth factor-regulated kinases was not decreased by cytotoxic stress, but rather increased within 24–48 h, reaching significance in four out of six cytotoxic stress conditions for p-IκBα (as a marker of IKK activity) and inactive p-GSK-3β (Fig. 6f). This compensatory regulation might be explained by the decline of the serine/threonine-protein phosphatase Pp2A[46], whose pleiotropic substrates include IKK and factors upstream of GSK-3β, but also Akt (at Thr308), ERK1/2, p70-S6K, and PKC isoenzymes[68,69]. Conclusively, RTK signaling to PI(3,4,5)P3 is efficiently suppressed by cytotoxic stress, which is in some cases also translated into decreased Akt phosphorylation at Ser473, whereas the phosphorylation of other survival kinases is not or inversely regulated. The already reduced p-Akt levels decline further at 48 h, when the expression of kinases (Akt, p70/p85-S6K and ERK1/2) and their substrates (MARCKS) decreases (Fig. 6d, f and Supplementary Fig. 22).

Akt is activated by a variety of signaling cascades, including the RTK-PI3K axis, which triggers a wide range of Akt-dependent and -independent signaling events. As the levels of p-Akt (used here as an indicator of the cytotoxic metabolic switch) decreased between 6 and 48 h for VAL, MC, TPG, and, by trend, serum depletion, the PUFA ratio in phospholipids increased proportionally for all six of the seven stressors and had a trend for the remaining (TPG) (Fig. 6b–d). The inverse correlation was strongest for PC species of the PUFA-enriched cluster (Fig. 6g). We also observed short-term Akt activation for various cytotoxic inducers within 10 min to 1 h. However, these effects were not linked to changes in the PUFA-PC ratio, which remained constant during this period (Fig. 6b). Note that changes in Akt signaling and phospholipid metabolism that precede fibroblast death are largely mediated by caspase-independent pathways, as suggested by the failure of the pan-caspase inhibitor (3S)-5-(2,6-difluorophenoxy)-3-[[(2S)-3-methyl-1-oxo-2-[(2-uinolinylcarbonyl)amino]butyl]amino]-4-oxo-pentanoic acid (Q-VD-OPh) to compensate for the cytotoxic effects (Supplementary Note 2 and Supplementary Fig. 24).

The inverse regulation of p-Akt and PUFA-containing phospholipids observed in specific cytotoxic settings seems also to be relevant in chemoresistant cancer cells, for example in sorafenib-resistant human Huh-7 hepatocarcinoma cells. These cells strongly upregulate Akt[70]

and have considerably lower proportions of PUFA-containing glycerophospholipids (PC, PE, PS, and PI) than the sensitive counterpart (Supplementary Fig. 25).

The Akt inhibitor GSK690693 promoted apoptotic PARP cleavage under cytotoxic conditions by trend (Supplementary Fig. 26a, b), increased the number of early apoptotic cells (Supplementary Fig. 26c, d), and tended to enhance necrotic and late apoptotic cell death (which has a necrotic component) under cytotoxic conditions within 48 h (Supplementary Fig. 26c, d), as described in Supplementary Note 3. Kinetic studies on cell death induction revealed comparable effects for ACC inhibition by TOFA on necrotic and late apoptotic cell death but only starting from 72 h (Supplementary Fig. 27). We conclude that the pro-survival activity of Akt is not necessarily dependent on ACC expression in early, but potentially in advanced cell death. Expression of constitutively active Akt (Supplementary Fig. 28a) reduced the number of cells in early apoptosis (48 h; Supplementary Figs. 28b and 29), as expected, while the effects on viable cells varied between the cytotoxic conditions, ranging from protective to supportive activities (Supplementary Figs. 28c and 29). Together, Akt contributes to the regulation of cell death under certain cytotoxic conditions, which seems to be independent of fatty acid biosynthesis at early stages.

### Regulation of membrane PUFA ratios by insulin/growth factors

To define the role of RTK signaling in shaping the phospholipid fatty acid composition, we stimulated fibroblasts with either insulin, platelet-derived growth factor (PDGF), or basic fibroblast growth factor (bFGF). These RTK ligands induced PI(3,4,5)P3 formation and Akt phosphorylation[71] and decreased the proportion of PUFAs in cellular phospholipids, specifically in PE, PI[72], and PS (Fig. 7a and Supplementary Fig. 30a, b). The availability of free PUFAs, instead, was not substantially altered (Fig. 7b and Supplementary Fig. 30c). Changes in the PUFA composition of phospholipids were evident in all subcellular fractions (nuclear, mitochondrial, membrane, cytosolic), as exemplified by PI in bFGF-treated Swiss-3T3 fibroblasts (Supplementary Fig. 30d). bFGF and PDGF (but not insulin) were also effective on NIH-3T3 fibroblasts and 3T3-L1 pre-adipocytes, whereas Hepa1-6 hepatocarcinoma cells were less responsive (Fig. 7c and Supplementary Fig. 30e).

Activation of the glucocorticoid receptor (by dexamethasone) partially attenuated the bFGF-induced decline in PUFA-PI, whereas inhibition of PKC (by Go6976) and, tendentially, mitogen-activated protein kinase kinase (MEK)1/2 (by U0126) enhanced the effect (Fig. 7d). 20:4-containing PI species (20:4-PI) were subject to even more extensive regulation: the bFGF-mediated decrease in 20:4-PI was partially compensated by inhibition of p38 mitogen-activated protein kinase (MAPK) (by SB203580), PI3K (by LY294002), and cPLA2 (by pyrrophenone) in addition to dexamethasone (Fig. 7d). In conclusion, two central signaling pathways of insulin/growth factor signaling, i.e., PI3K/Akt and MEK/ERK, shift the PUFA/20:4 composition of phospholipids in opposite directions and further pathways contribute to this regulation, which in total markedly decreases the PUFA proportion of cellular membranes.

### Consequences of the withdrawal of growth factors

Cytotoxic stress attenuated RTK signaling (Fig. 5) and interfered with lipid metabolism (Figs. 2–4) and survival (Supplementary Figs. 1 and 2). We expected comparable effects from the withdrawal of RTK ligands, i.e., insulin and growth factors. In fact, serum deprivation triggered apoptotic and necrotic cell death (Supplementary Figs. 1 and 2), decreased cellular PI(3,4,5)P3 levels (Fig. 6a), lowered the temporarily upregulated phosphorylation of Akt by trend (Fig. 6b–d and Supplementary Fig. 20a), contributing to the induction of cell death (Supplementary Fig. 26b), and reduced the nuclear availability of the lipogenic transcription factor SREBP1 (Fig. 7e). As a consequence, total

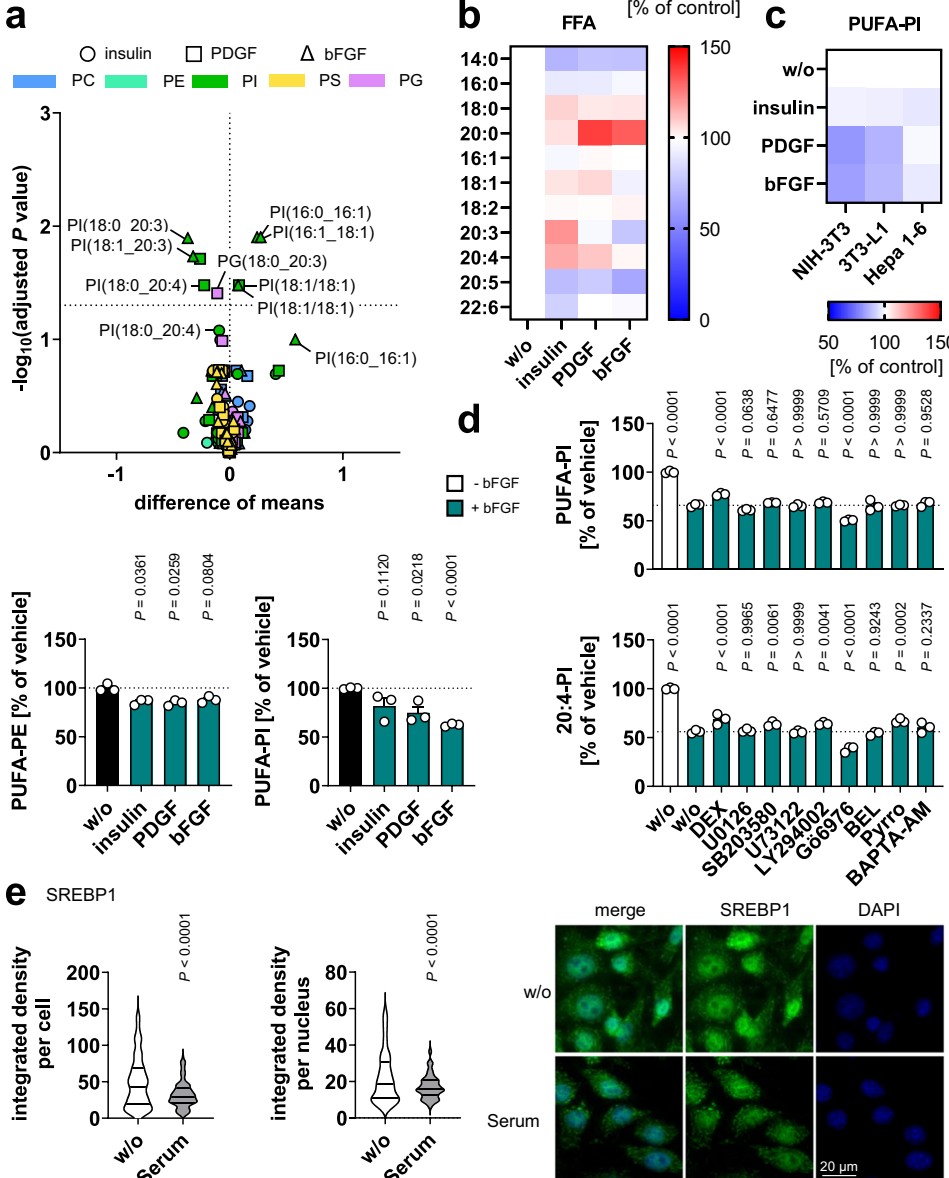

**Fig. 7 | RTK ligands lower the PUFA ratio in phospholipids. a**, **b** Swiss-3T3 fibroblasts were stimulated with vehicle, insulin (1 μM), PDGF (25 ng/ml), or bFGF (10 ng/ml) for 24 h. **a** Volcano plot showing the mean difference of relative phospholipid intensities and the negative $\log_{10}$(adjusted $P$ value) in insulin-, PDGF- or bFGF-treated cells. Adjusted $P$ values given vs. vehicle control; two-tailed multiple unpaired student $t$-tests from log-transformed data with correction for multiple comparisons (false discovery rate 5%). Bar charts show changes in the cellular proportion of PUFA-PE and PUFA-PI relative to vehicle control. **b** Heatmap showing percentage changes in the cellular FFA proportion relative to vehicle control. **c** NIH-3T3 fibroblasts, 3T3-L1 preadipocytes, and Hepa 1-6 hepatocarcinoma cells were stimulated with insulin, PDGF, or bFGF for 24 h. The heatmap shows percentage changes in the cellular proportion of PUFA-PI relative to vehicle control. **d** Swiss-3T3 fibroblasts were pre-incubated with vehicle or bFGF in combination with the glucocorticoid dexamethasone (1 μM), the MEK inhibitor U0126 (10 μM), the p38

MAPK inhibitor SB203580 (10 μM), the phospholipase C inhibitor U73122 (3 μM), the PI3K inhibitor LY294002 (10 μM), the PKC inhibitor Gö6976 (0.5 μM), the Ca²⁺-independent PLA$_2$γ inhibitor (R)-bromoenol lactone (BEL, 10 μM), the cPLA$_2$ inhibitor pyrrophenone (2 μM), and the Ca²⁺-chelator BAPTA-AM (15 μM) for 30 min (except dexamethasone for 60 min). **e** Immunofluorescence images of NIH-3T3 cells serum-starved for 48 are stained for SREBP1 and nuclei (DAPI). Images are representative of ≥ 255 single cells analyzed in n = 3 independent experiments (scale bar: 20 μm). Violin plots show the integrated density per cell or per nucleus. Mean (**a**–**c**), mean + s.e.m. and single data (**a**, **d**) or median and quartiles (**e**) from n = 3 independent experiments. $P$ values vs. vehicle control; two-tailed unpaired student $t$-test (**e**) of log-transformed data (**a**), repeated measures one-way ANOVA of log-transformed data (**d**) + Dunnett´s post hoc test. The exact $P$ values of panels **a** (for PUFA-PI), **d** and **e** are listed in the source data, which are provided with this paper.

phospholipid levels were decreased (Fig. 1a and Supplementary Fig. 3b), while total FFA levels (Fig. 1b and Supplementary Fig. 3b) and the PUFA ratio in PC and PE increased strongly (Fig. 1f, Supplementary Figs. 4, 5a, and 6), and the proportion of free PUFAs remained constant[46].

Mechanistically, serum depletion suppressed the expression of the SREBP1 target genes ACC (Fig. 3b, c), SCD1[46], and, by trend, FAS

(Fig. 3b, d), reduced the cellular energy status (Fig. 3h), increased the inhibitory phosphorylation of ACC by rapid AMPK activation (at 1 h; Fig. 3e), and induced a switch from de novo fatty acid biosynthesis (Fig. 2a) to degradation (Fig. 2b, c). Interestingly, serum depletion failed to decrease malonyl-CoA levels (Fig. 3f) despite significantly reduced ACC expression (Fig. 3c), which we ascribe to the increased availability of the substrate acetyl-CoA (Fig. 3g) and the low level of

inhibitory ACC phosphorylation at 48 h (Fig. 3e). Inhibitor studies further suggest that the decrease in fatty acid biosynthesis contributes to the increase in the PUFA-PC ratio (Fig. 2g and Supplementary Fig. 12a) and, like cytotoxic stressors, potentially promotes the induction of apoptotic PARP cleavage during serum deprivation (Supplementary Fig. 31). Together, insulin and growth factor withdrawal by serum depletion causes changes in fibroblast lipid metabolism and phospholipid composition comparable to small molecule cytotoxic stressors that interfere with RTK signaling.

Serum deprivation not only removes growth factors but also limits the availability of many other factors, including exogenous fatty acids. To investigate the role of serum fatty acids for preventing cytotoxic stress, we deprived cells of serum, but supplemented 18:1 to compensate for the lack of fatty acids in a serum-free culture medium, and studied the interference with RTK signaling at the proteome level. The consequences on stress-regulated proteins identified in RTK cascades were less severe for serum depletion than for treatment with VAL or MC but overall showed the same trend, with IGFBP6 and IDE levels increasing and IGF1R, NCK1, and FRS2 levels decreasing (Fig. 5c, d, h). The addition of 18:1 to the serum-free culture medium resulted in a marked enrichment of FGFR1 and NCK1 compared to the serum-free control but did not substantially enrich other RTK signaling proteins (Fig. 5c–h).

Proteomic changes in glucose and lipid metabolism differ between serum-starved and VAL- or MC-treated cells (Fig. 4a, b and Supplementary Figs. 16–19), despite similar adaptations in RTK signaling (Fig. 5) and PUFA metabolism (Figs. 1–3). In particular, serum depletion did not substantially reduce the availability of enzymes involved in fatty acid, phospholipid, and glucose metabolism, but rather induced the expression of proteins that convert fatty acids to energy, i.e., mitochondrial β-oxidation enzymes (e.g., HADH) (Fig. 4a, b and Supplementary Fig. 18), the TCA cycle (e.g., SUCLA2, SDHB), pyruvate decarboxylation (e.g., PDHA1, DLAT), and oxidative phosphorylation (e.g., NDUFC2, ATP5F1D, MTCO2) (Fig. 4a, b, Supplementary Fig. 16, and Supplementary Data 1). This switch from anabolic to catabolic pathways is also evident in the decrease of HK2 relative to HK1 (Fig. 4g, h and Supplementary Fig. 16), which is considered to favor glycolysis[61]. Serum starvation largely mediates these effects independent of fatty acid deprivation (Fig. 5c–h, Supplementary Figs. 15–18, and 19a).

Mechanistically, serum depletion strongly induced the expression of ACSL4 (Supplementary Fig. 32a), which has a preference for PUFAs[73], drives PUFA incorporation into membrane phospholipids, and promotes cell death by membrane peroxidation[74]. The induction of ACSL4 in serum-starved cells is dependent on low fatty acid levels and was blocked by excess 18:1 (Supplementary Fig. 32a). Serum starvation also moderately increased the availability of other ACSL isoenzymes (ACSL3, ACSL5) (Fig. 4c, Supplementary Fig. 18, and Supplementary Fig. 32b), with the exception of ACSL1, which prefers SFAs and MUFAs[73] and was weakly repressed (Supplementary Fig. 18 and Supplementary Fig. 32c). Note that fatty acids are not present in serum-free medium and are likely derived from intracellular sources during serum starvation, e.g., from autophagy, which is indeed modulated under our experimental conditions[46]. In summary, cytotoxic stress and serum deprivation trigger mechanisms that converge in i) impaired RTK signaling, ii) diminished de novo fatty acid biosynthesis, and iii) a maintained or enhanced capacity to incorporate PUFAs into membranes.

## Consequences for the oxidative component of cell death

There is increasing evidence that many cytotoxic stimuli that were previously described to induce apoptosis also trigger ferroptosis through poorly defined mechanisms[75]. This emerging link between cell death programs led us to speculate that the apoptotic increase in phospholipid PUFA ratios found here increases sensitivity to (ferroptotic) membrane peroxidation. In support of this hypothesis,

inhibition of fatty acid biosynthesis at the level of SCD1 raises the phospholipid PUFA ratio[46] and sensitizes cancer cells to GPX4 inhibition[30,36], as confirmed here for human MDA-MB-231 breast cancer cells (Supplementary Fig. 33a). We further confirmed that cells are more vulnerable to membrane peroxidation, ferroptotic cell death, and the sensitization to it when the membrane PUFA ratio is increased by supplementation of 20:4 (Supplementary Fig. 33b) and less vulnerable when the membrane PUFA ratio is decreased by deletion of LPLAT12/LPCAT3 (Fig. 8a, Supplementary Fig. 33c–e), consistent with previous studies[33,76–80]. Accordingly, we supplemented fibroblasts with 18:1 to block the cytotoxic accumulation of PUFAs in phospholipids (Supplementary Fig. 13a) and found that the fraction of cells undergoing necrosis (both with and without an apoptotic component) was substantially reduced, whereas the number of early apoptotic cells was hardly affected[46].

Motivated by these findings, we first systematically investigated whether the cytotoxic effect of different cell death inducers triggers ferroptosis per se, i.e., is attenuated by the selective ferroptosis inhibitor ferrostatin-1 (Fer-1), which acts as a lipophilic radical trap[81,82]. We found that only cell death induced by serum deprivation, and less so by VAL or MC, has a (per)oxidative component, as suggested by the ability of Fer-1 to reduce the decrease in fibroblast viability, measured as cellular dehydrogenase activity (Fig. 8b). Note that cells under these conditions activate AMPK (Fig. 3e), which has been implicated in ferroptosis resistance as a master regulator of cellular energy homeostasis[83]. Membrane peroxidation measured as 4-hydroxynonenal (4-HNE) production (Fig. 8c, d) and the level of oxidized arachidonoyl-PE (Fig. 8e, f) were greatly increased in VAL-treated cells, even more so in combination with RSL3, which is supposed to navigate cells towards ferroptosis[77]. Second, we show that cytotoxic stress, especially induced by VAL, MC, and CHX and less by ETO and TPG, sensitizes fibroblasts to ferroptosis (measured as loss of plasma membrane integrity; Fig. 8g and Supplementary Fig. 34a) with sublethal concentrations of the GPX4 inhibitor RSL3. Together, early apoptotic cells i) raise the membrane PUFA ratio (Fig. 1f, h), ii) seem to become more susceptible to ferroptosis (Fig. 8g and Supplementary Fig. 34a), and iii) under certain cytotoxic conditions, partially undergo cell death by lipid peroxidation (Fig. 8c–f).

Our hypothesis of a sequential induction of apoptosis that facilitates oxidative membrane damage is based on the assumption that apoptosis induction and membrane peroxidation actually occur in the same cell. To confirm this, we treated fibroblasts with VAL or the BCL-2 inhibitor ABT-737 (a selective apoptosis inducer) for 48 h and determined the levels and distribution of cleaved caspase-3 (an executioner caspase in apoptosis) and the lipid peroxidation product 4-HNE. In many cells, VAL and ABT-737 caused both caspase-3 cleavage and 4-HNE production, but there were also cells with elevated levels of 4-HNE only (Fig. 8g and Supplementary Fig. 34b). The ferroptosis inducer RSL3, used as a control, induced exclusively membrane peroxidation within 3 h, as expected (Fig. 8g).

Given the important role of (impaired) RTK signaling and fatty acid biosynthesis in mediating the cytotoxic switch in the membrane PUFA composition, we expected these pathways to be associated with ferroptosis sensitivity. Indeed, the expression of IGF1R, IGF2R, FGFR2-4, and de novo fatty acid biosynthetic enzymes (ACC1, FASN) correlates with ferroptosis resistance across 824 primary, adherent cancer cell lines (Fig. 8i and Supplementary Fig. 35a, b). The expression of other RTKs (FGFR1 and PDGFRA/B) is instead associated with ferroptosis susceptibility (Supplementary Fig. 35a), possibly due to a multi-faceted regulation of redox signaling[84]. Thus, the stress-induced upregulation of PDGFRA in fibroblasts (Fig. 5a–c) might further add to membrane peroxidation, whereas the depletion of FGFR1 (Fig. 5c, e) seems to be compensatory.

Whether an interference with fatty acid biosynthesis has the opposite effect and enhances cell death was investigated using the

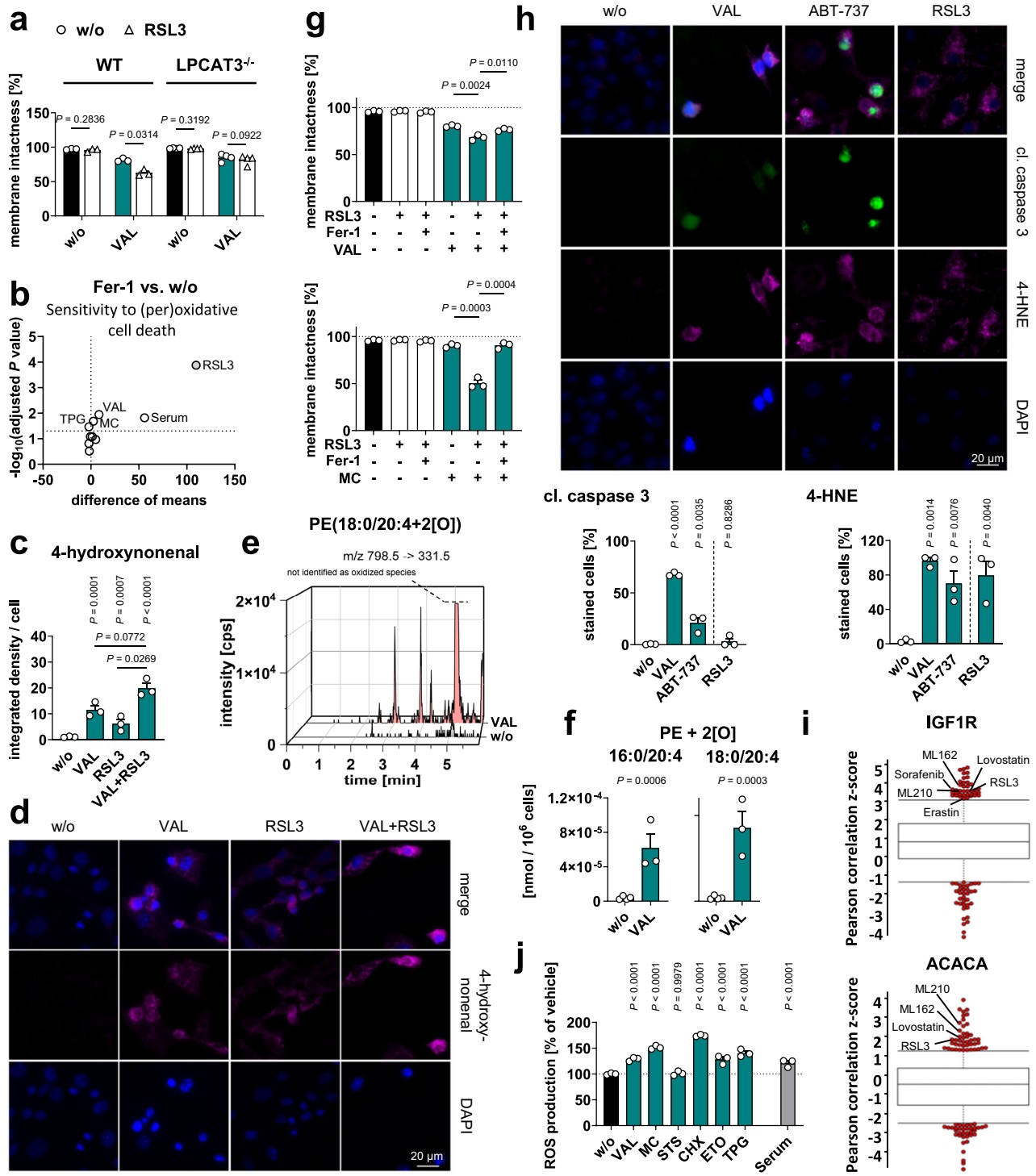

**Fig. 8 | Cytotoxic stress sensitizes cells towards (per)oxidative cell death.**
**a** Changes in membrane intactness (trypan blue exclusion) in VAL (10 μM, 48 h)-treated MDA-MB-231 cells (WT and LPCAT3$^{-/-}$). **b** NIH-3T3 fibroblasts were cultured (48 h) under cytotoxic stress conditions with or without ferrostatin-1 (Fer-1, 3 μM). Volcano plot showing the metabolic activity (MTT assay, % vehicle control) mean difference and negative log$_{10}$(adjusted $P$ value) of Fer-1-treated cells vs. respective treatments without Fer-1 (w/o); two-tailed multiple paired student $t$-tests with correction for multiple comparisons (false discovery rate 5%). **c, d, h** NIH-3T3 fibroblasts were pre-treated (45 h) with vehicle, VAL (10 μM) or ABT-737 (10 μM) and then with RSL3 (5 μM, 3 h). Immunofluorescence images (representative of ≥ 25 single cells, $n$ = 3, scale bar: 20 μm) stained for 4-HNE (**d, h**), cleaved (cl.) caspase 3 (**h**) and nuclei (DAPI, **d, h**) and integrated density/cell (**c**, upper panel) or % stained cells (**h**, lower panels). **e, f** PE hydroperoxides of fibroblasts treated with vehicle or VAL (10 μM, 48 h) were analyzed by UPLC-MS/MS. **g** Changes in membrane

intactness (trypan blue exclusion) of fibroblasts treated with vehicle, VAL (10 μM), MC (10 μM), RSL3 (0.05 μM) and/or Fer−1 (3 μM) for 48 h. **i** Pearson correlation Z-scores for each compound inform about the correlation of cancer cell resistance with IGF1R and ACACA (ACC1) expression (824 cancer cell lines; 481 small molecules). Data are available from the Cancer Therapeutics Response Portal[161−163] and are presented in box plots (interquartile multipliers: 0.75 [IGF1R], 0.5 [ACACA]). **j** Fibroblasts were cultured under cytotoxic stress conditions for 48 h before cellular ROS levels (normalized to average cell numbers) were determined. Mean (**b**) or mean + s.e.m. and single data (**a, c, f–h, j**) from $n$ = 3-4 independent experiments. Two-tailed paired or unpaired student $t$-test of log-transformed data or repeated measures one-way ANOVA of log-transformed data + Dunnett´s post hoc test. The exact number of experiments, details on statistical tests, and exact $P$ values (**c, h**, and **j**) are listed in the source data, which are provided with this paper.

selective ACC inhibitor TOFA, which induced PARP cleavage after 48 h of treatment (Supplementary Fig. 31) and specifically elevated the fraction of necrotic cells within 72 h (Supplementary Fig. 27). To assess the responsiveness of cells with high and low ACC activity to cytotoxic stimuli, we challenged vehicle- and TOFA-treated fibroblasts and determined the initiation of cell death by monitoring PARP cleavage (Supplementary Fig. 31). Cytotoxic treatment further increased the levels of cleaved PARP within 48 h of the establishment of pro-apoptotic conditions by TOFA (Supplementary Fig. 31). From 72 h onwards, TOFA also increased the number of necrotic (including late apoptotic) but not early apoptotic cells, as exemplarily shown for co-treatment with VAL (Supplementary Fig. 27). These findings imply that impaired fatty acid biosynthesis under cytotoxic stress, which increases the PUFA ratio of phospholipids, promotes necrotic/late apoptotic processes that compromise membrane integrity at later stages of cell death.

Our study provides strong evidence that the increased proportion of PUFA-containing phospholipids under cytotoxic stress sensitizes to oxidative, necrotic cell death. However, why the lethal activity of some cell death inducers benefits from this metabolic switch in the phospholipid PUFA composition even in the absence of an additional ferroptosis inducers and others do not is not fully understood. Our data rather exclude that the different cytotoxic efficacy is based on global oxidative stress, because the accumulation of reactive oxygen species (ROS) is not limited to those stressors, i.e., VAL, MC and serum depletion (Fig. 8j), that show the strongest (per)oxidative component in cell death (Fig. 8b).

NADH and NADPH are essential cofactors for various biological processes that contribute to redox balance and have both death-inducing and death-suppressing functions[85]. We used fluorescence lifetime imaging microscopy (FLIM) to investigate the effect of VAL on the total cellular fluorescence in the spectral range from 426 to 490 nm, which is representative of cellular NAD(P)H status[86]. In VAL-treated fibroblasts, NAD(P)H signals substantially increased in intensity (Fig. 9a), especially in the (peri)nuclear region (Fig. 9b). Our data indicate that VAL, possibly as a consequence of the metabolic switch from NADPH-consuming fatty acid biosynthesis to NADH-generating β-oxidation, increases the levels of NADH and/or NADPH (Fig. 9a, b).

To gain further insight into the regulation of redox metabolism by cytotoxic stress that not only sensitizes to ferroptosis but itself has a ferroptotic component, we compared the quantitative proteome of VAL- and MC-treated fibroblasts and searched for proteins that were regulated in the same direction by both cytotoxic stressors (Supplementary Fig. 36a). The focus was on overarching changes in the enzymatic and regulatory machinery that orchestrates the balance between redox homeostasis and membrane peroxidation. We explored the consequences of insulin/growth factor withdrawal (serum depletion) on this set of proteins, investigated whether inhibition of fatty acid biosynthesis (via SCD1) mimics their regulation, and addressed the role of fatty acids (18:1) and the stress-regulatory lipokine PI(18:1/18:1) in this context (Supplementary Fig. 36a–d).

Cytotoxic stress by VAL and MC seems to shift the balance from redox homeostasis to oxidative cell death by i) favoring ERO1-like protein alpha (ERO1a)-dependent hydrogen peroxide generation (Fig. 9c–e), ii) upregulating the iron-mobilizing heme oxygenase (HMOX)1 (Fig. 9c, d, f), iii) depleting cells of protective enzymes against oxidative stress such as superoxide dismutase (SOD)2 (Fig. 9c, d, g), and iv) suppressing the mevalonate pathway (Supplementary Fig. 37), among others by depletion of 3-hydroxy-3-methyl-glutaryl-CoA synthase 1 (HMGCS1) (Fig. 4a, b and Supplementary Fig. 19a). The latter is essential for the biosynthesis of the membrane-protective radical-traps ubiquinol[9] and 7-dehydrocholesterol[87]. Instead, serum depletion is associated with restrictions in glutathione biosynthesis, among others due to the depletion of glutamate-cysteine ligase modifier subunit (GCLM) (Fig. 9h and Supplementary Fig. 36a).

In parallel, counterregulatory mechanisms are triggered (Supplementary Fig. 19 and Supplementary Fig. 36a), which aim to keep membrane peroxidation and its deleterious consequences at bay. Whether the latter is successful, depends on the nature of the cytotoxic stressor (Fig. 8b). Supplementary Note 4 describes in more detail the cell death-induced changes of the fibroblast redox proteome and discusses the link to fatty acid metabolism and SCD1/PI(18:1/18:1) signaling.

## Discussion

Programmed cell death is induced by cytotoxic stress, counteracted by mitogenic signaling, and associated with changes in (lipid) metabolism and cellular lipid composition[1,2]. Functional implications are emerging and point towards a prominent role of phospholipid metabolism in cell cycle and cell death control[26,28,71]. Using comparative multiomics, we aimed to uncover overarching (phospho)lipid-regulatory mechanisms that determine cell fate. Initially, we identified the SCD1-derived lipokine PI(18:1/18:1), which reduces (adaptive) stress responses, such as ER stress, ERAD, autophagy, and programmed cell death[46,72,88,89]. Here, we report on a second set of bioactive phospholipids, i.e., PUFA-containing phospholipids, that are consistently regulated by mechanistically distinct cytotoxic stressors. The proportion of PUFA-containing PC preferentially increases throughout subcellular compartments during the initiation of cell death because SFA and MUFA biosynthesis is suppressed, while the machinery for incorporating exogenous fatty acids (including PUFAs) into membranes is partially maintained (Fig. 10). The incorporation of PUFAs into PC is mediated by LPCAT3/LPLAT12 and associated with altered expression of ACSL isoenzymes. This switch from MUFA- to PUFA-containing phospholipids seems to rely on impaired growth factor receptor-PI3K signaling, which drives anabolic lipid metabolism[21,31,32] and is, under some stress conditions, such as serum deprivation, mediated by suppressed Akt-SREBP1 activation. The stress-induced shift from MUFA- to PUFA-containing PC (and other phospholipids) adds a membrane-oxidative (and potentially immunomodulatory) component to apoptotic cell death (when sustained by oxidative stress) and contributes to ferroptosis sensitization at the intersection of the two cell death programs. Indeed, higher proportions of PUFAs in phospholipids are well known to increase the susceptibility of membranes to peroxidation[33], which facilitates induction of cell death[34,90], yields immunogenic factors[91,92], and produces (precursors of) lipid mediators[93]. MUFAs instead protect against oxidative cell death[35,36]. Our findings are consistent with recent in situ Raman spectroscopy studies proposing a crosstalk between ferroptosis and apoptosis[94]. We also noted specific effects of individual cytotoxic stressors on the phospholipid profile (Supplementary Note 5), as expected from the heterogeneity and the complex nature of cell death, but these were excluded from further investigation.

Our quantitative proteomics studies indicate that cytotoxic stress attenuates RTK/PI3K/Akt signaling by repressing RTKs (IGF1R, FGFR1) and downstream signaling components (e.g., CDH2, RPS6KA5, GSK-3β), while upregulating inhibitory factors (IGFBP6). Indeed, inhibition of IGF1R, PI3K, Akt, or SREBP1 substantially suppresses lipogenic gene expression[28,95,96] and increases the PUFA-PC ratio. This cascade is implicated in a myriad of (patho)physiological processes and has recently been identified, together with the target gene SCD1, as a critical regulator of apoptosis and ferroptosis sensitivity[30,97–99]. Accordingly, i) Akt activation by IGF1R suppresses ferroptosis by increasing creatine kinase B binding to GPX4[100], ii) tyrosine kinase inhibition by lorlatinib interferes with SCD expression and sensitizes melanoma to ferroptosis[101], and iii) initiation of cell death in murine macrophages by a α-tocopherol metabolite suppresses SREBP1 proteolytic maturation and SCD1 expression[102]. Selective apoptosis induction by pan-BCL-2 inhibition mimics the effect of cytotoxic stress on the PUFA-PC ratio, but the question remains by what seemingly caspase-independent mechanism apoptosis regulates the RTK-PI3K-axis and thus lipogenesis.

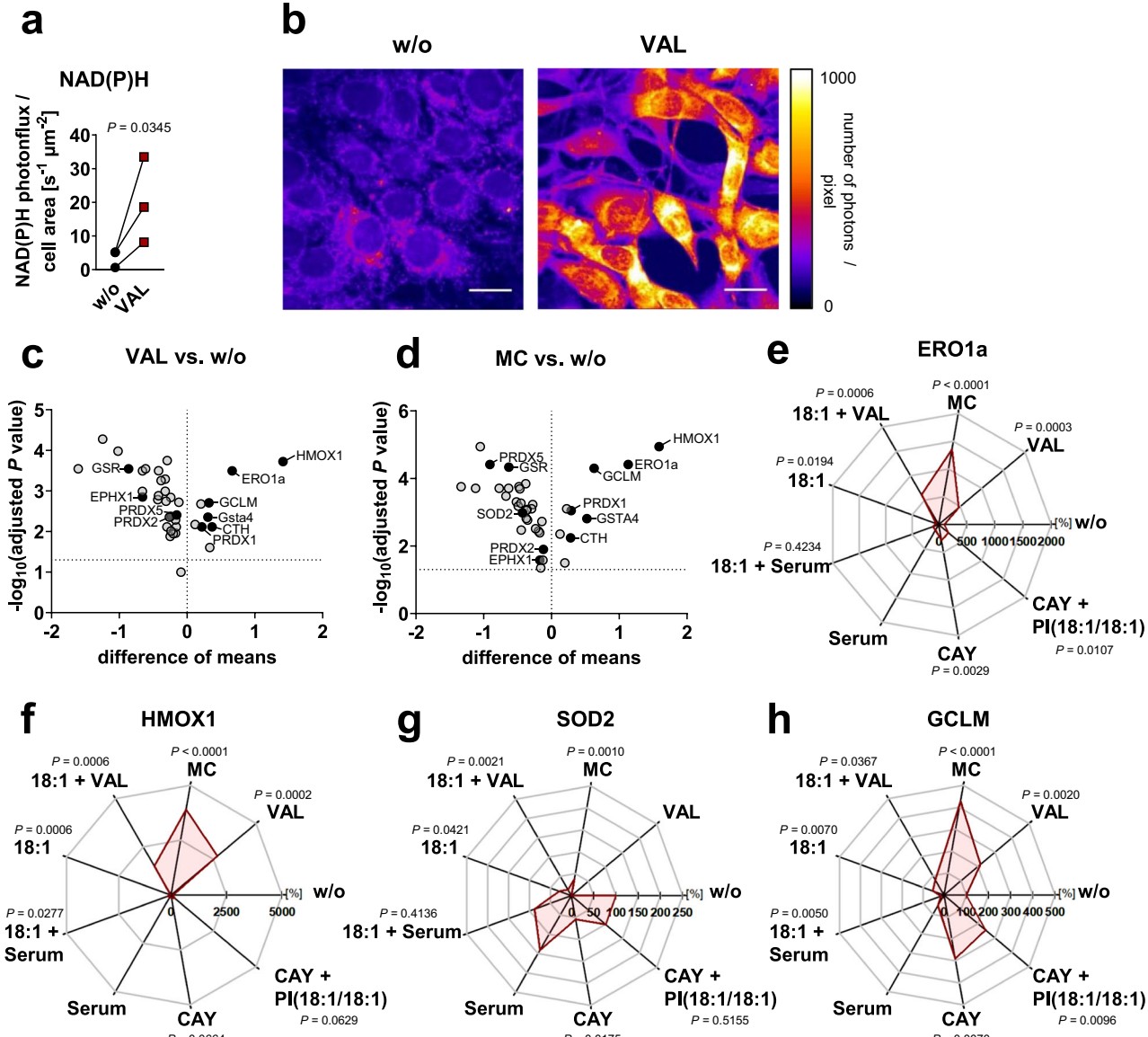

**Fig. 9 | Aberrant redox homeostasis under cytotoxic stress. a, b** Fibroblasts were treated with vehicle or VAL (10 µM) for 48 h. **a** Photonflux [$s^{-1}$ µm$^{-2}$] of NAD(P)H normalized to the cellular area as determined by FLIM. Paired data from $n = 3$ independent experiments. $P$ values vs. vehicle control; two-tailed paired student $t$-test of log-transformed data. **b** Subcellular distribution of NAD(P)H based on the total photon emission and normalized to equal acquisition time; scale bar: 20 µm. FLIM images are representative for 50 individual cells from three independent experiments. **c–h** Fibroblasts were treated with vehicle, VAL (10 µM), MC (10 µM), 18:1 (100 µM), VAL (10 µM) plus 18:1 (100 µM), CAY10566 (CAY, 3 µM), or CAY (3 µM) plus PI(18:1/18:1) (50 µM), or were serum starved (serum) in presence or absence of 18:1 (100 µM) for 48 h. Quantitative proteomics focused on proteins that contribute to redox homeostasis (and are either related to ROS regulation, antioxidant function, glutathione metabolism, iron metabolism, Nrf2 signaling, or ferroptosis) and are up- or downregulated by VAL and MC in the same direction ($\geq 20\%$). **c, d** Volcano plots highlight proteins that are regulated by VAL (**c**) or MC (**d**) compared to vehicle control. Comparisons of the indicated treatment groups show the difference of mean absolute intensities of log$_{10}$ data and the negative log$_{10}$(adjusted $P$ value). **e–h** Radar charts indicate the percentage changes of cellular ERO1a (**e**), HMOX1 (**f**), SOD2 (**g**), and GCLM (**h**) relative to vehicle control. Paired data (**a**) or mean from $n = 3$ (except $n = 2$ for serum) independent experiments. $P$ values given vs. vehicle control (**a**) or adjusted $P$ values given vs. vehicle control (**c–h**); two-tailed paired student $t$-tests from log-transformed data (**a**) or two-tailed multiple unpaired student $t$-tests from log-transformed data with correction for multiple comparisons using a two-stage linear step-up procedure by Benjamini, Krieger, and Yekutieli (false discovery rate 5%) (**c–h**). Exact $P$ values of panels **e**, **f** and **h** are listed in the source data. Source data are provided with this paper.

In support of RTK signaling being diminished by cytotoxic stress, cellular levels of PI(3,4,5)P$_3$ were reduced and, under some conditions, additionally Akt phosphorylation was inhibited and AMPK was activated, both of which interfere with SREBP1c maturation and transcriptional activity[97]. Small molecule inhibitor/activator studies also suggest a role in this process for the tumor suppressor and transcription factor p53, which promotes apoptosis and regulates lipid metabolism, including by interfering with SREBP1 maturation and transcriptional activity toward SCD1 expression[67,103]. Interestingly, the scaffold protein FAM120A/C9orf10, which supports the activation of PI3K by Src family members[104], is also strongly repressed by cytotoxic stressors and SCD1 inhibition, which might contribute to the decrease in PI(3,4,5)P$_3$ production and thus repression of lipogenic enzymes. Despite an overall lower availability, mitogenic and stress-activated kinases other than PI3K (and Akt for specific stress conditions) rather show a trend towards increased activating phosphorylation (although not consistently under the cytotoxic conditions used here), which we ascribe to the decrease in PI(18:1/18:1) that reduces PP2A levels[46].

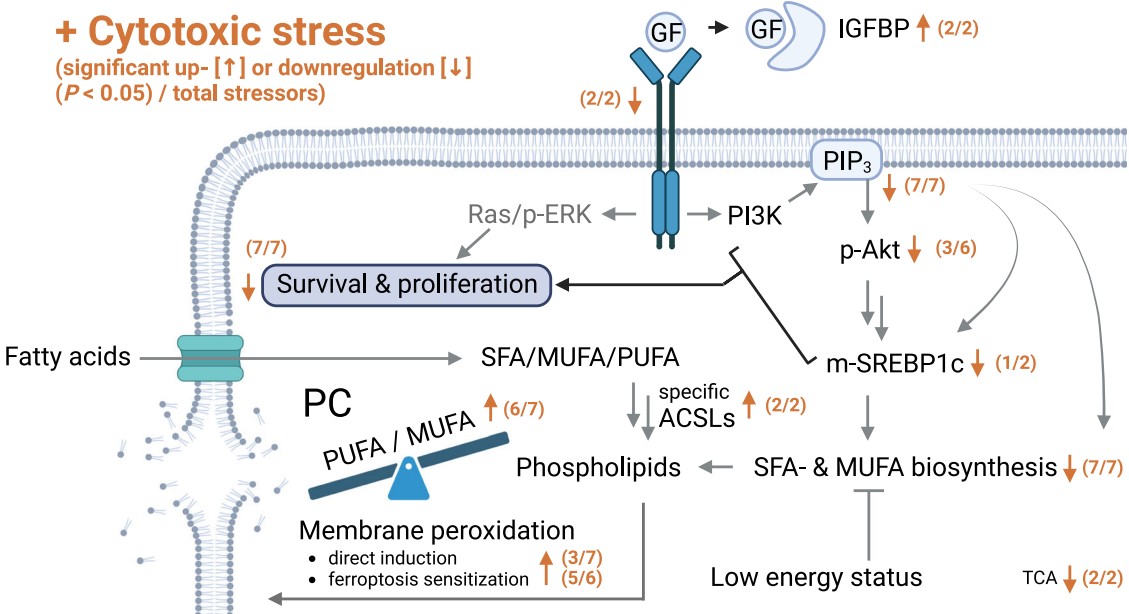

**Fig. 10 | Cytotoxic stress attenuates growth factor signaling and sensitizes membranes to oxidative damage.** Growth factors activate RTKs, which initiate phosphorylation cascades that activate ERK1/2 and trigger PI3K signaling, enhancing cell survival, growth, and proliferation, while inhibiting apoptosis, in part by promoting energy and glucose metabolism via Akt and SREBP1 and suppressing AMPK activation. Subsequent processing, nuclear translocation, and post-translational modification of the transcription factor SREBP1c drives lipogenic gene expression that sustains SFA- and MUFA-CoA biosynthesis. Free SFAs, MUFAs, and PUFAs are taken up or released by cells, converted into CoA esters by ACSL isoenzymes, and incorporated into phospholipids along with de novo synthesized acyl-CoAs. By adjusting the ratio of PUFAs to MUFAs in phospholipids, this machinery defines the susceptibility of membranes towards peroxidation. The latter is initiated when oxidative stress is induced in parallel, e.g., due to aberrant iron and/or redox metabolism, or by additional challenges (e.g., GPX4 inhibition). The first number in the brackets indicates the number of cytotoxic stressors showing significant effects ($P < 0.05$) in Figs. 1d–f, 2a, Fig. 3b–h, Fig. 4a–c, Fig. 5, Fig. 6a–d, Fig. 7e, Fig. 8b, g, Supplementary Figs. 1c, d, 2, 21b, 32, and 34a for at least one of the measured parameters and the second number states the total number of stressors investigated in the experiments. Created in BioRender[167].

Impaired RTK signaling thus provides an explanation for decreased SFA and MUFA biosynthesis, which is causative for the shift towards PUFA-containing phospholipids and is associated with (oxidative) membrane stress[30]. The latter depends in part on the depletion of the stress-buffering lipokine PI(18:1/18:1) as well as other SCD1-derived MUFA metabolites[46,72] and is expected to cause insulin resistance via stress-activated protein kinases, such as the PI(18:1/18:1)-regulated p38 MAPK[46,88,105]. We further speculate that the inverse relationship between SFA/MUFA biosynthesis and the phospholipid PUFA ratio contributes to the phenotypes of ACC isoenzyme-deficient mice, as discussed in Supplementary Note 6.

RTK ligands such as insulin and growth factors decrease the availability of free and phospholipid-incorporated PUFAs through multiple pathways, including PI3K/Akt and potentially MEK/ERK signaling, whereas growth factor withdrawal by serum deprivation had the opposite effect. As expected from this heterogeneity and detailed in Supplementary Note 7, the strategies by which stressed cells decrease SFA and MUFA biosynthesis differ between cytotoxic settings.

RTK signaling and MUFA biosynthesis not only keep apoptosis in check but are also central to ferroptosis[76]. While PUFA incorporation into cellular membranes increases ferroptosis sensitivity[106], MUFAs cause resistance through poorly understood mechanisms[35]. Thus, inhibition/silencing of PI3K, Akt, mTORC1, SREBP1, or SCD1 sensitizes (cancer) cells to ferroptosis[27,30,36,107] and activating mutations in *PIK3CA*, which encodes for the catalytic subunit of PI3K, cause ferroptosis resistance in cancer cells via the SREBP1-SCD1-axis[27,30]. Inactivating mutations or deletion of the PI(3,4,5)P₃-hydrolyzing phosphatase and tensin homolog (PTEN) have a likewise effect, whereas interference with fatty acid biosynthesis by activation of AMPK suppresses ferroptosis, among others, in mouse embryonic fibroblasts[83,108,109]. Consistent with these studies and the causal link proposed here between impaired growth factor signaling, fatty acid metabolism, and oxidative cell death, we show that specific RTKs and enzymes in fatty acid biosynthesis correlate with ferroptosis resistance across 824 cancer cell lines.

Activation of AMPK (as observed here under distinct cytotoxic conditions) simultaneously inhibits fatty acid biosynthesis and activates β-oxidation by phosphorylating ACC1 and ACC2[27], which lowers the availability of FFAs and phospholipids, largely indiscriminate of the fatty acid composition (SFA, MUFA, PUFA)[83]. AMPK deletion has the opposite effect. In addition, mTORC1 has recently been shown to coordinate the cellular lipid composition by lysosome-dependent hydrolysis of phospholipids, with the free fatty acids being then available for β-oxidation and TAG biosynthesis[110]. Proposed cell-protective mechanisms of AMPK-ACC1/2 relate to decreased levels of malonyl-CoA, which is required to elongate PUFAs[83], block β-oxidation[27], and possibly allow palmitoylation of an unknown ferroptosis regulator[27]. Interestingly, both ACC and SCD1 inhibition increase the PUFA composition of phospholipids in NIH-3T3 fibroblasts. However, while SCD1 inhibition sensitizes cells to ferroptosis[36,111], interference with ACC and FAS causes ferroptosis resistance in MEFs[83,108]. In contrast, ACC1 (ACACA) and FAS (FASN) seem to be protective in cancer cells: their expression correlates with resistance to different small molecule ferroptosis inducers (including ML210, ML162, RSL3, and erastin) in primary adherent cancer cell lines, as expected from the function of fatty acid biosynthetic enzymes in decreasing the phospholipid PUFA proportion described here. Our findings are consistent with a recent study by Bartolacci et al., who reported that selective inhibition of FAS favors incorporation of PUFAs into phospholipids via the remodeling pathway in KRAS-mutant lung cancer, thereby lowering the threshold for ferroptosis[29].

We also asked how exogenous or endogenously released PUFAs are incorporated into membrane phospholipids in stressed cells. Under serum starvation, the PUFA-CoA-specific isoenzyme ACSL4[73] is upregulated, while other lipid metabolic enzymes are downregulated. For small molecule cytotoxic stressors, such as VAL or MC, as well as the interference with IGF1R, PI3K, or Akt, PUFA incorporation into fibroblast phospholipids is associated with an upregulation of ACSL5, which has a broad substrate specificity[59], while other ACSL isoenzymes are rather repressed. Knockdown studies suggest that ACSL5 preferentially activates and thus incorporates SFAs and MUFAs into PC in non-stressed cells, whereas this effect is mitigated in stressed cells. While this finding does not exclude that ACSL5 participates in PUFA incorporation into PC, it shows that the VAL-induced upregulation of ACSL5 does not cause (but rather compensates for) the increase in PUFA-PC ratio under cytotoxic stress. Interestingly, ACSL5 also interacts with ceramide synthases and contributes to the formation of acylceramides, thereby alleviating ceramide-induced apoptosis[112]. Note that ceramide activates PP2A[113], which has been proposed under our settings to suppress the induction of (adaptive) stress reactions (autophagy, ER stress, apoptosis) in response to the decline in SCD1-derived PI(18:1/18:1)[46] and that these stress-triggered pathways are associated with ferroptosis[114–116]. LPLAT12 accepts PUFA-CoA from ACSL isoenzymes, incorporates the fatty acyls into PC, PE, and PS[53], maintains this function under cytotoxic stress (shown here), and sensitizes cells to ferroptosis[27]. Whether the initiation of cell death actively regulates LPLAT12 protein levels remains to be determined. As described in Supplementary Note 8, substantial changes in the fatty acid composition were also observed in CLs, which are abundant in the inner mitochondrial membrane and are externalized early in apoptosis.

The PUFA to MUFA ratio in phospholipids determines the susceptibility of cells to membrane peroxidation and ferroptosis[27], as confirmed here for fibroblasts and breast cancer cells. Accordingly, cytotoxic stress and serum depletion sensitize cells to phospholipid (hydroper)oxidation and ferroptosis (induced by the GPX4 inhibitor RSL3) by increasing the proportion of PUFA-containing phospholipids (most consistently PUFA-PC). While the seven cytotoxic treatments used in this study most strongly and consistently increased the cellular PUFA-PC ratio, they also markedly elevated the ratio of PUFAs in either PE (ETO, CHX, Serum), PS (VAL, CHX), or PI (ETO, TPG, Serum, STS). These latter phospholipids, especially PE but also PC, PS, and PI, have been proposed to play a predominant role in ferroptosis execution, mainly because they are sensitive to peroxidation and their (per)oxidation is associated with ferroptosis induction[74,77,117–119]. Whether their oxidation products actually execute ferroptosis or whether other phospholipid classes, such as the abundant PC, are mainly responsible is poorly understood[120], as is the question of whether the phospholipids involved differ between systems and (sub-)cellular membranes. Notably, an increasing number of specific phospholipids are becoming known for their contribution to ferroptosis, including di-PUFA species[121] and ether phospholipids[122,123]. Overall, there is active research on which phospholipid classes drive ferroptosis under which conditions, and our study adds to the controversial discussion.

In addition, some cytotoxic stress conditions (such as VAL and MC) induce membrane peroxidation and ferroptosis per se, which cannot be explained by membrane polyunsaturation alone but additionally requires an aberrant redox balance. Proteomics studies suggest that these cytotoxic stressors with an intrinsic ferroptotic component promote redox stress in fibroblasts by inducing ERO1-like protein alpha, depleting enzymes of the antioxidant response, upregulating HMOX1, suppressing the mevalonate pathway, or limiting glutathione biosynthesis. Since PI3K/Akt inhibits autophagy, including ferritinophagy[124] and we have previously confirmed that autophagy is regulated under our cytotoxic conditions[46], we also consider it likely that cytotoxic stress increases ROS production by mobilizing iron[125].

As exemplarily shown for VAL-treated fibroblasts, stressed cells accumulate NADH and/or NADPH, possibly by switching from fatty acid biosynthesis (NADPH consumption) to degradation (NADH generation) and by channeling glucose via HK2 into the pentose phosphate pathway (NADPH generation). Whether NADH or NADPH accumulation is dominant remains speculative. While both NADPH and NADH are required to sustain the antioxidant response[126–128], NADPH may also contribute to ROS production[129–133].

In conclusion, we have identified an overriding, caspase-independent mechanism by which fibroblasts and diverse other cell types increase the proportion of PUFAs in phospholipids under cytotoxic stress, rendering apoptotic cells more susceptible to membrane peroxidation and ferroptosis (Fig. 10). Specifically, cytotoxic stress impairs growth factor signaling through the RTK-PI3K-axis, in some cases resulting in attenuated Akt and SREBP1 signaling. In consequence, SFA and MUFA biosynthesis is suppressed, which allows exogenous or released PUFAs to be efficiently incorporated into phospholipids, driven by the changes in substrate availability and enabled by ACSL isoenzymes together with LPLAT12/LPCAT3. This mechanism applies to various cytotoxic stress conditions as well as serum starvation, is located at the crossroads of apoptosis and ferroptosis, and represents a central piece in our understanding of how these cell death programs cooperate when oxidative stress is induced in parallel.

# Methods
## Materials
STS, SB203580, LY294002, U73122, pyrrophenone, and Gö6976 were purchased from Calbiochem (Darmstadt, Germany). VAL, CHX, ETO, CAY10566, GSK690693, TOFA, (1S,3 R)-RSL3, and Fer-1 were obtained from Cayman Chemical (Ann Arbor, MI), TPG from Enzo Life Sciences (Farmingdale, NY), and Q-VD-OPh, dexamethasone, and bromoenol lactone from Sigma-Aldrich (St. Louis, MO). ABT-737, BAY3827, fatostatin, ipatasertib, laduviglusib, MHY1485, palbociclib, picropodophyllin, pifithrin-β hydrobromide, selisistat, skepinone L and venetoclax were purchased from MedChemExpress (Monmouth Junction, NJ). Metformin was bought from TCI (Eschborn, Germany), arachidonic acid from Biomol (Hamburg, Germany), BAPTA-AM from Dojindo Molecular Technologies (Kumamoto, Japan), and U0126 from Cell Signaling (Beverly, MA). Soraphen A and MC were kindly provided by Dr. Rolf Müller (Saarland University, Germany) and Dr. Johann Jauch (Saarland University, Germany), respectively.

Cytotoxic compounds and selective inhibitors were dissolved in DMSO, and stored in the dark at −20 °C under argon, and freeze-thaw cycles were kept to a minimum. Phospholipids were obtained from Avanti Polar Lipids (Alabaster, AL), dissolved in chloroform, aliquoted, and stored protected from light at −80 °C under argon.

Rabbit anti-acetyl-CoA carboxylase (1:1000; # 3662), mouse anti-Akt (pan; 40D4; 1:1000; # 2920); rabbit anti-caspase 3 (1:1000; # 9662), rabbit anti-cleaved caspase 3 (Asp175; 1:400; #9661), mouse anti-cleaved PARP (Asp214; 7C9; 1:1000; # 9548), rabbit anti-ERK1/2 (p44/42 MAPK; 137F5; 1:1000; # 4695), rabbit anti-fatty acid synthase (1:1000; # 3189), rabbit anti-GAPDH (14C10; 1:1000; # 2118), mouse anti-GAPDH (D4C6R; 1:1000; # 97166), mouse anti-GSK-3β (3D10; 1:1000; # 9832), rabbit anti-HA-Tag (C29F4; 1:1000; # 3724), mouse anti-IκBα (L35A5; 1:1000; # 4814), mouse anti-mTOR (L27D4; 1:1000; # 4517), rabbit anti-p70 S6 kinase (49D7; 1:1000; # 2708), rabbit anti-phospho-acetyl-CoA carboxylase (Ser79; D7D11; 1:1000; # 11818), rabbit anti-phospho-Akt (Ser473; 1:500 - 1:1000; # 9271), rabbit anti-phospho-AMPKα (Thr172; 40H9; 1:1000; # 2535), mouse anti-phospho-ERK1/2 (p44/42 MAPK; Thr202/Tyr204; E10; 1:1000; # 9106), rabbit anti-phospho-GSK-3β (Ser9; D85E12; 1:1000; # 5558), rabbit anti-phospho-IκBα (Ser32; 14D4; 1:1000; # 2859), rabbit anti-phospho-MARCKS (Ser152/156; 1:1000; # 2741), rabbit anti-phospho-mTOR (Ser2448; D9C2; 1:1000; # 5536), mouse anti-phospho-p70 S6 kinase

(Thr389; 1A5; 1:1000; # 9206), rabbit anti-β-actin (13E5; 1:1000; # 4970), mouse anti-β-actin (8H10D10; 1:1000; # 3700), rabbit anti-SCD1 (M38; 1:500; #2438), and rabbit anti-β-tubulin (9F3; 1:1000; # 2128) were obtained from Cell Signaling Technology (Danvers, MA). Rabbit anti-Lamin B1 (1:1000; # ab16048), rabbit anti-MARCKS (1:1000; # ab51100), rabbit anti-SREBP1 (1:1000; #ab28481), and mouse anti-plasma membrane $Ca^{2+}$ ATPase (PMCA; 5F10; 1:1000; # ab2825) were from Abcam (Cambridge, UK), Rabbit anti-ACSL3 (1: 5000; #20710-1-AP), rabbit anti-ACSL5 (1: 2000; #15708-1-AP), and rabbit anti-ACSL4 (1:2500; #22401-1-AP) were from Proteintech (Manchester, UK). Rabbit anti-LPCAT3 (1: 2000; #A17604) was from ABclonal (Woburn, MA) and rabbit anti-calnexin (H-70; 1:1000; # sc-11397) was from Santa Cruz (Dallas, TX). Mouse anti-4-HNE (12F7; 1:50; #MA5-27570) was obtained from Thermo Fisher Scientific (Waltham, MA). Secondary antibodies were from LI-COR Biosciences (Lincoln, NE) or Thermo Fisher Scientific (Waltham, MA).

## Cell culture and treatment

Cells were from the German Collection of Microorganisms and Cell Cultures (DSMZ, Braunschweig, Germany), the American Type Culture Collection (ATCC, Manassas, VA), or the Japanese Collection of Research Bioresources Cell Bank (JCRB Cell Bank, Ibaraki, Japan), cultivated at 37 °C and 5% $CO_2$, and reseeded before reaching confluence.

Adherent cell lines: Adherent cells were detached using trypsin/EDTA (GE Healthcare, Munich, Germany) for subculturing. Mouse NIH-3T3 fibroblasts (# ACC 59, DSMZ, $5 \times 10^5/25\ cm^2$), mouse Swiss 3T3 fibroblasts (# CCL-92, ATCC; $1.7 \times 10^5/8.8\ cm^2$), mouse 3T3-L1 pre-adipocytes (#CL-173, ATCC; $1.7 \times 10^5/8.8\ cm^2$), and mouse Hepa 1-6 hepatocarcinoma cells (#RCB1638, RIKEN, Tokyo; $1.7 \times 10^5/8.8\ cm^2$) were grown in DMEM high glucose medium (DMEM, 4.5 g/l; Merck, Darmstadt, Germany or GE Healthcare) supplemented with heat-inactivated fetal calf serum (FCS, 10%; Sigma-Aldrich, Thermo Fisher Scientific, or Nacalai Tesque, Kyoto, Japan). Human HeLa cervical carcinoma cells (# ACC 57, DSMZ; $4 \times 10^5/25\ cm^2$) and human MDA-MB-231 triple-negative breast cancer cells ($5 \times 10^3$ cells/$0.33\ cm^2$) stably transfected with shRNA directed against GFP (MDA-MB-231 (shGFP)) were cultured in DMEM containing FCS (10%) and penicillin/streptomycin (100 U/ml and 100 μg/ml; Sigma-Aldrich). The latter cell line was a generous gift from Simone and Thomas Brabletz (University of Erlangen-Nürnberg, Germany) and was cultivated in presence of puromycin (1 μg/ml) for 7 days every month to maintain stable transfection[134]. Human MDA-MB-231 triple-negative breast cancer cells ($5 \times 10^3$ cells/$0.33\ cm^2$) with *LPCAT3* knockout were cultured in DMEM containing FCS (10%) and penicillin/streptomycin (100 U/ml and 100 μg/ml; Sigma-Aldrich) in presence of 6-thioguanine (6-TG, 10 μg/mL; Sigma-Aldrich) to maintain stable transfection.

RPMI 1640 (Sigma-Aldrich) supplemented with 10% FCS was used for the cultivation of human HT-29 colon adenocarcinoma cells (# HTB-38, ATCC), human HEK-293 embryonic kidney cells (# CRL-1573, ATCC), human A549 lung carcinoma cells (# CCL-185, ATCC), and human HepG2 hepatocarcinoma cells (# ACC 180, DSMZ), which were each seeded at $4 \times 10^5/25\ cm^2$. Human MCF-7 breast adenocarcinoma cells (# HTB-22, ATCC; $4 \times 10^5/25\ cm^2$) were grown in DMEM containing FCS (10%), penicillin/streptomycin (100 U/ml and 100 μg/ml), and 0.1% insulin (Sigma-Aldrich). HepaRG cells (# HPR101, Biopredic International, Rennes, France; $1.5 \times 10^5/cm^2$) were cultured in William's Medium E (Biochrom, Berlin, Germany) containing FCS (10%; GIBCO, Darmstadt, Germany), 5 μg/ml insulin, 2 mM glutamine (GIBCO), 0.5 μM hydrocortisone hemisuccinate (Sigma-Aldrich), 100 U/ml penicillin (GIBCO), and 100 μg/ml streptomycin (GIBCO). Human HUH-7 hepatocarcinoma cells (# JCRB0403, JCRB Cell Bank; 4 to $5 \times 10^5/25\ cm^2$) were cultured in DMEM supplemented with FCS (10%) and penicillin/streptomycin (100 U/ml and 100 μg/ml). Chemoresistance was induced in HUH-7 cells by gradually increasing the concentration of sorafenib (BAY 43 9006, Enzo Life Sciences GmbH,

Lörrach, Germany) in the culture medium to 10 μM[46,70]. Follow-up culture was conducted at 10 μM sorafenib to maintain resistance.

Suspension cell lines: Human MM6 acute monocytic leukemia cells (# ACC 124, DSMZ; $1.2 \times 10^6$ cells/4 ml) were cultivated in RPMI 1640 supplemented with FCS (10%), penicillin/streptomycin (100 U/ml and 100 μg/ml), L-glutamine (2 mM), oxalic acid (1 mM; Sigma-Aldrich), sodium pyruvate (1 mM; GE Healthcare), and non-essential amino acids (1×; Sigma-Aldrich).

Primary cells: HUVECs were isolated from human umbilical cord veins by Dr. Alexander Mosig (University Hospital Jena, Germany)[135]. Experiments were performed with HUVECs ($1.5 \times 10^5$ cells/$cm^2$) cultured in Endothelial Cell Medium (ECM) (Promocell, Heidelberg, Germany) up to passage 4. Human PMNL and monocytes ($1 \times 10^7$ cells/4 ml, each) from healthy adult volunteers on informed consent were freshly isolated from leukocyte concentrates that were provided by the Institute for Transfusion Medicine of the University Hospital Jena (Germany)[136]. Venous blood was collected in heparinized tubes (16 I.E. heparin/ml blood) from healthy adult (18–65 years) male and female volunteers, following a 12-hour fasting period and with informed consent. Blood donors, who participated every 8–12 weeks, were free from apparent infections, inflammatory conditions, or current allergic reactions (as determined by prior clinical examination) and had not used antibiotics or anti-inflammatory drugs for at least 10 days before blood collection. Studies on PMNL and monocytes were approved by the ethical commission of the Friedrich-Schiller-University Jena.

Cell treatment: Cells were seeded as described above (unless indicated otherwise), cultured for 24 h, and then treated with vehicle, VAL (10 μM), MC (10 μM), STS (0.3 μM), CHX (20 μg/ml), ETO (10 μM), or TPG (2 μM), and/or inhibitors at 37 °C and 5% $CO_2$. For serum depletion of NIH-3T3 fibroblasts, the medium was changed to serum-free DMEM after 24 h. 18:1 (Cayman Chemical) at a concentration of 100 μM was supplemented. Stimulation with insulin from bovine pancreas (1 μM; Sigma-Aldrich), recombinant mouse platelet-derived growth factor (PDGF) BB (25 ng/ml; GIBCO, Billings, MT), and recombinant human basic fibroblast growth factor (bFGF, 10 ng/ml; R&D Systems, Minneapolis, MN) was performed on quiescent cells obtained by culturing confluent cells for 24 h in DMEM containing 2% BSA.

Cell harvesting procedures using trypsin/EDTA to collect adherent cells differed depending on the readout. To determine the number of viable, apoptotic, necrotic, and dead cells, detached cells were recovered from the cell culture medium (except for kinetic studies and the determination of membrane intactness). In all other experiments, cells were washed with PBS pH 7.4 to remove detached cells and to enrich for viable and early apoptotic cells before adherent cells were collected by trypsinization and counted using a Vi-CELL Series Cell Counter (Beckman Coulter, Krefeld, Germany). Freshly isolated PMNL and monocytes were incubated under cytotoxic conditions in RPMI 1640 medium supplemented with FCS (10%), penicillin/streptomycin (100 U/ml and 100 μg/ml), and L-glutamine (2 mM).

Cell lines were tested for mycoplasma and their morphology was regularly inspected. MCF-7 and HEK-293 cells were authenticated by Multiplexion (Friedrichshafen, Germany; December, 2020) using Single Nucleotide Polymorphism (SNP) profiling (Multiplex Cell Line Authentication, https://www.multiplexion.de/en/cell-line-testing-service/multiplex-human-cell-line-authentication). In brief, DNA for SNP profiling was isolated from cell pellets using an innuPREP DNA Mini Kit (Analytik Jena, Jena, Germany) according to the manufacturer's instructions. No other cell lines were authenticated. HEK-293 was reported to be a misidentified cell line and was used in this study due to high basal p-Akt levels.

## Determination of metabolic activity

NIH-3T3 fibroblasts ($5 \times 10^4$ –$1 \times 10^5$ cells/well of a 96-well plate) were exposed to vehicle or cytotoxic conditions in the presence or absence of Fer-1 (3 μM). Alternatively, MDA-MB-231 (shGFP) breast cancer cells

($5 \times 10^3$ cells/well of a 96-well plate) were incubated for 24 h at 37 °C and 5% $CO_2$ and then treated with vehicle (DMSO), CAY (3 μM), RSL3 (1 μM) or a combination of both. After 48 h at 37 °C and 5% $CO_2$, 3-(4,5-dimethylthiazol-2-yl)-2,5-diphenyltetrazolium bromide (MTT, 20 μl, 5 mg/ml; Sigma-Aldrich) was added and the incubation continued for 2-4 h. Cellular dehydrogenase activity was assessed by the conversion of MTT to the formazan product, which was solubilized in SDS (10% in 20 mM HCl, pH 4.5, ≥16 h)[137]. The absorbance was measured at 570 nm using a SpectraMAX iD3 spectrometer, which was operated by SoftMax Pro 7.1 (Molecular Devices, San José, CA).

### Subcellular fractionation of fibroblasts

To separate (peri)nuclear from non-nuclear membranes, NIH-3T3 fibroblasts were suspended in 500 μl ice-cold hypotonic fractionation buffer (10 mM HEPES pH 7.4, 2 mM $MgCl_2$, 0.1 mM EDTA, 0.1 mM EGTA, 10 mM KCl, 1 mM dithiothreitol) and passed through a 25-gauge needle ten times. After 45 min on ice, the (peri)nuclear (pellet) and non-nuclear fractions (supernatant) were separated by centrifugation (600 × g, 10 min, 4 °C). Non-nuclear membranes were obtained from the supernatant by centrifugation (100.000 × g, 1 h, 4 °C). The remaining intact cells in the (peri)nuclear fraction were disrupted by repeated homogenization in fractionation buffer (500 μl). Pellets were washed with PBS pH 7.4 prior to analysis by UPLC-MS/MS and Western Blot.

In a variation of the procedure described above, Swiss 3T3 fibroblasts were suspended in ice-cold subcellular fractionation buffer (20 mM Tris-HCl pH 7.4, 250 mM sucrose, 10 mM KCl, 1.5 mM $MgCl_2$, 2 mM mercaptoethanol, proteinase inhibitor cocktail Complete (Roche Applied Science, Basel, Switzerland)), passed through a 25-gauge needle 10 times[72], and incubated on ice for 20 min. Samples were centrifuged (720 × g, 20 min, 4 °C), the pellet was resuspended in subcellular fractionation buffer, and remaining intact cells were removed by repeated homogenization (25-gauche needle, 10 times). Differential centrifugation at 720 × g (20 min, 4 °C), 10,000 × g (20 min, 4 °C), and 100,000 × g (1 h, 4 °C) yielded fractions 1 (nuclear fraction), 2 (mitochondrial fraction), and 3 (membrane fraction) as pellet and fraction 4 (cytosolic fraction) as supernatant.

### Extraction and analysis of phospholipids and fatty acids

For the analysis of PC, PE, PI, PS, PG, CL, sphingomyelins (SM) and FFA, lipids were extracted from cells, subcellular fractions, or *C. elegans* by sequential addition of PBS pH 7.4, methanol, chloroform, and saline (final ratio: 14:34:35:17), unless otherwise noted[138,139]. The organic layer was evaporated using an Eppendorf Concentrator Plus System (Eppendorf, Hamburg, Germany) in the high vapor pressure application mode (V-HV). The extracted lipids were dissolved in methanol and stored at −20 °C prior to analysis. Internal standards (0.1-0.4 nmol): 1,2-dimyristoyl-*sn*-glycero-3-phosphatidylcholine (DMPC), 1-pentadecanoyl-2-oleoyl(d7)-*sn*-glycero-3-phosphocholine, 1,2-dimyristoyl-*sn*-glycero-3-phosphatidylethanolamine (DMPE), 1,2-diheptadecanoyl-*sn*-glycero-3-phosphatidylglycerol, and/or 1,2-diheptadecanoyl-*sn*-glycero-3-phosphatidylserine. Alternatively, for the analysis of PI (Fig. 7a and Supplementary Fig. 30b), PS (Fig. 7a and Supplementary Fig. 30a, b), PG (Fig. 7a and Supplementary Fig. 30a, b), and FFA (Fig. 7b and Supplementary Fig. 30c), lipids were extracted by successive addition of n-butyl alcohol, methanol, and chloroform (final ratio: 2:1:1) and separated by anion exchange chromatography on DEAE-cellulose columns[140]. Lipids were eluted with chloroform/methanol/28% aqueous ammonia/28% acetic acid (200:100:3:0.9), dried under vacuum, and dissolved in methanol.

For the analysis of PC, PE, PI, PS, PG, SM and FFA, lipids were separated on an Acquity UPLC BEH C8 column (130 Å, 1.7 μm, 2.1 × 100 mm, Waters, Milford, MA) using an Acquity Ultraperformance LC system (Waters) and detected by a QTRAP 5500 mass spectrometer (Sciex, Framingham, MA) equipped with a Turbo V Ion Source and a TurboIonSpray probe for electrospray ionization[71,88]. Alternatively, PC and PE were analyzed using an ExionLC AD UHPLC system (Sciex) that

was coupled to a QTRAP 6500$^+$ mass spectrometer (Sciex) equipped with an IonDrive Turbo V Ion Source and a TurboIonSpray probe for electrospray ionization (Figs. 1g, 2h–j, 4e, 6e, 8d, e, Supplementary Figs. 9a, b, 12c–e, 13, 19c, d, 21c, 33b, and 33e). Chromatographic separation with the latter system was performed at 45 °C and a flow rate of 0.75 ml/min using mobile phase A (acetonitrile/water, 95/5, 2 mM ammonium acetate) and mobile phase B (water/acetonitrile, 90/10, 2 mM ammonium acetate). The gradient was ramped from 75 to 85% mobile phase A within 5 min, further increased to 100% A within 2 min, and maintained isocratically for another 2 min. For the detection of PC and PE species using the QTRAP 6500$^+$ mass spectrometer, the curtain gas was set to 40 psi, the collision gas to medium, the ion spray voltage to -4500 V, the heated capillary temperature to either 350 °C (PC) or 650 °C (PE), the sheath gas pressure to 55 psi, and the auxiliary gas pressure to 75 psi. The declustering potential was set to −44 V (PC) or -50 V (PE), the entrance potential to -10 (PC and PE), the collision energy to −46 eV (PC) or -38 eV (PE), and the collision cell exit potential to −11 V (PC) or −12 V (PE). Glycerophospholipids were analyzed in the negative ion mode, and both fatty acid anion fragments were detected by multiple reaction monitoring (MRM). The more intensive transition was used for quantitation. FFAs were quantified by single ion monitoring in the negative ion mode.

Alternatively, lipids were separated on an Acquity UPLC BEH C8 column (130 Å, 1.7 μm, 1 × 100 mm, Waters) using an Acquity Ultraperformance LC system (Waters) and detected by a TSQ Vantage Triple Stage Quadrupole mass spectrometer (Thermo Fisher Scientific) equipped with a HESI-II electrospray ionization source (Fig. 7a–d and Supplementary Fig. 30)[72]. In brief, the chromatographic separation was performed at 45 °C and a flow rate of 0.1 ml/min using mobile phase A (20 mM aqueous ammonium bicarbonate) and mobile phase B (acetonitrile). The gradient was ramped from 20 to 95% mobile phase B within 20 min followed by isocratic elution for 10 min. For the mass spectrometric detection of lipids, the spray voltage was set to 3000 V (PC, PE) or −2500 V (PE, PS, PI, PG, and free fatty acids), the capillary temperature to 220 °C, the vaporizer temperature to 450 °C, the collision cell gas pressure to 0.7 mTorr, the sheath gas pressure to 50 psi, and the auxiliary gas pressure to 15 psi. The instrument was operated with a scan range of $m/z$ 150-1200. PC species were quantified as $[M + H]^+$ ions by $m/z = 184$ precursor ion scans (collision energy: 35 eV), PE species were determined by $m/z = 141$ neutral loss scans in the positive ion mode (collision energy: 25 eV), and PE, PS, PI, PG, and FFA were quantified as $[M − H]^-$ ions by full scans in the negative ion mode. The identity of the PE, PS, and PI headgroups was confirmed by neutral loss scans (PE, PS) or precursor ion scans (PC, PI)[139]. Fatty acid composition of phospholipids was determined by monitoring the transitions to fatty acid anions by MRM (collision energy: 40-45 eV)[139].

For the analysis of CL, the chromatographic separation was performed at 55 °C on an Acquity UPLC BEH C8 column (130 Å, 1.7 μm, 2.1 × 100 mm, Waters) using an ExionLC AD UHPLC system (Sciex). The system was operated at a flow rate of 0.6 ml/min using methanol with 2.5 mM ammonium acetate as mobile phase A and water with 2.5 mM ammonium acetate as mobile phase B. The gradient was ramped from 85 to 98% mobile phase A within 8 min, followed by isocratic elution for 1 min. Eluted CLs were detected in the negative ion mode by MRM using a QTRAP 6500$^+$ mass spectrometer (Sciex), which was equipped with an IonDrive Turbo V Ion Source and a TurboIonSpray probe for electrospray ionization. The curtain gas was set to 40 psi, the collision gas to medium, the ion spray voltage to −4000 V, the heated capillary temperature to 650 °C, the sheath gas pressure to 40 psi, the auxiliary gas pressure to 80 psi, the declustering potential to −44 V, the entrance potential to −10 V, the collision energy to −40 eV, and the collision cell exit potential to −12 V. Transitions from $[M-2H]^{2-}$ to fatty acid anions were monitored.

Mass spectra were acquired using either Analyst 1.6 (QTRAP5500, Sciex) or Analyst 1.7. (QTRAP6500$^+$, Sciex) and processed with Analyst

1.6 (Sciex)[71]. Alternatively, mass spectra recorded with the TSQ Vantage Triple Stage Quadrupole mass spectrometer were processed using Xcalibur 2.0 (Thermo Fisher Scientific)[72].

Publicly available targeted lipidomics datasets (Metabolomics Workbench: PR001114, ST001740; https://doi.org/10.21228/M8Q39V), based on the same independent experiments as used in this study, were slightly adapted and reanalyzed to allow consistent calculation of phospholipid subgroup levels across identical experimental systems with a focus on PUFA-containing species (Fig. 1c, Fig. 1f – PUFA-PI, Supplementary Fig. 3c – PI). This dataset was previously used by Thürmer et al. to determine total PI levels, the PI profile, and the proportion of MUFA-containing PC, PE, PI, PS, and PG[46]. In addition, we reanalyzed the non-public dataset of Koeberle et al., again with a focus on PUFA-containing phospholipids (Fig. 7a-d, Supplementary Fig. 30). The dataset was previously used to visualize changes in the PI profile (Swiss-3T3), the proportion of 16:1-containing PC, PE, PS, PI, PG, SM, and FFA (Swiss-3T3), and the proportion of 16:1-containing PI (NIH-3T3, 3T3-L1)[72]. The full dataset is now publicly available (https://doi.org/10.48323/22gty-xjr51). Data on the effect of metformin on the proportion of PUFAs in phospholipids from young and old *C. elegans* (Fig. 3i, https://doi.org/10.48323/22gty-xjr51) were reanalyzed from a study by Espada et al. that focused on triglycerides[58].

To calculate absolute amounts of PC and PE, signals were normalized to cell number and a subgroup-specific internal standard. Signals of PS, PG, and CL were normalized to cell number and DMPC (for the analysis of CL, reported as relative units / $10^6$ cells) and corrected by external calibration with standards (PS(17:0/17:0), PG(17:0/17:0)) for the respective lipid subclass [PS (factor: 2.224) and PG (factor 0.624)] in Fig. 1a, Supplementary Fig. 3b, and Supplementary Fig. 7a-d]. Total phospholipid levels represent the sum of the absolute amounts of all PC, PE, PS, and PG species analyzed. FFA signal intensities in Fig. 1b and Supplementary Fig. 3b were not normalized to an internal standard. For the calculation of relative intensities (i.e., the proportion of lipids), all signals analyzed within the subgroup were summarized (=100%), and the signals of individual lipid species or lipid subfractions are expressed as a percentage of this sum. The cellular proportion of SFA-, MUFA-, or PUFA-containing phospholipids was calculated by summarizing the relative intensities of phospholipid species containing only saturated fatty acids (SFAs), at least one fatty acid with one double bond (MUFA), or at least one fatty acid with ≥ two double bonds (PUFA). To calculate the absolute and relative amounts of CL species, the peak areas of the analyzed transitions were first summarized for each molecular species and the sum was then divided by the number of fatty acids that were detected within the respective transitions.

### Extraction and analysis of oxidized PE species

Oxidized PE was extracted with methanol, chloroform, and saline, chromatographically separated by an ExionLC AD UHPLC system (Sciex), and analyzed by MRM in the negative ion mode using the above specified QTRAP6500+ mass spectrometer (Sciex)[118,141]. Sample handling and instrument settings were identical to those described for the analysis of non-oxidized PE with two exceptions. First, lipid extracts were stored under argon at −80 °C after evaporation until analysis. Second, the collision energy was set at −45 eV. The instruments were operated using Analyst 1.7.1 (Sciex) and the mass spectra were processed using Analyst 1.6.3 (Sciex).

Identification of oxidized PE(16:0/20:4) and PE(18:0/20:4) was based on fragmentation of [M-H]- to the saturated fatty acid anion and either the 20:4 anion with one or two oxygens incorporated or its secondary fragment (Supplementary Table 1). Signals were analyzed only if the retention times applied to the effective carbon number model and were within defined ranges (Supplementary Table 2). To assess the retention time windows for 1[O] and 2[O] PE, we oxidized 1-palmitoyl-2-arachidonoyl-*sn*-glycero-3-phosphatidylethanolamine ((PE(16:0/20:4); 850759 C, Avanti Polar Lipids) with a lipoxidase

(type V) from *Glycine max* (soybean; L6632; batch number: SLCC4512; Sigma-Aldrich)[142] and used the product as external standard. The retention time windows for oxidized PE(16:0/20:4) (1[O]: 2.70-2.89 min; 2[O]: 2.79-2.96 min) were extended to include putative regioisomers[143], as specified in Supplementary Table 2. The most intense, specific transition of oxidized PE(16:0/20:4) and PE(18:0/20:4) was selected for quantitation. The individual signals for PE(18:0/20:4 + 2[O]) within the retention time window represent isomeric species and were summarized. Absolute intensities of oxidized species were normalized to the internal standard DMPE and the cell number.

For the enzymatic oxidation of PE(16:0/20:4), the phospholipid (5 mg) was taken up in an aqueous solution of sodium deoxycholate (DOC, 3%, 500 µl; Sigma-Aldrich) and combined with Tris pH 8.6 (200 mM, 24.43 ml) and DOC (3%, 1.93 ml). Oxidation was initiated by the addition of type V lipoxidase (*Glycine max*) (1 mg). The reaction mixture was stirred at room temperature for 20 min before being loaded onto a solid phase extraction cartridge (Sep-Pak C18 6 cc Vac Cartridge, 500 mg sorbent, Waters) and rinsed with water (10 ml). The oxidized PE was eluted with methanol (5 ml) and evaporated to dryness using a TurboVap LV Automated Solvent Evaporation System (Biotage Sweden AB, Uppsala, Sweden). The residue was dissolved in ethanol and stored under argon at −80 °C.

### Extraction and analysis of acyl-CoAs

For acyl-CoA extraction, NIH-3T3 cells were taken up in aqueous methanol (70%) and proteins were precipitated (−20 °C, 1 h)[51]. After centrifugation, the supernatant was evaporated to dryness and the residue was extracted with water. [$^{13}C_3$]-malonyl-CoA (1 nmol; Sigma-Aldrich) was used as internal standard.

Chromatography was carried out on an Acquity UPLC BEH C18 column (130 Å, 1.7 µM, 2.1 × 50 mm) using an Acquity Ultra Performance LC system, which was coupled to a QTRAP 5500 mass spectrometer[51]. After electrospray ionization, acyl-CoAs ([M + H]+) were detected by MRM based on the neutral loss of 2'-phospho-ADP ([M + H-507]+) in the positive ion mode. The declustering potential was set to 60 V and the collision energy to 30 eV for acetyl-CoA and butyryl-CoA or 45 eV for malonyl-CoA. Mass spectrometric data were processed using Analyst 1.6 (Sciex).

### Extraction of proteins and lipids for multiomics analysis

Changes in the proteome and lipidome profiles of cells exposed to cytotoxic stress were determined after simultaneous proteo-metabolome liquid-liquid extraction[46,144]. In brief, NIH-3T3 fibroblasts ($5 \times 10^5/25$ cm²) were grown for 24 h and treated with vehicle or cell death inducers for 48 h. Cell culture medium was then aspirated, and the cells were washed three times with ice-cold PBS pH 7.4 (2 ml). Methanol (500 µl, −20 °C) and water (500 µl) were added, and cells were harvested by scraping. The samples were mixed with chloroform (500 µl, −20 °C) and consecutively shaken (1,400 cycles/min, 4 °C) for 20 min. After subsequent centrifugation (16,100 × g, 4 °C, 5 min; Fresco 21 Microcentrifuge, Thermo Fisher Scientific), the polar and non-polar phases and the interphases were collected in reaction tubes for further processing.

The polar and non-polar phases were evaporated under vacuum at room temperature using an Eppendorf Concentrator Plus system (Eppendorf) in the aqueous application mode (V-HQ, polar phases) or the high vapor pressure application mode (V-HV, non-polar phases). After evaporation of the non-polar phases, the lipid residues were dissolved in methanol and subjected to UPLC-MS/MS analysis (Supplementary Figs. 13a and 19c, d). LC-MS/MS-based lipid analysis of PC and PE species was performed by a QTRAP 6500+ mass spectrometer (Sciex) after chromatographic separation on an Acquity UPLC BEH C8 column (130 Å, 1.7 µm, 2.1 × 100 mm, Waters) using an ExionLC AD UHPLC system (Sciex) as described above.

The collected interphases were washed with methanol (-20 °C) and centrifuged (16,100 × g, 4 °C, 10 min). The pellets were

resuspended in denaturation buffer (8 M urea and 100 mM ammonium bicarbonate (ABC) pH 8.3 in water; 60 μl), diluted (1:5) in aqueous ABC pH 8.3 (100 mM), and then sonicated at room temperature using a Branson Ultrasonics Sonifier Modell 250 CE (instrument parameters: 1 × 10 s, constant duty cycle, output control: 2; Branson Ultrasonics, Brookfield, CT). Protein concentrations of the samples were then determined using a Pierce Micro BCA Protein Assay Kit (Thermo Fisher Scientific) according to the manufacturer´s protocol.

Extracted proteins (50 μg in 285 μl 2 M urea, 100 mM ABC, pH 8.3) were reduced by addition of 3 μl of dithiothreitol (DTT)-containing reduction buffer (1 M DTT, dissolved in 100 mM triethylammonium bicarbonate (TEAB) pH 8.3) and incubation for 30 min on a thermoshaker (55 °C, 800 rpm). For alkylation, 12 μl of iodoacetamide (IAA)-containing alkylation buffer (0.5 M IAA, dissolved in 100 mM TEAB pH 8.3) was added and samples were incubated for 30 min in the dark at room temperature. The alkylation reaction was stopped by the addition of 3 μl of the reduction buffer (1 M DTT, dissolved in 100 mM TEAB pH 8.3). Afterwards, 282 μl 100 mM ABC (pH 8.3) was added and proteins were digested for 16 h at 37 °C using 2.5 μg of trypsin (dissolved in trypsin resuspension buffer, Promega, Walldorf, Germany). Tryptic digestion was stopped by acidification with formic acid (4.6 μl). The samples were centrifuged (5 min, 16,000 × $g$, room temperature) and supernatants were transferred to Sep-Pak 100 cartridges (sorbent: 30 mg, Waters) activated with methanol and acetonitrile/water/formic acid (95/4/1) for desalting. After sample loading, SPE cartridges were washed with 1 ml washing solution (1% formic acid in water), the peptides were eluted with acetonitrile/water/formic acid (70/29/1), dried under vacuum, and subjected to quantitative proteomics analysis.

## Quantitative proteomics analysis

For LC-MS/MS-based quantitative proteomics, the dried tryptic peptide samples were dissolved in 20 μl of 0.1% aqueous formic acid. Chromatographic separation of peptides was performed on a nanoEase M/Z CSH C18 separation column (130 Å, 1.7 μm, 75 μm × 250 mm, Waters) using a Dionex UltiMate 3000 RSLCnano pro flow nano-ultra pressure liquid chromatography system (Thermo Fisher Scientific). Briefly, samples were loaded onto an Acclaim PepMap C18 trapping column (5 μm, 1 × 5 mm, Waters) using a mobile phase consisting of 95% buffer A (0.1% formic acid in HPLC grade water) and 5% buffer B (80% acetonitrile, 0.1% formic acid in HPLC grade water), followed by washing with 5% buffer B for 5 min at a flow rate of 15 μl/min. Peptides were subsequently eluted onto a nanoEase M/Z CSH C18 separation column at a flow rate of 300 nl/min and separated with a gradient of 5% to 30% buffer B within 120 min.

Eluted peptides were detected by an Orbitrap Fusion mass spectrometer (Thermo Fisher Scientific; Orbitrap Tribrid Series Tune software 3), which was coupled with an electrospray ionization source (Nanospray Flex Ion Source, Thermo Fisher Scientific) to a nano-ultra pressure liquid chromatography system. The spray was generated from a stainless-steel emitter (40 mm, OD 1/32, ES542, Thermo Fisher Scientific). The capillary voltage was set to 1900 V and MS/MS measurements were performed in data dependent acquisition mode using a normalized HCD collision energy of 30% in full speed mode. Every second an MS scan was performed over a m/z range from 350 to 1600, with a resolution of 120,000 at m/z 200 (maximum injection time = 120 ms, AGC target = $2 \times 10^5$). MS/MS spectra were recorded in the ion trap (rapid scan mode, maximum injection time = 60 ms, maximum AGC target = $1 \times 10^4$, intensity threshold: $1 \times 10^5$, first m/z: 120), with a quadrupole isolation width of 1.6 Da and an exclusion time of 60 s.

Raw files were analyzed with Proteome Discoverer 2.4 (Thermo Fisher Scientific). For peptide and protein identification, the obtained LC-MS/MS information was searched against a mouse database (SwissProt, 17,023 entries, downloaded November 4, 2022) and a contaminant database (116 entries) by Sequest HT. The following parameters were used for the database search: mass tolerance MS1: 6 ppm, mass tolerance MS2: 0.5 Da, fixed modification: carbamidomethylation (Cysteine), variable modification: Oxidation (Methionine), variable modification at protein N-terminus: Acetylation, Methionine loss, Methionine loss + Acetylation.

Percolator was used for false discovery rate calculations and Minora Feature Detection (default settings) was used for feature detection. For label-free quantification, the Precursor Ions Quantifier was used with the following parameters: Peptides to use: unique peptides, Precursor Abundance Based On: Area, Minimum Replicate Features: 100%, Normalization Mode: Total Peptide Amount, Protein Abundance Calculation: Summed Abundances. For quantitative comparison, the reported protein intensities were used.

Metabolic pathway analysis was performed based on proteins listed in Reactome, Wiki, and Kegg pathways or recent review articles[145–147] (Supplementary Data 2). Pathways analyzed: 1) glycolysis, gluconeogenesis, the TCA cycle, fatty acid biosynthesis, uptake, and degradation, (phospho)lipid metabolism, peroxisome biogenesis and metabolism, pentose phosphate pathway, 2) insulin and growth factor (IGF1, PDGF, and FGF) signaling, and 3) redox homeostasis (ROS regulation, antioxidant function, glutathione metabolism, iron metabolism, Nrf2 signaling, and ferroptosis). Proteins included in pathways of interest were combined (exception: pentose phosphate pathway) and duplicates were removed. Subsequently, proteins were omitted that i) were not detected in cells treated with vehicle, ii) were identified by fewer than three unique peptides, or iii) were not up- or down-regulated by VAL and MC in the same direction (for 1) ≥ 20%, for 2) ≥ 50%, and for 3) ≥ 20%).

Data for VAL-, MC-, CAY-, CAY + PI(18:1/18:1)-, and CAY + PI(16:0/16:0)-treated cells were obtained from the reanalysis of data published by Thürmer et al.[46]. Supplementary Data 1 lists both the reanalyzed and the new data sets (18:1 + VAL, 18:1, Serum + 18:1, Serum).

Gene ontology (GO) term enrichment analysis of biological processes was performed by g:profiler (https://cran.r-project.org/web/packages/gprofiler2/index.html)[144,148], and $P$ values were corrected by the Benjamini-Hochberg procedure. Only biological processes that are among the top 100 enriched processes in VAL- and MC-treated cells and are listed for both stress conditions were considered for further analysis. Criteria for protein inclusion were fold changes of at least $\log_2(1.5)$ or -$\log_2(1.5)$, respectively, and an adjusted $P$ value < 0.05. The remaining biological processes were ranked separately for VAL and MC treatment according to the adjusted $P$ values from low to high. The average positions of biological processes from both rankings (Supplementary Data 3) reflect the order, in which the biological processes are listed. Adjusted $P$ values were log-transformed for visualization in heatmaps.

## Metabolic flux studies

Mouse NIH-3T3 fibroblasts ($5 \times 10^5$/25 cm$^2$) were seeded and grown for 24 h at 37 °C and 5% CO$_2$ and treated with vehicle (DMSO), TOFA (5 μM), VAL (10 μM), or both VAL and TOFA. Metabolic flux studies were initiated by supplementation with acetate-d$_3$ (30 μM; Santa Cruz Biotechnologies) or 20:4-d$_8$ (1 μM; Cayman Chemical). Note that concentrations selected[149] are below the physiological plasma concentrations of acetate (50-200 μM)[150] and 20:4 (3 μM)[149]. After 48 h, lipids were extracted from the cells and subjected to UPLC-MS/MS analysis as described above. PC species carrying 20:4-d$_8$ or its elongation product 22:4-d$_8$ were detected by MRM in the negative ion mode as transitions from [M + 8 + CH$_3$COO]$^-$ parental ions to the respective deuterated and non-deuterated fatty acid anions. The incorporation of one or two acetate-d$_3$ into 20:4, 22:4, 22:5, and 22:6 was analyzed for the PC subclass in the negative ion mode by monitoring the transitions from [M + 1 + CH$_3$COO]$^-$ or [M + 2 + CH$_3$COO]$^-$ ions to the non-deuterated fatty acid anion and either to PUFA anions of the d$_1$-series (20:4-d$_1$, 22:4-d$_1$, 22:5-d$_1$, or 22:6-d$_1$) or d$_2$-series (22:4-d$_2$, 22:5-d$_2$, or 22:6-d$_2$). In parallel, non-labeled PC was analyzed to calculate the M + 1

and M + 2 isotopic patterns from the monoisotopic signals using the Mass (m/z) calculation tool from Lipid Maps® (https://www.lipidmaps.org/tools/structuredrawing/masscalc.php). These isotopic signals were then subtracted from the corresponding signals of the deuterium-labeled species. In addition, signals were background corrected based on control incubations without acetate-d$_3$. The amount of PUFA-d$_{1/2}$ in PC was calculated by summarizing the concentrations of PC(16:0_PUFA-d$_1$), PC(18:0_PUFA-d$_1$), PC(18:1_PUFA-d$_1$), PC(16:0_PUFA-d$_2$), PC(18:0_PUFA-d$_2$), and PC(18:1_PUFA-d$_2$).

## Analysis of cellular PI(3,4,5)P$_3$ levels

NIH-3T3 fibroblasts ($1.5 \times 10^6/75$ cm$^2$) grown for 24 h at 37 °C and 5% CO$_2$ were treated with vehicle (DMSO) or cell death inducers for 48 h. Cell culture medium was removed and the cells were immediately exposed to ice-cold 0.5 M trichloric acid. Cell debris was collected by centrifugation, washed twice with 5% trichloric acid/1 mM EDTA, and neutral lipids were removed by methanol/chloroform (2:1). Acidic lipids, including PI(3,4,5)P$_3$, were extracted from the pellet with methanol/chloroform/12 N HCl (80:40:1) and then from the supernatant with chloroform/0.1 N HCl (34:64)[46]. PI(3,4,5)P$_3$ was quantified using a PIP$_3$ Mass ELISA Kit (K-2500s; Echelon Biosciences Inc., Salt Lake City, UT) according to the manufacturer's instructions. Absorbance was measured using a SpectraMax iD3 Microplate Reader operated by SoftMax Pro 7.1 (Molecular Devices). Concentrations of PI(3,4,5)P$_3$ were normalized to the protein content, which was determined using a DC protein assay kit (Bio-Rad Laboratories GmbH)[46].

## Immunoprecipitation of Akt

NIH-3T3 cells ($5 \times 10^5/25$ cm$^2$) were grown for 48 h and then serum starved for 24 h in DMEM plus 0.2% FCS. Washed cells were scraped in 500 µl ice-cold lysis buffer (20 mM Tris-HCl pH 7.5, 1 mM EDTA, 1 mM sodium vanadate, 1 mM EGTA, 2.5 mM sodium pyrophosphate, 1 µg/ml leupeptin, 1 mM phenylmethanesulphonyl fluoride) and sonicated on ice ($3 \times 5$ s). After centrifugation of the lysate ($14,000 \times g$, 10 min, 4 °C), an aliquot (200 µl, 0.55 mg/ml total protein) was pre-cleared by incubation with 20 µl of protein A magnetic bead slurry (# 73778, Cell Signaling Technology) for 20 min at room temperature. The beads were removed with a magnet, and the pre-cleared lysate was incubated with mouse anti-pan-Akt (40D4; 1:200; # 2920, Cell Signaling Technology) for 16 h at 4 °C under rotation. Washed protein A magnetic beads (20 µl of slurry) were added to the formed immunocomplexes, and the incubation was continued for 20 min at room temperature. After magnetic separation and repeated washing of the beads with lysis buffer, phospholipids were detached from the pelleted immunoprecipitates with methanol (365 µl) and subjected to phospholipid extraction and analysis by UPLC-MS/MS as described for cells.

## Incorporation of phospholipids into fibroblasts

Lipid vesicles were formed by vigorous mixing of PC(16:0/16:0) (360 nmol) or PC(16:0/20:4) (360 nmol) in 3 ml DMEM (for serum depletion) plus FCS (10%) (for all other experiments) and sonication at 35 kHz and 50 °C (above the phase transition temperature of PC(16:0/16:0)) using a Sonorex RK512H ultrasonic bath (Bandelin, Berlin, Germany). Alternatively, PI(18:1/18:1) (150 nmol) or PI(16:0/16:0) (150 nmol) was resuspended in DMEM containing 10% FCS, mixed for 30 s, sonicated at 50 °C for 10 min at 45 kHz (Ultrasonic Cleaner USC100TH, VWR, Radnor, PA), mixed again for 30 s, and sonicated at 50 °C for another 10 min (Ultrasonic Cleaner USC100TH, VWR, Radnor, PA) to form phospholipid vesicles[46].

## Cell numbers, membrane intactness, cell viability, and cell morphology

Cell number and membrane intactness (indicative of cell viability) were measured after trypan blue staining using a Vi-CELL Series Cell Counter (Beckman Coulter; software: Vi-Cell XR Cell Viability Analyzer, version 2.03 or 2.06.3)[46]. For fibroblast morphology studies, cells were imaged by an Axiovert 200 M microscope with a Plan Neofluar × 100/1.30 Oil (DIC III) objective (Carl Zeiss, Jena, Germany) and an AxioCam MR3 camera (Carl Zeiss) using AxioVision 4.8 (Carl Zeiss)[46].

## Sample preparation, SDS-PAGE, and Western blotting

Cell pellets were resuspended in ice-cold Tris-HCl (pH 7.4, 20 mM), NaCl (150 mM), EDTA (2 mM), Triton X-100 (1%), sodium fluoride (5 mM), leupeptin (10 µg/ml), soybean trypsin inhibitor (60 µg/ml), sodium vanadate (1 mM), phenylmethanesulfonyl fluoride (1 mM), and sodium pyrophosphate (2.5 mM). The lysates were sonicated ($2 \times 5$ s, on ice), centrifuged ($12,000 \times g$, 5 min, 4 °C), and the protein concentrations of the supernatants were determined using a DC protein assay kit (Bio-Rad Laboratories GmbH, Munich, Germany). Aliquots were adjusted to equal protein concentrations and mixed with 1× SDS/PAGE sample loading buffer [125 mM Tris-HCl pH 6.5, 25% (m/v) sucrose, 5% SDS (m/v), 0.25% (m/v) bromophenol blue, and 5% (v/v) β-mercaptoethanol] and heated at 95 °C for 5 min. Aliquots of 10-20 µg of protein were resolved on 8–12% SDS-PAGE gels, and proteins were transferred onto Hybond ECL nitrocellulose membranes (GE Healthcare) or Amersham Protran 0.45 µm NC nitrocellulose membranes (Carl Roth, Karlsruhe, Germany), which were blocked with BSA (5%, m/v) or skim milk (5%, m/v) for 1 h at room temperature, washed and then incubated with primary antibodies overnight at 4 °C. Membranes were subsequently washed and incubated with secondary antibodies for 60 min. Proteins were detected using an Odyssey infrared imager (LI-COR Biosciences, Lincoln, NE) operated with Odyssey Infrared Imaging System Application Software Version 3.0 (LI-COR Biosciences) or a Fusion FX7 Edge Imaging System (spectra light capsules: C680, C780; emission filters: F-750, F-850; VILBER Lourmat, Collegien, France) operated with Evolution-Capt Edge Software Version 18.06 (VILBER Lourmat)[46,71]. Secondary antibodies used: IRDye 800CW-conjugated anti-rabbit IgG (1:10,000; # 926-32211, LI-COR Biosciences), anti-mouse IgG (1:10,000; # 926-32210, LI-COR Biosciences), IRDye 680LT-conjugated anti-rabbit IgG (1:80,000; # 926-68021, LI-COR Biosciences), anti-mouse IgG (1:80,000; # 926-68020, LI-COR Biosciences), DyLight® 800 anti-rabbit IgG (1:10,000; # SA510036, Thermo Fisher Scientific), and/or DyLight® 680 anti-rabbit IgG (1:10,000, # 35569, Thermo Fisher Scientific).

Data from densitometric analysis were background corrected and analyzed using an Odyssey Infrared Imaging System Application Software Version 3.0, and protein levels were normalized to β-actin, GAPDH, or, in case of phospho-proteins, to total protein (Odyssey infrared imager). Densitometric analysis using the Fusion FX7 Edge Imaging System was performed using Bio-1D imaging software Version 15.08c (Vilber Lourmat) with background subtraction based on the valley-to-valley approach. Protein levels were normalized to β-actin or total protein.

Uncropped versions of the Western blots from Figs. 2, 4 and 6 are shown in the corresponding source data files, while uncropped versions of the blots shown in the Supplementary Figs. are presented in Supplementary Figs. 38–54.

## Real-time quantitative PCR

Total RNA was isolated using E.Z.N.A Total RNA Kit (Omega Bio-tek, Norcross, GA), and equal amounts were used for cDNA synthesis with SuperScript III (Invitrogen, Carlsbad, CA). PCR was carried out with cDNA (1.25 µl), Maxima SYBR Green/ROX qPCR Master Mix (1×, Thermo Fisher Scientific), and forward and reverse primers (0.5 µM, each; TIB MOLBIOL, Berlin, Germany) in Mx3000P 96-well plates using a Stratagene Mx 3005 P qPCR system (Agilent Technologies, Santa Clara, CA). The thermal cycle started at 95 °C for 10 min, followed by 45 cycles of 95 °C for 15 s, 61 °C for 30 s, and 72 °C for 30 s. mRNA expression was analyzed using MXPro™ - Mx3005P v4.10 software (Agilent Technologies) and normalized to the amount of total RNA.

Primer sequences for *Acc1* (forward: 5′-CCTCCGTCAGCTCAGATA-CACTTT-3′; reverse: 5′-AATTCTGCTGGAGAAGCCACAG-3′) and *Fasn* (forward: 5′-CTGTTGGAAGTCAGCTATGAAGC-3′; reverse: 5′-AAGAA-GAAAGAGAGCCGGTTGG-3′) were selected based on previous studies[72,139]. Data on *Actb* and *Gapdh* mRNA expression (analyzed in parallel) can be found in Thürmer et al.[46].

## Knockdown of ACC1
NIH-3T3 fibroblasts ($5 \times 10^5/25$ cm$^2$) in DMEM plus FCS (10%) were incubated for 24 h at 37 °C and 5% $CO_2$ and then transfected with Acaca (ID 107476) Trilencer-27 Mouse *Acc1* siRNA (15 nM, OriGene Technologies, Inc., Rockville, MD) using Lipofectamine RNAiMax transfection reagent (10 μl; Invitrogen) according to the manufacturer's protocol[46]. The three unique 27mer siRNA duplexes targeted the sequences 5′-UAGAUAGUCAUGCAGCUACACUGAA-3′, 5′-AAGCUACUUUGGUUGAG CAUGGCAT-3′, and 5′-GAACUGAGAUUGCCAUAUUACUGT-3′. Universal scrambled negative control siRNA duplex (OriGene Technologies, Inc.) was used as control.

## Knockdown of LPLAT12/LPCAT3
NIH-3T3 fibroblasts ($9 \times 10^5/75$ cm$^2$) were cultivated for 24 h at 37 °C and 5% $CO_2$ in DMEM plus FCS (10%) and transfected with *Lpcat3* siRNA (15 nM; 5′-CACGGGCCTCTCAATTGCTTA-3′; Entrez Gene ID: 14792; Mm_Grcc3f_2 FlexiTube siRNA; Qiagen, Venlo, Netherlands) using Lipofectamine RNAiMax transfection reagent (30 μl; Invitrogen) according to the manufacturer´s instructions[151]. Non-targeting siRNA (15 nM; #1022076, Qiagen) was transfected as control. After 4 to 6 h, the culture medium was replaced, and the cells were treated with vehicle (DMSO) or VAL (10 μM) for 48 h.

## Knockdown of ACSL5
NIH-3T3 fibroblasts ($1 \times 10^5/9.6$ cm$^2$) were cultivated for 24 h at 37 °C and 5% $CO_2$ in DMEM plus FCS (10%) and transfected with *Acsl5* siRNA (15 nM; #SR419096; 3 unique 27mer siRNA duplexes; Locus ID 433256; OriGene Technologies, Inc., Rockville, MD) using Lipofectamine RNAiMax transfection reagent (7.5 μl; Invitrogen) according to the manufacturer´s instructions[151]. Non-targeting siRNA (15 nM; Trilencer-27 Universal Scrambled Negative Control siRNA Duplex; #SR30005; OriGene Technologies, Inc., Rockville, MD) was transfected as control. After 6 h, the cells were treated with vehicle (DMSO) or VAL (10 μM) for 48 h.

## CRISPR/Cas9-mediated gene knockout of LPLAT12/LPCAT3
LPCAT3 was deleted in human MDA-MB-231 triple-negative breast cancer cells using the GENF (gene co-targeting with non-efficient conditions) strategy according to Harayama et al.[152]. Single guide (Sg) RNAs were cloned into the BpiI-digested pX330-U6-Chimeric_BB-CBh-hSpCas9 (pX330) vector (Addgene plasmid #42230; https://www.addgene.org/42230; RRID:Addgene_42230; a gift from Feng Zhang)[153] or the BpiI-digested pUC-U6-sg vector[154] and designed to target *LPCAT3* (pX330_LPCAT3; sgRNA sequence: GCGCGTCAGAA-CAGGCGCTG) or *HPRT1* (pUC-U6-sg_HPRT1_sg2 for experiment #11; sgRNA sequence: GTCTTGCTCGAGATGTGATGA; pUC-U6-sg_HPRT1_sg3 for experiment #5 and #24; sgRNA sequence: GTAGCCCTCTGTGTGCTCAA; pUC-U6-sg_HPRT1_sg6; sgRNA sequence: GTATAATCCAAAGATGGTCA for experiment #16). Cells ($5 \times 10^4$ cells/1.9 cm$^2$) were co-transfected with the pX330_LPCAT3 and the competing pUC-U6-sg_HPRT1 vectors using TurboFectin 8.0 (# TF81001, OriGene Technologies, Inc.). In brief, the individual combinations of the pX330_LPCAT3 vector with one of the pUC-U6-sg_HPRT1 vectors (0.5 μg DNA in total; vector ratio: 49:1) were diluted in Opti-MEM (Gibco) immediately before transfection, gently mixed with Turbofectin 8.0 (1 μl), and incubated for 15 min at room temperature. The mixture was added dropwise to the cells and evenly distributed. As positive control to confirm selection by 6-TG, cells were transfected with pX330_LPCAT3 alone. 11 days post transfection, the cells were seeded in 24-well plates ($5 \times 10^4$ cells/1.9 cm$^2$) and incubated with 6-TG (10 μg/ml) for 13 days to select resistant cells lacking HPRT1. Semi-confluent cells were passaged routinely (1:5 to 1:20), with the cell culture medium and 6-TG (10 μg/ml) being refreshed every 3–4 days. Resistant polyclonal *LPCAT3*$^{-/-}$ cells (#5, #11, #16, #24) were expanded in routine cell culture medium containing 6-TG (10 μg/ml) and used for further experiments.

## Expression of constitutively active Akt
NIH-3T3 fibroblasts ($2.5 \times 10^5/25$ cm$^2$) were cultured for 24 h at 37 °C and 5% $CO_2$ in DMEM plus FCS (10%). For transient expression of constitutively activated Akt (T308D and S473D), cells were transfected with the expression vector pEGFP-Akt-DD using TurboFectin 8.0. (# TF81001, OriGene Technologies, Inc.) according to the manufacturer's instructions. The pmEGFP-C1 vector was transfected as a negative control. In brief, plasmid DNA was diluted in Opti-MEM (Gibco) immediately before transfection, gently mixed with Turbofectin 8.0 (Turbofectin:DNA ratio: 6:2), and incubated for 15 min at room temperature. The mixture was added dropwise to the cell culture flasks and evenly distributed. After 24 h at 37 °C and 5% $CO_2$, the medium was changed to fresh DMEM plus FCS (10%), and cells were exposed to cytotoxic stress for 48 h. The expression of constitutively activated Akt was confirmed 24 h and 72 h after transfection in vehicle-treated cells by Western blotting using an antibody directed against the hemagglutinin (HA)-tag of the mutant kinase (Supplementary Fig. 27a).

The expression vector pEGFP-Akt-DD was a gift from Julian Downward (Addgene plasmid # 39536; https://n2t.net/addgene:39536; RRID: Addgene_39536)[155]. pmEGFP-C1 was a gift from Benjamin Glick (Addgene plasmid # 36412; http://n2t.net/addgene:36412; RRID: Addgene_36412). Plasmids were amplified in bacteria (DH5alpha) that were grown in Lennox LB medium (Carl Roth) with kanamycin sulfate (50 μg/ml, Thermo Fisher Scientific) and extracted with a GeneJET Plasmid Midiprep Kit (Thermo Fisher Scientific) according to the manufacturer's instructions (protocol for low-speed centrifuges).

## Incorporation of [1-$^{14}$C]-acetate into cellular lipids
NIH-3T3 fibroblasts ($5 \times 10^5/25$ cm$^2$) were grown at 37 °C and 5% $CO_2$ for 24 h and pre-incubated with vehicle (DMSO) or cell death inducers. After 44 h, [1-$^{14}$C]-acetate (56 mCi/mmol, 1 mCi/ml, Hartmann Analytic, Braunschweig, Germany) was added (final concentration in medium: 0.5 μCi/ml) and the incubation continued for 4 h. Cells were harvested by trypsinization, washed twice with PBS pH 7.4 (500 μl), and the cell suspension was separated into two aliquots. Lipids were extracted from the first aliquot (400 μl) of the cell suspension according to the protocol for the analysis of phospholipids and FFA. The lipid film obtained by evaporation was dissolved in methanol (400 μl), and the incorporated radioactivity was counted after the addition of Rotiszint® eco plus (10 ml) using a Wallac 1414 WinSpectral Liquid Scintillation counter (Perkin Elmer, Waltham, MA). The total protein content of the second aliquot (100 μl) was determined using the DC protein assay kit (Bio-Rad Laboratories) after centrifugation and lysis of the cell pellet with 0.4% NaOH, 2% $Na_2CO_3$, 1% SDS (1 min trituration). The concentration of [1-$^{14}$C]-acetate (in pmol) was calculated from the measured radioactivity and normalized to the protein content; vehicle control: $12.2 \pm 2.0$ pmol [1-$^{14}$C]-acetate/mg protein.

## β-Oxidation of [1-$^{14}$C]-palmitate
NIH- 3T3 fibroblasts ($5 \times 10^4$/well; 24-well plate) were cultured for 24 h at 37 °C and 5% $CO_2$ before the addition of vehicle (DMSO) or cell death inducers. After 45 h, [1-$^{14}$C]-palmitate (55 mCi/mmol, 0.1 mCi/ml, Hartmann Analytic) complexed to fatty acid-free bovine serum albumin was added in the presence of unlabeled palmitate (final concentration

in medium: 1 μCi/ml and 109 μM). The wells were covered with Whatman filter paper grade 3 (Sigma-Aldrich) and the incubation was continued for 3 h. The filter papers were subsequently moistened with aqueous NaOH (3 M, 200 μl) to capture the [$^{14}$C]-CO$_2$ released after addition of perchloric acid (70%, 60 μl/well). Following another incubation for 16 h at 4 °C, the filter papers were collected, and their radioactivity was analyzed in Rotiszint® eco plus (2 ml) using a Wallac 1414 WinSpectral Liquid Scintillation counter (Perkin Elmer). Oxidized palmitate was calculated from the measured [$^{14}$C]-CO$_2$ radioactivity and normalized to cellular protein levels detected after lysis of parallel samples subjected to the same experimental procedure (see previous paragraph for lysis and protein assay); vehicle control: 415 ± 40 pmol [$^{14}$C]-CO$_2$/mg protein.

### Complexation of radioactive palmitate to BSA
[1-$^{14}$C]-Palmitic acid (10.5 μCi/105 μl) and non-labeled palmitic acid (9 mM, 100 μl), both dissolved in ethanol, were combined and concentrated to 120 μl by evaporation of the solvent with nitrogen. After adding 120 μl aqueous KOH (13.55 mM) to obtain water-soluble palmitate, the samples were again concentrated under a nitrogen stream at 45 °C to 60-80 μl and then adjusted to 100 μl by adding water. The palmitate solution was then added dropwise to fatty acid-free BSA (25 mg/ml, 0.4 ml, 37 °C) in DMEM at a molar ratio of 6:1.

### Determination of the cellular ADP/ATP ratio
NIH-3T3 fibroblasts (1 × 10$^4$/well; 96-well plate) were grown at 37 °C and 5% CO$_2$ for 24 h before treatment with vehicle or cell death inducers for 48 h. Cell death-induced changes in the ADP/ATP ratio were analyzed using an ADP/ATP Ratio Assay Kit (Sigma-Aldrich) according to the manufacturer's instructions. Luminescence was measured using a NOVOstar microplate reader (BMG Labtech, Ortenberg, Germany) operated with NOVOstar software version 1.30 (BMG Labtech).

### Flow cytometric analysis of apoptotic and necrotic cells
NIH-3T3 fibroblasts (5 × 10$^5$/25 cm$^2$) were exposed to cytotoxic stress for 48 h and then harvested and stained with propidium iodide and annexin-V using Annexin V Apoptosis Detection Kit FITC (Thermo Fisher Scientific) according to the manufacturer's protocol. Cells were analyzed with a BD LSR Fortessa flow cytometer (BD Biosciences, Franklin Lakes, NJ). Data were acquired by BD FACSDiva 8.0.1 (BD Biosciences) and processed by Flowlogic 7.3 (Miltenyi Biotech, Bergisch Gladbach, Germany). Alternatively, cells were analyzed with a Guava easyCyte 8HT flow cytometer (Merck Millipore) operated with guavaSoft 3.1.1 and the acquired data were processed by FlowJo™ v10.10 (Ashland, OR).

NIH3-3T3 fibroblasts (2.5 × 10$^5$/25 cm$^2$) were grown for 24 h, transfected with pEGFP-Akt-DD or pmEGFP-C1 expression vectors, and exposed to cytotoxic stress for 48 h. Cells were harvested and stained with DRAQ7 (Deep Red Anthraquinone 7, 564904, BD Biosciences) and Annexin V Alexa Fluor 555 conjugate (# A35108, Thermo Fisher Scientific). Briefly, harvested cells were washed with PBS pH 7.4, resuspended in Annexin V binding buffer (1×, 500 μl), and an aliquot (100 μl) was incubated with DRAQ7 (5 μl) and Annexin V Alexa Fluor 555 conjugate (5 μl) for 15 min at room temperature. After dilution in Annexin V binding buffer, cells were analyzed by a Guava easyCyte 8HT flow cytometer (Merck Millipore) operated with guavaSoft 3.1.1, and the acquired data were processed by Flowlogic 8.7 (Miltenyi Biotech).

The gating strategies are depicted in Supplementary Fig. 54.

### Flow cytometric analysis of cell cycle phases
NIH-3T3 fibroblasts (5 × 10$^5$/25 cm$^2$) were exposed to cytotoxic stress for 48 h and then harvested, washed in PBS (pH 7.4) and fixed with cold 70% (v/v) ethanol for 30 min at 4 °C. Fixed cells were washed twice with PBS (pH 7.4), treated with bovine pancreas RNase A (100 μg/mL, 50 μL, HY-129046, MedChemExpress) and subsequently stained with propidium iodide (50 μg/mL, 200 μL, HY-D0815, MedChemExpress). Cells were analyzed by a Guava easyCyte 8HT flow cytometer (Merck Millipore) operated with guavaSoft 3.1.1, and the acquired data were processed by FlowJo™ v10.10 (Ashland, OR).

### NADH/NADPH imaging by two-photon excited FLIM
NIH-3T3 cells (8 × 10$^4$/well; 12-well plate) were seeded on glass coverslips and, after 24 h, treated with vehicle or VAL for another 48 h. Cells were fixed with paraformaldehyde (4%, 20 min) and permeabilized with ice-cold acetone for 5 min. After blocking with normal goat serum (10%; Life Technologies, Carlsbad, CA), samples were incubated with rabbit anti-p-Akt (Ser472; Cell Signaling, #9271, 1:400) for 16 h at 4 °C, followed by goat anti-rabbit IgG Alexa Fluor 488 (Thermo Fisher Scientific, A32731, 1:500) for 45 min at room temperature[88].

The optical setup used has been described elsewhere[156] and is detailed in Supplementary Note 9. Briefly, a 2 ps Titanium sapphire laser (Mira HP, Coherent, Santa Clara, CA) was used at 830 nm for two photon excited FLIM imaging. The sample was illuminated using a water immersion objective (LD C-Apochromat, 40×, NA 1.1, Carl Zeiss), and FLIM signal was collected in Epi-direction. The FLIM signal was separated from the laser light using spectral filters (600 nm long pass dichroic mirror, Carl Zeiss, short pass filter 650 nm, bandpass filter 458 nm central wavelength, 64 nm bandwidth, Semrock, Rochester, NY fitting the NAD(P)H window) before being detected by a hybrid detector (HPM100, Becker & Hickl, Berlin, Germany) in combination with a fast AD-conversion card (SPC-150 v. 9.71 64 bit, Becker & Hickl) operated with SPCimage version 5.3 (Becker & Hickl).

### Immunofluorescence microscopy
The subcellular distribution of SREBP1, cleaved caspase 3 and 4-HNE was visualized by immunofluorescence microscopy. NIH 3T3 cells (70,000 per well, 6-well plate) were seeded onto glass coverslips following pre-treatment with Gibco Attachment Factor Protein (1X) (10308363, Thermo Fisher Scientific) (30 min, 37 °C) and subsequent drying of the coverslips (15 min, room temperature). Cells were cultured for 24 h prior to addition of vehicle (DMSO, 0.1%), VAL (10 μM), ABT-737 (10 μM), arachidonic acid (20 μM) (48 h), or withdrawal of serum (48 h) (37 °C, 5% CO$_2$). Alternatively, cells were exposed to RSL3 (5 μM) added after 45 h for 3 h or RSL3 (0.5 μM) added after 24 h for another 24 h. Then, the culture media were discarded, cells were washed twice with PBS pH 7.4 (1 ml), fixed with 4% paraformaldehyde in PBS pH 7.4 (Sigma Aldrich, 20 min, room temperature), permeabilized with 0.1% Triton X-100 in PBS pH 7.4 (Thermo Fisher Scientific, 15 min, room temperature), and blocked with 10% Normal Goat Serum in PBS pH 7.4 with 0.1% sodium azide (Thermo Fisher Scientific, 30 min, room temperature). The cells were then incubated overnight (4 °C) with rabbit anti-SREBP1 (1:100, #ab28481, Lot: 1051053-6; Abcam), rabbit anti-cleaved caspase-3 (Asp175) (1:400, #9661, Lot: 47; Cell Signaling Technology) or mouse anti-4-HNE (12F7) (1:50, # MA5-27570, Lot: ZC4192871; Thermo Fisher Scientific). 4-HNE was stained with Alexa Fluor™ 555 goat anti-mouse IgG (1:1000, # A21422, lot: 10143952; Invitrogen), and SREBP1 or cleaved caspase-3 was stained with Alexa Fluor™ 488 goat anti-rabbit IgG (1:500, # 11034, lot: 10729174; Invitrogen) for 30 min (room temperature). Samples were mounted with ProLong™ Gold Antifade Mountant with DAPI (15260719, Thermo Fisher Scientific) and images were captured using a BZ-X800 Microscope (Keyence, Mechelen, Belgium) operated with a BZ-X800 Viewer (Keyence) and equipped with a Plan-Apochomat 40x BZ-objective (Keyence) and a BZ-X800 camera (Keyence). The exposure time was kept constant for all acquired images between biological replicates. Omnifocal images of Z stacks (11 slices, Z = 0.5 μm) were exported with BZ-X800 Analyzer (Keyence), and fluorescence images were quantitatively analyzed using Fiji[157].

## Culture and treatment of *C. elegans*

The *C. elegans* strain N2 Bristol was obtained from the Caenorhabditis Genetics Center (University of Minnesota, Minneapolis, MN) and cultured under standard conditions[58]. In brief, *Escherichia coli* OP50 was seeded on LB agar plates and maintained at 4 °C. Bacterial cultures were grown overnight at 37 °C in antibiotic-free LB medium and provided to *C. elegans* on standard nematode growth medium (NGM) agar. Metformin hydrochloride (Tokyo Chemical Industry, Eschborn, Germany) was given to the worms in NGM agar. After 24 h at 20 °C, control and metformin-exposed *C. elegans* cultures were collected in M9 buffer, centrifuged, and worm pellets were snap frozen.

## Analysis of cellular ROS levels

Cellular ROS levels in NIH-3T3 fibroblasts exposed to cytotoxic stress were determined by a Cellular ROS Assay Kit (Red; ab186027, Abcam). Briefly, cells ($1 \times 10^4$/well) were cultured at 37 °C and 5% $CO_2$ in DMEM plus FCS (10%) for 24 h and then treated with cytotoxic compounds or serum starved. After 46 h, fibroblasts were stained with ROS Red Working Solution and incubated for another 2 h to complete the 48 h of cytotoxic stress exposure. Fluorescence intensity (excitation: 520 nm, emission: 605 nm) was monitored in bottom read mode by a SpectraMAX iD3 spectrometer operated by SoftMax Pro 7.1 (Molecular Devices). Changes of cellular ROS levels were determined relative to vehicle control by normalizing fluorescence intensities to the average cell number under each treatment condition.

## Co-regulated lipid networks

Lipid co-regulation is defined as Pearson correlation coefficient (calculated using CorrelationCalculator v1.0.1) with $0.7 \leq r \leq 1.0$ and is visualized by the MetScape v3.1.3 App (http://metscape.ncibi.org/)[158] in an Edge-weighted Spring-Embedded layout implemented in Cytoscape 3.9.1 (Cytoscape Consortium)[159,160]. Networks, generated from mean cellular proportions, visualize individual lipid species as nodes and co-regulations as edges, and exclude lipid species that do not correlate with others. Negative correlations with average p-Akt levels (Pearson correlation values < −0.6) are highlighted in blue.

## Data analysis and statistics

Data are presented as paired data, mean, mean + s.e.m. or mean + s.e.m. and single data of *n* independent experiments. Sample size was not pre-determined by statistical methods, and samples were not blinded. Normal distribution of the data with similar variance between groups was investigated using the Shapiro-Wilk test. Statistical evaluation of the non-transformed or log-transformed data was performed by one-way or two-way ANOVA or mixed-effects model (REML) for independent or correlated samples, followed by Dunnett's or Tukey's HSD *post hoc* tests, or by two-tailed student *t*-tests for paired or unpaired samples. Tests were performed using a two-sided $\alpha$ level of 0.05. *P* values < 0.05 were considered statistically significant. Volcano plots show the mean difference of percentage changes and the negative $\log_{10}$(adjusted *P* value). Adjusted *P* values were calculated by two-tailed, multiple unpaired student *t*-tests with correction for multiple comparisons using a two-stage linear step-up procedure by Benjamini, Krieger, and Yekutieli (false discovery rate 5%). Violin plots show the median and quartiles of the integrated density per cell or per nucleus. Outliers were detected using Grubb's (significance level of alpha = 0.05) or ROUT (Q = 1%) test.

Data were analyzed using Excel (Microsoft 365, Microsoft, Redmond, WA) and statistical calculations were performed using Graph-Pad Prism 9 and 10 for Windows (GraphPad Software, Boston, MA, www.graphpad.com). Exact *P* values can be found in the Source Data or, for the Supplementary Information, in Supplementary Data 4.

Line graphs, dot plots (Pearson correlation), violin plots, and bar graphs were either created using GraphPad Prism 9 and 10 (GraphPad Software), or OriginPro 2020 (OriginLab, Northampton, MA). Volcano plots were created using GraphPad Prism 9 and 10 (GraphPad Software). Heatmaps were created using GraphPad Prism 9 (GraphPad Software) from relative or absolute intensities that were normalized to control. Radar plots were prepared using SigmaPlot 14 (Systat Software GmbH). Principal component analysis was performed with OriginPro 2020.

Correlations between the expression of genes of interest (e.g., IGF1R) and resistance to small molecule probes and drugs were calculated across cancer cell lines based on datasets from the Cancer Therapeutics Response Portal (https://portals.broadinstitute.org/ctrp.v2.1/; 09/2022) and are visualized by Pearson correlation z-scores in Box-whisker-plots[161–163].

## Ethics & inclusion statement

Research for this manuscript is part of a local priority program (CMBI) and was conducted partly by local researchers and local collaborators in Innsbruck, Austria, and Jena, Germany, who are co-authors and partially share data ownership. Risk management and safety measures comply with national laws and regulations of the University of Innsbruck, the University of Jena, or partner organizations. Local ethics committee approval was obtained for studies involving human primary innate immune cells and does not apply to studies involving cell lines or *C. elegans*. Biological materials are shared upon reasonable request unless this would violate the rights of third-parties. Local research was not explicitly considered for citation.

## Reporting summary

Further information on research design is available in the Nature Portfolio Reporting Summary linked to this article.

## Data availability

Source data are provided with this paper. The mass spectrometric lipidomics data generated in this study have been deposited in the Metabolomics Workbench database (an international repository for metabolomics data and metadata, metabolite standards, protocols, tutorials and training, and analysis tools[164]) under accession codes PR001114 [https://doi.org/10.21228/M8Q39V][165] (ST001740) and PR002065 [https://doi.org/10.21228/M8PR7B] (ST003318, ST003319, ST003320, ST003321, ST003336, ST003337) or in the repository for research data of the University of Innsbruck powered by InvenioRDM (https://doi.org/10.48323/22gty-xjr51). The mass spectrometric proteomics data have been deposited to the ProteomeXchange Consortium via the PRIDE[166] partner repository with the dataset identifier PXD025396 and PXD053866. The processed mass spectrometric data are available at the above-mentioned repositories, the Source Data or the Supplementary Data. The correlation data between gene expression and cellular resistance to small molecules used in this study are available at the Cancer Therapeutics Response Portal (https://portals.broadinstitute.org/ctrp.v2.1/; 09/2022). All other data generated or used in this study are provided in the Source Data, Supplementary Information, or Supplementary Data. Source data are provided with this paper.

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

## Acknowledgements

The authors thank Felix Benscheidt, Katrin Fischer, Gabriel Knoll, Elif Gelmez, Laura Miek and Maria Völkel for technical assistance in performing experimental methods or data analysis. Research activities of A.K. related to the subject of this article were funded in part by the Austrian Science Fund (FWF) (10.55776/P36299). A.K. was further supported by the German Research Foundation (GRK 1715 and KO 4589/4-1), the Phospholipid Research Center Heidelberg (AKO-2015-037/1-1, AKO-2019-070/2-1, AKO-2O22-100/2-2), and the University of Jena (DRM/2013-05 and 2.7-05). A.K., T.M., and J.P. were supported by a Strategy and Innovation Grant from the Free State of Thuringia (41-5507-2016) and the Leibniz ScienceCampus InfectoOptics (SAS-2015-HKI-LWC). T.M., M.S., and J.P. acknowledge financial support from the "Thüringer Ministerium für Wirtschaft, Wissenschaft und Digitale Gesellschaft" (2015FOR-001), the "Europäische Fonds für regionale Entwicklung (EFRE)" (2015FOR-001), the "Thüringer Aufbaubank", the German Federal Ministry of Education and Research (BMBF), and the German Research Foundation (CRC SFB 1278 Polytarget). K.T. acknowledges support from European Union's Horizon 2020 research and innovation programme (MESI-STRAT Grant Agreement No. 754688), from the European Partnership for the Assessment of Risks from Chemicals PARC (Grant Agreement No. 101057014) and European Research Council (ERC AdG BEYOND STRESS, Grant Agreement No. 101054429) which have received funding from the European Union's Horizon Europe research and innovation programme. Views & opinions are those of the authors. Research of H.Sch. was supported by grants from the German Research Foundation (INST 337/15-1, INST 337/16-1, INST 152/837-1, INST 152/947-1 FUGG and SCHL 406/21-1). C.K. and R.H. were also supported by grants from the German Research Foundation (GRK 1715: C.K. and R.H.; GRK 2155: R.H.). R.W. received a Carl-Zeiss stipend and a Walter Benjamin fellowship (project number 498523371) from the German Research Foundation. Z.R. was funded by the Tyrolean Science Fund (TWF) (F.33467/7-2021). The authors acknowledge the financial support by the University of Graz. For the purpose of open access, the authors have applied a CC BY public copyright license to any Author Accepted Manuscript version arising from this submission. The Department of Lipid Life Science, National Center for Global Health and Medicine is collaborating with ONO PHARMACEUTICAL Co., Ltd. (Osaka, Japan) and Shimadzu Corp. (Kyoto, Japan) and the Department of Lipid Signaling, National Center for Global Health and Medicine is collaborating with ONO PHARMACEUTICAL Co., Ltd. (Osaka, Japan) without financial support for studies related to this manuscript.

## Author contributions

A.K. designed and L.B., A.G., J.G., K.L., V.J., H.P., Z.R., F.S., M.T., F.W., and L.W. performed and analyzed cell-based experiments and conducted lipidomic studies. K.L. generated subcellular fractions and determined subcellular lipid profiles. C.K., G.B., R.G., H.S., T.S., and K.T. provided flow cytometric or mass spectrometric methodology. A.G., K.L., A.M., and R.W performed and analyzed flow cytometric studies. A.G., S.H., M.H., and M.K. recorded and processed proteomics data, and A.G. and A.K. analyzed the results. H.P., S.G., and R.H. measured and evaluated experiments with radiolabeled lipids. J.P., M.S., and T.M. provided Non-

Linear Multimodal Imaging techniques, and K.L., T.M., and M.S. recorded and analyzed data from experiments that A.K. and M.S. designed. L.E., M.E., J.P., S.R., and C.W. provided model systems. T.H. provided CRISPR/Cas9 methodology. A.G., A.K., H.P., H.S., K.L., T.M., M.S., and J.P. interpreted the data. A.K. conceived the project, and A.K., H.P., and A.G. wrote the manuscript, which was edited and approved by all authors. Z.R., L.W., and L.B. contributed equally to the study.

## Competing interests

The authors declare no competing interests.

## Additional information

[1]Michael Popp Institute and Center for Molecular Biosciences Innsbruck (CMBI), University of Innsbruck, 6020 Innsbruck, Austria. [2]Chair of Pharmaceutical/Medicinal Chemistry, Institute of Pharmacy, Friedrich-Schiller-University Jena, 07743 Jena, Germany. [3]Institute of Pharmaceutical Sciences and Excellence Field BioHealth, University of Graz, Graz, Austria. [4]Institute of Physical Chemistry and Abbe Center of Photonics, Friedrich-Schiller-University Jena, 07743 Jena, Germany. [5]Leibniz Institute of Photonic Technology Jena e.V., Member of Leibniz Health Technology, 07745 Jena, Germany. [6]Department of Biochemistry, Center for Molecular Biomedicine (CMB), Friedrich-Schiller-University Jena, 07745 Jena, Germany. [7]Josep Carreras Leukaemia Research Institute (IJC), Campus Can Ruti, 08916 Badalona, Spain. [8]Institute of Molecular Cell Biology, Center for Molecular Biomedicine (CMB), Jena University Hospital, 07745 Jena, Germany. [9]Institute of Biochemistry and Center for Molecular Biosciences Innsbruck, University of Innsbruck, 6020 Innsbruck, Austria. [10]Institute of Clinical Chemistry and Laboratory Medicine, Section Mass Spectrometry and Proteomics, University Medical Center Hamburg-Eppendorf, 20246 Hamburg, Germany. [11]Leibniz Institute on Aging - Fritz Lipmann Institute (FLI), 07745 Jena, Germany. [12]ADSI-Austrian Drug Screening Institute, University of Innsbruck, 6020 Innsbruck, Austria. [13]Institute of Transfusion Medicine, University Hospital Jena, 07747 Jena, Germany. [14]Institute of Pharmacy, Paracelsus Medical University, 5020 Salzburg, Austria. [15]Institut de Pharmacologie Moléculaire et Cellulaire, Université Côte d'Azur - CNRS UMR7275 - Inserm U1323, 06560 Valbonne, France. [16]Department Metabolism, Senescence and Autophagy, Research Center One Health Ruhr, University Alliance Ruhr & University Hospital Essen, University Duisburg-Essen, 45141 Essen, Germany. [17]Freiburg Materials Research Center FMF, Albert-Ludwigs-University of Freiburg, 79104 Freiburg, Germany. [18]Laboratory of Pediatrics, Section Systems Medicine of Metabolism and Signaling, University of Groningen, University Medical Center Groningen, 9713 GZ Groningen, The Netherlands. [19]German Cancer Consortium (DKTK), partner site Essen/Duesseldorf, a partnership between German Cancer Research Center (DKFZ) and University Hospital Essen, 45147 Essen, Germany. [20]Department of Lipid Signaling, National Center for Global Health and Medicine, Shinjuku-ku, Tokyo, Japan. [21]Institute of Microbial Chemistry, Tokyo 141-0021, Japan. [22]Department of Lipid Life Science, National Center for Global Health and Medicine, Shinjuku-ku, Tokyo, Japan. [23]Department of Medical Lipid Science, Graduate School of Medicine, The University of Tokyo, Bunkyo-ku, Tokyo, Japan. [24]These authors contributed equally: André Gollowitzer, Helmut Pein. ✉e-mail: andreas.koeberle@uni-graz.at

