## [Transparent Peer Review file · Nature Communications]

Attenuated growth factor signaling during cell death initiation sensitizes membranes towards peroxidation

Corresponding Author: Professor Andreas Koeberle

Version 0:

Reviewer comments:

Reviewer #1

(Remarks to the Author)

See attached file, comments for authors.

Reviewer #2

(Remarks to the Author)

In this work Pein et al consider how alterations in fatty acid metabolism during apoptosis influence the activation of Akt. This work builds on previous studies from the same group on the relationship between phospholipid composition and Akt signaling (i.e. Pein et al., 2017). The present report is technically sound, contains many interesting pieces of data, and is generally well-written. However, much of the data is currently correlational, and several key questions concerning the model would need to be substantiated prior to publication, as described below:

Major:

-Does TOFA treatment or ACC siRNA (Figure 6) actually accelerate apoptosis? Figures 6h and 6i show effects on PUFA-PC levels and Akt, but what about the actual effects on cell death? This would seem to be the key point of the whole paper and it is not addressed. Maybe all of the observed changes in PUFA-PC and Akt are merely correlates of a dying cell with no functional impact on the extent or kinetics of cell death?

-Serum contains many free fatty acids and other lipids. How do serum lipids impact the remodeling of PCs during apoptosis? One might assume, for example, that the uptake of serum SFAs could compensate for the inhibition of ACC1/2 or FASN. Practically, one prediction of the proposed model would be that supplementing cells with excess SFA or MUFAs would counteract the enrichment of PUFA-PCs and thereby prevent the inhibition of Akt and the onset of apoptosis.

-In general, this work demonstrates a lot of interesting correlations (e.g. Figure 1-5), but has few interventional experiments of the types noted in the two points above that would strongly support the proposed model linking the observed changes in PUFA-PC levels and Akt activation to actual cell death. The possible experiments are numerous, but here are other concrete suggestions: (i) is cell death per se altered in the liposome experiments where the levels of PUFA-PC versus non-PUFA-PC are directly modulated (Figure S6)? Does introduction of a constitutively active Akt mutant protein overcome the effects of PUFA-PC enrichment on cell death in response to the various inducers studies here? Does the GSK Akt inhibitor used in this paper sensitize to or accelerate apoptosis?

-ACC1/2 activity and malonyl-CoA is not only essential for de novo palmitate synthesis, but also for the elongation of other fatty acids, including potentially various PUFA species. Does ACC1/2 inhibition interfere with PUFA synthesis and, if not, is this because all pro-death PUFAs are being obtained from the serum (back to the question above)?

-The manuscript lacks clarity about how the authors envision that PUFA-PCs are enriched in the membrane. Is it ultimately entirely due to the shutdown of de novo palmitate synthesis, or activation of an enzyme that preferentially acylates lyso-lipids with PUFAs? Some of this speculation is included in the Discussion, but the authors model could be more clearly stated, if

not investigated.

-A central aspect of the model is that ACC1/2 inhibition leads to defects in apoptotic regulation through effects on phospholipid composition and Akt signaling. The phenotype of mice lacking Acc1 or Acc2 might be mentioned in the Discussion. Acc2-KO mice are viable while Acc1-KO mice did not survive embryogenesis. Might these phenotypes inform on the present model?

Minor:

-Page 2, lines 8-11. Re-phrase for greater clarity.

-Page 6, line 16-17 contains the statement that "Together, PUFA-PC accumulates in early apoptotic fibroblasts throughout cytotoxic mechanism." This line needs to be re-written for clarity, but also perhaps care should be taken here and elsewhere in the manuscript to differentiate between the accumulation of PUFA-PC versus the relative depletion of other non-PUFA-PCs. Do PUFA-PC levels increase on an absolute scale in apoptotic cells, or is it that other types of PCs are vastly decreased, such that PUFA-PC levels are relatively increased? This is explained perhaps more clearly on page 17, lines 16-17.

-Page 12, line 3. Re-phrase.

-Page 16, lines 3-3 Re-phrase.

-Many of the Figure legends are organized in a confusing manner. Multiple panels are referenced out of order and in various configurations in the same line. The legend for Figure 6 is particularly egregious in this regard. This is probably done in an effort to save space, but perhaps more clarity could be introduced without elongating the text too extensively.

Reviewer #3

(Remarks to the Author)

In this paper, by using different apoptotic stimuli, the authors demonstrated that changes in phospholipid composition induced early in apoptosis attenuate AKT activation and enhance apoptosis. They showed that inhibition of fatty acid biosynthesis and a shift to beta-oxidation results in the accumulation of phosphatidylcholine containing polyunsaturated fatty acids species (PUFA-PC) in (peri)nuclear membranes. Consequently, PUFA-PC inhibits AKT recruitment and activation. 1. Phospholipids are the main structural components of plasma and intracellular membranes. The authors demonstrated that serum deprivation and exposure of fibroblasts to cytotoxic compounds such as staurosporine, cycloheximide, etoposide, valinomycin, thapsigargin, myricetin, and indirubin-3'-monoxime, results in a huge decrease of total phospholipid content. Particularly, after exposure, the amount of total phospholipids decreased to 25% as compared to those in control cells? Is this huge reduction of phospholipids associated with apoptosis? What was the viability of the cells after exposure? Will inhibitors of apoptosis or other cell death pathways prevent this consumption of phospholipids?

2. The observed decrease of phospholipid content can be due to activation of phospholipases, particularly phospholipase A2. Did the authors observe the accumulation of lysophospholipids? The involvement of lysophospholipids in the regulation of the PI3K-AKT pathway have been reported (PMID:25275042, PMID:24510253).

3. For some reason, almost all data in the paper are presented as a %, ratio or fold change. For the sets of phospholipid data, no absolute values are provided. For the analysis of phospholipids, since the authors used internal standards, this should allow them to make an estimation of the amounts of phospholipids in cells. The data could then be normalized to total protein or cell number.

4. The authors demonstrate that exposure to cytotoxic compounds results in a decrease of PC containing saturated and monounsaturated fatty acids localized in the sn-2 position of the PC molecule. However, the content of PUFA-PC under the same conditions was increased. This increase in the content of PUFA-PC molecular species may result from calculation error. It is very well may be that these "changes" in the content of PUFA-PC are related to the decrease of total PC that was used as the denominator for their calculation. For this reason, it is important to present normalized absolute values for the phospholipid content for all phospholipids.

5. Is the phospholipid-remodeling process involved in PUFA-PC accumulation during apoptosis? What will be the effect of knocking down of lysophosphatidylcholine-acyltransferase (LPCAT3) or acyl-CoA-synthetase 4 (ACSL4) on the accumulation of PUFA-PC and suppression of the PI3K/Akt pathway?

6. Using liposomes the authors demonstrated that liposomes composed of arachidonoyl-PC suppressed AKT phosphorylation. What was the amount of integrated arachidonoyl-PC and where in the cell it was localized? What will be the effect of other exogenous PUFA-PC that have linoleic, eicosatetraenoic, docosapentaenoic and docosahexaenoic in the sn-2 position? Will it be similar to the results obtained for arachidonoyl-PC?

7. It is not clear how PUFA-PC affects the activation of AKT. Do any protein/lipid interactions between AKT and PUFA-PC play a role in the suppression of AKT phosphorylation and therefore acceleration of apoptosis?

8. The phosphatidylinositol-3-kinase -AKT (PI3K-AKT) is an important intracellular signaling pathway in regulating cell proliferation, differentiation and apoptosis. Upon stimulation, AKT is recruited to the plasma membrane through binding to PI-3,4,5-P3, where it can be phosphorylated that results in its activation. In addition, the anti-apoptotic effect of the PI3K-AKT pathway through translocation of phosphorylated AKT to mitochondria has been documented (PMID:30165359).

Mitochondria are a major player in apoptosis. One of the early events of apoptosis is the oxidation (PMID:16408039) and externalization (PMID:24300280) of the mitochondrial phospholipid - cardiolipin that results in the release of cytochrome c to the cytosol and subsequent events for the execution of apoptosis. In addition, apoptosis associated with the translocation of

cardiolipin into the plasma membrane was also documented (PMID:26491702, PMID: 15181455). Similar to PI-3,4,5-P3, cardiolipin is a negatively charged phospholipid that can make its interaction with AKT possible. Were any changes in cardiolipin content detected? Moreover, it is not clear why the study was focused on (peri-)nuclear membranes while completely ignoring the plasma membrane and mitochondria.

9. The method for liposome preparation is not sufficiently detailed. The conditions, such as the amount of phospholipids, the frequency used for sonication procedure as well as the model of the sonication unit needs to be provided. It is not clear why the sonication procedure was performed at 50°C. It has been demonstrated that phospholipids are susceptible to degradation during sonication. These include oxidation of PUFA chains and hydrolysis of ester bonds that results in the generation of lysophospholipids and oxygenated PUFA (PMID:1814643). Thus, the LC/MS analysis of prepared liposomes has to be performed before addition to cells. In addition, the amount of lipids integrated into plasma membranes or intracellular compartments needs to be estimated and presented in the paper.

10. "PL-PUFA" has to be used instead of "PL-bound PUFA".

Reviewer #4

(Remarks to the Author)

I have read with pleasure the submitted manuscript which is a significant embodiment of work about lipid metabolism, apoptosis and signalling. I had difficulties in understanding part of the work; however, probably just because of my expertise is tangential to this work. Thus, I will focus my report on the analysis of imaging data, as originally asked by the Editor. I am rather acquainted with non-linear and time-resolved microscopy. However, I do not believe that the specific assay the authors implemented for this report is very common and I have missed justifications for several experimental choices the authors have done. I have also read Ref. 69 to find additional details but I could not find satisfactory explanations — this lack of clarity obstacles my capability to interpret the results.

From material and methods, I understand that cells were FA-fixed, immunostained for phospho-Akt, stained with DAPI and mounted. Excitation of fluorescence was done at 830nm and the fluorescence emission was detected with a 458/64 nm bandpass filter. Is this correct?

The bandpass filter is optimal for NADH imaging but poorly configured for Alexa488 with a cutoff wavelength of the filter at 490nm and a peak of Alexa488 further shifted to the red. Furthermore, DAPI is also emitting in the same spectral band. The two-photon cross-section of DAPI is negligible. However, the same is true for Alexa488. I am inclined to think I am misunderstanding the setup, but after several readings of the material and methods and supplementary notes I cannot see how. To summarize so far, I have the impression that three fluorophores (the endogenous NADH, DAPI and Alexa488) are emitting fluorescence in the analysed bandwidth, with NADH being naturally dim, Alexa488 being detected in the wrong emission band, hence emitting a low fluorescence signal and a possible contaminant of the signal coming from DAPI. I will neglect the contribution from DAPI, as nuclei are correctly dim in the images submitted. However, I struggle to understand the choice of filter. Are material and methods correct? A diagrammatic representation of the setup might be useful.

Also, the authors utilize FLIM for unmixing, where FLIM-based unmixing seems unnecessary. As the samples are fixed, why not doing sequential imaging of NADH and phospho-Akt, utilizing a different more red-shifted secondary antibody? If hardware limitations did not permit the authors this arrangement, at least the bandpass filter could have been replaced (swapped) for appropriate imaging of the phospho-Akt signal.

In the data analysis, the authors are then forced to perform a multi-exponential fit trying to unmix the signals. I have additional concerns on regard of the data analysis. The authors declare to require a three exponential fit to achieve unmixing of NADH (bound/unbound) and Alexa488, which seems then to exhibit also a double exponential decay.

The number of photons required to fit a double exponential decay is approximately 10,000 (see Koellner&Wolfrum in ' How Many Photons Are Necessary For Fluorescence-Lifetime Measurements' and related literature). The authors declare to stop acquisition of images whenever the brightest pixel reaches 1,000 photon counts. It is however unclear which is the typical threshold they used, i.e. it is unclear which is the minimum photon count they allowed to fit the data.

I miss information that is necessary to understand the quality of the data, as typical exposure time, pixel resolution, and it is unclear if a binning factor of 2 was the parameter that can be setup in B&H software or the literal meaning of a 2x2 kernel. I will assume that this was the value for B&H, which should result in a 5x5 convolution with a summing kernel of the image. The boost of 25 fold of the signal, would bring the max signal fitted to about 25,000 photon counts, i.e. sufficient for bi-exponential analysis.

All these details should be described.

In Fig. 2 and Supp. Info, I do not see LUTs so it is difficult to interpret the meaning of the blue colour. In light of the significant binning and lack of other details, I am therefore incapable to understand the data. At a first read, the images seem nice and to support the authors' interpretation, but at a more in-depth analysis I am not capable to shape an opinion around it.

Please clarify the nature of the representation and provide not overlaid images at least in Supp. Info.

Although the signal seems sufficient for the two-exponential fit, it is clearly insufficient to resolve 3 to 4 molecular species. The following section of the Supp. Note 3 is very difficult to understand:

“In order to localize the antibody signal, which is on average lower than the NAD(P)H signal, we used the co-localization approach plotting the 2D-histogram of the lifetime images. The 2D-histogram of the lifetimes t_1 and t_2 is characterized by two pixel clouds characteristic for NAD(P)H emission ($t_1 \sim 300$ ps, $t_2 \sim 2300$ ps) and Alexa488 ($t_1 \sim 1000$ ps, $t_2 \sim 4100$ ps). The pixel cloud characteristic for antibody emission has been back projected onto the fluorescence intensity image using Fiji7.”

The 2D histograms of Tau1 and Tau2 should be shown. It is impossible to assess the technique otherwise. However, this technique would work only if the signals are spatially segregated, which is not the case. The signals are mixed and unmixable with the use of a single spectral detection band, limited photon budget and a 2-exponential fit. The authors should provide a robust validation of the technique as this is new and potentially contradicting the experience of those familiar with similar approaches.

The validation seems to be limited to Supp. Fig. 8. It is not clear to me the histogram in panel c. The green curves are marked as ‘without anti-p-Akt’, but histograms very similar to the others are shown.

Eventually, the authors use this data only to state that the chemical environment of ‘p-Akt is modulated’ by chemical treatment. However, the authors cannot infer what these changes are reporting about. Cells are fixed and stained with an antibody. Therefore, the authors should have excluded that these shifts are unspecific by analysing fluorescence emitted by one antibody targeting a different protein. Even in this case, I am unsure about the value of reporting such a shift of biochemical environment with no clear relation to the rest of the work.

I am aware of how much work this type of assays require, and I do not write this lightly. Without additional information and validation, I do not have confidence in the methodology, and I question its biological relevance. Therefore, I would recommend omitting these results as they might be weakening the manuscript, overall, rather than strengthening it.

Version 1:

Reviewer comments:

Reviewer #5

(Remarks to the Author)

The manuscript “Attenuated growth factor signaling during cell death initiation sensitizes membranes towards peroxidation” by Gollowitz et al. suggested that fibroblasts stimulated with a variety of apoptosis-inducing agents display an increase in cellular PUFA-phospholipids and become susceptible to peroxidation under (associated) redox stress. This reprogramming in fibroblasts is caused by RTK-PI3K-Akt axis, which is recognized as a central pathway promoting lipid anabolism and cell survival. Cytotoxic agents inhibit growth factor receptor tyrosine kinase (RTK) and phosphatidylinositol-3-kinase (PI3K)/Akt signaling and results in a switch from fatty acid biosynthesis to degradation. Under these conditions, cells become dependent on exogenous or intracellularly released fatty acids for phospholipid remodeling, likely involving long-chain acyl-CoA synthetase (ACSL)5 and lysophospholipid acyltransferase (LPLAT)12/LPCAT3, thereby accumulating PUFAs in membranes and increasing susceptibility to peroxidation. Authors used cytotoxic inducers with different modes of action to claim that this mechanism works in all kind of stress and involved all phospholipids present in cells.

Overall topic of the paper is very exciting and some of the data are intriguing, but results obtained in different type of experiments demonstrated, that cell responses sometimes didn’t follow mechanism suggested by authors. It appears that author’s hypothesis works only for some cytotoxic inducer and involve only particular lipids. Presence of data, that doesn’t support authors idea don’t help reader to understand what is going on. I think that it’s reasonable to concentrate on stress agents that worked and on lipid that really changed in these conditions.

Major comments:

1. Results of experiments to study the induction of programmed cell death in fibroblasts by cytotoxic agents covering a broad mechanistic range presented at Supplementary Figures 1 and 2.

Measurements of cell death by staining with PI and annexin-V demonstrated, that not all cytotoxic agents used in these experiments are effective in induction of cell death. Treatment with VAL MC, ETO, STS, serum deprivation induced very small effect (~15%), while TNF alpha, TPG CHX, I3M were not effective. Other results presented by authors demonstrated that cytotoxic agents decrease cell numbers, and viability, but it can be the result of decreased proliferation,

Results presented in supplementary Fig 25 b also demonstrated very low percentage of cell death. In most cases reading is not more than 6%, which is usually considered normal result for control cells. Only GSK+VAL produce~12% of apoptotic cells and 6% of necrotic).

I also have similar concern about results presented on supplemental Fig.26

2. One of the main ideas of this manuscript is suggestion, that cells stimulated with different apoptosis-inducing agents follow the same rule and display an increased level of cellular PUFA-phospholipids. Results presented on several figures demonstrated, that opposite to authors suggestions changes in phospholipids under stress conditions are not similar and

PUFA content increased not in all lipids.

i) Supplementary Fig. 3b summarizes the kinetic changes in the cellular content of individual lipid subclasses, i.e., phosphatidylcholine (PC), phosphatidylethanolamine (PE), PS, phosphatidylglycerol (PG), and FFA.

This picture contains a lot of information (different cytotoxic agents, different time points, different lipids) and it's difficult to understand what is going on. But it can be clearly seen effects of stress agents on amount of different lipids are not similar and can involve different mechanisms.

ii) Fig. 1d To demonstrate, that cytotoxic stress elevates the proportion of PUFAs in phospholipids authors present Volcano plots indicating changes in lipid composition after treatment with VAL or MC. These results demonstrated, that changes only in certain lipids (mostly PC and PE) follow this suggestion, while changes in biggest part of lipids are not statistically significant.

iii) Supplementary Fig. 4 presents effects of cytotoxic agents other than VAL and MC on phospholipid composition. Results demonstrated, that again changes involve mostly PC and PE and can be found only for samples treated with CHX, STS and TPG, while other agents didn't induce statistically significant changes.

iv) Results presented on Supplementary Fig. 6 demonstrated, that cytotoxic stress-induced increase in amount of 20:3 20-4 22-5 22-6 fatty acids in PC, while amount of Linoleic acid, which is one of the major polyunsaturated fatty acid didn't change.

In general it can be suggested, that authors statement about increase of PUFA amount in all lipids under stress conditions is not true and describe behavior of mostly PC and PE.

3. Several conclusions made by authors in the individual experiments are in my opinion- not supported by the data.

i) Describing Fig. 1a authors stated: "Multiple cytotoxic conditions decreased the cellular content of glycerophospholipids, reaching significance for TPG and VAL."

It's not completely true. Only results obtained with VAL and serum deprivation can be taken into consideration. Differences obtained after treatment with other agents are not statistically significant. Results after treatment with TPG having $p=0.0640$ are also not significantly different from control. So, authors statement is true only for two conditions out of nine.

ii) Describing Fig 2 authors stated: "De novo fatty acid biosynthesis was moderately to severely reduced for all cell death conditions after 48 h (Fig. 2a), whereas the rate of β -oxidation (Fig. 2b) as well as the cellular levels of the β -oxidation intermediate butyryl-CoA were markedly increased (Fig. 7 2c).

Claimed by authors changes occurred not in all conditions tested. Changes are not significant for TNF α , STS, MC, I3M on Fig 2a and for TNF and I3M on Fig 2c

iii) In description of Fig. 3b authors wrote: "The mRNA levels of ACC and FAS were reduced after 24 and/or 48 h under different cytotoxic conditions"

This statement is also not true. In these experiments authors tested nine cytotoxic agents. Level of mRNA for FAS was not reduced in four and seven cases after 24 h and 48 h of incubation respectively.

iv) Results obtained in experiments to study phosphorylation of ACC and AMP protein kinase and changes in ADP/ATP demonstrated, that only half of stress agents used in this experiments (TPG, VAL, serum depletion, and MC) worked according to authors description.

v) Describing changes in phosphorylation of AKT authors stated, that many cytotoxic agents diminished Akt phosphorylation at Ser473 within 6 to 48 h (Fig. 6b-d, Supplementary Fig. 19a, b).

Results presented on fig. 6d demonstrated that four agents (TNF, CHX, ETO, serum deprivation) from total nine had no effects. That constituted ~40 % of tested conditions.

vi). Another inconsistency is present in description of the effects of cytotoxic agents on proteins involved in insulin and growth factor signaling including direct and indirect targets of Akt, such as glycogen synthase kinase (GSK)-3 β , mechanistic target of rapamycin (mTOR), and 70 kDa ribosomal S6 kinase (p70-S6K) (Fig. 6e).

Again, not in all cases author's statement is true. Amount of GSK-3 β was increased after addition of only four agents (TPG, VAL, MC and serum deprivation).

vii). The same in Fig. 6b, c where authors wrote: "As levels of p-Akt (used here as an indicator of the cytotoxic metabolic switch) decreased between 6 and 48 h, the PUFA ratio in phospholipids increased proportionally".

Good correlation was present only for TPG, VAL, serum deprivation, MC, I3M (five agents from nine used in experiment) and only in 48h.

viii) "... RTK ligands ... decreased the proportion of PUFAs in cellular phospholipids, specifically in PE, PI, and PS (Fig 7a and Supplementary Fig. 30a, b)".

Fig 7a statistically significant is changes only in PI. Changes in other phospholipids are not statistically significant.

4. To unravel the metabolic network behind the cytotoxic enrichment of PUFA-containing phospholipids, authors analyzed the proteome of fibroblasts challenged with either VAL or MC (Supplementary Data 1 and 2)48. Supplementary Fig. 14 .

They presented data about several groups of proteins such as proteins related to glycolysis, gluconeogenesis, and the TCA cycle (Supplementary Fig.15) , proteins related to the pentose phosphate cycle (Supplementary Fig.16), proteins related to fatty acid metabolism.(Supplementary Fig.17) proteins related to (phospho-)lipid metabolism, (Supplementary Fig.18), proteins involved in growth factor signaling. Presented data demonstrated, that treatment of VAL and MC inhibited expression of almost all studied proteins opposite to effects of CAY and serum deprivation, which effect only some proteins. Almost complete inhibition of protein expression by VAL and MC looks unnatural. It can be suggested that inhibitory effect of VAL and MC can be induced by direct suppression of protein synthesis and has nothing to do with general mechanisms of stress.

Result presented in literature demonstrated, that both VAL and MC can inhibit protein synthesis. Inhibition of protein synthesis at the level of elongation was proven to be another mode of action of valinomycin in addition to dissipation of membrane potential. Myrtoicommulone A (MC) can also act on protein synthesis, as it binds to chaperone Hsp60 and inhibits the refolding activity of the Hsp60-Hsp10 complex assisting in the correct folding of most proteins i

Suggestion about inhibition of protein synthesis by VAL can explain results obtained in the experiments to study subcellular fractionation of VAL-treated fibroblasts (Supplementary Fig. 8). These results demonstrated, that Ca²⁺ ATPase (PMCA), used as a marker of plasma membrane disappeared in non nuclear fraction after VAL treatment.

5.In Supplementary Fig. 38 authors present results about stress-induced regulation of p-Akt in cells with active and inactive ACC.

Effects of ETO TNF CHX and serum in different independent experiment are not consistent

6. In Fig. 8 authors presented data about possible involvement of increased amount of PUFA in phospholipids in development of ferroptosis.

I have several concerns about this data.

i) Authors claimed that selective ferroptosis inhibitor Fer-1 can inhibit cell death in apoptotic cells. In these experiments they studied cell population contained thousands of cells. It is not clear from their results whether simultaneous development of ferroptosis and apoptosis occur in the same cell or in different cells present in sample.

ii) Authors found, that the level of oxidized arachidonoyl-PE was greatly increased in VAL-treated cells (Fig 8b), and suggested, that accumulation of this lipid will navigate cells towards ferroptosis.

It's better to present results demonstrating increase in the amount of PE-AA-OOH not in percentage but in absolute value (pg/mg protein or pg/106 cells)

Several different compounds with similar m/z can be found in cells, so to prove that they studied oxidized AA-PE authors need to provide results of fragmentation.

lii)To demonstrate involvement of ferroptosis in cell death induced by VAL authors used selective ferroptosis inhibitor ferrostatin-1 (Fer-1) (Fig. 8c), but it looks like Fer-1 was not effective in preventing cell death induced by VAL.

7. Authors stated that NADPH is an essential cofactor for maintaining redox balance. NADPH has been associated with aberrant ROS generation and induction of cell death via NADPH oxidases (NOX)88 and oxidoreductases (POR). That is why they investigated the effect of VAL on the total cellular fluorescence in the spectral range from 426 to 490 nm, which is representative of cellular NAD(P)H status91.

Amount of NADPH was involved not only in maintaining of ROS balance. Besides being involved in the ROS defense, NADPH is a product of the pentose phosphate pathway. Accordingly, alterations in cellular growth rate could also influence NADPH levels via this pathway.

I am not sure, that measurements of fluorescence can accurately enough assess amount of NADPH. NADP(H) cannot be separated spectrally from NADH autofluorescence. As a result, measured fluorescence is always a mixture of both NADPH and NADH. Concentrations of the NAD(H) are usually about 10 times higher than the levels of NADP(H). Measurement of NAD(P)H fluorescence does not provide absolute values. In cells intensity of NAD(P)H fluorescence is confounded by scattering and the presence of other fluorophores that cannot be distinguished from NAD(P)H or that influence NADH autofluorescence such as the protein composition of a cell. Some lipid droplets or lipofuscin have similar autofluorescence spectra as NADH and can also interfere with measurements.

Reviewer #6

(Remarks to the Author)

In this manuscript, Gollowitzer et al. explore the crosstalk between various cell death programs and lipid metabolism. They show how a reprogramming of fatty acid channeling and incorporation is affected under some of these conditions and they

finally relate this phenotype to impaired RTK signaling which would be associated with membrane stress. While they describe a mechanism by which cells increase the proportion of PUFAs in phospholipids under certain cytotoxic stresses the manuscript fails to prove the direct effect on membrane peroxidation. In general, the manuscript is way too long and hard to follow with too many small details that distract from the mayor point which is also not clearly demonstrated. I believe it should be simplified as Nat.Comms targets a generalistic audience.

My main concern is related to the interpretation of the lipidomics data. First, normalizing it by the number of cells does not seem appropriate when cytotoxic conditions are analyzed basically because cell confluency affects the lipidome of cells (<https://doi.org/10.1038/nature14429>). It would be interesting to see how the lipid changes precede the cytotoxicity and for that, it would be more appropriate to harvest the cells 6, 12 or 24 h after the treatments, when the number of cells is comparable between the conditions. Perhaps a control could be added were the lipidomic changes at different confluencies is described, together with some of the treatments at timepoints where the confluency is still similar to that of control cells. Interestingly, the only treatment that doesn't follow the tendency to increase the PUFA-containing phospholipids is the one with TNF-alpha, the only one where the confluency is also higher than in the control. Moreover, PUFA's are taken up from the media and thus, they decrease with time and when the confluency goes up. If cells are under cytotoxic conditions, there will be less cells to consume the PUFA's and they would go up. In summary, I believe that even if interesting, the detected changes in lipids and the reprogramming of lipid metabolism that is described in this paper might be solely due to cell confluency.

- The lipidomics data is normalized by the cell number. When are the cells counted? The protocol does not explain how cells are harvested for the lipidomics analysis. If total cell number is used to normalize, this should be a crucial step to explain.
- What does "Total phospholipids" mean? How is it calculated? During the explanation of the protocol the text mentions all the time "PC, PE, PI, PS, PG, and FFA were extracted". "PC, PE, PI, PS, PG, and FFA were separated". In general, through the mentioned protocol one cannot be specific for the extraction of such lipids, the samples would be composed of many kinds of phospholipids including others such as PA or sphingolipids for example.
- The use of internal standards for PI or cardiolipins is not mentioned, is this correct? Please, state the amount and specific name/formula of the used internal standards for each lipid subclass in the methods section, in the form of a table. If an internal standard for PI was not used, how was the amount of different PI in cells calculated?
- The methods section for the lipidomics analysis is extremely confusing. What does "Reanalysis of lipidomics datasets" mean? In which context were all these datasets reanalyzed and for what purpose? Are those datasets being reused for main figures in this manuscript? This should be clearly stated in each figure.
- Fig 1g and 1h. Why is this such an important information? It could be represented in supplementary data.
- What is the effect of TOFA in cell number? This treatment should induce cell death because is increasing the amount of PUFA-containing phospholipids in membranes. Also, is inhibition of SCD1 cytotoxic?
- Which is the cell confluency when cells are treated with TOFA plus the other cytotoxic agents such as VAL?
- The data upon supplementation with MUFA's and PUFA's suggests that the effects seen are indeed dependent on the availability of these lipids in the media, which can vary with changes in confluency. A good experiment would be to replace the media with fresh media after 24 hours with the cytotoxic agents and check for the increase in PUFA containing phospholipids to see if the tendency seen in Fig 1f, changes.
- I do not find the data for SCD1 suppression among the cytotoxic conditions. It would be good to also check for this phenotype upon supplementation with fresh media. The effects seen in the active synthesis of MUFA's and degradation of fatty acids could be homeostatic.
- Fig 8c. Please describe what do the authors mean by "membrane intactness", how they measure it. Also, please demonstrate that membranes enriched in arachidonic acid containing phospholipids are more prone to lipid peroxidation and thus less viable. All these things are not connected.

- Different cytotoxic conditions are mixed throughout the paper. While I believe they might have some phenotypes in common, their effects are very different. Thus, sentences like the following, do not give any important information and I do not understand the intended message: Page 16: "While ACC protein expression decreased up to 1.7-fold (serum, 48 h) and p-ACC levels increased up to 2.5-fold (VAL, 1h), malonyl-CoA levels decreased as much as 14.5-fold (MC, 48 h)." In this section it is also stated that "Decreased levels of malonyl-CoA might therefore explain why fatty acid catabolism is greatly increased during cell death" but this is not proven and is not further investigated. Thus, the hints to understand how the metabolism is reprogrammed are merely descriptive and again, all these responses could be homeostatic. While homeostatic responses are interesting, I believe it is not the aim of this manuscript since the authors claim to have a mechanism.

Reviewer #7

(Remarks to the Author)

In this manuscript, Gollowitzer, et al. used various mechanistically distinct cytotoxic stress inducers to trigger apoptosis in the fibroblast and found that lipid metabolism program was switched in the early apoptotic cells: SFA and MUFA biosynthesis decreased and synthesis of phospholipids with PUFA (PUFA-PLs) increased. They claimed that the lipid metabolism reprogram was attributed to impaired growth factor receptor-PI3K-Akt signaling. The accumulation of PUFA-PLs caused by apoptosis inducing agents sensitized cells to ferroptosis, revealing a hidden relationship between apoptosis and ferroptosis. Overall, this study provides some interesting viewpoints. However, mechanism studies in this version are weak and a lot of conclusions in this manuscript are inferred from the correlation of gene/protein expression profiles without further evaluation by gene interference experiments. The evidence to support the relationship between apoptosis and ferroptosis is also weak.

1. This manuscript is way too long with 55 figures in total. The authors need to consolidate the figures and texts to highlight

their findings. The working model in Figure 10 is too complicative.

2. PL-PUFAs were generally increased upon different treatments from 24-48h (Fig S5A), however, the cell death upon different treatments varied a lot from 24-48h (Fig S1 and S2). It seemed the accumulation PL-PUFAs did not correlate with apoptosis status. It is questionable whether apoptosis per se or some pre-apoptotic status is related to the lipid metabolism programming, for example cell cycle arrest. A recent study suggested that cell cycle arrest caused PL-PUFAs accumulation through a PL remodeling process (PMID: 37963466). The authors need to rule out this possibility. And it is better to include some more specific apoptosis inhibitors, e.g. BCL2/BCL-xL inhibitors in the experiments.

3. The mechanisms how different apoptosis inducing agents inhibited de novo SFA/MUFA synthesis is not very clear. The authors showed that ACC and FAS mRNA level generally decreased, while protein level did not significantly decrease (up to 48h, Fig 3B, C,D). Since ACC activity was controlled by AMPK-mediated phosphorylation, the authors questioned that whether energy stress triggered AMPK activation and inhibited ACC activity. However, the result is controversial here. Only VAL and MC showed consistently increase of p-ACC (inhibition marker) as well as p-AMPK from 10min-6h (Fig 3E), while ADP/ATP showed that VAL and MC did not have energy stress (Fig 3H, time point unknown). General malonyl-CoA depletion might indicate that beta-oxidation may be the reason for SFA/MUFA catabolism. However, the authors did not have further experiment to confirm the function of beta-oxidation in apoptosis inducing agents-related lipid metabolism reprogramming. Moreover, there is a gap between RTK/PI3K/AKT signaling and SFA/MUFA synthesis/beta oxidation upon apoptosis inducing agents' treatment.

4. VAL and MC treatment upregulated ACSL5 expression (Fig 5A,B, C). The authors inferred that ACSL5 might be required for exogenous PUFA incorporation upon VAL and MC treatment. However, the authors did not provide experimental evidence to support ACSL5's function. Do other stimuli also upregulate ACSL5? Interestingly, serum starvation, which was used to mimic RTA inactivation, increased both ACSL3 (selective for MUFA-CoA) and ACSL4 (selective for PUFA-CoA, Sup Fig 32A). Does RTK/AKT/PI3K signaling regulate ACSL3/4/5 expression? And how? How to explain why PUFA-PL synthesis increased upon serum starvation? Again, experimental evidence is important.

5. RTK ligands selectively increase PUFA-PLs (Fig 7), while apoptosis inducing agents and serum depletion mainly increase PUFA-PCs (Fig 1), which suggested that RTK signaling and cytotoxic stress may regulate lipid metabolism through different mechanisms.

6. The link of PUFA-PLs accumulation upon apoptosis inducing agents to sensitize ferroptosis is weak. Fer1 (ferroptosis inhibitors) did not obviously affect apoptosis inducing agents-induced cell death comparing to RSL3 (GPX4 inhibitor, classical ferroptosis inducer, Fig 8A). There is some controversial result in Fig 8C that RSL3 alone did not inhibit fibroblast cell viability. Despite that, it also confirms that Fer1 did not obviously restore VAL-decreased cell viability (Fig 8C). Very delicate experiments are required here to confirm the function of ferroptosis in late apoptosis. For example, do apoptosis inducing agents have different potency on wt and ACSL4 ko cells?

Reviewer #8

(Remarks to the Author)

My focus in reviewing this manuscript is specifically on the proteomic data. In "NCOMMS-19-00380A-Z, the authors use a standard label free-quantitative proteomic approach of cells exposed to cytotoxic stress. I found the proteomic section to be very well written and the data presented is high quality. The method section is sufficiently detailed in accordance with proteomic standards with one exception. I do not see where the data was uploaded to a shared data repository like PRIDE. Please upload the data as required to ProteomExchange or PRIDE.

Reviewer #9

(Remarks to the Author)

The authors put a lot of effort and new experimental setups to complement and expand the relevance, robustness and significance of the study. Multiomics approaches were added to support the mechanistic findings and interpretation. However, my expertise only tangentially includes pathway analyses, therefore, I will focus on providing feedback on the imaging part as requested by the editors.

1) Proteomic & pathway analyses indicate alterations in glycolysis & oxidative phosphorylation (Fig 4 a,b, Supplements). It would have been expected that these changes might have been reflected in the free vs bound NADP(H) ratio (Fig. 9c), however different experimental batches indicate opposite trends in free vs bound NADP(H) ratios. As FLIM is highly sensitive to environmental and culture condition the robustness of the FLIM measurements has to be confirmed. In the methods, it is stated that cells were fixed before NADH imaging, was there a particular reason for this and were the times between fixation and data acquisition the same between all experimental batches? Furthermore, the authors removed the section on p-Akt imaging; does this imply that FLIM measurements were repeated without the AF488 co-staining?

2) CARS of the CH₂ band (2850 cm⁻¹) was used to determine overall lipid content in VAL-treated cells. Based on the images provided in Suppl. Figure 3c, it is difficult to evaluate the localization of CARS signals. Would it be possible to provide a CARS heatmap at a better contrast. Similarly, the graph plotting CARS intensities (Suppl Fig. 3c) is difficult to interpret regarding a 'perinuclear' localization of lipids.

Version 2:

Reviewer comments:

Reviewer #5

(Remarks to the Author)

Authors performed substantial amount of work answering reviewers comments and significantly improve quality of manuscript. But I still have several comments.

1. Authors demonstrated, that agents inducing cytotoxic stress increased PUFA ratio in different phospholipids, especially PC, while less pronounced increase in PUFA ratio were found in PE, PS, and PI. This phenomenon promoted oxidative membrane damage and sensitize cells to ferroptotic stimuli such as GPX4 inhibitor RSL3.

It was indeed demonstrated in many publications, that development of ferroptosis is accompanied by increase in lipid peroxidation. At the same time PE but not PC has been identified as a crucial PL oxidized during ferroptosis. For instance, studies of different types of cell death in bone-marrow-derived macrophages (BMDMs) demonstrates, that while in apoptotic, necroptotic, and pyroptotic BMDMs mostly oxidation of cardiolipin and/or phosphatidylcholine were detectable, BMDMs undergoing ferroptosis revealed mainly oxidized phosphatidylethanolamine (oxPE) species followed by oxidized phosphatidylserine (oxPS) and phosphatidylinositol (oxPI). (Bartosz Wiernicki et.al. Cell Death Dis .2020 Oct 27;11(10):922). I think it will be very useful if authors can provide some considerations explaining how their results about predominant accumulation of PUFA PC coordinate with significant role of PE in ferroptosis.

Minor concerns:

2. Fig.8b. The x-axis on it has name "difference of means ". but it is not clear what units was used. Figure has title "Fer-1 vs. w/o" and contained mostly readings from 0 to100. If Fer-1 serve as inhibitor, differences between readings with Fer-1 vs readings without it needs to be negative, or <1. Maybe its w/o vs Fer-1.

3. Picture in lower left corner of Fig.8 is not marked. It looks like 8c.

4. On Fig. 10 authors use indexes demonstrated changes in protein/metabolite availability or protein phosphorylation induced by cytotoxic stressors. The first number in the brackets indicates the number of cytotoxic stressors showing significant effects ($P < 0.05$) the second number states the total number of stressors investigated in the experiments. In this classification RAS/p-ERK has index (0/6) while high energy status low ADP/ATP and low-AMPK have indexes(1/7) and (2/7) respectively. Those indexes indicates that practically no stressors effected those parameters. May be its reasonable to remove them from schema?

5.Line 985 authors stated:" inhibition of IGF1R, PI3K, Akt or SREBP1 substantially induces lipogenic gene expression and increases the PUFA-PC ratio".

At the same time the main idea of this manuscript is decrease of SFA and MUFA biosynthesis during cytotoxic stress causing the shift towards PUFA-containing phospholipids.

Reviewer #6

(Remarks to the Author)

My comments only reate to the lipidomics data which is actually used to make the main conclusion of the paper. Unless NatComm editorial board thinks otherwise, already published data should not be used again in another manuscript. "Reanalysis of lipidomics datasets previously published by us to achieve consistency of lipid species between different figures" is not a valid justification. Moreover, specially in lipidomics analysis, one should be careful when using old data to compare datasets and draw conclusions from there.

The authors say that they did not trust their PI standard for quantifications because of the differences with long chain PIs in ionization and thus they used DMPC which does not make sense because DMPC is even more difefrent structurally. Then they corrected the values by external calibration with PI(15:0/d7-18:1)stating that "At the time this was measured, we did not have a deuterated PI standard". This raises again the same concern of using these old data to compare and make statements with newly acquired data. In lipidomics this is not trustable nor acceptable.

If the manuscript is finally accepted, the authors must upload ALL the lipidomics data related to this paper (new and already published) together to a repository prior to final acceptance and publication.

I find the manuscript still too long and specific for the scope of this journal.

Reviewer #7

(Remarks to the Author)

In this revised version, the authors have generated some new data to address the reviewer's comments. However, this reviewer still has some major concern regarding the quality of this study.

There is not a consensus and clear mechanism of how different cytotoxic stressors suppressed RTK-PI3K-AKT-lipogenesis pathways to increase the incorporation of PUFAs into phospholipids, which greatly limits the significance of this study. Importantly, the authors showed that the remodeling of phospholipids by these cytotoxic stressors is caspase independent. This raised an important question that whether apoptosis initiation per se is required generally for the suppression of RTK-PI3K-AKT-lipogenesis pathways. The authors can test whether pan BCL inhibitors and BH3 mimetics can induce the similar phospholipid alteration as cytotoxic stressors.

If yes, the authors need to dissect very carefully how apoptosis initiation regulates RTK-PI3K-AKT-lipogenesis pathways in upstream. If not, the authors need to spend more efforts to elucidate how those cytotoxic stressors inactivate RTK-PI3K-AKT pathways in common (ER stress, redox stress, cell cycle arrest, etc). Otherwise, the conclusion that attenuated RTK signaling during cell death initiation increases cells' susceptibility to oxidative membrane damage at the interface of apoptosis and alternative cell death programs is way too broad.

Response to the editor and reviewers

We thank the editor and the reviewers for their critical evaluation and very constructive comments. All points raised by the reviewers have been carefully addressed, and extensive additional experiments have been performed over the last few years (as requested by the reviewers, but also beyond), including:

- i) targeted lipidomics approaches (Fig. 7a-c, Supplementary Fig. 3a, b, 5b, 7a-e, 18b, c, 30a-e),
- ii) metabolic flux studies (Fig. 2h, i, Supplementary Fig. 11b-d),
- iii) quantitative proteomics (Fig. 4a-f, 5a-h, 9d-i, Supplementary Fig. 14-17, 18a, d-g, 32, 43a-d, Supplementary Data 1-3),
- iv) proof-of-concept studies in *C. elegans*, for chemoresistant cancer cells, and based on public databases correlating gene expression with cancer cell sensitivity (Fig. 3i, 8d, Supplementary Fig. 24, 40),
- v) and functional investigations delving deeper into the mechanism (Fig. 2j, 6a, 7d, 8a-c, e, 10, Supplementary Fig. 1c (right panel), d, 25c, d, 26a, b, 27a-c, 28, 36a-c, 37, 39, 41, 44, 45).

In addition, we have revised and expanded the presentation of previously acquired data as suggested by the reviewers: Fig. 1d, e, f, g (lower panel), i, 2g, Supplementary Fig. 4, 19a, 20, 23a-d, 25b, 31, 38a.

These new findings greatly expanded our mechanistic understanding of the lipidomic changes in stressed cells and led us to reinterpret our previous findings. Furthermore, we linked lipid metabolic changes to membrane alterations and (per)oxidative cell death, taking into account recent advances in cancer metabolism and ferroptosis. Large sections of the Introduction, Results, and Discussion have been restructured and rewritten accordingly, as indicated in the track-change version of the manuscript. In the following, we provide a detailed point-by-point list of the changes made (“answer”) in response to the referees’ suggestions (“comment”).

Reviewer 1

Synopsis

Comment: The manuscript “A switch in fatty acid metabolism during apoptosis attenuates subcellular AKT survival signalling” by Pein *et al.* reports that changes in the subcellular lipid composition of apoptotic cells attenuate AKT survival signalling. The authors show that fibroblasts stimulated with a variety of apoptosis-inducing agents display an increase in cellular PUFA-PC (poly-unsaturated fatty acids-containing phosphatidylcholines) levels after 48h, which the authors attribute predominantly to the (peri)nuclear region of these cells. The authors also investigate the levels of S473-phosphorylated AKT (p-AKT), which they use as a proxy for AKT activity. They find that the increase in PUFA-PC correlates with a decrease in p-AKT and conclude that an increase in PUFA-PC levels decreases AKT survival signalling in these cells. The authors claim that the reduction in p-AKT is predominantly in the (peri)nuclear region of cells. Next, the authors attempt to provide evidence that the lipid metabolism is dramatically changed in apoptotic fibroblasts, showing that *de novo* lipid synthesis is downregulated and β -oxidation is upregulated in early apoptosis. They show that apoptotic cells are in a low energy state, which they assess with the ratio of ATP to ADP and the overall cellular NADPH levels. The authors also report that the inhibition of lipid synthesis leads to a similar phenomenon of increased PUFA-PC levels and decreased p-AKT. Using a caspase inhibitor the authors propose that these observations are independent of caspase activity. Ultimately, the authors report that the correlation between increased PUFA-PC and reduced p-AKT can also be made in a number of cancer cell lines and to a lesser extent in primary human cells. In conclusion, the authors suggest that there exists a general mechanism of

lipid metabolism (“feed forward loop”) that accelerates apoptosis by attenuating AKT dependent survival signalling.

Answer: We thank the reviewer for the very constructive and critical comments on our manuscript and would like to point out that our findings from the original manuscript have been placed in a larger context in the revised version and the conclusions have been adapted accordingly. Inspired by the reviewer, we have performed extensive additional experiments, as described above. Our data indicate that cytotoxic stress impairs growth factor/PI3K/Akt signaling and shifts SFA and MUFA biosynthesis towards fatty acid catabolism. PUFAs from external or endogenous sources are then incorporated into phospholipids along with other fatty acids via ACSL4 or ACSL5 and LPCAT3, which elevates the proportion of PUFA-containing species and renders membranes more susceptible to peroxidation. As a consequence, the susceptibility to ferroptosis is increased, in particular when oxidative stress is induced in parallel. Suppression of Akt activation by elevated PUFA-containing phospholipids (as described in the first version) contributes to the induction of cell death under specific cytotoxic conditions. Taken together, we report a previously unknown mechanism at the crossroads of apoptosis and ferroptosis.

Comment: Overall judgement: Whilst the overall topic of the paper is very interesting, and some of the data is intriguing I would **not** recommend publication of this work in Nature Communications. My biggest concern with the manuscript is the handling and interpretation of data which appears to be very selective and biased. Also, several conclusions drawn from the individual experiments are - in my opinion - not supported by the data.

Answer: Driven by the reviewer’s concerns, we have continued our (experimental) studies on this project over the last few years. As described in the following sections, we have thoroughly revised the data handling, data interpretation and conclusions and provide additional mechanistic insights (see page 1 of the response letter). The Introduction, Results and Discussion have been restructured and large sections have been rewritten, as highlighted in the track-change version.

Comment: Concerning the overall novelty of this paper, many of the findings presented in this manuscript are not entirely new, but have been reported in similar settings before. 1) The fact that apoptosis reprograms the lipid composition of the cell has long been known (eg, see a brief review by Esposito, Cell Death and Differentiation 2002). However, the finding that apoptosis leads to an increase in PUFA-PC is indeed novel and very interesting. 2) It is also long known that the inhibition of PI3K/AKT signalling has been shown to induce apoptosis and therefore AKT signalling is considered to be anti-apoptotic. So it is no surprise to see that p-AKT levels are downregulated in apoptosis. It is rather more surprising that p-AKT is not downregulated in all the apoptotic settings the authors report. 3) It is also known that AKT signalling is not only dependent on PIP3 as an activating lipid but also dependent on the lipid composition of the membrane. To my knowledge, a thorough analysis of PUFA-PC species on their effect on AKT activity has not been reported. The claim that PUFA-PC can abolish AKT activation is therefore also interesting, but has been previously proposed by the same author (Andreas Koeberle et al., PNAS, 2013). However, the current manuscript does not provide any new mechanistic insight into how PUFA-PC inhibits AKT activation. The novelty of this manuscript is thus limited to the claims that a general feed forward loop of anti-correlated PUFA-PC and p-AKT drives apoptosis.

Answer: Based on the reviewer’s critical assessment of the novelty, we made strong efforts to further (experimentally) advance our study and redefined our claims. In the following, we briefly highlight the most important novel aspects and indicate the new insights gained during the revision (“**new**”). Please note that we have analyzed multiple cytotoxic stressors in parallel, which allows for the discovery of overriding mechanisms and distinguishes our study from those investigating a single cytotoxic stimulus. Moreover, we would like to emphasize that it is important to put independently studied signaling events into a larger context to understand how cell death programs are interconnected.

Novel aspects of the revised manuscript:

1. Cytotoxic stress and serum starvation increase the proportion of PUFAs in phospholipids (as appreciated by the reviewer), whereas insulin and growth factors have the opposite effect **[new]**.
2. Subcellular information on PUFA-PC in cell death has been lacking. In fact, there are only very few lipidomic studies at all that have investigated subcellular lipid profiles.
3. Mechanistically, cytotoxic stress reduces growth factor signaling (e.g., by decreasing Igfr availability or upregulating Igfbp6) **[new]** and Akt phosphorylation, which suppresses SFA and MUFA biosynthesis, promotes β -oxidation, leading to preferential incorporation of exogenous or released fatty acids (including PUFAs) into phospholipids via ACSL4/5 **[new]** and LPLAT12 (LPCAT3) **[new]**. This mechanism is consistent with previously published studies. For example, the central role of receptor tyrosine kinases and PI3K/Akt in regulating fatty acid metabolism and survival is well-established. It has also been reported that ACSL4 and LPLAT12 contribute to the incorporation of PUFAs into phospholipids. However, it has not been described that these individual steps are interconnected and that they are regulated by cytotoxic stress to increase the PUFA ratio of membranes **[new]**.
4. Regarding Akt activation (mentioned by the reviewer), we would like to point out that the phosphorylation of other survival kinases (e.g., ERK1/2) was not affected (at least not during this early stage of cell death), likely due to the concomitant decrease in protein phosphatase (PP)2A levels.
5. Multiomics studies (lipidomics and now also proteomics) link overarching changes in fatty acid and phospholipid metabolism with receptor tyrosine kinase signaling and mechanisms contributing to membrane peroxidation **[new]**. We have also performed compensatory studies **[new]** to ascribe individual changes at the proteome level to i) fatty acid biosynthesis (via SCD1), ii) the availability of de novo synthesized fatty acids (i.e., oleic acid/18:1), and iii) the stress-limiting lipokine PI(18:1/18:1) that we recently discovered (PMID: 35624087).
6. While elevated phospholipid PUFA ratios are known to render membranes susceptible to peroxidation and many studies describe small molecules and other settings that trigger both apoptosis and ferroptosis, here we describe for the first time that apoptotic changes in lipid metabolism (accompanied by an altered redox homeostasis) contribute to this enigmatic co-induction of cell death pathways **[new]**.
7. In addition, it has until recently never been proposed that the induction of cell death is regulated by survival kinases through fatty acid metabolism (which, however, is limited to specific cytotoxic stressors **[new]**).

In our opinion, these insights represent a strong conceptual advance as is now clearly pointed out in the restructured and rewritten manuscript. Supplementary Fig. 45 illustrates how we unraveled the link between apoptosis and (per)oxidative forms of cell death and highlights the mechanistic studies that were performed (originally and additionally). Our mechanistic concept is summarized in the new Fig. 10.

Comment: Unfortunately, the data presented in this manuscript only report correlations between these two measurements in certain settings or certain cell lines. Sometimes these correlations do not hold true in the majority of apoptotic agents or cell lines investigated. There is no evidence provided that such a feed forward loop would actually accelerate apoptosis or is just a “by-product of apoptosis”. Overall, I am very doubtful of the findings, conclusions, and claims of this manuscript.

Answer: We took these criticisms very seriously and conducted additional mechanistic studies on the previously proposed apoptosis-sensitizing pathway (new Supplementary Fig. 25a-d, 26a, b, 27a-d, 28, 36a-c, 37). On the one hand, our data clearly show that inhibition of fatty acid biosynthesis or PUFA-PC supplementation impairs Akt activation and decreases the viability of non-stressed cells (see also:

PMID: 23359699). On the other hand, an increased PUFA-PC ratio desensitized fibroblasts to a further decrease in Akt activation only under certain cytotoxic conditions, but not under others (new Fig. 36a-c). Moreover, we compared the kinetics of apoptosis and necrosis induction by Akt and fatty acid inhibition (Supplementary Fig. 25a-d, 26a, b, Supplementary Note 6) and investigated the effect of cytotoxic stress on cells expressing constitutively active Akt (Supplementary Fig. 27a-d, 28). Our data indicate that Akt contributes to the regulation of cell death under cytotoxic conditions, but seemingly independently from fatty acid biosynthesis at early stages. The manuscript has been reinterpreted accordingly (page 45, last paragraph – page 46, first paragraph).

Major comments:

Fig. 1-2

Comment: The authors make a quantitative assessment of the PUFA-PC levels, by reporting the ratio of PUFA-PC to total PC (e.g. Figure 1e). This is reasonable given that the level of total phospholipids, including PC, decreases during apoptosis (Figure 1a; Supplementary Figure 3a). However, when looking at p-AKT it appears that the authors only report the p-AKT density determined by western blot, without normalizing it to the total levels of AKT (e.g. Figure 2d). This is problematic because AKT levels seem also to go down during apoptosis in some cases (Figure 2d, Supplementary Figure 15). The drop in AKT could be due to the overall decrease in cell number (as compared to untreated cells, Supplementary Figure 1c) or simply due to the degradation of AKT in apoptosis. In any case, this must be taken into account. For instance, if the p-AKT signal in figure 2d were to be normalized by the AKT signal, the effect would most probably be less pronounced for all apoptotic agents (but ETO).

Answer: Agreed, we now additionally show the p-Akt/Akt ratio in the new Supplementary Fig. 19a to demonstrate that both a decrease in Akt phosphorylation and total Akt levels contribute to impaired Akt signaling in stressed fibroblasts (page 23, last paragraph). The y-axis labels describe whether “p-Akt / β -actin” or “p-Akt / Akt” is shown. In addition, on page 68, last paragraph and page 69, last paragraph, we explicitly mention that equal protein aliquots were loaded and that p-Akt, Akt, and other protein levels were normalized to β -actin or, in the case of phospho-proteins, additionally to total protein to correct for different cell numbers and degradation of Akt protein.

Comments: The authors make an important, but not validated assumption, which is that PIP3/PI(3,4)P2 levels are unaffected. It is well established that Akt depends on at least one of these lipids for both activation and activity. Given that the authors observe a dramatic decrease in total phospholipids during apoptosis, it is perhaps not surprising that they might also see reduced pAkt levels (although the evidence for this is also problematic, as described above).

Answer: Point well taken by the reviewer and now addressed by us. Despite the overall decrease in total phospholipids in early cell death, the absolute PI levels were decreased only by STS (reaching significance), VAL, and serum depletion but not or hardly affected by ETO, MC, and I3M, and even slightly increased for CHX and TPG under our experimental conditions [PMID: 35624087]. Regardless of the cellular PI content, PI(3,4,5)P₃ levels consistently decreased throughout the cytotoxic settings (new Fig. 6a), consistent with the revised interpretation of the data and supporting that cytotoxic stress impairs RTK-PI3K-Akt signaling.

Comment: Regarding the subcellular localization of PUFA-PC, the authors use CARS and TPEF microscopy (Figure 1f). From the data provided it is not clear how the authors can conclude that PUFA-PC is enriched in the perinuclear or nuclear region. The CARS signal intensity should be proportional to the C-H₂ symmetric vibrations (at 2845 cm⁻¹) (Folick A. et al, 2011), which is dependent on the length and the degree of unsaturation of the acyl chain. How do the authors distinguish between shorter fatty

acids (with less CH₂ groups) and polyunsaturated fatty acids? It seems that with CARS microscopy only lipid-rich regions of the cell can be distinguished from parts that contain less lipids. Given that membrane density is high in the perinuclear region, it seems questionable what has been learned here.

Answer: As the reviewer correctly points out, the image visualizes lipid-rich regions. The former Figure 1f shows where total lipids are reduced in fibroblasts during cell death, but does not distinguish between lipid subgroups (e.g., saturated and unsaturated lipids). We have moved the figure to the supplementary information (Supplementary Fig. 3c) and now emphasize on page 7, first paragraph and Supplementary Note 2 that total lipid levels are reduced in the (peri)nucleus and at other cellular membranes. Please note that we tried to detect unsaturated alkyl chains at 3050 cm⁻¹ but could not obtain a specific signal due to an overlay with the water signal.

Comment: Similar criticism applies for the “subcellular localization” of p-AKT (Figure 2e, Supplementary Figure 8a). The localization in untreated cells seems very different in Figure 2e and Supplementary Figure 8a, when in fact it should look the same. This could be due to a difference in confocal plane. It seems that both VAL treatment and LY294022 treatment reduces the overall intensity of p-AKT, but it is not justified to claim that there is a change in subcellular localization of p-AKT signal as the change between the two untreated cell pictures (compare Figure 2e and Supplementary Figure 8a) appears to be more pronounced.

Answer:

We agree with the reviewer that the way of presentation was confusing and, given the new direction and focus of the manuscript, have decided to delete these figures.

Comment: Regarding the subcellular fractionation (peri-nuclear vs non-nuclear membranes) it is also doubtful whether one can infer that there is any difference in membrane composition on the perinuclear membrane. In fact, it seems that the effect of increased PUFA-PC levels holds true for both (peri)nuclear and non-nuclear membranes when induced with VAL compared to untreated cells (Supplementary Figure 5b). In this respect, one of the key claims of the paper, which is that subcellular Akt signalling is suppressed by PUFA accumulation in the perinuclear region, is actually contradicted by their data. In summary, there is no convincing evidence provided for any subcellular change of PUFAPC nor p-AKT.

Answer: Thank you for highlighting this apparently confusing point. We do not claim that PUFA-PC accumulates selectively at the (peri)nuclear membrane during cell death. In fact, as mentioned by the reviewer, we show that PUFA-PC is also enriched at non-nuclear membranes, as is now clearly stated on page 10, second paragraph.

Comment: Regarding Fig 2c; How did the authors quantify the data to obtain a more than 100% decrease in p-AKT signal for VAL treated fibroblasts? What does a more than 100% decrease in p-AKT even mean?

Answer: Fig. 2c (now Fig. 6c) shows the difference in percentages (relative to vehicle control) between 6 and 48 h, the relevant period in which reduced fatty acid biosynthesis leads to an increase in the PUFA-PC ratio. p-Akt levels increase to 156% of vehicle control after 6 h and decrease to 26% of vehicle control after 48 h (see heatmap in Fig. 6b). The difference between 156% (6h) and 26% (48 h) is 130%. The way of calculation is now described in the legend of Fig. 6.

Comment: One final comment to pg. 9, ln 11-13: the authors use an Akt inhibitor to exclude that decreased Akt activity during apoptosis does not lead to PUFA-PC accumulation (Supplementary Fig. 7b). However, they propose a feed-forward mechanism in which decreased Akt signalling promotes apoptosis, which leads to PUFA-PC accumulation (Supplementary Fig. 16). The authors should think carefully about the apparent contradiction between their data and their model.

Answer: Thank you for your advice; we agree that the experiment is difficult to interpret. Given the central role of Akt in activating the lipogenic transcription factor SREBP1c and inducing de novo fatty acid biosynthesis, we would expect Akt inhibition to increase the cellular PUFA-PC ratio. However, the selective inhibitor GSK690693 (which is active as indicated by hyperphosphorylation of Akt; Supplementary Fig. 29b) neither affected the PUFA-PC ratio (Supplementary Fig. 29a) nor induced (apoptotic) cell death in non-stressed fibroblasts (new Supplementary Fig. 25). Thus, we can exclude that PUFA-PC levels are altered secondary to cell death (point raised by the reviewer). What is not fully understood is why GSK690693 enhances apoptotic PARP cleavage and cell death in stressed cells (that are challenged with cytotoxic stimuli) (Supplementary Fig. 25b-d and new Supplementary Note 6). It seems that additional adaptations besides impaired Akt signaling are required to elevate the cellular proportion of PUFA-containing phospholipids and that these adaptations are triggered by cytotoxic stress. Our quantitative proteomics study points to potential candidate proteins, including acyl-CoA synthetases (ACSLs) and factors in RTK signaling, such as GSK-3 β and S6K (Fig. 4 and 5, Supplementary Fig. 14-18, Supplementary Data 1 and 2). In further support of this hypothesis, using selective inhibitors, we show that in addition to PI3K/Akt, other signaling routes (cPLA₂, p38 MAPK, PKC, MEK1/2) contribute to shaping the phospholipid composition in response to RTK activation (Fig. 7d). These aspects are now described on page 25, last paragraph to page 26, second paragraph and page 28, first paragraph.

Fig. 4

Comment: The authors show that ACC and FAS are transcriptionally downregulated during apoptosis on the mRNA level. Again this could be due to a generally lower amount of cells or degradation of mRNAs or lower transcription. Interpretation of the authors' findings is complicated by the fact that the two "house-keeping" genes (GAPDH, β -actin) are also highly variable over time and according to treatment (Fig 4e).

Answer: Since GAPDH and β -actin mRNA are reduced in a time-dependent manner for different cytotoxic settings, the mRNA levels were normalized to the total RNA content, thus excluding that the observed effects depend on an overall degradation of RNA or a reduced cell number, as now described in the result section on page 14, lines 21-24, in the legend of Fig. 3, and in the method section on page 69, lines 5-7.

Comment: In any case, the observed downregulation on the mRNA level is not manifested on the protein level with any statistical significance. Only in serum deprived cells is ACC significantly downregulated and only in VAL-treated cells is FAS significantly downregulated (Figure 4f, 4g). This data can therefore not explain the overall switch in lipid metabolism. Moreover, how should a change in mRNA levels but not protein levels affect enzymatic activity? One possibility the authors explore is the change in ACC phosphorylation state. ... Overall, the p-ACC is quite different depending on which stimulus was used and never increased compared to untreated cells. Therefore, this cannot explain the decrease in fatty acid biosynthesis. In summary the authors conclude that "fatty acid biosynthesis was strongly reduced for many apoptotic settings with varying kinetics and mechanisms that include i) the decrease of ACC and FAS expression, ii) ACC inhibition by posttranslational modification, iii) limitations in ACC substrate supply, iv) energy depletion. However the data presented show i) ACC and FAS protein levels are only downregulated in one treatment each, ii) ACC is not inhibited by phosphorylation (it is not more phosphorylated/inhibited, than in the untreated control), iii) the ACC substrate acetyl-coA is not significantly reduced, only in TPG treated cells (however, the ACC product malonyl-CoA is reduced in 6 out of 8 cases), iv) the ATP/ADP levels are the same in almost all cells (treated/untreated), however the NADPH levels seem to be increased (Figure 5).

Answer: Our statements and conclusions are now more clearly described on page 3, lines 8-11 and page 14, second paragraph to page 16, first paragraph. From Fig. 2a-c, it is clear that the vast majority of cytotoxic settings strongly decrease the incorporation of acetate into fatty acids (fatty acid biosynthesis), induce the release of CO₂ from palmitate (following β -oxidation), and increase the cellular concentration of butyryl-CoA (an intermediate of β -oxidation). Thus, we show that many cell death stimuli suppress *de novo* fatty acid biosynthesis and induce β -oxidation. All further studies were aimed at finding out how individual cell death settings accomplish this task. We have, to the best of our knowledge, addressed key enzymes and regulatory pathways that might be affected. As pointed out by the reviewer, we did not find a single mechanism by which all cytotoxic settings inhibit fatty acid biosynthesis. Instead, some cell death inducers decrease the expression of ACC and/or SCD1 (PMID: 35624087), while others diminish FAS expression or induce ACC phosphorylation (see also comment below). Still others decrease the ATP/ADP ratio or reduce the availability of the substrate acetyl-CoA. Several of the cytotoxic settings have multiple effects. We consider it as one of the most intriguing findings of our study that cell death inducers have strikingly different strategies for perturbing fatty acid metabolism, but consistently shift the balance from fatty acid biosynthesis to β -oxidation.

Notably, many of the cell death inducers reduce malonyl-CoA levels (as marker of ACC activity), which can only partially be explained by reduced ACC availability, as the reviewer correctly pointed out (see also comment below).

Comment: The authors claim that the altered levels of malonyl-CoA that they clearly see (Figure 4i) is due to ACC inhibition by phosphorylation of ACC (p-ACC) which is reportedly carried out by AMPK, but the blots in Supplementary Figure 11 don't support this conclusion. After 48h p-ACC is most strikingly found in the untreated cells and TNF α (Supplementary Figure 11a).

Answer: Malonyl-CoA levels do not necessarily reflect ACC phosphorylation at 48 h, but depend on the cumulative ACC activity up to that time point. We therefore consider it likely that intermediate AMPK and ACC phosphorylation (up to 6 h for VAL and MC; Fig. 3e and Supplementary Fig. 13) contributes to the decrease in malonyl-CoA and PUFA-PC levels at later stages, as now explicitly described on page 14, lines 24 to page 15, line 5. Please note that we do not claim that this mechanism applies to all cytotoxic settings.

Fig. 5

Comment: The authors report a change in ADP/ATP levels when fibroblasts are treated with apoptotic agents. However, only 2 out of the 8 treated fibroblasts have an elevated ADP/ATP ratio (Figure 5a).

Answer: We agree with the author that ADP/ATP levels are substantially increased by TPG and serum depletion, while other cell death inducers are not or barely effective. As explained above, we neither expect nor intend to claim that all cytotoxic conditions decrease fatty acid biosynthesis by depleting ATP (at least that early in cell death). What we are saying is that ATP depletion seems to be one of the mechanisms by which different cytotoxic settings (i.e., TPG and serum depletion) impair fatty acid biosynthesis. This point has been clarified on page 16, last paragraph.

Comment: More worrying though, is that the ADP/ATP ratio in cells that are untreated is reported to be 1, which is not considered physiological for healthy cells. The authors cite a paper (reference 36) that indicates that in normal proliferating cells the ADP/ATP ratio is below 0.11, whereas in apoptotic cells the levels increase to 0.1-1.0. This paper also reports a ratio of ADP/ATP of 15 for heat-shocked necrotic cells. If the authors agree with this study, they must conclude that all their ADP/ATP ratios indicate apoptosis (also the untreated and TNF α treated cells) and that treatment only affects ADP/ATP ratios for 2 out of 8/9 treated cells. Perhaps the authors have artificially set the ADP/ATP ratio w/o

treatment to 1, in which case all other values are reported relative to the control cells, but this is extremely confusing and misleading.

Answer: Agreed and corrected (see revised Fig. 3h). The ADP/ATP was actually given as a fold-change of the vehicle control, as assumed by the referee. The average ADP/ATP ratio of our vehicle control is 0.25 and increases up to 0.67 for TPG. Please note that it is not possible to quantitatively compare ADP/ATP ratios between different cell lines. ADP/ATP ratios have been reported to vary by a factor up to 6 even between different fibroblast cell lines (Suhr et al., 2010, PLoS One, 23;5(11):e14095). We have therefore removed our semi-quantitative statement and reference to ADP/ATP levels in apoptosis and necrosis.

Fig. 6

Comment: Figure 6h,i,k reports the change in PUFA-PC/total PC, p-AKT density, and cl.PARP density for cells that are untreated or “pre-treated” with TOFA, before treatment with apoptosis inducing agents. It seems that the grey and green bars (non-pretreated) cells were normalized to the w/o cells without TOFA, and that all the white bars represent normalized values to TOFA-pretreated cells. In my opinion it is not valid to statistically compare these two quantities (between green and white bars) that are normalized to different values. Figure 6b shows that TOFA treatment changes the PUFA-PC ratio (Figure 6b), but the values for TOFA-treated and untreated cells are shown to be equal in Figure 6h. Similarly, p-AKT density with TOFA treatment decreases in Figure 6c, but is again shown to be equal in Figure 6i. Finally, Figure 6j shows a big difference in cleaved PARP levels between TOFA treated cells, but the values of treated and untreated cells are again set to be 100% in Figure 6k. The conclusion that TOFA does not have an additional effect to the apoptosis stimulating agents is flawed, at least on the basis of how the data has been treated/presented.

Answer: Many thanks for the comment. Motivated by the criticism, we thoroughly reflected on how to best visualize that TOFA diminishes the additional effect of cytotoxic stress on the three readouts. When we normalize the data within the bar charts to the vehicle control (100%), the graphs contain maximum information. However, it is difficult to estimate whether cell death inducers have the same percentage effect on the readout in the presence as in the absence of TOFA because TOFA already has an effect on the PUFA-PC ratio, p-Akt levels, and PARP cleavage *per se*. If we normalize the effects of the cell death inducers in the absence of TOFA to the vehicle control (100%, black bar) and the effects of the cell death inducers + TOFA to the TOFA-treated control (100%, white bar) (as done in the previous version), the graph shows whether the susceptibility of the cells to the cell death inducers was altered by TOFA. Therefore, to answer this particular question, it is appropriate to compare percentages that have been normalized to different controls. In other words, we do not want to compare absolute values but percentage changes (+/- cell death inducers) starting from different absolute values (+/- TOFA). Nevertheless, we agree with the reviewer that the differentially normalized series of bars is not optimal and can be misleading. We have therefore decided to visualize the data (all normalized to the vehicle control) using radar plots (new Supplementary Fig. 11a and 31) or multiple line charts connecting data from the same independent experiment (new Fig. 2g and Supplementary Fig. 38a). Statistical calculations for the latter compare vehicle vs. cytotoxic treatment and TOFA vs. TOFA plus cytotoxic treatment. The figure legends explain the design of the graphs and the underlying statistical calculations.

Fig. 7

Comment: Figure 7a shows that the pan-caspase inhibitor (Q-VD) either does not affect the western blot density of caspase 3 or even increases the level of uncleaved caspase 3. This is in striking contrast

to Supplementary Figure 14a, representative for the data presented in Fig. 7a, which displays a marked decrease in caspase 3 signal in response to Q-VD. It is therefore questionable whether Q-VD is actually working in the authors' experiments.

Answer: We thank the reviewer for pointing out this error. The label in the former Supplementary Fig. 14a was inadvertently reversed.

Comment: Again, it is not exactly clear what the values in Figure 7b-d have been normalized to but it seems that all the white bars were obtained by normalization to Q-VD pre-treated samples whereas the green bars are normalized to the untreated control (black bar). The p-tests are therefore not valid and a general conclusion cannot be inferred from these data.

Answer: Done, we now follow the same data visualization strategy as described above for experiments in the presence and absence of TOFA (see new Supplementary Fig. 23a-d).

Comment: Of particular concern is that the authors do not show the effect of the caspase inhibitor on PUFA-PC levels (the values in Fig. 7c have been artificially set to 100% with and without the inhibitor), despite claiming that it has no effect (pg. In 6-9). If the authors' claim of a feed-forward loop is to be believed, the inhibition of apoptosis should lead to a reduction of PUFA-PC (as the authors graphically illustrate in their model, Supplementary Fig. 16).

Answer: Thank you for highlighting this issue; we now present the requested data (new Supplementary Fig. 23c). Inhibition of caspase-induced apoptosis by Q-VD-OPh neither affected the PUFA-PC ratio nor substantially altered p-Akt levels in fibroblasts with or without cytotoxic treatment (Supplementary Fig. 23b, c). Thus, our data indicate that caspases do not play a major role in either decreasing the levels of active Akt or increasing the cellular proportion of PUFA-PC. Whether the progression of cell death results in a further increase of the PUFA-PC ratio cannot be answered from this experiment because cell death is initiated only after 24-48 h (Supplementary Fig. 2a and 26a, b) and another ≥ 24 h are required from the inhibition of fatty acid biosynthesis until the PUFA-PC ratio changes markedly (Fig. 2e). These aspects are now described in Supplementary Note 5. In addition, we have deleted the previous conclusion figure and rearranged our model as described above.

Fig. 8

Comment: Supplementary Figure 15 appears to be the basis for the quantification in Figure 8. However, it is not clear that any normalisation of the p-Akt signal to total Akt signal has taken place. Visual inspection of the raw Western blots suggests that the decrease in p-AKT would be only significant for MCF-7 cells (when treated with MC). So whilst the decrease in p-AKT might hold true for some apoptotic treatments, it does not appear to be the case for most of the cell lines investigated. To derive a common mechanism based on this data is misleading.

Answer: Agreed, we now state in the main text that cytotoxic stress consistently increases the PUFA-PC ratio in diverse cell lines and primary cells (new Fig. 1i). The data on active Akt levels (p-Akt) have instead been moved to Supplementary Fig. 22a, b and are described in Supplementary Note 4. We explicitly mention the limitations of the p-Akt analysis, including the use of concentrations pre-optimized for fibroblasts, and indicate that "the decrease in p-Akt levels is not necessarily predictive of the extent to which the PUFA-PC ratio has increased, and" that "certain cell lines/types are non-responsive". Note that we are more interested in p-Akt levels rather than Akt phosphorylation (normalized to total Akt) because the former determines signal transmission through this cascade when both phosphorylation and Akt expression are regulated (as it is the case during initiation of cell death). p-Akt levels were therefore normalized to β -actin.

Reviewer 2

Comment: In this work Pein et al consider how alterations in fatty acid metabolism during apoptosis influence the activation of Akt. This work builds on previous studies from the same group on the relationship between phospholipid composition and Akt signaling (i.e. Pein et al., 2017). The present report is technically sound, contains many interesting pieces of data, and is generally well-written. However, much of the data is currently correlational, and several key questions concerning the model would need to be substantiated prior to publication, as described below:

Answer: We thank the reviewer for the constructive assessment, which helped us a lot in developing the manuscript. Over the past few years, we have made great efforts to experimentally address the points raised by the reviewer and have performed many additional experiments to gain more detailed mechanistic insights. By combining multiomics (quantitative lipidomics and proteomics) with metabolic flux studies and experiments designed to confirm functional links within the proposed cascade, we have largely expanded our understanding of the underlying mechanisms. We also realized that the originally described mechanism is part of a bigger picture, which we present in the revised manuscript. In brief, we show that cytotoxic stress interferes with growth factor/PI3K/Akt signaling and causes a switch from SFA and MUFA biosynthesis towards β -oxidation. Under these conditions, cells are highly dependent on exogenous or released fatty acids for phospholipid biosynthesis/remodeling, and PUFAs (along with other fatty acids) are incorporated into phospholipids via ACSL isoenzymes (ACSL4 or ACSL5) and LPLAT12 (LPCAT3). As a consequence, the proportion of PUFA-containing phospholipids increases and membranes become more sensitive to (per)oxidation, which seems to be facilitated by the concomitant disturbance of redox homeostasis. Accordingly, apoptotic cell death gains a (per)oxidative component in later stages. The originally described negative regulation of Akt by increasing PUFA-PC ratios complements this mechanism, but is limited to specific cytotoxic settings. In conclusion, we have uncovered a mechanism that contributes to the emerging link between apoptotic and ferroptotic cell death.

Major:

Comment: - Does TOFA treatment or ACC siRNA (Figure 6) actually accelerate apoptosis? Figures 6h and 6i show effects on PUFA-PC levels and Akt, but what about the actual effects on cell death? This would seem to be the key point of the whole paper and it is not addressed. Maybe all of the observed changes in PUFA-PC and Akt are merely correlates of a dying cell with no functional impact on the extent or kinetics of cell death?

Answer: Point well taken; we have now performed kinetic studies on VAL-treated fibroblasts and demonstrate by PI/annexin V staining that co-treatment with TOFA actually enhances (not accelerates) cell death starting at 72 h (new Supplementary Fig. 26a, b). Notably, TOFA induces necrotic rather than (early) apoptotic cell death in stressed fibroblasts, consistent with our hypothesis that the elevated PUFA fraction in membrane phospholipids facilitates membrane peroxidation.

Comment: -Serum contains many free fatty acids and other lipids. How do serum lipids impact the remodeling of PCs during apoptosis? One might assume, for example, that the uptake of serum SFAs could compensate for the inhibition of ACC1/2 or FASN. Practically, one prediction of the proposed model would be that supplementing cells with excess SFA or MUFAs would counteract the enrichment of PUFA-PCs and thereby prevent the inhibition of Akt and the onset of apoptosis.

Answer: Many thanks, we completely agree that this is a straightforward and valuable experiment. Since saturated fatty acids, such as 16:0, cause lipotoxic cell stress depending on the concentration, we decided to supplement fibroblasts with 18:1 and confirmed that the addition of the MUFA fully prevented the increase in the PUFA-PC ratio upon treatment with VAL (new Supplementary Fig. 12a).

Moreover, we point out that “the fraction of cells undergoing necrosis (both with and without an apoptotic component) was substantially reduced, whereas the number of early apoptotic cells was hardly affected” (page 35, second paragraph), as we show in our recent manuscript on the discovery of a stress-limiting lipokine (PMID: 35624087). In addition, we investigated the effect of 18:1 on the proteome of VAL-treated and serum-starved fibroblasts and identified multiple proteins involved in growth factor-PI3K-Akt signaling, fatty acid and phospholipid metabolism, and redox homeostasis, whose levels are responsive to this treatment (Fig. 4c-f, 5c-h, 9h-i, Supplementary Fig. 14-17, 18a, d, f, 43a, d).

Comment: -In general, this work demonstrates a lot of interesting correlations (e.g. Figure 1-5), but has few interventional experiments of the types noted in the two points above that would strongly support the proposed model linking the observed changes in PUFA-PC levels and Akt activation to actual cell death. The possible experiments are numerous, but here are other concrete suggestions: (i) is cell death per se altered in the liposome experiments where the levels of PUFA-PC versus non-PUFA-PC are directly modulated (Figure S6)?

Answer: Done, we investigated the effect of the PUFA-PC species PC(16:0/22:6) on membrane intactness (trypan blue staining) (Fig. C1a), cell viability (MTT assay) (Fig. C1b), cell death (PI staining) (Fig. C1c), and PS externalization (Annexin V staining) (Fig. C1c).

Fig. C1. Cellular uptake of PC(16:0/22:6) on survival. Fibroblasts were treated with liposomal PC(16:0/16:0) or PC(16:0/22:4) (120 μ M, each) for 48 h. **a** Membrane intactness determined by trypan blue staining. **b** Cell viability determined by MTT assay. **c** Annexin V and propidium iodide (PI) staining. Cytograms are representative of five independent experiments. Mean + s.e.m. and single data from $n = 14$ (**a**), $n = 5$ (**b**, **c**) independent experiments. * $P < 0.05$ vs. vehicle control; repeated measures one-way ANOVA + Tukey HSD *post hoc* tests or two-tailed paired student *t*-test.

However, we would prefer to publish these data in a separate manuscript (in preparation) describing the mechanisms by which PC(16:0/22:6) helps to overcome tumor resistance by cytoskeleton-targeting drugs. The cellular uptake, subcellular distribution, and effect of PC(16:0/22:6) on p-Akt levels are shown below in Fig. C2.

Comment: Does introduction of a constitutively active Akt mutant protein overcome the effects of PUFA-PC enrichment on cell death in response to the various inducers studies here?

Answer: Thank you for this suggestion. We expressed the constitutively active Akt mutant pEGFP-Akt-DD in fibroblasts (new Supplementary Fig. 27a) and found that the number of cells in early apoptosis (Annexin V positive, PI negative) was markedly suppressed (Supplementary Fig. 27b and 28). However, the consequences on the number of viable cells varied between the cytotoxic settings, and the whole range from protective to (by trend) pro-death effects was observed (new Supplementary Fig. 27b and 28). We conclude that “Akt contributes to the regulation of cell death under cytotoxic conditions, which seems to be independent of fatty acid biosynthesis at early stages.”

Comment: Does the GSK Akt inhibitor used in this paper sensitize to or accelerate apoptosis?

Answer: Akt inhibition by GSK690693 sensitizes rather than accelerates VAL-induced cell death in fibroblasts, as shown by monitoring membrane intactness and PS externalization in kinetic studies (see new Supplementary Fig. 25c, d). This aspect is now correctly described on page 25, last paragraph, and in Supplementary Note 6. Considering the results of the kinetic studies on TOFA-induced cell death, we conclude that “the pro-survival activity of Akt is not necessarily dependent on ACC expression in early, but potentially in advanced cell death.”

Comment: -ACC1/2 activity and malonyl-CoA is not only essential for de novo palmitate synthesis, but also for the elongation of other fatty acids, including potentially various PUFA species. Does ACC1/2 inhibition interfere with PUFA synthesis and, if not, is this because all pro-death PUFAs are being obtained from the serum (back to the question above)?

Answer: To address this important question, we supplemented fibroblasts with deuterium-labelled acetate or deuterium-labelled arachidonic acid (20:4) and performed metabolic flux studies. The ACC inhibitor TOFA and the cytotoxic agent VAL did not affect the elongation of 18:2 or 18:3 to 20:4, but induced further elongation towards 22:4 and 22:5 (new Supplementary Fig. 11c-d), albeit at low levels. Instead, the incorporation of exogenous 20:4 into PC was strongly increased by TOFA and even more by VAL, and synergistic effects were observed when the two compounds were combined (new Fig. 2h, i and Supplementary Fig. 11b). Together, these data further support our hypothesis that “the reduced availability of SFAs and MUFAs upon ACC inhibition decreases the competition between acyl-CoA species in favor of PUFA-CoAs, which are efficiently incorporated into the overall depleting membrane phospholipids.” These aspects are now described in detail on page 13, first paragraph to page 14, first paragraph.

Comment: -The manuscript lacks clarity about how the authors envision that PUFA-PCs are enriched in the membrane. Is it ultimately entirely due to the shutdown of de novo palmitate synthesis, or activation of an enzyme that preferentially acylates lyso-lipids with PUFAs? Some of this speculation is included in the Discussion, but the authors model could be more clearly stated, if not investigated.

Answer: Done, we have addressed this point experimentally by i) the metabolic flux studies described above (new Fig. 2h, i and Supplementary Fig. 11b-d), ii) analysis of lipid metabolism at the proteome level (new Fig. 4a-d, Supplementary Fig. 14, 17, 18a, d-g, 32a-c), iii) quantitation of lyso-PC and lyso-PE (Supplementary Fig. 18b, c), and iii) silencing of LPLAT12/LPCAT3 (Fig. 2j and Supplementary Fig. 12b). Our findings and hypothesis are now described in detail on page 14, lines 5-14, page 17, second paragraph to page 18, first paragraph, page 19, lines 17-19, page 30, last paragraph, and page 45, second paragraph. In brief, the cytotoxic stress-induced increase of the PUFA-PC ratio (Fig. 1f) is accompanied by an enhanced expression of ACSL5 (for VAL and MC) (Fig. 4a-c and Supplementary Fig. 17) or ACSL4 (for serum depletion) (Supplementary Fig. 32a and Supplementary Fig. 17). Acyl-CoA synthetases (ACSLs) are essential for the activation of fatty acids as acyl-CoA esters, which are then incorporated into phospholipids by lysophospholipid acyltransferases. While ACSL5 accepts a wide

range of fatty acid substrates, ACSL4 has a preference for PUFAs. We speculate that the reduced de novo fatty acid biosynthesis lowers the competition of fatty acid species from the cellular free fatty acid pool (replenished by fatty acid uptake and/or degenerative processes such as autophagy) in favor of PUFAs, which are then incorporated into phospholipids under cytotoxic stress (along with the remaining SFAs and MUFAs) via ACSL5/ACSL4 and LPCAT3 (which prefers PUFA-CoAs).

Comment: -A central aspect of the model is that ACC1/2 inhibition leads to defects in apoptotic regulation through effects on phospholipid composition and Akt signaling. The phenotype of mice lacking Acc1 or Acc2 might be mentioned in the Discussion. Acc2-KO mice are viable while Acc1-KO mice did not survive embryogenesis. Might these phenotypes inform on the present model?

Answer: Thank you for pointing this out. We now write in the discussion that “we further speculate that the inverse relationship between SFA/MUFA biosynthesis and the phospholipid PUFA ratio contributes to the phenotypes of ACC isoenzyme-deficient mice” and explain potential links in detail in the new Supplementary Note 9.

Minor:

Comment: -Page 2, lines 8-11. Re-phrase for greater clarity.

Answer: Done, the entire section has been rewritten (page 10, second paragraph).

Comment: -Page 6, line 16-17 contains the statement that “Together, PUFA-PC accumulates in early apoptotic fibroblasts throughout cytotoxic mechanism.” This line needs to be re-written for clarity, but also perhaps care should be taken here and elsewhere in the manuscript to differentiate between the accumulation of PUFA-PC versus the relative depletion of other non-PUFA-PCs. Do PUFA-PC levels increase on an absolute scale in apoptotic cells, or is it that other types of PCs are vastly decreased, such that PUFA-PC levels are relatively increased? This is explained perhaps more clearly on page 17, lines 16-17.

Answer: Agreed, we now show the time-dependent changes in the total PUFA-PC content under the cytotoxic settings (new Fig. 1e and Supplementary Fig. 7c, d), rewrote the respective section (page 9, second paragraph to page 10, first paragraph), and carefully rephrased other parts of the manuscript to make clear that the proportion of PUFA-PC increases most consistently in stressed cells.

Comment: -Page 12, line 3. Re-phrase.

Answer: Done, we now write: “Cytotoxic stress downregulated further central proteins in insulin and growth factor signaling (Fig. 5) with metabolic and pro-survival functions (Fig. 6e), including direct and indirect targets of Akt, such as glycogen synthase kinase (GSK)-3 β , mechanistic target of rapamycin (mTOR), and 70 kDa ribosomal S6 kinase (p70-S6K).”

Comment: -Page 16, lines 3-3 Re-phrase.

Answer: Done, we now write: “TPG and serum depletion, but not other cytotoxic agents, increased the ADP/ATP ratio (Fig. 3h), indicating a low cellular energy status.”

Comment: -Many of the Figure legends are organized in a confusing manner. Multiple panels are referenced out of order and in various configurations in the same line. The legend for Figure 6 is particularly egregious in this regard. This is probably done in an effort to save space, but perhaps more clarity could be introduced without elongating the text too extensively.

Answer: Indeed, aspects common to several panels have been grouped in the legends to meet the requirements of the author instructions (max. 350 words; mentioning all *p*-values; giving the exact number of experiments, etc.). Note that the legend of the former Fig. 6 was already at the word limit.

We have now carefully revised and rearranged the figure legends, but we still have to compress them, and we feel that it is not possible to describe each panel individually without losing information.

Reviewer 3

Comment: In this paper, by using different apoptotic stimuli, the authors demonstrated that changes in phospholipid composition induced early in apoptosis attenuate AKT activation and enhance apoptosis. They showed that inhibition of fatty acid biosynthesis and a shift to beta-oxidation results in the accumulation of phosphatidylcholine containing polyunsaturated fatty acids species (PUFA-PC) in (peri)nuclear membranes. Consequently, PUFA-PC inhibits AKT recruitment and activation.

1. Phospholipids are the main structural components of plasma and intracellular membranes. The authors demonstrated that serum deprivation and exposure of fibroblasts to cytotoxic compounds such as staurosporine, cycloheximide, etoposide, valinomycin, thapsigargin, myrto-commulon A, and indirubin-3'-monoxime, results in a huge decrease of total phospholipid content. Particularly, after exposure, the amount of total phospholipids decreased to 25% as compared to those in control cells? Is this huge reduction of phospholipids associated with apoptosis?

Answer: Thank you for raising this important issue. We must apologize that, contrary to what was stated, the data in the previous (now revised) Fig. 1a were not normalized to cell number. The strong decrease in phospholipid levels was therefore mainly due to lower cell numbers under cytotoxic conditions. The actual decrease in phospholipid (revised Fig. 1a) and PC, PE, PS, and PG content per cell (Supplementary Fig. 3b) is instead less pronounced and significant only for TPG, VAL, and/or serum depletion at 48 h. Together, cell death induction results in an overall decrease in cellular phospholipids, but the effects are still moderate at this early stage. For potential mechanisms that reduce phospholipid levels in stressed cells, see new Fig. 4a-d and Supplementary Fig. 14, 17, 18a, d-g, 32a-c and page 17, last paragraph to page 19, first paragraph and page 30, last paragraph.

Comment: What was the viability of the cells after exposure?

Answer: The membrane intactness of washed adherent cells (without detached and dead cells) was either not or only weakly impaired by the cell death inducers within 48 h, as determined by trypan blue staining (Supplementary Fig. 2d). Note that this fraction of viable cells was also used to analyze lipid profiles, assess proteomic changes, and study signaling cascades (as indicated in the method section on page 52, first paragraph). We show by MTT assay (which does not discriminate between attached and detached cells) that the mitochondrial dehydrogenase activity is reduced for our cell death settings (new Supplementary Fig. 1d). Like other cell viability assays that are based on the formation of chromogenic products, the MTT assay does not discriminate between effects on cell viability and cell number, and we found that the latter actually decreased (Supplementary Fig. 1c). In addition, PI/Annexin-V staining of total cells indicates a decrease in the number of viable cells between 2 and 51% within 48 h, depending on the cytotoxic stimulus (Supplementary Fig. 2b and c).

Comment: Will inhibitors of apoptosis or other cell death pathways prevent this consumption of phospholipids?

Answer: For the majority of cytotoxic conditions, the pan-caspase inhibitor Q-VD-OPh neither attenuated the increase in PUFA-PC ratio (Supplementary Fig. 23c) nor the decrease in PC content (new Supplementary Fig. 23d). In recent years, we have made great efforts to elucidate the mechanisms that increase the proportion of PUFA-containing phospholipids. Our data indicate that cytotoxic stress attenuates growth factor-PI3K-Akt signaling (Fig. 5-7, Supplementary Fig. 19-32, Supplementary Data 1 and 2) and causes a shift from SFA and MUFA biosynthesis to fatty acid catabolism. In consequence, exogenous or released fatty acids (including PUFAs) are preferentially incorporated into phospholipids via ACSL isoenzymes (i.e., ACSL5 or ACSL4) and LPCAT3 (Fig. 2-4, Supplementary Fig. 10-18,

Supplementary Data 1 and 2), which elevates the PUFA ratio in membranes (Fig. 1, Supplementary Fig. 3-9) and increases their sensitivity to peroxidation (Fig. 8 and 9, Supplementary Fig. 39-44). Accordingly, apoptotic cells become more susceptible to ferroptosis (Fig. 8a-d and 10, Supplementary Fig. 39 and 40). Together, the systematic investigations guided by the reviewers led to the discovery of a mechanism that adds to the emerging link between apoptosis and ferroptosis. The negative regulation of Akt activation by PUFA-PC (on which the previous version of the manuscript was based) seems to support this mechanism, but is restricted to specific cytotoxic conditions (Supplementary Fig. 33-38).

Comment: 2. The observed decrease of phospholipid content can be due to activation of phospholipases, particularly phospholipase A2. Did the authors observe the accumulation of lysophospholipids? The involvement of lysophospholipids in the regulation of the PI3K-AKT pathway have been reported (PMID:25275042, PMID:24510253).

Answer: Point well taken and done. Quantitative proteomics indicates that cytosolic phospholipase A_{2α} (Pla2g4a) is upregulated under cytotoxic stress (Fig. 4a, b, d and Supplementary Fig. 17) and that free fatty acid levels increase (Fig. 1b and Supplementary Fig. 3b). However, lysophospholipids were not consistently enriched under the applied stress conditions (Fig. 18b, c), and we speculate that “Pla2 cleavage products”, i.e., lysophospholipids, might undergo “a rapid turnover under steady-state conditions”.

Comment: 3. For some reason, almost all data in the paper are presented as a %, ratio or fold change. For the sets of phospholipid data, no absolute values are provided. For the analysis of phospholipids, since the authors used internal standards, this should allow them to make an estimation of the amounts of phospholipids in cells. The data could then be normalized to total protein or cell number.

Answer: Done, we now provide absolute values of phospholipid species, subgroups and classes in figures (Fig. 1a, b, 2h, i, 6a, Supplementary Fig. 3a, b, 7a-c, e, 11b-d, 23d) or in the Source Data. Data are normalized to the cell number (unless otherwise noted).

Comment: 4. The authors demonstrate that exposure to cytotoxic compounds results in a decrease of PC containing saturated and monounsaturated fatty acids localized in the sn-2 position of the PC molecule. However, the content of PUFA-PC under the same conditions was increased. This increase in the content of PUFA-PC molecular species may result from calculation error. It is very well may be that these “changes” in the content of PUFA-PC are related to the decrease of total PC that was use as the denominator for their calculation. For this reason, it is important to present normalized absolute values for the phospholipid content for all phospholipids.

Answer: Done, we now additionally show in the new Fig. 1e and Supplementary Fig. 7c, d that the effect of cytotoxic stress on the total amount of PUFA-PC in fibroblasts ranges from a moderate increase to decrease. It should be noted that total phospholipid levels are indicative of the availability of cellular membranes. More critical for membrane architecture and peroxidation is the relative composition of phospholipids, or in other words, the density at which PUFAs are packed in the membrane. Therefore, we mostly refer to relative intensities (also called ‘proportion’ or ‘ratio’) normalized to the total amount of the phospholipid subclass. The relative intensity of PUFA-PC increased strongly and consistently under cytotoxic stress conditions (Fig. 1f). To avoid confusion, we replaced misleading phrases such as “accumulation of PUFA-PC” with “increase in [the] PUFA-PC ratio”.

Comment: 5. Is the phospholipid-remodeling process involved in PUFA-PC accumulation during apoptosis? What will be the effect of knocking down of lysophosphatidylcholine-acyltransferase (LPCAT3) or acyl-CoA-synthetase 4 (ACSL4) on the accumulation of PUFA-PC and suppression of the PI3K/Akt pathway?

Answer: Thank you for bringing this up. We have previously reported that LPCAT3 (LPLAT12) knockdown decreases PUFA-PC levels and enhances Akt phosphorylation in fibroblasts (PMID: 28687611). Here, we additionally show that LPCAT3 silencing attenuates the VAL-induced increase in PUFA-PC proportion (new Fig. 2j and Supplementary Fig. 12b). Since LPCAT3 knockdown per se slightly decreased the number of viable cells (new Supplementary Fig. 41), we did not follow this approach to investigate whether the proportion of PUFA-containing phospholipids is functionally linked to cell death induction, but instead lowered the cellular PUFA-PC ratio by supplementation of oleic acid (18:1) (new Supplementary Fig. 12a) and refer to our previous study (PMID: 35624087) showing that the fraction of cells undergoing necrosis (both with and without apoptotic component) was substantially reduced. In addition, we performed metabolic flux studies and now demonstrate that deuterium-labeled arachidonic acid (a preferred substrate of ACSL4, and after conversion to the CoA ester, of LPCAT3) is efficiently incorporated into membrane phospholipids of stressed cells and that this incorporation is further enhanced by inhibition of de novo fatty acid biosynthesis (new Fig. 2h, i and Supplementary Fig. 11b).

Comment: 6. Using liposomes the authors demonstrated that liposomes composed of arachidonoyl-PC suppressed AKT phosphorylation. What was the amount of integrated arachidonoyl-PC and where in the cell was it localized? What will be the effect of other exogenous PUFA-PC that have linoleic, eicosatentaenic, docosapenaenoic and docosahexaenoic in the sn-2 position? Will it be similar to the results obtained for arachidonoyl-PC?

Answer: The amount of PC(16:0/20:4) incorporated into fibroblasts was 0.16 ± 0.01 nmol/ 10^6 cells, as now indicated in the legend of Supplementary Fig. 33. We have previously published that PC(16:0/18:2) suppresses Akt activation in fibroblasts (Pein et al., 2017, *FASEB J.*) and are currently preparing a manuscript describing a role of PC(16:0/22:6) in overcoming tumor resistance by drugs targeting the cytoskeleton. In this project, we investigated the effect of PC(16:0/22:6) on Akt activation and obtained results comparable to those of 18:2- and 20:4-containing PC (Fig. C2a, b). Moreover, supplemented PC(16:0/22:6) was taken up by fibroblasts (0.058 ± 0.003 nmol/ 10^6 cells) (Fig. C2c) and distributed throughout intracellular membranes (Fig. C2d). Notably, PC(16:0/22:6) was most enriched in the (peri)nuclear cell fraction (Fig. C2d). PC species containing 20:5 and 22:5 are not commercially available.

Fig. C2. Cellular uptake and subcellular distribution of PC(16:0/22:6) and impact on Akt phosphorylation. NIH-3T3 cells were treated with liposomes consisting of PC(16:0/22:6), PC(16:0/16:0), PC(17:0/17:0), PC(16:0/18:1), PC(16:0/18:2), or PC(16:0/20:4) (120 μM , each, if not indicated otherwise) for 48 h. **a, b** Akt phosphorylation; PL, phospholipid, n.s., not significant. Western Blots are representative of five independent experiments. **c** Cellular uptake of PC(16:0/22:6) and the saturated control PC(16:0/16:0). **d** Distribution of 22:6-containing PC species in subcellular fractions. Mean + s.e.m. (**d**) and single data (**a, b, c**) from $n = 17$ (**a**, left panel), $n = 3$ (**a**, right panel, PC(16:0/16:0), **c, d**), $n = 4$ (**a**, right panel, PC(16:0/22:6)), $n = 5$ (**b**) independent experiments. One-way ANOVA + Tukey HSD post hoc tests (**a-c**) or student *t*-test (**a**).

Comment: 7. It is not clear how PUFA-PC affects the activation of AKT. Do any protein/lipid interactions between AKT and PUFA-PC play a role in the suppression of AKT phosphorylation and therefore acceleration of apoptosis?

Answer: It is known that PUFA-PC decreases the affinity of Akt for its membrane anchor PI(3,4,5)P₃, thereby suppressing the membrane translocation of Akt upon stimulation (PMID: 23359699). In a cell-free system, we investigated the binding of purified Akt to liposomes containing PI(3,4,5)P₃ and varying proportions of PUFA-PC. Furthermore, we reported that Akt membrane translocation is impaired in the cellular context and excluded a role of PI(3,4,5)P₃ in the phospholipid-dependent regulation of Akt (PMID: 23359699). Here, we investigated whether PUFA-PC interacts directly with the kinase by co-immunoprecipitation of Akt followed by analysis of protein-bound phospholipids. The proportion of PUFA-containing PC species was substantially reduced on immunoprecipitated Akt

compared to total cells, whereas saturated PC species were enriched (Supplementary Fig. 35). It is therefore unlikely that PUFA-PC interacts directly with Akt, and we hypothesize that it rather disrupts micro- or nanodomains that are essential for Akt organization and function (PMID: 18641634). These aspects are now discussed on page 31, first and third paragraphs, and in Supplementary Note 13.

Comment: 8. The phosphatidylinositol-3-kinase -AKT (PI3K-AKT) is an important intracellular signaling pathway in regulating cell proliferation, differentiation and apoptosis. Upon stimulation, AKT is recruited to the plasma membrane through binding to PI-3,4,5-P3, where it can be phosphorylated that results in its activation. In addition, the anti-apoptotic effect of the PI3K-AKT pathway through translocation of phosphorylated AKT to mitochondria has been documented (PMID:30165359). Mitochondria are a major player in apoptosis. One of the early events of apoptosis is the oxidation (PMID:16408039) and externalization (PMID:24300280) of the mitochondrial phospholipid - cardiolipin that results in the release of cytochrome c to the cytosol and subsequent events for the execution of apoptosis. In addition, apoptosis associated with the translocation of cardiolipin into the plasma membrane was also documented (PMID: 26491702, PMID: 15181455). Similar to PI-3,4,5-P3, cardiolipin is a negatively charged phospholipid that can make its interaction with AKT possible. Were any changes in cardiolipin content detected?

Answer: Thank you for sharing this hypothesis with us. We analyzed cardiolipins of stressed fibroblasts using our recently established UPLC-MS/MS method. In contrast to other phospholipids, the amount of cardiolipins (both saturated and unsaturated) was markedly increased under our cytotoxic settings (new Supplementary Fig. 3a and 7e), which was accompanied by substantial changes in the cardiolipin fatty acid composition (new Supplementary Fig. 5b). The potential impact of an altered cardiolipin content on Akt signaling is now discussed on page 45, lines 19-21 and in Supplementary Notes 12 and 13.

Comment: Moreover, it is not clear why the study was focused on (peri-)nuclear membranes while completely ignoring the plasma membrane and mitochondria.

Answer: Done, we have rewritten and restructured the relevant sections (page 10, second paragraph, page 26, lines 22-24) and deleted our speculations based on the former Fig. 2e. The restructured manuscript no longer focuses on the subcellular distribution of p-Akt. Our original aim was to investigate whether cytotoxic stress (i.e., VAL treatment) increases the PUFA-PC ratio at the subcellular site where p-Akt levels are reduced, which turned out to be the (peri)nucleus (former Fig. 2e). Changes in p-Akt levels were less dramatic for the plasma membrane and not obvious for mitochondrial structures. Subcellular fractionation then confirmed that PUFA-PC ratios did indeed decrease in the (peri)nuclear fraction (Fig. 1g, h and Supplementary Fig. 8). Note that this effect is not specific to the (peri)nuclear fraction, as the non-nuclear fraction experienced a comparable decrease in PUFA-PC (Fig. 1g, h). Taken together, in the first version we focused on the subcellular locale where we observed a decrease in p-Akt levels.

In an independent project, we investigated the subcellular distribution of PUFA-PC in fibroblasts that were supplemented with the PUFA-PC species PC(16:0/22:6). The proportion of PUFA-PC was highest in the (peri)nuclear fraction, but also increased substantially in fractions containing mitochondria and the plasma membrane upon treatment with PC(16:0/22:6) (see Fig. C2 above).

Comment: 9. The method for liposome preparation is not sufficiently detailed. The conditions, such as the amount of phospholipids, the frequency used for sonication procedure as well as the model of the sonication unit needs to be provided. It is not clear why the sonication procedure was performed at 50°C.

Answer: Done, we now describe on page 66, last paragraph, that “lipid vesicles were formed by vigorous mixing of PC(16:0/16:0) (360 nmol) or PC(16:0/20:4) (360 nmol) in 3 ml DMEM (for serum depletion) plus FCS (10%) (for all other experiments) and sonication at 35 kHz and 50°C (above the

phase transition temperature of PC(16:0/16:0)) using a Sonorex RK512H ultrasonic bath (Bandelin, Berlin, Germany).” In addition, we describe an alternative procedure for PI(18:1/18:1) (150 nmol) and PI(16:0/16:0) (150 nmol) using an Ultrasonic Cleaner USC100TH at 45 kHz and 50°C.

Comment: It has been demonstrated that phospholipids are susceptible to degradation during sonication. These include oxidation of PUFA chains and hydrolysis of ester bonds that results in the generation of lysophospholipids and oxygenated PUFA (PMID:1814643). Thus, the LC/MS analysis of prepared liposomes has to be performed before addition to cells.

Answer: Done, we analyzed potential degradation and oxidation products of PC(16:0/22:6) after sonication. Hydrolysis of PC(16:0/22:6) to the lysophospholipid PC(0:0/22:6) is marginal (Fig. C3a), and only a small fraction of the phospholipid is oxidized with incorporation of 1 to 6 oxygen atoms (< 3%) (Fig. C3b). Note that the lysophospholipids and oxidation products are not exclusively derived from PC(16:0/22:6) degradation, but are also present in the supplemented FCS. In particular, we detected substantial amounts of PC(16:0/0:0), which is 20-fold more abundant than PC(0:0/22:6) in plasma (PMID: 20671299). The characterization of PC(16:0/22:6) vesicles shall be included in a separate manuscript that is currently in preparation.

Fig. C3. Hydrolysis and oxidation products of PC(16:0/22:6) upon sonication. Lipid vesicles were prepared by sonication of a PC(16:0/22:6) solution (120 μ M) in DMEM (for serum depletion) plus FCS (10%). Lysophospholipids [PC(0:0/22:6) and PC(16:0/0:0)] (a) and oxidation products (+14 Da: [1O-2H]; +16 Da: [1O]; +32 Da: [3O]; +64 Da: [4O]; +96 Da: [6O]) (b) were analyzed by UPLC-MS/MS. Mean (a, b) + s.e.m. (b) from $n = 3$ technical replicates.

Comment: In addition, the amount of lipids integrated into plasma membranes or intracellular compartments needs to be estimated and presented in the paper.

Answer: Done, the legend of Supplementary Fig. 33a now indicates the increase in the total amount of PC(16:0/16:0) and PC(16:0/20:4) in fibroblasts that were exposed to the respective liposomes. In addition, Fig. C2c shows the amount of phospholipids incorporated into fibroblasts, and Fig. C2d provides insight into the distribution of PUFA-PC species in subcellular fractions. As mentioned above, these data shall be included in a manuscript describing the tumor-sensitizing mechanisms of PC(16:0/22:6).

Comment: 10. “PL-PUFA” has to be used instead of “PL-bound PUFA”.

Answer: Done, we have replaced the term “PL-bound PUFA” with “PUFA-containing phospholipids”.

Reviewer 4

Comment: I have read with pleasure the submitted manuscript which is a significant embodiment of work about lipid metabolism, apoptosis and signalling. I had difficulties in understanding part of the work; however, probably just because of my expertise is tangential to this work. Thus, I will focus my report on the analysis of imaging data, as originally asked by the Editor. I am rather acquainted with non-linear and time-resolved microscopy. However, I do not believe that the specific assay the authors implemented for this report is very common and I have missed justifications for several experimental choices the authors have done. I have also read Ref. 69 to find additional details but I could not find satisfactory explanations — this lack of clarity obstacles my capability to interpret the results.

Answer: Thank you for this kind assessment. While we consider the technology to be very valuable for studying the localization of low abundant phospho-proteins, we agree that our approach is uncommon. Therefore, we have decided not to show our p-Akt imaging data in the current manuscript (as suggested by the reviewer), but to further improve the experimental setup based on the constructive feedback.

Comment: From material and methods, I understand that cells were FA-fixed, immunostained for phospho-Akt, stained with DAPI and mounted. Excitation of fluorescence was done at 830nm and the fluorescence emission was detected with a 458/64 nm bandpass filter. Is this correct? The bandpass filter is optimal for NADH imaging but poorly configured for Alexa488 with a cutoff wavelength of the filter at 490nm and a peak of Alexa488 further shifted to the red. Furthermore, DAPI is also emitting in the same spectral band. The two-photon cross-section of DAPI is negligible. However, the same is true for Alexa488. I am inclined to think I am misunderstanding the setup, but after several readings of the material and methods and supplementary notes I cannot see how. To summarize so far, I have the impression that three fluorophores (the endogenous NADH, DAPI and Alexa488) are emitting fluorescence in the analysed bandwidth, with NADH being naturally dim, Alexa488 being detected in the wrong emission band, hence emitting a low fluorescence signal and a possible contaminant of the signal coming from DAPI. I will neglect the contribution from DAPI, as nuclei are correctly dim in the images submitted. However, I struggle to understand the choice of filter. Are material and methods correct? A diagrammatic representation of the setup might be useful.

Answer: We highly appreciate the reviewer's feedback. As mentioned above, we have removed all data related to p-Akt imaging, including the former Fig. 2e and f and Supplementary Fig. 8. In addition, we corrected the method section, which mistakenly stated that cells were stained with DAPI, which was, however, not the case.

Comment: Also, the authors utilize FLIM for unmixing, where FLIM-based unmixing seems unnecessary. As the samples are fixed, why not doing sequential imaging of NADH and phospho-Akt, utilizing a different more red-shifted secondary antibody? If hardware limitations did not permit the authors this arrangement, at least the bandpass filter could have been replaced (swapped) for appropriate imaging of the phospho-Akt signal.

Answer: As mentioned above, we have removed the section related to p-Akt imaging.

Comment: In the data analysis, the authors are then forced to perform a multi-exponential fit trying to unmix the signals. I have additional concerns on regard of the data analysis. The authors declare to require a three exponential fit to achieve unmixing of NADH (bound/unbound) and Alexa488, which seems then to exhibit also a double exponential decay. The number of photons required to fit a double exponential decay is approximately 10,000 (see Koellner&Wolfrum in ' How Many Photons Are Necessary For Fluorescence-Lifetime Measurements' and related literature). The authors declare to stop acquisition of images whenever the brightest pixel reaches 1,000 photon counts. It is however

unclear which is the typical threshold they used, i.e. it is unclear which is the minimum photon count they allowed to fit the data.

Answer: As mentioned above, we have removed the section related to p-Akt imaging. Still, we have used a 5x5 kernel for fitting the data, which results in up to 25000 photons per decay trace, which is sufficient for a biexponential fit. The fitting procedure has been focused on bright pixels only by using a threshold parameter of 5 or 8 (as now described in Supplementary Note 16). This means, that pixels have been excluded from the fit, in which after summing of the photons by 5x5 binning less than 5 or 8 photons are in the time bin with the maximum number of photons.

Comment: I miss information that is necessary to understand the quality of the data, as typical exposure time, pixel resolution, and it is unclear if a binning factor of 2 was the parameter that can be setup in B&H software or the literal meaning of a 2x2 kernel. I will assume that this was the value for B&H, which should result in a 5x5 convolution with a summing kernel of the image. The boost of 25 fold of the signal, would bring the max signal fitted to about 25,000 photon counts, i.e. sufficient for bi-exponential analysis. All these details should be described.

Answer: As mentioned above, we have removed the section related to p-Akt imaging. But the referee is correct. The binning parameter of 2 refers to the B&H software, resulting in a 5x5 convolution to enhance the number of photons for the biexponential fitting procedure. For fitting, a multi-exponential decay model has been used. Pixels have been excluded from the fit, in which after summing of the photons by 5x5 binning less than the threshold parameter number of photons are in the time bin with the maximum number of photons. The pixel resolution of the data shown in the new manuscript is 220 nm using an image size of 512 x 512 pixels and 112.5 μm field of view (as now stated in the legend of Supplementary Fig. 42). The diffraction limited resolution using an objective of NA 1.1 and 670 nm excitation is about 300 nm. The pixel dwell time was 2.56 μs per frame. Many frames have been averaged until 1000 photons have been collected in the brightest pixels. This corresponds to accumulating 648 frames (figure 9 b left) or 263 frames (9 b right).

Comment: In Fig. 2 and Supp. Info, I do not see LUTs so it is difficult to interpret the meaning of the blue colour. In light of the significant binning and lack of other details, I am therefore incapable to understand the data. At a first read, the images seem nice and to support the authors' interpretation, but at a more in-depth analysis I am not capable to shape an opinion around it. Please clarify the nature of the representation and provide not overlaid images at least in Supp. Info.

Answer: As mentioned above, we have removed the section related to p-Akt imaging. The LUTs have been added to the figures shown in the new submission.

Comment: Although the signal seems sufficient for the two-exponential fit, it is clearly insufficient to resolve 3 to 4 molecular species.

Answer: As mentioned above, we have removed the section related to p-Akt imaging.

Comment: The following section of the Supp. Note 3 is very difficult to understand: "In order to localize the antibody signal, which is on average lower than the NAD(P)H signal, we used the co-localization approach plotting the 2D-histogram of the lifetime images. The 2D-histogram of the lifetimes t_1 and t_2 is characterized by two pixel clouds characteristic for NAD(P)H emission ($t_1 \sim 300$ ps, $t_2 \sim 2300$ ps) and Alexa488 ($t_1 \sim 1000$ ps, $t_2 \sim 4100$ ps). The pixel cloud characteristic for antibody emission has been back projected onto the fluorescence intensity image using Fiji7."

Answer: As mentioned above, we have removed the section related to p-Akt imaging.

Comment: The 2D histograms of Tau1 and Tau2 should be shown. It is impossible to assess the technique otherwise. However, this technique would work only if the signals are spatially segregated, which is not the case. The signals are mixed and unmixable with the use of a single spectral detection band, limited photon budget and a 2-exponential fit. The authors should provide a robust validation of

the technique as this is new and potentially contradicting the experience of those familiar with similar approaches.

Answer: As mentioned above, we have removed the section related to p-Akt imaging.

Comment: The validation seems to be limited to Supp. Fig. 8. It is not clear to me the histogram in panel c. The green curves are marked as 'without anti-p-Akt', but histograms very similar to the others are shown.

Answer: As mentioned above, we have removed the section related to p-Akt imaging.

Comment: Eventually, the authors use this data only to state that the chemical environment of 'p-Akt is modulated' by chemical treatment. However, the authors cannot infer what these changes are reporting about. Cells are fixed and stained with an antibody. Therefore, the authors should have excluded that these shifts are unspecific by analysing fluorescence emitted by one antibody targeting a different protein. Even in this case, I am unsure about the value of reporting such a shift of biochemical environment with no clear relation to the rest of the work.

Answer: As mentioned above, we have removed the section related to p-Akt imaging.

Comment: I am aware of how much work this type of assays require, and I do not write this lightly. Without additional information and validation, I do not have confidence in the methodology, and I question its biological relevance. Therefore, I would recommend omitting these results as they might be weakening the manuscript, overall, rather than strengthening it.

Answer: Thank you very much for your constructive comments. Following your suggestion, we omitted the criticized data (former Fig. 2e and f and Supplementary Fig. 8) from the manuscript and will continue to work on the validation of this powerful method.

Response to the reviewers

Reviewer #5:

Comment: The manuscript “Attenuated growth factor signaling during cell death initiation sensitizes membranes towards peroxidation” by Gollowitzer et al. suggested that fibroblasts stimulated with a variety of apoptosis-inducing agents display an increase in cellular PUFA-phospholipids and become susceptible to peroxidation under (associated) redox stress. This reprogramming in fibroblasts is caused by RTK-PI3K-Akt axis, which is recognized as a central pathway promoting lipid anabolism and cell survival. Cytotoxic agents inhibit growth factor receptor tyrosine kinase (RTK) and phosphatidylinositol-3-kinase (PI3K)/Akt signaling and results in a switch from fatty acid biosynthesis to degradation. Under these conditions, cells become dependent on exogenous or intracellularly released fatty acids for phospholipid remodeling, likely involving long-chain acyl-CoA synthetase (ACSL)5 and lysophospholipid acyltransferase (LPLAT)12/LPCAT3, thereby accumulating PUFAs in membranes and increasing susceptibility to peroxidation. Authors used cytotoxic inducers with different modes of action to claim that this mechanism works in all kind of stress and involved all phospholipids present in cells.

Answer: Thank you for your careful review and constructive feedback. While our main findings are accurately summarized above, we disagree with the derived claims. Our data clearly show that the described mechanism applies to many (but not all) cytotoxic stress conditions and that there are strong differences in the responsiveness between phospholipid classes. These findings are critically described in the Results section and Supplementary Notes, but later translated into more accessible conclusion statements in an effort to reduce complexity. We apologize that these statements may have caused confusion and have carefully revised them throughout the manuscript. For example, we have replaced "phospholipids" with "phosphatidylcholine" in these statements, and we have more frequently mentioned the concrete cytotoxic stress conditions or the number of different stresses that cause the effect, or more clearly stated that only "some/many/multiple" cytotoxic stress conditions cause the described effect. In addition, we now more carefully distinguish between the consistent suppression of RTK-PI3K signaling under cytotoxic stress and the suppressive effect on downstream Akt signaling found for some (but not all) cytotoxic stressors.

Comment: Overall topic of the paper is very exciting and some of the data are intriguing, but results obtained in different type of experiments demonstrated, that cell responses sometimes didn't follow mechanism suggested by authors. It appears that author's hypothesis works only for some cytotoxic inducer and involve only particular lipids. Presence of data, that doesn't support authors idea don't help reader to understand what is going on. I think that it's reasonable to concentrate on stress agents that worked and on lipid that really changed in these conditions.

Answer: We appreciate the positive feedback and the kind suggestion to improve the clarity. While we consider it very important to also show that cytotoxic stressors, despite a common mechanism, remain very heterogeneous (as expected from multiple targets and metabolic cross-links), we agree with the reviewer that the focus should only be placed on those stressors that actually consistently induce apoptosis to avoid confusion (see also our responses to the comments below). Specifically, we have now removed throughout the manuscript the control TNF α (which sensitizes to apoptosis but does not induce apoptosis per se) and the data on I3M, which reduced cell number (former Supplementary Fig. 1c) but did not efficiently induce apoptosis (former Supplementary Fig. 2b, c).

Major comments:

Comment: 1. Results of experiments to study the induction of programmed cell death in fibroblasts by cytotoxic agents covering a broad mechanistic range presented at Supplementary Figures 1 and 2.

Measurements of cell death by staining with PI and annexin-V demonstrated, that not all cytotoxic agents used in these experiments are effective in induction of cell death. Treatment with VAL MC, ETO, STS, serum deprivation induced very small effect (~15%), while TNF alpha, TPG CHX, I3M were not effective. Other results presented by authors demonstrated that cytotoxic agents decrease cell numbers, and viability, but it can be the result of decreased proliferation,

Answer: Thank you for sharing your concern. We were interested in the very early (almost pre-apoptotic) effects of cytotoxic stress on the cellular lipidome. Therefore, the concentration of cytotoxic agents was deliberately chosen in the sensitive range where cell death is not fully underway but barely initiated. To this end, we have performed concentration-dependent studies and selected optimized concentrations based on cell morphological features such as blebbing, nuclear shrinkage and/or detachment, as now mentioned on page 6, lines 12-14. This careful selection of concentration is important to avoid late apoptotic cells with disrupted membranes, as they release lysosomal hydrolases that degrade phospholipids with massive effects on phospholipid content and composition. To minimize such confounding effects, we even intensively washed the adherent fibroblasts for lipidomic analysis and signaling studies to remove any (partially) detached cells (indicating apoptosis induction), as described on page 53, second paragraph. To better illustrate the validity of this approach, we now performed concentration-dependent experiments on vehicle- and VAL-treated fibroblasts and analyzed apoptosis induction by PI/annexin V staining (new Supplementary Fig. 2d, e). Consistent with our expectations, the concentration of VAL used here (10 μ M) represents a threshold concentration above which cell death is effectively induced.

Comment: Results presented in supplementary Fig 25 b also demonstrated very low percentage of cell death. In most cases reading is not more than 6%, which is usually considered normal result for control cells. Only GSK+VAL produce ~12% of apoptotic cells and 6% of necrotic). I also have similar concern about results presented on supplemental Fig.26

Answer: As described in the previous comment, our focus was on the onset of cell death, when cells are still largely viable - a time point that dictates the further fate of the cells. In addition, to further enrich for cells in the early phase of cell death, we removed dead, detached cells by washing prior to flow cytometric analysis. Although the number of apoptotic/necrotic cells is still low, the increase in their proportion is already highly significant. Statistics versus vehicle control for each time point have now been added to Supplementary Figs. 26c and 27a.

Comment: 2. One of the main ideas of this manuscript is suggestion, that cells stimulated with different apoptosis-inducing agents follow the same rule and display an increased level of cellular PUFA-phospholipids. Results presented on several figures demonstrated, that opposite to authors suggestions changes in phospholipids under stress conditions are not similar and PUFA content increased not in all lipids.

Answer: It was never our intention to make claims that encompass all cytotoxic stress conditions or all phospholipids (see also our comment above). What we do see is that cytotoxic stressors consistently (though to varying degrees) increase the proportion of PUFAs in phosphatidylcholine (PC), whereas the non-cytotoxic control TNF α (a sensitizer but not an inducer of apoptosis in NIH-3T3 cells; PMID: 12119420; not shown anymore) has no effect. We have now carefully revised the manuscript to remove any ambiguous or misleading statements.

Comment: i) Supplementary Fig. 3b summarizes the kinetic changes in the cellular content of individual lipid subclasses, i.e., phosphatidylcholine (PC), phosphatidylethanolamine (PE), PS, phosphatidylglycerol (PG), and FFA. This picture contains a lot of information (different cytotoxic agents, different time points, different lipids) and it's difficult to understand what is going on. But it can be clearly seen effects of stress agents on amount of different lipids are not similar and can involve different mechanisms.

Answer: Exactly, this is the main message of Supplementary Fig. 3b. In combination with Fig. 1a, it shows that some cytotoxic conditions decrease the total amount of phospholipids within 48 h, while others are hardly affected, and that there are also differences between phospholipid groups. Therefore, changes in the total amount of phospholipids cannot easily explain the consistent increase in PUFA-PC levels between 6 and 48 h shown in Fig. 1f, left panel. We have adjusted the text on page 6, last paragraph to make this point clearer.

Comment: ii) Fig. 1d To demonstrate, that cytotoxic stress elevates the proportion of PUFAs in phospholipids authors present Volcano plots indicating changes in lipid composition after treatment with VAL or MC. These results demonstrated, that changes only in certain lipids (mostly PC and PE) follow this suggestion, while changes in biggest part of lipids are not statistically significant.

Answer: Yes, this is the main finding of our initial lipidomic analysis, as now more clearly indicated on page 8, last paragraph. All cytotoxic stress conditions result in an increased proportion of PUFA-PC between 6 and 48 h of exposure (Fig. 1f, left panel). Several of the stress conditions (but not all) have a similar effect on PE and some also on PS and PI composition (Fig. 1f). The volcano plots in Fig. 1d illustrate which phospholipid species are most strongly and significantly regulated. It should be noted that we used adjusted p-values corrected for multiple comparisons for this untargeted presentation. Consequently, moderate changes in individual phospholipids (significant by two-tailed paired t-test) are no longer significant after correction to avoid over-interpretation of the results. This limitation does not apply to the hypothesis-driven data presentation used in Fig. 1f, where the subset of PUFA-containing species is shown, explaining why PUFA-PS and PUFA-PI can significantly increase in Fig. 1f, although only one major PUFA-PS species is highlighted in Fig. 1d.

Comment: iii) Supplementary Fig. 4 presents effects of cytotoxic agents other than VAL and MC on phospholipid composition. Results demonstrated, that again changes involve mostly PC and PE and can be found only for samples treated with CHX, STS and TPG, while other agents didn't induce statistically significant changes.

Answer: Correct, this is again a matter of how the data are presented and which statistics are used, as described above. When the phospholipid species are grouped according to individual fatty acids (Supplementary Fig. 6) or all PUFA-containing species are combined (Fig. 1f) and an ANOVA is used for statistical calculation, the effects of ETO and I3M (now removed) are also highly significant. The volcano plots in Supplementary Figure 4 show significant effects for at least two PUFA-containing phospholipid species for STS, CHX, TPG and serum depletion, but not for ETO and I3M treatment (not shown anymore) nor for the negative control TNF α (not shown anymore), because p-values were corrected for multiple comparisons with a false discovery rate of 5%. Please note that we could have accepted a false discovery rate of 10% and would have obtained significant results for all cytotoxic stressors. The second point of the reviewer's comment regarding the specific role of PC, which is preferentially enriched in PUFAs, has already been addressed in our response to the previous comment.

Comment: iv) Results presented on Supplementary Fig. 6 demonstrated, that cytotoxic stress-induced increase in amount of 20:3 20:4 22:5 22:6 fatty acids in PC, while amount of Linoleic acid, which is one of the major polyunsaturated fatty acid didn't change.

Answer: Indeed, it is the proportion of PUFAs with at least three double bonds that increases in PC under cytotoxic stress, whereas the proportion of linoleic acid (18:2) even tends to decrease slightly for some treatments (Supplementary Figure 6). This finding is of particular interest because these are the PUFAs that are most sensitive to membrane lipid peroxidation, whereas 18:2 plays a minor role in this context (PMID: 19705847). We now highlight this aspect more prominently on page 9, lines 10-12. Our decision to include 18:2 in the PUFA-phospholipid scores therefore results in an underestimation

of the effect of cytotoxic stress on the PC fraction with highly unsaturated fatty acids, again intentionally made to avoid over-interpretation of the peroxidation-sensitizing effect.

Comment: In general it can be suggested, that authors statement about increase of PUFA amount in all lipids under stress conditions is not true and describe behavior of mostly PC and PE.

Answer: We fully agree with the reviewer's conclusion, which is exactly what we wanted to express in our previous version. It was never our intention to claim that PUFAs increase in all phospholipid classes. We stated that PUFAs increase in phospholipids (to keep it simple) and provided the details of which phospholipids are affected in the results section. As described above, we have now carefully revised statements throughout the manuscript that were intended to simplify the main findings for the non-expert readership.

Comment: 3. Several conclusions made by authors in the individual experiments are in my opinion- not supported by the data. i) Describing Fig.1a authors stated: "Multiple cytotoxic conditions decreased the cellular content of glycerophospholipids, reaching significance for TPG and VAL.." It's not completely true. Only results obtained with VAL and serum deprivation can be taken into consideration. Differences obtained after treatment with other agents are not statistically significant. Results after treatment with TPG having $p=0.0640$ are also not significantly different from control. So, authors statement is true only for two conditions out of nine.

Answer: Done, we revised the statement and now write: "Of the six cell death inducing stress conditions, VAL significantly and three others (MC, STS, TPG) tendentially reduced the cellular content of glycerophospholipids (Fig. 1a)". As mentioned above, $TNF\alpha$ is a negative control (not shown anymore) and cannot be counted as a cytotoxic stressor, and I3M (also not shown anymore) must also be taken with caution since it did not (yet) initiate cell death according PI/annexin V staining (former Supplementary Fig. 2b, c). We considered these two treatments as controls and valuable for comparison, and had given them a different color than the six core cytotoxic stressors in the previous version of the bar charts. We have also not mentioned serum starvation here because we discuss it separately in the result section 'Consequences of the withdrawal of growth factors'.

Comment: ii) Describing Fig 2 authors stated: " De novo fatty acid biosynthesis was moderately to severely reduced for all cell death conditions after 48 h (Fig. 2a), whereas the rate of β -oxidation (Fig. 2b) as well as the cellular levels of the β -oxidation intermediate butyryl-CoA were markedly increased (Fig. 7 2c). Claimed by authors changes occurred not in all conditions tested. Changes are not significant for $TNF\alpha$, STS, MC, I3M on Fig 2a and for TNF and I3M on Fig 2c.

Answer: Done, all effects are now significant for the refined dataset. As described in our responses to the reviewer's comments above, $TNF\alpha$ is a negative control (not shown anymore), and I3M (also not shown anymore) does not readily induce cell death and therefore cannot be unambiguously classified as a cytotoxic stressor under our experimental conditions. Thus, all seven core cytotoxic stressors (including serum deprivation, which is described later) significantly decrease fatty acid biosynthesis in Fig. 2a and upregulate butyryl-CoA levels in Fig. 2c. The new Fig. 10 now provides a quantitative overview of the number of stressors with significant effects on the displayed parameters.

Comment: iii) In description of Fig.3b authors wrote: " The mRNA levels of ACC and FAS were reduced after 24 and/or 48 h under different cytotoxic conditions". This statement is also not true. In these experiments authors tested nine cytotoxic agents. Level of mRNA for FAS was not reduced in four and seven cases after 24 h and 48 h of incubation respectively. *Comment:* iv) Results obtained in experiments to study phosphorylation of ACC and AMP protein kinase and changes in ADP/ATP demonstrated, that only half of stress agents used in this experiments (TPG, VAL, serum depletion, and MC) worked according to authors description.

Answer: Many thanks for this comment. We have now performed statistical analysis for ACC and FAS mRNA expression at 24 and 48 h (new Fig. 3b with statistics added), revealing significant repression at

24 h for 6 (ACC) or 5 (FAS) out of 7 cytotoxic conditions, which may make the trends for VAL, MC, CHX, ETO, TPG, and serum depletion on ACC and/or FAS protein expression shown in Fig. 3c, d more reliable. We now write: "The mRNA levels of ACC or FAS were reduced after 24 and/or 48 h under the cytotoxic conditions (except for STS; Fig. 3b)". Nevertheless, the point of the reviewer is well taken. Only some cytotoxic stressors decrease either ACC or FAS (protein) levels, induce ACC and AMPK phosphorylation, deplete substrates for fatty acid biosynthesis, or cause changes in the ADP/ATP ratio, but there is definitely no common effect shared by all stressors. Somewhat more common to the stressors is the decrease in the protein expression of SCD1 (significant for CHX, TPG, VAL, by trend for ETO, serum depletion, and MC; Thürmer et al. - Fig. 3e), which we published in 2022 (PMID: 35624087) for the same set of cytotoxic agents under exactly the same experimental conditions. SCD1 is a $\Delta 9$ -desaturase that converts saturated to monounsaturated fatty acids.

Fig. 3e in Thürmer et al., 2022 (PMID: 35624087).

Thus, significant effects on one of the above-mentioned parameters were observed for CHX, VAL, TPG, serum depletion and MC, but not for STS and ETO, although the latter tended to reduce ACC, FAS and SCD1 expression along with malonyl-CoA levels. Again, it is ETO that is only moderately active in this context, as it also showed only moderate effects on Akt phosphorylation and the increase in PUFA-PC ratios, in line with an important role of reduced de novo fatty acid biosynthesis in increasing PUFA-PC ratios. This leaves STS, which significantly but not efficiently reduces de novo fatty acid biosynthesis (Fig. 2a) and increases the cellular PUFA-PC ratio through unknown mechanisms. In the sentence of the previous version quoted by the reviewer, we wanted to condense this observation by referring to "different" rather than "all" stressors. To make this important section clearer, we now explicitly mention on page 17, first paragraph, which stressors significantly interfere with one of these multiple steps, which do so only by trend, and which must have a different mechanism of action to increase the PUFA-PC ratio.

In addition, we now also show in the new Fig. 2e that selective inhibition of SCD1 increases the proportion of PUFA-PC.

Comment: v) Describing changes in phosphorylation of AKT authors stated, that many cytotoxic agents diminished Akt phosphorylation at Ser473 within 6 to 48 h (Fig. 6b-d, Supplementary Fig. 19a, b). Results presented on fig.6d demonstrated that four agents (TNF, CHX, ETO, serum deprivation) from total nine had no effects. That constituted ~40 % of tested conditions.

Answer: Correct, CHX and serum depletion decreases Akt phosphorylation by trend only and etoposide has no effect. In addition to TNF α as a negative control (not shown anymore), STS cannot be used to study phosphorylation because, as a pan-kinase inhibitor, STS affects phosphorylation independently of cell death induction. As a result, three out of six stress conditions significantly decreased Akt phosphorylation (50%). We previously wrote that "many" stressors "decreased Akt phosphorylation at Ser473 within 6 to 48 h" and have now revised this statement to precisely refer to "three of six stressors" that significantly decrease Akt phosphorylation. In addition, the manuscript now correctly

states that cytotoxic stress decreases PI3K signaling (Fig. 6a) and that only specific stress conditions additionally interfere with Akt activation in fibroblasts.

Comment: vi). Another inconsistency is present in description of the effects of cytotoxic agents on proteins involved in insulin and growth factor signaling including direct and indirect targets of Akt, such as glycogen synthase kinase (GSK)-3 β , mechanistic target of rapamycin (mTOR), and 70 kDa ribosomal S6 kinase (p70-S6K) (Fig. 6f). Again, not in all cases author's statement is true. Amount of GSK-3 β was increased after addition of only four agents (TPG, VAL, MC and serum deprivation).

Answer: Yes, GSK-3 β phosphorylation is significantly increased in four of in total six cytotoxic stress conditions, when ignoring the negative control TNF α (not shown anymore), the pan-kinase inhibitor STS and the inconclusive non-cytotoxic stressor I3M (as detailed in previous comments; not shown anymore). To be more precise, we now write that "Regulatory phosphorylation or substrate turnover of other insulin/growth factor-regulated kinases was not decreased by cytotoxic stress, but rather increased within 24 to 48 h, reaching significance in four out of six cytotoxic stress conditions for p-IkBa (as a marker of IKK activity) and inactive p-GSK-3 β (Fig. 6f)."

Comment: vii). The same in Fig. 6b, c where authors wrote: "As levels of p-Akt (used here as an indicator of the cytotoxic metabolic switch) decreased between 6 and 48 h, the PUFA ratio in phospholipids increased proportionally". Good correlation was present only for TPG, VAL, serum deprivation, MC, I3M (five agents from nine used in experiment) and only in 48h.

Answer: As explained above, TNF α (not shown anymore), STS, and I3M (not shown anymore) cannot be considered here, leaving good correlations for four of the six stressors. Nevertheless, the reviewer is absolutely right that changes in Akt phosphorylation cannot be the complete answer (at least not for all stressors) for increased PUFA-PC levels. While we now show that selective inhibition of IGF1R, PI3K, Akt, and SREBP1 increases the cellular PUFA-PC ratio (new Fig. 6e), it was never our claim that all cytotoxic stressors mediate this effect via Akt, as now more clearly stated throughout the manuscript. To more accurately describe the results, we now write: "As the levels of p-Akt (used here as an indicator of the cytotoxic metabolic switch) decreased between 6 and 48 h for VAL, MC, TPG and, by trend, serum depletion, the PUFA ratio in phospholipids increased proportionally for all six of the seven stressors and had a trend for the remaining (TPG) (Fig. 6b-d)". Note that the remaining stressor, ETO, which did not decrease Akt phosphorylation, was also the weakest in increasing PUFA-PC ratios, supporting rather a critical (though not exclusive) role of Akt suppression in increasing PUFA-PC ratios. Moreover, we speculate on additional signaling pathways that might contribute to this effect on page 27, paragraph 2 to page 28, paragraph 1.

Comment: viii) "... RTK ligands ...decreased the proportion of PUFAs in cellular phospholipids, specifically in PE, PI, and PS (Fig 7a and Supplementary Fig. 30a, b)". Fig 7a statistically significant is changes only in PI. Changes in other phospholipids are not statistically significant.

Answer: In the fibroblast cell line used for these experiments (Swiss 3T3 instead of NIH-3T3 cells), i) insulin significantly decreased the proportion of PUFA-PE (Fig. 7a, bar chart), ii) PDGF significantly decreased the proportion of PUFA-PE (Fig. 7a, bar chart), PUFA-PI (Fig. 7a, bar chart), and PUFA-PS (Supplementary Fig. 30a), and iii) bFGF significantly decreased the proportion of PUFA-PI (Fig. 7a, bar chart), PUFA-PS (Supplementary Fig. 30a) and PUFA-PG (Supplementary Fig. 30a). The volcano plot in Fig. 7a highlights the most strongly and significantly regulated species, which are indeed mainly PI species. The criteria for significantly regulated species were intentionally made restrictive by setting the false discovery rate for multiple comparison correction to 5%, as explained in our response to the comment regarding Fig. 1d. Overall, we believe that our original statement is correct and would prefer to keep it as short as possible without listing exactly which RTK ligand has significant effects on which phospholipid subclass. Please note that other reviewers asked us to streamline the manuscript.

Comment: 4. To unravel the metabolic network behind the cytotoxic enrichment of PUFA-containing phospholipids, authors analyzed the proteome of fibroblasts challenged with either VAL or MC

(Supplementary Data 1 and 2)48. Supplementary Fig. 15. They presented data about several groups of proteins such as proteins related to glycolysis, gluconeogenesis, and the TCA cycle (Supplementary Fig.16) , proteins related to the pentose phosphate cycle (Supplementary Fig.17), proteins related to fatty acid metabolism. (Supplementary Fig.18) proteins related to (phospho-)lipid metabolism, (Supplementary Fig.19), proteins involved in growth factor signaling. Presented data demonstrated, that treatment of VAL and MC inhibited expression of almost all studied proteins opposite to effects of CAY and serum deprivation, which effect only some proteins. Almost complete inhibition of protein expression by VAL and MC looks unnatural. It can be suggested that inhibitory effect of VAL and MC can be induced by direct suppression of protein synthesis and has nothing to do with general mechanisms of stress. Result presented in literature demonstrated, that both VAL and MC can inhibit protein synthesis. Inhibition of protein synthesis at the level of elongation was proven to be another mode of action of valinomycin in addition to dissipation of membrane potential. Myrtucommulone A (MC) can also act on protein synthesis, as it binds to chaperone Hsp60 and inhibits the refolding activity of the Hsp60-Hsp10 complex assisting in the correct folding of most proteins.

Answer: Thank you for the opportunity to address the concerns raised. First of all, we would like to emphasize that the proteomic data were normalized to protein abundance, i.e. the total amount of protein analyzed was the same in control, VAL- and MC-treated fibroblasts. Accordingly, VAL and MC led to a depletion of some of the 5400 proteins analyzed, while enriching others. This can best be seen in the volcano plots and heatmap already published for the dataset in Thuermer et al., 2022 (Supplementary Fig. 23, PMID: 35624087).

Supplementary Fig. 23 in Thürmer et al., 2022 (PMID: 35624087). Impact of SCD1 and SCD1 products on the proteome of stressed cells. Fibroblasts were treated with vehicle, VAL (10 μ M), MC (10 μ M), CAY10566 (CAY, 3 μ M) in presence or absence of either PI(18:1/18:1 (50 μ M) or PI(16:0/16:0) for 48 h. a-d Volcano plots, where the log₂ fold change (log₂(FC)) is plotted against the negative log₁₀ of adjusted P value, highlight proteins that are regulated by VAL (a), MC (b), or CAY (c) relative to vehicle control or by PI(18:1/18:1) supplementation in CAY-treated cells (d). For statistical analysis, an unpaired, two-tailed Welch t-test was used and adjusted P-values (correction for multiple comparisons) were calculated using Benjamini-Hochberg correction. Proteins that are consistently down- (blue) or upregulated (green) ($\geq 20\%$) by all three stressors (VAL, MC, CAY) versus vehicle control are indicated by color. e Heatmap of proteins identified in the independent experiments of vehicle control, VAL-, MC-, CAY-, CAY+PI(18:1/18:1)- and CAY+PI(18:1/18:1)-treated cells. Protein intensities were log₂ transformed and are colored based on their z-score ranging from -3 to 3.

It is true that a higher number of proteins are downregulated than upregulated, which is not surprising when studying mechanisms related to apoptosis, which per se impairs protein biosynthesis (PMID: 10889505). Not only does apoptosis rapidly downregulate protein biosynthesis, but inhibition of

protein biosynthesis (e.g. by CHX) also induces apoptosis. The two processes are inextricably linked. However, due to our focus on the onset of cell death (see comment above), total protein levels determined by a BCA protein assay and normalized to cell number were not yet decreased for the cytotoxic stress conditions investigated (Fig. X1), likely due the dominance of structural proteins with longer half-life.

Fig. X1. Protein content of NIH-3T3 fibroblasts under cytotoxic stress. Fibroblasts were cultured under cytotoxic stress conditions for 24 or 48 h before the protein content was determined using a Pierce Micro BCA Protein Assay Kit (Thermo Fisher Scientific). Mean \pm s.e.m. and single data from $n = 3$ independent experiments. P values vs. vehicle control; repeated measures one-way ANOVA of log-transformed data + Dunnett's post hoc test.

VAL has been reported to inhibit leucin incorporation into polypeptides (e.g., PMID: 4715678, PMID: 240122, PMID: 7207469). The authors excluded that VAL interacts directly with ribosomes and show that the ATP/AMP ratio decreases, but the exact mechanisms remain unclear and it cannot be excluded that the inhibition of protein biosynthesis is secondary to the induction of apoptosis. The inhibition of protein biosynthesis by VAL is now mentioned on page 6, lines 5-6.

It has been reported by my former colleagues in Jena that MC directly interacts with HSP60 (PMID: 28457707), which, together with its cochaperone HSP10, is mainly localized in mitochondria and is responsible for the correct folding of proteins imported into mitochondria (PMID: 35609646). Thus, HSP60 does not contribute significantly to ribosomal protein biosynthesis in the cytosol or at the rough ER (except in cancer cells, where HSP60 is often overexpressed and may also be localized in the cytosol to protect against protein unfolding stress). Accordingly, MC has been shown to specifically induce the aggregation of two mitochondrial proteins (PMID: 28457707), which can hardly account for an overall decrease in protein biosynthesis. The mechanism of MC is now discussed in more detail on page 6, lines 6-8.

Regarding Supplementary Fig. 15, we only show those biological processes here that are downregulated by both VAL and MC in order to filter for biological processes that are generally suppressed by cytotoxic stress. This means that all of the pathways shown in Supplementary Fig. 15 must be downregulated by the two stressors because that is what we selected. There are several other biological processes that are upregulated by either VAL or MC (as listed below), but interestingly, none are upregulated by both VAL and MC.

Upregulated biological processes for VAL:

- RNA splicing, via transesterification reactions
- RNA splicing, via transesterification reactions with bulged adenosine as nucleophile
- mRNA splicing, via spliceosome
- Macroautophagy

Upregulated biological processes for MC:

- Biological process involved in symbiotic interaction
- Negative regulation of peptidase activity
- Positive regulation of DNA-templated transcription, elongation
- Cellular response to iron ion
- DNA-templated transcription, elongation

Comment: Suggestion about inhibition of protein synthesis by VAL can explain results obtained in the experiments to study subcellular fractionation of VAL-treated fibroblasts (Supplementary Fig. 8). These results demonstrated, that Ca²⁺ ATPase (PMCA), used as a marker of plasma membrane disappeared in non nuclear fraction after VAL treatment.

Answer: Thank you for mentioning this potential mechanism. While we cannot completely exclude that the general suppression of protein biosynthesis by apoptosis induction (and possibly additional mechanisms of VAL) contributes to the marked decrease in PMCA levels in the nuclear fraction upon treatment with VAL, we consider this scenario unlikely for two reasons. First, according to our proteomics studies (as described above), VAL does not reduce the expression of various other proteins, at least not as much as it does for PMCA. It could be argued that inhibition of protein biosynthesis would most strongly affect the levels of short-lived proteins. However, with a half-life of approximately 12 hours (PMID: 9266850), PCMA does not qualify as a specific short-lived protein. Second, PMCA is known to be a caspase substrate and is cleaved early in apoptosis (at 6-9 h) resulting in a 120 kDa fragment that remains intact for several hours (PMID: 17446484). We checked whether this fragment is still visible after 48 h of treatment, but the uncropped Western blot does not show a corresponding band (anymore?). In conclusion, the general suppression of protein biosynthesis may contribute to but, in our opinion, cannot explain the rather selective strong decrease of nuclear PMCA levels in VAL-treated cells.

Comment: 5. In Supplementary Fig. 38 authors present results about stress-induced regulation of p-Akt in cells with active and inactive ACC. Effects of ETO TNF CHX and serum depletion in different independent experiment are not consistent

Answer: Although the former Supplementary Fig. 38 has been removed in our efforts to focus the manuscript on the most important findings (as requested by other reviewers), we would like to address the reviewer's concern. First, the reviewer's comment is correct; we do not claim otherwise. Only some stress conditions (i.e., TPG, VAL, MC) were less efficient in further reducing Akt activation in ACC-inactivated fibroblasts. Note that TNF α was a negative control and not a cytotoxic stressor, and that the standard errors for ETO- or CHX-treated or serum-starved cells, based on eight independent datasets, were still in an acceptable range at 11-14%. As a result, CHX showed a clear trend to decrease p-Akt levels in fibroblasts with active ACC by 20% ($p = 0.08$), while increasing p-Akt levels in fibroblasts with inactive ACC by 18% ($p = 0.09$). This leaves ETO (the cytotoxic stressor that least exploits the mechanisms described here in many settings) and serum depletion.

Comment: 6. In Fig. 8 authors presented data about possible involvement of increased amount of PUFA in phospholipids in development of ferroptosis. I have several concerns about this data. i) Authors claimed that selective ferroptosis inhibitor Fer-1 can inhibit cell death in apoptotic cells. In these experiments they studied cell population contained thousands of cells. It is not clear from their results whether simultaneous development of ferroptosis and apoptosis occur in the same cell or in different cells present in sample.

Answer: Excellent point, which we have now addressed experimentally by immunofluorescence staining of cleaved caspase 3 (the executioner caspase in apoptosis) and the lipid peroxidation product 4-hydroxynonenal (4-HNE). In many cases, VAL induced both caspase 3 cleavage and 4-HNE production in cells, although there were exceptions, with cells preferentially producing 4-HNE (Fig. 8g and

Supplementary Fig. 34b). The selective apoptosis inducer ABT-737 (a BCL-2 family inhibitor) had a similar effect, whereas the ferroptosis inducer RSL3 exclusively induced membrane peroxidation, as expected (Fig. 8g).

Comment: ii) Authors found, that the level of oxidized arachidonoyl-PE was greatly increased in VAL-treated cells (Fig 8b), and suggested, that accumulation of this lipid will navigate cells towards ferroptosis. It's better to present results demonstrating increase in the amount of PE-AA-OOH not in percentage but in absolute value (pg/mg protein or pg/106 cells).

Answer: Agreed, the amount of PE(16:0/20:4+2[O]) and PE(18:0/20:4+2[O]) is now given in nmol/10⁶ cells (Fig. 8e).

Comment: Several different compounds with similar m/z can be found in cells, so to prove that they studied oxidized AA-PE authors need to provide results of fragmentation.

Answer: The data in Fig. 8d, e have already been obtained using a fragmentation-based scan mode, i.e. multiple reaction monitoring (MRM). Thus, the specificity for the detection of PE(16:0/20:4+2[O]) and PE(18:0/20:4+2[O]) is derived from the retention time, the m/z ratio of the parent ion, and the oxidized fatty acid fragment (20:4+2[O]-4H) [used for quantification; the signal is shown in Fig. 8d as an extracted chromatogram]. To further confirm that VAL induces lipid peroxidation, we now visualized the lipid peroxidation product 4-hydroxynonenal (4-HNE) by immunofluorescence microscopy using a selective antibody (new Fig. 8c, g and Supplementary Fig. 33b and 34b).

Comment: lii) To demonstrate involvement of ferroptosis in cell death induced by VAL authors used selective ferroptosis inhibitor ferrostatin-1 (Fer-1) (Fig. 8c), but it looks like Fer-1 was not effective in preventing cell death induced by VAL.

Answer: Good point, while Fer-1 reduced the decrease in cell viability measured by MTT assay (an early event in ferroptosis) (Fig. 8b), it did not prevent VAL-induced disruption of the plasma membrane (a late event in ferroptosis) (former Fig. 8c). These data are indeed consistent with our hypothesis that initial apoptosis sensitizes cells to membrane peroxidation later on. According to this hypothesis, VAL first induces apoptosis, resulting initially in a low number of dead cells (a process that cannot be prevented by Fer-1; former Fig. 8c). Then, sensitization to lipid peroxidation is accelerated and early ferroptotic events (such as a decrease in mitochondrial functionality – a process that can be attenuated by Fer-1; Fig. 8b) become evident. More important for our conclusion, however, is that VAL and several other stressors (i.e., MC, TPG, CHX, and ETO) sensitize cells to ferroptotic stimuli such as RSL3 (new Fig. 8f and Supplementary Fig. 34a). We have rewritten the paragraph (page 35, second paragraph) to more clearly distinguish between ferroptosis sensitization (5 of 6 stressors) and direct ferroptosis induction (mainly by serum depletion and less by VAL and MC).

Comment: 7. Authors stated that NADPH is an essential cofactor for maintaining redox balance. NADPH has been associated with aberrant ROS generation and induction of cell death via NADPH oxidases (NOX)88 and oxidoreductases (POR). That is why they investigated the effect of VAL on the total cellular fluorescence in the spectral range from 426 to 490 nm, which is representative of cellular NAD(P)H status91. Amount of NADPH was involved not only in maintaining of ROS balance. Besides being involved in the ROS defense, NADPH is a product of the pentose phosphate pathway. Accordingly, alterations in cellular growth rate could also influence NADPH levels via this pathway. I am not sure, that measurements of fluorescence can accurately enough assess amount of NADPH. NADP(H) cannot be separated spectrally from NADH autofluorescence. As a result, measured fluorescence is always a mixture of both NADPH and NADH. Concentrations of the NAD(H) are usually about 10 times higher than the levels of NADP(H). Measurement of NAD(P)H fluorescence does not provide absolute values. In cells intensity of NAD(P)H fluorescence is confounded by scattering and the presence of other fluorophores that cannot be distinguished from NAD(P)H or that influence NADH autofluorescence such as the protein composition of a cell. Some lipid droplets or lipofuscin have similar autofluorescence spectra as NADH and can also interfere with measurements.

Answer: Yes, there is no question that NADPH has pleiotropic activities other than maintaining redox balance, and the pentose phosphate pathway is a major source. In fact, we show that cytotoxic stress does not substantially interfere with the oxidative part of the pentose phosphate pathway that produces NADPH, which seems to allow continuous NADPH production in stressed cells, as described on page 22, last paragraph to page 23, first paragraph. However, enzymes of the pentose phosphate pathway are not up-regulated, which rather rules out regulation at this level (Fig. 4a, b, Supplementary Fig. 17). What we did find is a shift from hexokinase-1 (which is downregulated) to hexokinase-2 (which is upregulated), which may indicate that glucose-6-phosphate is more efficiently channeled into the pentose phosphate pathway (page 22, last paragraph to page 23, first paragraph). Overall, we would like to keep the focus of this manuscript on redox homeostasis, following the advice of the editor and other reviewers to concentrate the manuscript on the most important findings.

Regarding the analysis of NAD(P)H, we want to emphasize that we did not simply detect fluorescence, but the fluorescence lifetime (by FLIM = fluorescence lifetime imaging microscopy), which requires a special microscopic setup and better discriminates between fluorophores. Thus, FLIM allows to distinguish NAD(P)H from autofluorescence originating from proteins, lipid droplets or other cellular sources, and even to distinguish between NADH and NADPH [PMID: 24874098]. In the current study, we had a more conservative setup that does not distinguish between NADH and NADPH. Accordingly, the manuscript always refers to both redox cofactors as NAD(P)H.

It is true that NADH is present in cells in much higher concentrations than NADPH, and we cannot exclude that the observed changes in NAD(P)H are mainly dependent on NADH (as now mentioned on page 48, second paragraph). Please note that NADH also has a protective value against peroxidation, e.g., as a cofactor of FSP1 in the regeneration of CoQ10-H₂ [PMID: 31634899]. However, based on our data (including the proteomics analysis), we consider it more likely that the inhibition of fatty acid biosynthesis reduces the consumption of NADPH and that the increase in β -oxidation generates NADH, thereby increasing the sum of NADPH and NADH, as discussed on page 48, second paragraph .

Table X1 Fluorescence lifetimes of NADH and NADPH in free and protein-bound form

Free NAD(P)H	400 ps
Protein-bound NAD(P)H	2.5 ns
Intracellular NADH	1.5±0.2 ns (PMID: 24874098)
Intracellular NADPH	4.4±0.2 ns (PMID: 24874098)

In support of this hypothesis, and as expected from a balanced increase in both NADPH and NADH levels, re-analysis of our FLIM data revealed that the fluorescence lifetime (an indicator of the NADPH/NADH ratio, Table X1, PMID: 24874098) is not substantially altered (Table X2). Note that a shift towards NADPH would be associated with an increase in fluorescence lifetime and a shift towards NADH would be associated with a decrease in fluorescence lifetime, both of which were not observed.

Table X2 Fluorescence lifetimes calculated using the bi-exponential decay fitting model

	Vehicle (DMSO)	VAL (10 μ M)
t_1 (ps)	803 \pm 93	886 \pm 55 ($p = 0.608^2$)
t_2 (ps)	4418 \pm 232	3810 \pm 284 ($p = 0.228$)
t_m^1 (ps)	1387 \pm 111	1408 \pm 104 ($p = 0.921$)

¹weighted mean fluorescence lifetime; ²two-sided paired student *t*-test

Reviewer #6:

Comment: In this manuscript, Gollowitzer et al. explore the crosstalk between various cell death programs and lipid metabolism. They show how a reprogramming of fatty acid channeling and incorporation is affected under some of these conditions and they finally relate this phenotype to impaired RTK signaling which would be associated with membrane stress. While they describe a mechanism by which cells increase the proportion of PUFAs in phospholipids under certain cytotoxic stresses the manuscript fails to prove the direct effect on membrane peroxidation.

Answer: We are grateful for your careful review and constructive criticism of our manuscript and now provide direct evidence that VAL induces membrane peroxidation per se (new Fig. 8c-e, g) and that this effect is enhanced by the ferroptosis inducer RSL3 (new Fig. 8c). To provide also more direct evidence that the increased proportion of PUFAs in phospholipids sensitizes cells to membrane peroxidation and ferroptosis, we manipulated the cellular PUFA-phospholipid ratio in fibroblasts by PUFA supplementation or genetic means and assessed directly membrane peroxidation (for PUFA supplementation) and the susceptibility to ferroptosis induction. Arachidonic acid/20:4 supplementation increased the PUFA-PC ratio by 1.6-fold (new Supplementary Fig. 33b), and deletion of LPLAT12/LPCAT3 (an enzyme involved in the proposed mechanism) decreased the PUFA-PC ratio by 1.4-fold (new Supplementary Fig. 33e), similar to changes induced by cytotoxic stress conditions. As expected, 20:4 supplementation sensitized cells to the selective ferroptosis inducer RSL3 while favoring lipid peroxidation (new Supplementary Fig. 33b), and deletion of LPCAT3 had a desensitizing effect, even under cytotoxic stress or serum depletion (new Fig. 8a, new Supplementary Fig. 33c). In addition, we have previously shown that supplementation of fibroblasts with oleic acid/18:1 prevented the accumulation of PUFA-PC upon treatment with cytotoxic inducers (Supplementary Fig. 13a) and reduced the fraction of cells undergoing necrosis (PMID: 35624087). These findings are in line with a large number of studies in different systems showing ferroptosis-sensitizing (rather than ferroptosis-inducing) activity for PUFAs (e.g., PMID: 34118189, PMID: 27506793, PMID: 35216678) and LPCAT3 (e.g., PMID: 35658397, PMID: 27842066, PMID: 37407304).

Comment: In general, the manuscript is way too long and hard to follow with too many small details that distract from the mayor point which is also not clearly demonstrated. I believe it should be simplified as Nat.Comms targets a generalistic audience.

Answer: Agreed; we have removed several descriptions/figures that are not central to the main conclusions of the manuscript (i.e., Fig. 9c, Supplementary Fig. 3c, 18d-g, 22, 29, 33-38, 41, 42, 45 and Supplementary Note 2-4, 11, 13-15). We also no longer show the negative control TNF α (which only sensitizes to apoptosis but does not induce it per se) and the unambiguous I3M (which decreases cell numbers, but does not seem to induce apoptosis under our experimental conditions). However, we would also like to emphasize that many details were added during the first round of peer review to address the reviewers' concerns, which significantly increased the length of the manuscript. We hope that you will agree to keep these sections.

Comment: My main concern is related to the interpretation of the lipidomics data. First, normalizing it by the number of cells does not seem appropriate when cytotoxic conditions are analyzed basically because cell confluency affects the lipidome of cells (<https://doi.org/10.1038/nature14429>). It would be interesting to see how the lipid changes precede the cytotoxicity and for that, it would be more appropriate to harvest the cells 6, 12 or 24 h after the treatments, when the number of cells is comparable between the conditions. Perhaps a control could be added were the lipidomic changes at different confluencies is described, together with some of the treatments at timepoints where the confluency is still similar to that of control cells. Interestingly, the only treatment that doesn't follow the tendency to increase the PUFA-containing phospholipids is the one with TNF-alpha, the only one where the confluency is also higher than in the control. Moreover, PUFA's are taken up from the media and thus, they decrease with time and when the confluency goes up. If cells are under cytotoxic

conditions, there will be less cells to consume the PUFA's and they would go up. In summary, I believe that even if interesting, the detected changes in lipids and the reprogramming of lipid metabolism that is described in this paper might be solely due to cell confluency.

Answer: Thank you for raising these important points, which we now carefully discuss (also based on additional experiments) below and in the revised manuscript.

Normalization of the phospholipid data to cell number or protein content is, in our opinion, the only way to compare absolute data (e.g. in Fig. 1a). Otherwise, the effects on lipid amounts would largely reflect cell number, because more cells mean more membranes and thus more phospholipids. However, the absolute data are not very informative when interpreting the consequences of altered membrane fatty acid composition on membrane properties, because changes in e.g. peroxidation sensitivity do not depend on whether a cell has more or less membranes, but on the density of PUFAs within them. Taking this into account, the most important results on the phospholipidome that we present here (e.g. in Fig. 1d, f, Supplementary Fig. 8b, c) are not absolute but relative data (proportions, ratios) that were not normalized to the number of cells but to the total amount of phospholipids.

Kinetic studies of changes in the phospholipidome (Fig. 1f) showed that cytotoxic stress elevated the PUFA-PC ratio starting at 24 h (significant for STS and CHX) and further increased the ratio at 48 h. At 24 h, trends toward increased PUFA-PC ratios were additionally observed for ETO, serum depletion, and MC. Our data show (consistent with our hypothesis) that the increase in PUFA-PC ratio coincides with the onset of apoptosis (Supplementary Fig. S26c, d) and that at least two stressors increase the PUFA-PC ratio at a time point (24 h) (Fig. 1f) when cell number/confluency is not yet substantially reduced (Supplementary Fig. 1c). As a negative control, TNF α (not shown anymore) was used, which sensitizes to apoptosis but does not induce apoptosis per se (former Supplementary Fig. 1 and 2). Accordingly, TNF α neither decreased cell number (former Supplementary Fig. 1c) nor increased the proportion of PUFA-PC. At 6 h, none of the cytotoxic stress conditions caused a substantial increase in the PUFA-PC ratio (Fig. 1f), which is not unexpected since we propose that changes in the PUFA-PC ratio are secondary to the induction of apoptosis.

To exclude that the increased PUFA content in phospholipids of stressed cells is simply related to the increased availability of PUFAs, we followed the advice of the reviewer in a later comment and replaced the cell culture medium after 24 h of treatment, before harvesting the cells for lipidomic analysis after a total of 48 h. Refreshing the medium slightly raised the basal levels of PUFA-PC, but did not reduce the responsiveness to VAL, which was still able to decrease SCD1 expression (new Supplementary Fig. S14d) and increase the PUFA-PC ratio efficiently (new Supplementary Fig. 9b). These results strongly suggest that confluence-controlled PUFA availability is not the main reason for the relative cytotoxic increase in the PUFA-PC ratio at lower cell densities.

In addition, selective inhibition of de novo fatty acid biosynthesis (via ACC) strongly increased the cellular PUFA-PC ratio (in support of the mechanism proposed here), but only moderately reduced cell number by trend (new Fig. 2d, e). Similarly, only the selective induction of apoptosis by the BCL-2 family inhibitor ABT-737, but not the induction of cell cycle arrest by the selective CDK-4/6 inhibitor palbociclib, increased the PUFA-PC ratio (new Fig. 1g), although both compounds were comparably effective in reducing viable cell numbers (new Supplementary Fig. 9d).

Consistent with these new results and the overall findings of our manuscript, the cellular fraction of PUFA-PC under cytotoxic stress significantly correlates with the availability of enzymes in lipid metabolism, e.g., positively with ACSL5 (new Supplementary Fig. 19b) and negatively with enzymes in de novo fatty acid biosynthesis (i.e., ACC) (new Supplementary Fig. 19b).

Nevertheless, cell density seems to be important in this context, possibly via cell-cell contacts or humoral factors. When the decrease in cell number induced by VAL within 48 h of treatment is mimicked by seeding fewer cells, both approaches result in markedly increased PUFA-PC ratios, although VAL is significantly more effective (new Supplementary Fig. 9a). In the revised manuscript,

we now point out that a decrease in cell density is likely to contribute to the cytotoxic increase in PUFA-PC ratios (page 10, lines 18-20). Notably, cell-cell contacts and RTK signaling are interrelated [PMID: 28716887]. For example, cell-cell contacts can control the spatial distribution and activity of RTKs, the latter in part through direct association of RTKs with junction components and also indirectly through changes in cell mechanics [PMID: 28716887].

We now describe these important aspects on page 10, third paragraph and page 11, first paragraph.

The reviewer's concern about the influence of PUFA consumption as a function of cell confluency is also closely related to in vivo relevance. In an independent project that will be published separately, we investigated the phospholipidome of tumors from mice grafted with human HL60 leukemia cells and treated with the cell death-inducing anticancer agent vinblastine and found a significantly elevated proportion of PUFA-PC in the tumor, in line with our expectations (Fig. X2).

Fig. X2. Vinblastine increases the proportion of PUFAs in tumor phospholipids in HL60-grafted mice. 6-week-old female immunodeficient athymic nude mice (strain: Rj:ATHYM-Foxn1nu/nu) received subcutaneous injections of HL-60 cells (4.5×10^6). After 14 days, vehicle (0.5% DMSO in PBS pH 7.4, 200 μ l, i.p.) or VIN (0.5 mg/kg, i.p.) were administered every three days over another 10 days. The mice were killed at day 25. **a** Proportion of PUFA-PC in the tumor. Mean \pm s.e.m. from $n = 4$ tumors. P values vs. vehicle control; two-tailed unpaired student t -test of log-transformed data. **b** Volcano plot showing the mean difference in the phosphatidylcholine (PC) and phosphatidylethanolamine (PE) profile in VIN- vs. control-treated tumors. Adjusted P values given vs. vehicle control; two-tailed multiple unpaired student t -tests with correction for multiple comparisons using a two-stage linear step-up procedure by Benjamini, Krieger, and Yekutieli (false discovery rate 5%).

Comment: - The lipidomics data is normalized by the cell number. When are the cells counted? The protocol does not explain how cells are harvested for the lipidomics analysis. If total cell number is used to normalize, this should be a crucial step to explain.

Answer: Done, we extended our description on how cells were harvested for individual experiments in the method section 'Cell culture and treatment' on page 53, second paragraph: "Cell harvesting procedures using trypsin/EDTA to collect adherent cells differed depending on the readout. To determine the number of viable, apoptotic, necrotic, and dead cells, detached cells were recovered from the cell culture medium (except for kinetic studies and the determination of membrane intactness). In all other experiments, cells were washed with PBS pH 7.4 to remove detached cells and to enrich for viable and early apoptotic cells before adherent cells were collected by trypsinization and counted using a Vi-CELL Series Cell Counter (Beckman Coulter, Krefeld, Germany)."

Comment: - What does "Total phospholipids" mean? How is it calculated? During the explanation of the protocol the text mentions all the time "PC, PE, PI, PS, PG, and FFA were extracted". "PC, PE, PI, PS, PG, and FFA were separated". In general, through the mentioned protocol one cannot be specific for the extraction of such lipids, the samples would be composed of many kinds of phospholipids including others such as PA or sphingolipids for example.

Answer: Total phospholipids refer to the sum of the absolute amounts of all phospholipid species analyzed, as now described on page 59, lines 8-10. Thus, the phospholipid subclasses considered for

Fig. 1a are PC, PE, PS, and PG, with the calculation shown in the source data. In the Methods section, we wanted to express that the respective extraction and separation protocols were used for the analysis of PC, PE, PI, PS, PG and FFA, which were extracted and separated together with other lipophilic molecules. To avoid confusion, we have simplified the sentence and now write "lipids were extracted/separated ..." and specify separately which lipid classes this procedure applies to.

Comment: - The use of internal standards for PI or cardiolipins is not mentioned, is this correct? Please, state the amount and specific name/formula of the used internal standards for each lipid subclass in the methods section, in the form of a table. If an internal standard for PI was not used, how was the amount of different PI in cells calculated?

Answer: Thank you for bringing this oversight to our attention. In the main and supplementary figures, we show the relative proportions of PI subgroups and PI species (normalized to the signal of all PI species analyzed, 100%), which describes their density within membranes. This is the information we need to draw conclusions about the peroxidation sensitivity of membranes. Absolute amounts, on the other hand, are strongly influenced by whether cells have more or less membranes. Information on absolute PI levels under these experimental conditions can be found in Thürmer et al. (PMID: 35624087). However, the reviewer is correct. The source data for Fig. 3i list the absolute amounts of PI in addition to the relative proportions. At the time this was measured, we did not have a deuterated PI standard and were not satisfied with PI(8:0/8:0) as an internal standard due to differences in extraction and ionization efficiencies compared to PI species with long-chain fatty acids. Therefore, we normalized PI to DMPC and corrected the values by external calibration with PI(15:0/d7-18:1), as now mentioned on page 59, lines 6-8.

Since Nature Communications only allows 10 display items (figures or tables), we would rather not show the internal standards in a table, because we would have to omit one of the main figures (or transfer this important information to the Supplementary information). Instead, we have expanded the section on the calculation of absolute lipid amounts and now explicitly refer to the phospholipid classes for which a class-specific internal standard was used (page 59, first paragraph).

The amount of the internal standards added is now specified (page 55, line 20).

Comment: - The methods section for the lipidomics analysis is extremely confusing. What does "Reanalysis of lipidomics datasets" mean? In which context were all these datasets reanalyzed and for what purpose? Are those datasets being reused for main figures in this manuscript? This should be clearly stated in each figure.

Answer: With reanalysis of lipidomics datasets, we want to express that we reused our own previously published datasets and slightly adjusted the analyzed species for better consistency throughout the manuscript, as now stated on page 58, lines 14-18. For Fig. 3i, we had already mentioned the reuse/reanalysis in the figure legend. For Fig. 1c (data for PI), Fig. 1f (panel 3) and Supplementary Fig. 3c (data for PI), we have now added a note in the figure legends. The datasets for Fig. 7 and Supplementary Fig. 30 have never been made fully available, but parts of them were previously published by us in a different article, as now more clearly described on page 58, lines 18-22.

Comment: - Fig 1g and 1h. Why is this such an important information? It could be represented in supplementary data.

Answer: We agree and have moved these panels to the Supplementary Information (new Supplementary Fig. 8b, c).

Comment: - What is the effect of TOFA in cell number? This treatment should induce cell death because is increasing the amount of PUFA-containing phospholipids in membranes.

Answer: Treatment of NIH-3T3 fibroblasts with the ACC inhibitors TOFA (5 μ M) and Soraphen A (100 nM), and the SCD1 inhibitor CAY10566 (3 μ M) for 48 h reduces the cell number by 25%, 23%, and 64%, respectively (new Fig. 2d), which we ascribe to impaired proliferation (and which is not surprising given the restriction of fatty acid supply and thus membrane biogenesis). While weak PARP cleavage can be detected already at 48 h (Supplementary Fig. 31), TOFA per se is not sufficient to efficiently induce apoptosis within 48 h, as shown by Annexin V/PI staining; first trends appear at 72 h (Supplementary Fig. 27). These findings are consistent with the mechanism outlined in the manuscript, which proposes that the cytotoxic switch to elevated PUFA-PC ratios (at 24-48 h) sensitizes cells to membrane peroxidation/ferroptosis and contributes directly to cell death only when the cytotoxic stressor additionally induces redox stress (which is not the case for TOFA). For clarity, we have rewritten the respective section on page 35, second paragraph.

Comment: Also, is inhibition of SCD1 cytotoxic?

Answer: The SCD1 inhibitor CAY10566 (3 μ M) reduced cell numbers by 64% after 48 h of treatment (new Fig. 2d) along with the induction of PARP cleavage (Thürmer et al. - Fig. 4h, PMID: 35624087) and, to a lesser extent, increases the proportion of apoptotic/necrotic cells (Thürmer et al. - Fig. 4i) under our experimental conditions. The ferroptosis-sensitizing activity of SCD1 inhibition is well documented [PMID: 37029018], as confirmed by us for the GPX4 inhibitors RSL3 and ML210 in human MDA-MB-231 breast cancer and A549 lung cancer cells in a separate, recently published manuscript (Schwab et al., 2024, Nat. Cell. Biol., PMID: 39009641).

Comment: - Which is the cell confluency when cells are treated with TOFA plus the other cytotoxic agents such as VAL?

Answer: Combined treatment with TOFA and cytotoxic stressors (including VAL) further reduced the absolute number of NIH-3T3 fibroblasts within 48 h compared to treatment with VAL, MC, STS, and ETO alone (new Supplementary Fig. S12b). The combination of TOFA with CHX, TPG or serum depletion was hardly effective.

Comment: - The data upon supplementation with MUFA's and PUFA's suggests that the effects seen are indeed dependent on the availability of these lipids in the media, which can vary with changes in confluency. A good experiment would be to replace the media with fresh media after 24 hours with the cytotoxic agents and check for the increase in PUFA containing phospholipids to see if the tendency seen in Fig 1f, changes.

Answer: Done, as described in our response to the reviewer's second comment, we performed this important control experiment. Refreshing the medium after 24 h slightly increased the basal level of PUFA-PC, but did not markedly interfere with the VAL-induced repression of SCD1 expression (new Supplementary Fig. 14d) (originally shown by us in Thürmer et al., 2022 without interim medium exchange) and the increase in PUFA-PC ratio (new Supplementary Fig. 9b), suggesting that the VAL effect is largely independent of confluence-controlled PUFA availability.

Comment: - I do not find the data for SCD1 suppression among the cytotoxic conditions. It would be good to also check for this phenotype upon supplementation with fresh media. The effects seen in the active synthesis of MUFA's and degradation of fatty acids could be homeostatic.

Answer: Indeed, that's a very relevant point in this context. The effect of cytotoxic stressors on SCD1 expression has already been published by us in Thürmer et al, 2022 (PMID: 35624087) under identical experimental conditions as now more clearly mentioned on page 17, lines 2-4. Of the seven cytotoxic stressors that efficiently induce apoptosis, three significantly decreased and three tended to decrease SCD1 expression (Thürmer et al. - Fig. 3e). Following the suggestion of the reviewer, we further investigated the consequences of refreshing the cell culture medium after 24 h on this effect after 48 h. Changing the culture medium still resulted in an efficient suppression of SCD1 expression, as shown

exemplarily for VAL (new Supplementary Fig. 14d). Taken together, the decrease in SCD1 expression is shared by many (but not all) cytotoxic stressors.

Comment: - Fig 8c. Please describe what do the authors mean by “membrane intactness”, how they measure it.

Answer: Membrane intactness (or membrane integrity) simply refers to the viable/dead staining by trypan blue, which is based on the exclusion of the DNA-binding dye by cells with an intact plasma membrane and is used by us to differentiate between different cell death readouts. We define the term in the legend of Fig. 8 ("Changes in membrane intactness (trypan blue exclusion) were measured") and in the Methods section on page 69, first paragraph: "Cell number and membrane intactness (indicative of cell viability) were measured after trypan blue staining using a Vi-CELL Series Cell Counter (Beckman Coulter; software: Vi-Cell XR Cell Viability Analyzer, version 2.03 or 2.06.3)".

Comment: Also, please demonstrate that membranes enriched in arachidonic acid containing phospholipids are more prone to lipid peroxidation and thus less viable. All these things are not connected.

Answer: Good point, as described in our response to comment 1, we have now additionally manipulated the membrane PUFA ratio either by supplementation of arachidonic acid/20:4 or deletion of LPLAT12/LPCAT3 and confirmed the central role of the membrane PUFA ratio for peroxidation and sensitization to ferroptotic cell death (new Fig. 8a, Supplementary Fig. 33b, c).

Comment: - Different cytotoxic conditions are mixed throughout the paper. While I believe they might have some phenotypes in common, their effects are very different. Thus, sentences like the following, do not give any important information and I do not understand the intended message: Page 16: “While ACC protein expression decreased up to 1.7-fold (serum depletion, 48 h) and p-ACC levels increased up to 2.5-fold (VAL, 1h), malonyl-CoA levels decreased as much as 14.5-fold (MC, 48 h).”

Answer: Agreed, we have revised the sentence as follows: "The levels of the ACC product, malonyl-CoA, were efficiently decreased by VAL, MC, CHX and by trend by ETO and TPG (Fig. 3f). Given the limited ACC repression (Fig. 3c) and inhibitory phosphorylation (Fig. 3e), additional mechanisms may contribute to the depletion of malonyl-CoA, ...".

Comment: In this section it is also stated that “Decreased levels of malonyl-CoA might therefore explain why fatty acid catabolism is greatly increased during cell death” but this is not proven and is not further investigated.

Answer: Done, we have now deleted this speculation, which was not explicitly confirmed by our experiments, and we agree that it would therefore fit better in the Discussion section than in the Results section. Actually, we consider this mechanism very likely, since malonyl-CoA is a universal metabolic regulator of β -oxidation (PMID: 11814524). It inhibits the carnitine acyltransferase shuttle, which is key for limiting mitochondrial fatty acid import and thus β -oxidation, to avoid parallel fatty acid biosynthesis and degradation. Further studies in this direction are planned, but shall be published separately.

Comment: Thus, the hints to understand how the metabolism is reprogrammed are merely descriptive and again, all these responses could be homeostatic. While homeostatic responses are interesting, I believe it is not the aim of this manuscript since the authors claim to have a mechanism.

Answer: We are very grateful for the critical evaluation and valuable guidance, which allowed us to further strengthen the manuscript and fill in mechanistic gaps. In addition, we would like to point out that we have also put a lot of effort into mechanistic studies on the metabolic reprogramming, which we consider to be much more than "merely descriptive".

We had already performed the following mechanistic studies to show that:

- Inhibition of ACC (the rate-limiting enzyme in fatty acid biosynthesis) by two selective inhibitors or knockdown is sufficient to increase PUFA-PC ratios (Fig. 2e, f)
- The ability of several cytotoxic stressors to increase PUFA-PC ratios is reduced when ACC is inhibited (Fig. 2g)
- Exogenous PUFAs and, to a lesser extent, PUFA biosynthesis contribute to the increase in PUFA-PC levels under cytotoxic treatment, with the former effect being enhanced by inhibition of ACC (Fig. 2h, i, Supplementary Fig. 12c, d).
- Mimicking fatty acid biosynthesis (by supplementation of MUFAs) prevents the increase in PUFA-PC in response to cytotoxic stressors (Supplementary Fig. 13a).
- Silencing of LPCAT3 attenuates the cytotoxic shift to increased PUFA-PC ratios (Fig. 2j)
- Growth factor withdrawal (by serum depletion) mimics the increase in PUFA-PC ratios (Fig. 1f) and metabolic adjustments (e.g. ACC repression, Fig. 3c)
- RTK agonists (growth factors) instead decrease the proportion of PUFAs in phospholipids (Fig. 7a), an effect modulated by PKC, MEK-1/2, p38 MAPK, PI3K and cPLA2 (Fig. 7d).
- The cytotoxic increase of PUFA-PC is independent of caspases (Supplementary Fig. 24)

In addition, we have now performed further experimental studies to explore the mechanisms increasing PUFA-PC ratios and inducing lipid peroxidation in more detail:

- Induction of cell cycle arrest (new Supplementary Fig. 9c) and decreasing cell confluence (thereby less depleting exogenous PUFAs) (new Supplementary Figs. 9a, b) may contribute to, but cannot explain, the decrease in SCD1 expression and increase in PUFA-PC ratio in stressed cells.
- Pan-inhibition of BCL-2 family members (which selectively induces apoptosis) increases the PUFA-PC ratio, whereas selective inhibition of BCL-2 or CDK4/6 (which induces cell cycle arrest) had no effect, even in combination (new Fig. 1g and Supplementary Fig. 9d).
- VAL and the BCL-2 inhibitor ABT-737 often induce apoptosis (caspase-3 cleavage) and lipid peroxidation (4-HNE formation) in the same cell (new Fig. 8g, Supplementary Fig. 34b).
- The nuclear availability of the lipogenic transcription factor SREBP1 (with ACC, FAS and SCD1 as target genes) is reduced by serum depletion (new Fig. 7e) but not by VAL treatment (new Supplementary Fig. 21b).
- Screening of small molecule inhibitors/activators of SREBP1 regulatory pathways shows that p53 (pifithrin- β) and AMPK (BAY3827), which is activated by VAL, contribute (to a limited extent) to the VAL-induced increase in PUFA-PC ratio (new Supplementary Fig. 21c)
- Inhibition of SCD1 also increases the PUFA-PC ratio, as found for inhibition of ACC (new Fig. 2d, e).
- Silencing of ACSL5 increases the PUFA-PC ratio more in vehicle-treated cells than in VAL-treated cells, in which ACSL5 is upregulated (new Fig. 4e).
- Inhibition of IGF1R, PI3K, Akt or SREBP1 increases the PUFA-PC ratio (new Fig. 6e), with IGF1R and Akt inhibition also inducing the expression of ACSLs, preferentially ACSL5, and PI3K inhibition moderately suppressing SCD1 (new Supplementary Fig. 21a).
- Adjusting cellular PUFA-PC ratios to those of stressed cells by supplementation of arachidonic acid (20:4) sensitizes non-stressed cells to ferroptosis induction by RSL3 (new Supplementary Fig. 33b).
- Deletion of LPCAT3 (which incorporates PUFAs into membrane phospholipids) reduces the sensitizing effect of cytotoxic stress and serum deprivation on ferroptosis (new Fig. 8a, Supplementary Fig. 33c-e).

While we are convinced that metabolic changes upon apoptosis induction are accompanied by homeostatic regulations, here we have unraveled a mechanism by which cytotoxic stress increases the proportion of PUFAs in major membrane phospholipids, thereby sensitizing cells to membrane peroxidation. Whether the apoptotic cell has an intrinsic advantage in this regulation or whether it is a coincidental (homeostatic) regulation remains unclear. This question is also related to the ongoing debate about the physiological relevance of cell death induction by ferroptosis [PMID: 35803244]. Nevertheless, the outcome is the same: apoptotic cells undergo this metabolic adaptation and become vulnerable to redox stress, which we consider of high interest.

Reviewer #7:

Comment: In this manuscript, Gollowitzer, et al. used various mechanistically distinct cytotoxic stress inducers to trigger apoptosis in the fibroblast and found that lipid metabolism program was switched in the early apoptotic cells: SFA and MUFA biosynthesis decreased and synthesis of phospholipids with PUFA (PUFA-PLs) increased. They claimed that the lipid metabolism reprogram was attributed to impaired growth factor receptor-PI3K-Akt signaling. The accumulation of PUFA-PLs caused by apoptosis inducing agents sensitized cells to ferroptosis, revealing a hidden relationship between apoptosis and ferroptosis. Overall, this study provides some interesting viewpoints. However, mechanism studies in this version are weak and a lot of conclusions in this manuscript are inferred from the correlation of gene/protein expression profiles without further evaluation by gene interference experiments. The evidence to support the relationship between apoptosis and ferroptosis is also weak.

Answer: Thank you for the critical feedback and valuable advice. We have (experimentally) addressed the points of criticism raised by the reviewer, as described below. The challenge in this particular study is that the proposed mechanism covers a very broad range from cytotoxic stress to RTK signaling and changes in lipid metabolism to PUFA incorporation into phospholipids to membrane peroxidation. Although we are making great efforts to fill in the gaps with mechanistic and functional studies (not only by gene interference, but also by small molecule probes, ligands, and direct metabolite supplementation) before and during the revision, it will be impossible to confirm all these multiple links in a single study. We are therefore grateful for the reviewer's specific suggestions, which we have all investigated in more detail.

Comment: 1. This manuscript is way too long with 55 figures in total. The authors need to consolidate the figures and texts to highlight their findings.

Answer: We agree that this manuscript is extensive and have removed parts that do not directly support the main conclusions (i.e., the former Fig. 9c, Supplementary Fig. 3c, 18d-g, 22, 29, 33-38, 41, 42, 45 and Supplementary Note 2-4, 11, 13-15). We also simplified the figures and text by removing data on the negative control TNF α (which only sensitizes to apoptosis but does not induce it per se) and the unambiguous I3M (which decreases cell numbers, but does not seem to induce apoptosis under our experimental conditions). In fact, the length of the manuscript increased significantly from 25 to 55 figures when we included the experimental data requested by the reviewers from the first review round. For this reason, we are a bit hesitant to delete this information, which was considered important by the previous reviewers and for which we made great efforts.

Comment: The working model in Figure 10 is too complicative.

Answer: Done, we have now simplified the scheme and focused more on the key findings.

Comment: 2. PL-PUFAs were generally increased upon different treatments from 24-48h (Fig S5A), however, the cell death upon different treatments varied a lot from 24-48h (Fig S1 and S2). It seemed the accumulation PL-PUFAs did not correlate with apoptosis status. It is questionable whether apoptosis per se or some pre-apoptotic status is related to the lipid metabolism programming, for example cell cycle arrest. A recent study suggested that cell cycle arrest caused PL-PUFAs accumulation through a PL remodeling process (PMID: 37963466). The authors need to rule out this possibility. And it is better to include some more specific apoptosis inhibitors, e.g. BCL2/BCL-xL inhibitors in the experiments.

Answer: We thank the reviewer for this well taken point. First, we compared the kinetics of increasing PUFA-PC ratios (Fig. 1f) with cell cycle arrest (new Supplementary Fig. 9c) and with apoptosis induction (Supplementary Fig. 2a [PARP cleavage] and Supplementary Fig. 26c, d [PI/annexin V staining]) for VAL. The increase in PUFA-PC ratio (at 48 h) upon VAL treatment coincides with apoptosis induction (at 48 h) and is preceded by G1 cell cycle arrest (at 24 h). Importantly, cell cycle arrest is no longer evident at

48 h, strongly suggesting that VAL-induced upregulation of the PUFA-PC ratio is associated with apoptosis induction. Other cell death inducers start to increase the PUFA-PC ratio between 24 and 48 h of treatment (Fig. 1f), which is in the same range when the early apoptotic marker PARP is cleaved (Supplementary Fig. 2a). However, the reviewer is correct that when broken down to individual cytotoxic stressors, the increase in PUFA-PC ratio and PARP cleavage are not always in perfect agreement. We now point out this discrepancy in the manuscript (page 11, lines 4-8) and discuss that pre-apoptotic processes, such as cell cycle arrest, may contribute to the increase in the PUFA-PC ratio for certain stress conditions (page 11, first paragraph). To provide further evidence that apoptosis induction leads to an increase in PUFA-PC ratios, we treated fibroblasts with the BCL-2 family protein inhibitor ABT-737 and the selective BCL-2 inhibitor venetoclax, and, for comparison, with the CDK4/6 inhibitor palbociclib. In support of our hypothesis, the induction of apoptosis by the inhibition of BCL-2 family members (BCL-2, BCL-xL, Bcl-w) strongly increased the PUFA-PC ratio, whereas the selective inhibition of BCL-2 or CDK4/6 was not effective (new Fig. 1g and Supplementary Fig. 9d).

Comment: 3. The mechanisms how different apoptosis inducing agents inhibited de novo SFA/MUFA synthesis is not very clear. The authors showed that ACC and FAS mRNA level generally decreased, while protein level did not significantly decrease (up to 48h, Fig 3B, C,D). Since ACC activity was controlled by AMPK-mediated phosphorylation, the authors questioned that whether energy stress triggered AMPK activation and inhibited ACC activity. However, the result is controversial here. Only VAL and MC showed consistently increase of p-ACC (inhibition marker) as well as p-AMPK from 10min-6h (Fig 3E), while ADP/ATP showed that VAL and MC did not have energy stress (Fig 3H, time point unknown).

Answer: Thank you for bringing this obviously incorrect description to our attention. We have revised the sentence and now write the following on page 35, lines 10-12: "Note that cells under these conditions activate AMPK (Fig. 3e), which has been implicated in ferroptosis resistance as a master regulator of cellular energy homeostasis." AMPK is indeed a central energy sensor activated by allosteric binding to AMP, but it can also be phosphorylated at Thr172 by non-canonical kinases, i.e. CAMKK2, independent of energy stress [PMID: 28622524]. Our data suggest the latter, although this conclusion remains speculative and we have not discussed it further.

The reviewer is also absolutely right that the mechanism by which different stress conditions inhibit de novo fatty acid biosynthesis is quite diverse. The most consistent is the decrease in SCD1 expression (significant for CHX, TPG, VAL, trend for ETO, serum depletion, and MC), which we have already published under identical experimental conditions (Thürmer et al., 2022, Fig. 3e) and show here again for VAL (new Supplementary Fig. 14d). However, there are exceptions (STS) and it becomes clear that the different cytotoxic stressors have different effects on lipid metabolism, which is not very surprising. Much more intriguing is that, despite these many differences, they all increase the PUFA-PC ratio between 6 and 48 h.

Comment: General malonyl-CoA depletion might indicate that beta-oxidation may be the reason for SFA/MUFA catabolism. However, the authors did not have further experiment to confirm the function of beta-oxidation in apoptosis inducing agents-related lipid metabolism reprogramming.

Answer: Indeed, malonyl-CoA is a universal metabolic regulator of β -oxidation that inhibits the carnitine acyltransferase shuttle and thus the import of fatty acids into the mitochondrial matrix (PMID: 11814524). However, although very likely, we do not provide direct evidence that this regulation applies under our experimental conditions and have therefore deleted this speculative sentence in the Results section. In our efforts to streamline the manuscript (as recommended by the reviewer), we have decided not to perform experiments in this direction, thus opening up another aspect of research and further expanding the manuscript.

Comment: Moreover, there is a gap between RTK/PI3K/AKT signaling and SFA/MUFA synthesis/beta oxidation upon apoptosis inducing agents' treatment.

Answer: We agree and have now addressed this missing link. That the RTK-PI3K-Akt axis controls lipogenesis (including SCD1 expression) has been extensively described in the literature (PMID: 23000402), including our recent review (PMID: 38036934). This pro-lipogenic activity of RTK-PI3K-Akt is largely dependent on the activation of the transcription factor SREBP1, which is regulated at multiple levels, including proteolytic processing, nuclear transport and posttranslational modification. Accordingly, inhibition of the IGF1 receptor (IGF1R) by picropodophyllin, PI3K by LY294002, Akt by ipatasertib, or SREBP1 by fatostatin increased the PUFA-PC ratio (new Fig. 6e).

To further investigate the role of SREBP1 (with ACC, FASN and SCD1 as target genes) in mediating cytotoxicity-induced changes in lipid metabolism, we challenged fibroblasts with VAL and determined the expression and nuclear translocation of SREBP1 by immunofluorescence microscopy. In contrast to serum depletion (see answer to comment below), VAL did not markedly decrease SREBP1 expression, and even promoted its distribution to the nucleus (new Supplementary Fig. 21b). We speculated that VAL may regulate SREBP1 activity through posttranslational modification, as previously described for acetylation and phosphorylation (PMID: 38036934). In support of this hypothesis, an inhibitor/activator screen for SREBP1 regulatory pathways suggests a limited contribution of AMPK (BAY3827) and p53 (pifithrin- β) to the VAL-induced increase in PUFA-PC ratio (new Supplementary Fig. 21c).

Comment: 4. VAL and MC treatment upregulated ACSL5 expression (Fig 5A,B, C). The authors inferred that ACSL5 might be required for exogenous PUFA incorporation upon VAL and MC treatment. However, the authors did not provide experimental evidence to support ACSL5's function. Do other stimuli also upregulate ACSL5?

Answer: Thank you for this important question. We have now silenced ACSL5 in fibroblasts, challenged them with VAL for 48 hours and determined the PUFA-PC ratio. Knockdown of ACSL5 substantially increased the PUFA-PC ratio preferentially in vehicle-treated cells and less so in VAL-treated cells (new Fig. 4e). These data exclude that upregulation of ACSL5 is the driving force for increasing the PUFA-PC ratio. We suggest that ACSL5, together with other remaining ACSL isoenzymes (although lacking substrate specificity for PUFAs), sustains the efficient incorporation of exogenous fatty acids (including PUFAs) into PC, as indicated by our metabolic flux studies with d8-AA (Fig. 2i).

In addition to VAL and MC, ACSL5 expression is strongly upregulated by TPG (new Fig. 4d).

Comment: Interestingly, serum starvation, which was used to mimic RTA inactivation, increased both ACSL3 (selective for MUFA-CoA) and ACSL4 (selective for PUFA-CoA, Sup Fig 32A). Does RTK/AKT/PI3K signaling regulate ACSL3/4/5 expression? And how?

Answer: This is an important point that we have now addressed by inhibiting major components in RTK signaling. Inhibition of IGF1R by picropodophyllin, PI3K by LY294002, Akt by ipatasertib, and SREBP1 by fatostatin all increased the PUFA-PC ratio (new Fig. 6e), and picropodophyllin and ipatasertib additionally induced the expression of ACSLs, preferentially of ACSL5 (new Supplementary Fig. 21a), the major ACSL upregulated under cytotoxic stress (Fig. 4c).

Comment: How to explain why PUFA-PL synthesis increased upon serum starvation? Again, experimental evidence is important.

Answer: We do not claim that serum starvation increases the biosynthesis of PUFA-containing phospholipids. What we do claim is that under serum starvation (and cytotoxic stress in general), cells retain the cellular capacity to incorporate PUFAs into phospholipids, while losing the capacity to provide de novo synthesized fatty acids for phospholipid biosynthesis. This metabolic reprogramming is expected to shift the fatty acid composition of phospholipids from SFAs and MUFAs to PUFAs (without further regulation), as the fatty acid composition of phospholipids is highly dynamic and subject to continuous remodeling (PMID: 37611890). Changes in the availability of fatty acid substrates

will therefore, over time, be reflected in the fatty acid composition of phospholipids (unless there is active counterregulation).

Here, we used serum starvation as a control to deprive growth factors and found that a transient Akt activation at 6 h was followed by Akt inactivation at 24 and 48 h (Fig. 6b), resulting in a net increase in the critical period from 6 to 48 h (Fig. 6c). Correspondingly, the proportion of PUFA-PC first significantly decreases at 6 h and then significantly increases again at 24 and 48 h (Fig. 1f). The mechanistic insights (including new experimental data) underlying this regulation are summarized on page 32, second paragraph and page 33, last paragraph to page 34, first paragraph. The most important seems to be that serum starvation decreases the expression of the SREBP1 target genes ACC (Fig. 3c), SCD1 expression (as shown by us under exactly the same experimental conditions in Thürmer et al., 2022, PMID: 35624087) and, as a trend, FAS (Fig. 3d), apparently by decreasing the expression and nuclear availability of SREBP1 (new Fig. 7e), an RTK-PI3K-Akt controlled lipogenic transcription factor (PMID: 38036934). Note that we confirmed that inhibition of ACC or SCD1 is sufficient to increase the PUFA-PC ratio (new Fig. 2e).

In addition, serum depletion leads to energy depletion (Fig. 3h) and decreases the expression of SFA- and MUFA-activating ACSL1, while simultaneously upregulating other ACSLs, including PUFA-activating ACSL4 (Fig. 4c, Supplementary Fig. 32), which could lead to an imbalance of SFA/MUFA-CoAs vs. PUFA-CoAs. To confirm this hypothesis, we analyzed acyl-CoA levels by UPLC-MS/MS, but the method was not sensitive enough to detect PUFA-CoAs. Supplementation studies with the SCD1 product 18:1 further showed that 18:1 prevented the upregulation of ACSLs (Fig. 4c, Supplementary Fig. 32), suggesting that their regulation is downstream of de novo fatty acid biosynthesis.

Comment: 5. RTK ligands selectively increase PUFA-PIs (Fig 7), while apoptosis inducing agents and serum depletion mainly increase PUFA-PCs (Fig 1), which suggested that RTK signaling and cytotoxic stress may regulate lipid metabolism through different mechanisms.

Answer: Thank you for raising this point, which requires further explanation. First, RTK ligands do not selectively increase the proportion of PUFA-PI in Swiss 3T3 cells. Instead, they decrease the cellular proportion of PUFA-PI (Fig. 7a - bar chart), and secondly, this is not selective either. The proportions of PUFA-PE (Fig. 7a - bar chart), PUFA-PS (Supplementary Fig. 30a) and PUFA-PG (Supplementary Fig. 30a - for bFGF) are also significantly reduced. However, the reviewer is correct that in NIH-3T3 cells cytotoxic stress preferentially increases the PUFA-PC ratio, whereas in Swiss 3T3 cells RTK ligands have little effect on the PUFA-PC ratio, but rather increase the ratio of PUFAs in other phospholipid classes. It is also true that phosphatidylinositols are the phospholipid species most significantly regulated by RTK ligands in Swiss 3T3 cells (Fig. 7a - volcano plot). In fact, several PUFA-PI species are highlighted in the volcano plot even when the p-values from the multiple *t*-tests were corrected for false discovery (5%).

We were not actually surprised by these findings, because the mechanism we describe here is mainly based on the suppression of de novo fatty acid biosynthesis, which shifts the balance from SFAs/MUFAs to PUFAs. Since these fatty acids compete for enzymes in phospholipid biosynthesis, such a shift is translated into an increased proportion of PUFAs in phospholipids. Which phospholipid classes are most affected by this shift in fatty acid availability is determined by the expression pattern of phospholipid-metabolizing isoenzymes, mainly lysophospholipid acyltransferases and phospholipases A, which often have broad, partially overlapping substrate specificities (PMID: 29410529, PMID: 37611890, PMID: 21910409). On the one hand, the expression, distribution and/or activity of these enzymes, lipid transport proteins and other contributing factors is likely to differ between cell lines (NIH-3T3 vs. Swiss 3T3). On the other hand, the cellular state also has an enormous impact on the expression pattern of phospholipid metabolic proteins, as confirmed here by quantitative proteomics for cytotoxic stress induction (Supplementary Fig. 19a, Supplementary Data 2). This means that cytotoxic stress and RTK activation have complex effects on the enzymatic machinery that decides in which phospholipid classes PUFAs are preferentially enriched. Elucidating these regulations in detail is

beyond the scope of the current study and would require a detailed understanding of the substrate specificities of these isoenzymes along with their contribution to the biosynthesis of individual phospholipids in a cellular context – a frontier research area with many open questions.

Comment: 6. The link of PUFA-PLs accumulation upon apoptosis inducing agents to sensitize ferroptosis is weak. Fer1 (ferroptosis inhibitors) did not obviously affect apoptosis inducing agents-induced cell death comparing to RSL3 (GPX4 inhibitor, classical ferroptosis inducer, Fig 8A). There is some controversial result in Fig 8C that RSL3 alone did not inhibit fibroblast cell viability. Despite that, it also confirms that Fer1 did not obviously restore VAL-decreased cell viability (Fig 8C). Very delicate experiments are required here to confirm the function of ferroptosis in late apoptosis. For example, do apoptosis inducing agents have different potency on wt and ACSL4 ko cells?

Answer: Thank you for pointing out this seemingly misleading section; we have performed the suggested experiment and revised the text for clarity. In fact, we do not claim that cytotoxic stressors necessarily induce ferroptosis. Inhibitor studies with Fer-1 show that this is only the case for serum depletion and less so for VAL and MC (Fig. 8b). What we are claiming is that the increased PUFA-PC ratio acquired during apoptosis sensitizes cells to membrane peroxidation. This is a completely different statement because it means that cells do not directly undergo ferroptosis, but become more vulnerable to membrane peroxidation/ferroptosis induced by an additional stress. We induced such a stress using the GPX4 inhibitor RSL3 at sublethal concentrations and indeed found that the cytotoxic stressors VAL, MC, CHX, ETO, and TPG significantly sensitize cells to RSL3, an effect that was completely prevented by Fer-1 (Fig. 8f, Supplementary Fig. 34a). Note that we deliberately set the RSL-3 concentration at a level that does not induce cell death under our experimental conditions in order to better detect sensitizing effects of the cytotoxic stressors and to distinguish between synergistic and additive effects.

The reviewer is correct that Fer-1 partially protected cells against VAL-induced decrease in cell viability as measured by MTT assay (an early event in ferroptosis; Fig. 8b), but not against VAL-induced disruption of the plasma membrane (a late event in ferroptosis; former Fig. 8c). These data are consistent with our hypothesis that early apoptosis sensitizes cells to later membrane peroxidation. According to this hypothesis, VAL first induces apoptosis, resulting initially in a low number of dead cells (a process that cannot be prevented by Fer-1; former Fig. 8c). Then, sensitization to lipid peroxidation is accelerated and early ferroptotic events (such as a decrease in mitochondrial functionality - a process that can be attenuated by Fer-1; Fig. 8b) become apparent, probably because VAL induces some ferroptotic stress (independent of the mechanism described here). At the same time, VAL (and other cytotoxic stress conditions) sensitizes cells to ferroptotic stimuli such as RSL3 (new Fig. 8c, f). We have now removed the former Fig. 8c and replaced it with data from new experiments, not only for VAL but also for other cytotoxic stressors (new Fig. 8f and Supplementary Fig. 34a).

These findings are entirely in line with the role of the membrane PUFA ratio in ferroptosis as described in the literature: higher membrane PUFA ratios do not induce ferroptosis per se, but sensitize cells to membrane peroxidation and ferroptosis induced by redox stress (e.g. inhibition of GPX4) [PMID: 35459217], now also confirmed for our experimental system in Supplementary Fig. 33b.

According to the here described mechanism, the deletion of enzymes that contribute to the incorporation of PUFAs into phospholipids, such as ACSL4 or LPCAT3, should not alter the sensitivity to VAL, but should reduce the sensitivity of VAL-treated cells to ferroptotic inducers, such as RSL3. Indeed, we were able to confirm this hypothesis by deleting LPCAT3 followed by exposure of stressed and non-stressed cells to RSL3 and determination of membrane intactness and cell viability (new Fig. 8a and Supplementary Fig. 33c-e).

Reviewer #8:

Comment: My focus in reviewing this manuscript is specifically on the proteomic data. In "NCOMMS-19-00380A-Z, the authors use a standard lable free-quantitative proteomic approach of cells exposed to cytoxic stress. I found the proteomic section to be very well written and the data presented is high quality. The method section is sufficiently detailed in accordance with proteomic standards with one exception. I do not see where the data was uploaded to a shared data repository like PRIDE. Please upload the data as required to ProteomExchange or PRIDE.

Answer: Many thanks for the evaluation of the proteomics data now uploaded to PRIDE (dataset identifier PXD053866; username: reviewer_pxd053866@ebi.ac.uk, password: fjQBHBezGvzw), as indicated in the Data Availability section.

Reviewer #9:

Comment: The authors put a lot of effort and new experimental setups to complement and expand the relevance, robustness and significance of the study. Multiomics approaches were added to support the mechanistic findings and interpretation. However, my expertise only tangentially includes pathway analyses, therefore, I will focus on providing feedback on the imaging part as requested by the editors. 1) Proteomic & pathway analyses indicate alterations in glycolysis & oxidative phosphorylation (Fig 4 a,b, Supplements). It would have been expected that these changes might have been reflected in the free vs bound NADP(H) ratio (Fig. 9c), however different experimental batches indicate opposite trends in free vs bound NADP(H) ratios. As FLIM is highly sensitive to environmental and culture condition the robustness of the FLIM measurements has to be confirmed.

Answer: Thank you for the valuable feedback on the imaging section. Please note that the former Fig. 9c does not show opposing trends between independent datasets, but only no effect of VAL on the ratio of free and protein-bound NAD(P)H (3.3 ± 0.3 vs. 3.2 ± 0.3). Thus, the SEM is in the normal range for cellular treatment studies and there is no trend to be seen for the three independent experiments (1x no effect, 1x slight increase, 1x slight decrease). However, since we were asked to shorten the manuscript and Fig. 9c and Supplementary Fig. 42 are not central to the proposed mechanism, we decided to delete these figures.

Comment: In the methods, it is stated that cells were fixed before NADH imaging, was there a particular reason for this and were the times between fixation and data acquisition the same between all experimental batches?

Answer: We preferred to fix cells because sample preparation and imaging were performed in different locations, making (stress-free) transfer of live cells a challenge. The times between fixation and data acquisition were the same for samples from independent experiments. There was less than two days between independent experiments. We have verified that autofluorescence signals in fixed cells remain stable for up to 1 week when cells are stored at 4°C until imaging experiments are performed. Live cell experiments are much more prone to error. We have observed changes in live cells within 30 minutes of imaging due to cellular stress. Performing the experiments on fixed cells therefore provides more robust results. Please note that we are not interested in absolute quantification, but in the direct comparison of vehicle and VAL-treated cells. Therefore, small differences in analytical readouts should be negligible compared to the differences due to biological parameters, both of which are (partially) compensated by the paired statistical test used to compare the two groups. In summary, the significant effects in Fig. 9a become even more reliable (because more difficult to achieve) the greater the differences between individual datasets.

Comment: Furthermore, the authors removed the section on p-Akt imaging; does this imply that FLIM measurements were repeated without the AF488 co-staining?

Answer: No, we just removed the criticized part about p-Akt imaging and left the section on NAD(P)H, as indicated in the method section on p. 78, first paragraph.

Comment: 2) CARS of the CH₂ band (2850 cm⁻¹) was used to determine overall lipid content in VAL-treated cells. Based on the images provided in Suppl. Figure 3c, it is difficult to evaluate the localization of CARS signals. Would it be possible to provide a CARS heatmap at a better contrast. Similarly, the graph plotting CARS intensities (Suppl Fig. 3c) is difficult to interpret regarding a 'perinuclear' localization of lipids.

Answer: Since these data are not supportive for the main conclusions of the manuscript and we were recommended to streamline the manuscript, we have decided to delete Supplementary Fig. 3c.

Response to the reviewers

Reviewer #5:

Comment: Authors performed substantial amount of work answering reviewers comments and significantly improve quality of manuscript. But I still have several comments.

Answer: Many thanks for the positive feedback and appreciating our efforts.

1. Authors demonstrated, that agents inducing cytotoxic stress increased PUFA ratio in different phospholipids, especially PC, while less pronounced increase in PUFA ratio were found in PE, PS, and PI. This phenomenon promoted oxidative membrane damage and sensitize cells to ferroptotic stimuli such as GPX4 inhibitor RSL3. It was indeed demonstrated in many publications, that development of ferroptosis is accompanied by increase in lipid peroxidation. At the same time PE but not PC has been identified as a crucial PL oxidized during ferroptosis. For instance, studies of different types of cell death in bone-marrow-derived macrophages (BMDMs) demonstrates, that while in apoptotic, necroptotic, and pyroptotic BMDMs mostly oxidation of cardiolipin and/or phosphatidylcholine were detectable, BMDMs undergoing ferroptosis revealed mainly oxidized phosphatidylethanolamine (oxPE) species followed by oxidized phosphatidylserine (oxPS) and phosphatidylinositol (oxPI). (Bartosz Wiernicki et.al. *Cell Death Dis* .2020 Oct 27;11(10):922). I think it will be very useful if authors can provide some considerations explaining how their results about predominant accumulation of PUFA PC coordinate with significant role of PE in ferroptosis.

Answer: Thank you for this excellent point, which addresses a controversial issue in ferroptosis research. Firstly, we would like to point out that the studies on BMDM are difficult to compare with our studies on fibroblasts. BMDM express higher levels of 15-lipoxygenase, which is recruited to membranes by PEBP1 and accepts phospholipids as substrates, thereby contributing to PE peroxidation [PMID: 29053969]. Other cells, such as fibroblasts, express much lower levels of 15-LOX and the role of lipoxygenases in ferroptosis induction has been questioned and may be negligible compared to non-enzymatic lipid peroxidation (e.g. via the Fenton reaction). Nevertheless, we and others have observed that even in such cells, strong ferroptosis inducers lead to (per)oxidation of PE, PS and PI more than PC. However, the question of which phospholipids are responsible for ferroptosis sensitivity remains a hot topic in ferroptosis research and is still controversial due to the lack of in-depth functional studies. In favor of PC would be its high abundance in various membranes, and in fact it is considered by leading researchers in the field as a relevant phospholipid class that can be sensitive to peroxidation and can contribute to ferroptosis [PMID: 38908072]. Since ferroptosis can spread from different intracellular membranes to the ER and plasma membrane [PMID: 36747055], the question arises whether the same phospholipid classes contribute to ferroptosis in all these membranes. Other studies have focused on di-PUFA phospholipid species [PMID: 38366593] or ether phospholipids [PMID: 32939090]. Taken together, there is still active research in this field and it is still unclear which phospholipid classes drive ferroptosis under which conditions. Our study will be another piece of the puzzle towards a better understanding of these processes. Note that these considerations do not contradict our main finding that cytotoxic stress shifts the membrane phospholipid composition towards higher ferroptosis sensitivity. Indeed, all seven cytotoxic treatments used in this study not only increased the PUFA-PC ratio, but also significantly elevated the ratio of PUFAs in either PE, PS or PI (PE: ETO, CHX, Serum, STS; PS: VAL, CHX; PI: ETO, TPG, Serum, STS). The effects on the latter phospholipid classes seem to be weaker than for PC, but this is not necessarily related to a less severe response to metabolic changes, but may also depend on the higher basal PUFA proportion of PE, PS and PI compared to PC, which per se reduces the percentage increase by a fixed amount. We now discuss these points on page 47, last paragraph – page 48, first paragraph.

Comment: Minor concerns: 2. Fig.8b. The x-axis on it has name "difference of means". but it is not clear what units was used.

Answer: Done, the figure legend now states that for each cytotoxic condition we calculated the difference between Fer-1- and vehicle-treated cells in mean metabolic activity (% of vehicle control) measured by MTT assay.

Comment: Figure has title "Fer-1 vs. w/o" and contained mostly readings from 0 to100. If Fer-1 serve as inhibitor, differences between readings with Fer-1 vs readings without it needs to be negative, or <1. Maybe its w/o vs Fer-1.

Answer: Based on your comment, we have carefully checked the "Fer-1 vs. w/o" label again, but cannot find any inconsistencies. Positive values in the x-direction indicate that the metabolic activity of the cells is higher in cells treated with Fer-1 and exposed to the indicated stress conditions compared to cells that were only exposed to the respective stress (difference > 0) [Calculation: Metabolic activity of Fer1-treated and stress-exposed cells in % of vehicle control - Metabolic activity of stress-exposed cells in % of vehicle control], which is significantly the case for the control ferroptosis inducer RSL3, serum depletion, and less so for VAL and MC. Note that this experiment is not related to our conclusion that cytotoxic stress sensitizes cells to ferroptosis, but goes one step further and shows which cytotoxic stressors directly induce ferroptosis, at least to some extent.

Comment: 3. Picture in lower left corner of Fig.8 is not marked. It looks like 8c.

Answer: Yes, the previous Figure 8c showed both the microscopic images and the quantitative calculation. To avoid confusion, we have now labeled these two panels independently.

Comment: 4. On Fig. 10 authors use indexes demonstrated changes in protein/metabolite availability or protein phosphorylation induced by cytotoxic stressors. The first number in the brackets indicates the number of cytotoxic stressors showing significant effects ($P < 0.05$) the second number states the total number of stressors investigated in the experiments. In this classification RAS/p-ERK has index (0/6) while high energy status low ADP/ATP and low-AMPK have indexes(1/7) and (2/7) respectively. Those indexes indicates that practically no stressors effected those parameters. May be its reasonable to remove them from schema?

Answer: Agreed, these parameters / indices have been removed.

Comment: 5.Line 985 authors stated:" inhibition of IGF1R, PI3K, Akt or SREBP1 substantially induces lipogenic gene expression and increases the PUFA-PC ratio". At the same time the main idea of this manuscript is decrease of SFA and MUFA biosynthesis during cytotoxic stress causing the shift towards PUFA-containing phospholipids.

Answer: Thank you for bringing this error to our attention. The sentence should read "inhibition of IGF1R, PI3K, Akt or SREBP1 substantially **suppresses** lipogenic gene expression and increases the PUFA-PC ratio."

Reviewer #6:

Comment: My comments only reate to the lipidomics data which is actually used to make the main conclusion of the paper. Unless NatComm editorial board thinks otherwise, already published data should not be used again in another manuscript.

Answer: We appreciate the reviewer's careful consideration of this important issue, but disagree with the conclusion. On the contrary, we strongly believe that re-analysis of large datasets to extract new information is consistent with the purpose of public repositories and with the Nature portfolio editorial policy on plagiarism and duplicate publication (<https://www.nature.com/nature-portfolio/editorial->

policies/plagiarism). These state that “Duplicate (or redundant) publication occurs when an author reuses substantial parts of their own published work without providing the appropriate references. This can range from publishing an identical paper in multiple journals, to only adding a small amount of new data to a previously published paper.” These criteria do not apply to our manuscript for several reasons:

First, the reused/reanalyzed lipidomics datasets (provided in previous manuscripts or repositories) are limited to Fig. 1c (1 panel), Fig. 1f (for one of the four line plots shown → 1/4 panel), Fig. 3i (→ 1 panel), and Supplementary Fig. 3c (for one of the five phospholipid classes shown in the PCA → 1/5 panel); a total of 2.45 panels out of 166 panels, i.e. 1.5% of the total data). In addition, we mention on page 59, last paragraph – page 60, first paragraph, that the lipidomics dataset behind Fig. 7a-d (→ 4 panels) and Supplementary Fig. 30a-e (→ 5 panels), which has not been publicly available, was used 14 years ago to extract other information (not related to the current manuscript). Since the entire dataset has never been made public (until now), the information we provide here is largely original and does not fall into the category of “reuse”. Nevertheless, we explicitly mention it in the manuscript to allow the reader to better compare these two datasets from the two manuscripts, knowing that the data refer to exactly the same independent experiments and lipidomic analysis.

Second, the manuscript clearly indicates in the figure legends and/or the methods section all datasets that have been reused/reanalyzed (page 59, last paragraph – page 60, first paragraph).

Third, and most importantly, we do not simply republish data as it was, but use data from large datasets, show aspects that have not been considered before, and interpret them in the new light of current knowledge. Thus, Fig. 1c highlights a different cluster of phospholipids than previously focused on, Fig. 1f visualizes for the first time the effect of cytotoxic stressors on the PUFA-PC ratio, and Supplementary Fig. 3c shows for the first time the results of a multivariate analysis across four new phospholipid classes and one that was reanalyzed to get an overall picture of the lipidomic changes.

Although the reviewer focuses solely on our lipidomics data, for completeness we would also like to mention that we re-used treatment group data from a proteomics dataset and combined it with new groups that were measured in the same run but were not needed for the previous project. Again, the data that we present in the main and SI figures are not a repetition of what has been analyzed before, but address a completely different research question, provide new insights, and focus on proteins that were not considered in the previous manuscript.

To make it clearer what lipidomics data is actually being reused/reanalyzed and what the new insights are, we have revised the description on page 59, last paragraph – page 60, first paragraph.

Comment: "Reanalysis of lipidomics datasets previously published by us to achieve consistency of lipid species between different figures" is not a valid justification.

Answer: We do not agree with this statement. Since we used a targeted approach, it is reasonable to make minor adjustments in the selection of lipid species, e.g. to calculate the proportion of the PUFA-containing fraction in a particular cell line based on the same set of species in different figures. On page 59, last paragraph, we now describe more clearly our intention to reanalyze previous datasets.

Comment: Moreover, specially in lipidomics analysis, one should be careful when using old data to compare datasets and draw conclusions from there.

Answer: There seems to be a misunderstanding here. The manuscript does not make comparisons between old and new datasets. For example, the "new" lipidomics data on PC, PE, PS, PG come from exactly the same samples as the "old" PI data, measured in the same LC-MS/MS batch. The same is true for the proteomics data. Consequently, statistical comparisons are only made between groups derived from the same set of experiments.

Comments: The authors say that they did not trust their PI standard for quantifications because of the differences with long chain PIs in ionization and thus they used DMPC which does not make sense

because DMPC is even more different structurally. Then they corrected the values by external calibration with PI(15:0/d7-18:1) stating that "At the time this was measured, we did not have a deuterated PI standard". This raises again the same concern of using these old data to compare and make statements with newly acquired data. In lipidomics this is not trustable nor acceptable.

Answer: We understand the reviewer's concern and have now removed these values from the source data, although we respectfully disagree with the general criticism of the PI data normalization method. This method, while impacting only the source data of a single subpanel of one figure, follows a standard analytical procedure using an internal standard to correct for extraction efficacy and external calibration to normalize the mass spectrometric signal intensities. The data calculated in this way are not even shown in the figure, but have only been added to the source data as additional information for the interested reader. However, the reviewer is correct that in this case we had to use "new" measurements of the standard to normalize "old" datasets, which might slightly affect the absolute values of the data (notably in the same way for all groups being compared).

We also disagree that PI(8:0/8:0) should be a better standard than DMPC, as normalizing the data to PI(8:0/8:0) would falsely suggest that PI is the most abundant phospholipid in cellular systems. Such a conclusion would contradict what is known from the literature as well as our own results from later studies based on an internal deuterated PI standard. The main problem is the large difference in lipophilicity. While the abundant PI species in biological membranes (e.g. PI(18:0/20:4) or PI(18:1/18:1)) have a similar lipophilicity (and thus retention time on reversed-phase columns) compared to DMPC, PI(8:0/8:0) is much less lipophilic. As a result, the extraction yield is considerably lower compared to phospholipids with longer chain fatty acids (Fig. X1), resulting in an overestimation of the amount of these PI species. When bypassing the extraction and injecting equal amounts of these lipids directly into the UPLC-MS/MS system, the difference in signal intensities between the standards is far less pronounced.

Fig. X1 Ratio of internal standard signal intensities before and after extraction. The internal standards PI(8:0/8:0), PI(15:0/18:1-d7) and DMPC were analyzed by UPLC-MS/MS before and after extraction from aqueous solutions according to the analytical setup described. Ratios are indicative of extraction efficiency. Mean \pm s.e.m. and single data.

The considerations of the reviewer regarding the ionization and fragmentation efficiency are correct to the extent that both PI(8:0/8:0) and DMPC are not ideal internal standards. PI(8:0/8:0) differs strongly in retention time from other PI species and DMPC has a different polar head group which also has an influence. However, unlike the extraction yield, the effects on ionization and fragmentation efficiency are corrected by the external calibration. Therefore, such an approach does not lead to inaccurate data, although it may suffer from higher variance compared to our preferred and meanwhile applied approach using a deuterated internal PI standard.

Comment: If the manuscript is finally accepted, the authors must upload ALL the lipidomics data related to this paper (new and already published) together to a repository prior to final acceptance and publication.

Answer: Done, the lipidomic data has been uploaded to repositories and the permanent IDs (of both new and previously published data) are listed in the data availability section. Please note that the new data are not yet released, but will be made publicly available before the manuscript is published. Reviewers can already gain access using the login information provided in the "Data availability" section.

Comment: I find the manuscript still too long and specific for the scope of this journal.

Answer: Thank you for sharing your opinions with us. Indeed, we have received a lot of valuable feedback from a total of nine reviewers, which has greatly improved the now very comprehensive and information-dense manuscript. Actually, we are very grateful to the reviewers and see the length and also the specific details (mainly in the SI) as a strength of the manuscript.

Reviewer #7:

Comment: In this revised version, the authors have generated some new data to address the reviewer's comments. However, this reviewer still has some major concern regarding the quality of this study. There is not a consensus and clear mechanism of how different cytotoxic stressors suppressed RTK-PI3K-AKT-lipogenesis pathways to increase the incorporation of PUFAs into phospholipids, which greatly limits the significance of this study.

Answer: We appreciate this feedback but do not fully understand the criticism. It seems that the reviewer is concerned that different stressors all affect growth factor and PI3K signaling, affect fatty acid biosynthesis through different mechanisms, and then consistently increase the PUFA-PC ratio. These results are beyond our control and we find them intriguing.

Comment: Importantly, the authors showed that the remodeling of phospholipids by these cytotoxic stressors is caspase independent. This raised an important question that whether apoptosis initiation per se is required generally for the suppression of RTK-PI3K-AKT-lipogenesis pathways. The authors can test whether pan BCL inhibitors and BH3 mimetics can induce the similar phospholipid alteration as cytotoxic stressors.

Answer: This is indeed a very relevant question, but we would like to respectfully point out that this is exactly what we show in Figure 1g. Pan-inhibition of BCL-2 family members increased the PUFA-PC ratio as expected, whereas selective inhibition of BCL-2 had no effect.

Comment: If yes, the authors need to dissect very carefully how apoptosis initiation regulates RTK-PI3K-AKT-lipogenesis pathways in upstream. If not, the authors need to spend more efforts to elucidate how those cytotoxic stressors inactivate RTK-PI3K-AKT pathways in common (ER stress, redox stress, cell cycle arrest, etc). Otherwise, the conclusion that attenuated RTK signaling during cell death initiation increases cells' susceptibility to oxidative membrane damage at the interface of apoptosis and alternative cell death programs is way too broad.

Answer: We now mention this interesting area for future research on page 44, lines 10-12. However, without information on which pathway(s) might be involved, identifying and studying the mechanism by which apoptosis regulates the RTK-PI3K axis would require substantial additional work, which may be more appropriate for a follow-up project rather than the current, already extensive manuscript.

Synopsis:

The manuscript “A switch in fatty acid metabolism during apoptosis attenuates subcellular AKT survival signalling” by Pein *et al.* reports that changes in the subcellular lipid composition of apoptotic cells attenuate AKT survival signalling.

The authors show that fibroblasts stimulated with a variety of apoptosis-inducing agents display an increase in cellular PUFA-PC (poly-unsaturated fatty acids-containing phosphatidylcholines) levels after 48h, which the authors attribute predominantly to the (peri)nuclear region of these cells. The authors also investigate the levels of S473-phosphorylated AKT (p-AKT), which they use as a proxy for AKT activity. They find that the increase in PUFA-PC correlates with a decrease in p-AKT and conclude that an increase in PUFA-PC levels decreases AKT survival signalling in these cells. The authors claim that the reduction in p-AKT is predominantly in the (peri)nuclear region of cells.

Next, the authors attempt to provide evidence that the lipid metabolism is dramatically changed in apoptotic fibroblasts, showing that *de novo* lipid synthesis is downregulated and β -oxidation is upregulated in early apoptosis. They show that apoptotic cells are in a low energy state, which they assess with the ratio of ATP to ADP and the overall cellular NADPH levels. The authors also report that the inhibition of lipid synthesis leads to a similar phenomenon of increased PUFA-PC levels and decreased p-AKT. Using a caspase inhibitor the authors propose that these observations are independent of caspase activity.

Ultimately, the authors report that the correlation between increased PUFA-PC and reduced p-AKT can also be made in a number of cancer cell lines and to a lesser extent in primary human cells. In conclusion, the authors suggest that there exists a general mechanism of lipid metabolism (“feed forward loop”) that accelerates apoptosis by attenuating AKT-dependent survival signalling.

Overall judgement:

Whilst the overall topic of the paper is very interesting, and some of the data is intriguing I would **not** recommend publication of this work in Nature Communications. My biggest concern with the manuscript is the handling and interpretation of data which appears to be very selective and biased. Also, several conclusions drawn from the individual experiments are - in my opinion - not supported by the data.

Concerning the overall novelty of this paper, many of the findings presented in this manuscript are not entirely new, but have been reported in similar settings before.

- 1) The fact that apoptosis reprograms the lipid composition of the cell has long been known (eg, see a brief review by Esposito, Cell Death and Differentiation 2002). However, the finding that apoptosis leads to an increase in PUFA-PC is indeed novel and very interesting.
- 2) It is also long known that the inhibition of PI3K/AKT signalling has been shown to induce apoptosis and therefore AKT signalling is considered to be anti-apoptotic. So it is no surprise to see that p-AKT levels are downregulated in apoptosis. It is rather more surprising that p-AKT is not downregulated in all the apoptotic settings the authors report.
- 3) It is also known that AKT signalling is not only dependent on PIP3 as an activating lipid but also dependent on the lipid composition of the membrane. To my knowledge, a thorough analysis of PUFA-PC species on their effect on AKT activity has not been reported. The claim that PUFA-PC can abolish AKT activation is therefore also interesting, but has been previously proposed by the same author (Andreas Koeberle et al., PNAS, 2013). However, the current manuscript does not provide any new mechanistic insight into how PUFA-PC inhibits AKT activation.

The novelty of this manuscript is thus limited to the claims that a general feed forward loop of anti-correlated PUFA-PC and p-AKT drives apoptosis. Unfortunately, the data presented in this manuscript only report correlations between these two measurements in certain settings or certain cell lines. Sometimes these correlations do not hold true in the majority of apoptotic agents or cell lines investigated. There is no evidence provided that such a feed forward loop would actually accelerate apoptosis or is just a “by-product of apoptosis”. Overall, I am very doubtful of the findings, conclusions, and claims of this manuscript.

Major comments:

Fig. 1-2

The authors make a quantitative assessment of the PUFA-PC levels, by reporting the ratio of PUFA-PC to total PC (e.g. Figure 1e). This is reasonable given that the level of total phospholipids, including PC, decreases during apoptosis (Figure 1a; Supplementary Figure 3a). However, when looking at p-AKT it appears that the authors only report the p-AKT density determined by western blot, without normalizing it to the total levels of AKT (e.g. Figure 2d). This is problematic because AKT levels seem also to go down during apoptosis in some cases (Figure 2d, Supplementary Figure 15). The drop in AKT could be due to the overall decrease in cell number (as compared to untreated cells, Supplementary Figure 1c) or simply due to the degradation of AKT in apoptosis. In any case, this must be taken into

account. For instance, if the p-AKT signal in figure 2d were to be normalized by the AKT signal, the effect would most probably be less pronounced for all apoptotic agents (but ETO).

The authors make an important, but not validated assumption, which is that PIP3/PI(3,4)P2 levels are unaffected. It is well established that Akt depends on at least one of these lipids for both activation and activity. Given that the authors observe a dramatic decrease in total phospholipids during apoptosis, it is perhaps not surprising that they might also see reduced pAkt levels (although the evidence for this is also problematic, as described above).

Regarding the subcellular localization of PUFA-PC, the authors use CARS and TPEF microscopy (Figure 1f). From the data provided it is not clear how the authors can conclude that PUFA-PC is enriched in the perinuclear or nuclear region. The CARS signal intensity should be proportional to the C-H₂ symmetric vibrations (at 2845 cm⁻¹) (Folick A. et al, 2011), which is dependent on the length and the degree of unsaturation of the acyl chain. How do the authors distinguish between shorter fatty acids (with less CH₂ groups) and polyunsaturated fatty acids? It seems that with CARS microscopy only lipid-rich regions of the cell can be distinguished from parts that contain less lipids. Given that membrane density is high in the perinuclear region, it seems questionable what has been learned here.

Similar criticism applies for the “subcellular localization” of p-AKT (Figure 2e, Supplementary Figure 8a). The localization in untreated cells seems very different in Figure 2e and Supplementary Figure 8a, when in fact it should look the same. This could be due to a difference in confocal plane. It seems that both VAL treatment and LY294022 treatment reduces the overall intensity of p-AKT, but it is not justified to claim that there is a change in subcellular localization of p-AKT signal as the change between the two untreated cell pictures (compare Figure 2e and Supplementary Figure 8a) appears to be more pronounced.

Regarding the subcellular fractionation (peri-nuclear vs non-nuclear membranes) it is also doubtful whether one can infer that there is any difference in membrane composition on the perinuclear membrane. In fact, it seems that the effect of increased PUFA-PC levels holds true for both (peri)nuclear and non-nuclear membranes when induced with VAL compared to untreated cells (Supplementary Figure 5b). In this respect, one of the key claims of the paper, which is that subcellular Akt signalling is suppressed by PUFA accumulation in the perinuclear region, is actually contradicted by their data.

In summary, there is no convincing evidence provided for any subcellular change of PUFA-PC nor p-AKT.

Regarding Fig 2c; How did the authors quantify the data to obtain a more than 100% decrease in p-AKT signal for VAL treated fibroblasts? What does a more than 100% decrease in p-AKT even mean?

One final comment to pg. 9, ln 11-13: the authors use an Akt inhibitor to exclude that decreased Akt activity during apoptosis does not lead to PUFA-PC accumulation (Supplementary Fig. 7b). However, they propose a feed-forward mechanism in which decreased Akt signalling promotes apoptosis, which leads to PUFA-PC accumulation (Supplementary Fig. 16). The authors should think carefully about the apparent contradiction between their data and their model.

Fig. 4

The authors show that ACC and FAS are transcriptionally downregulated during apoptosis on the mRNA level. Again this could be due to a generally lower amount of cells or degradation of mRNAs or lower transcription. Interpretation of the authors' findings is complicated by the fact that the two "house-keeping" genes (GAPDH, β -actin) are also highly variable over time and according to treatment (Fig 4e). In any case, the observed downregulation on the mRNA level is not manifested on the protein level with any statistical significance. Only in serum deprived cells is ACC significantly downregulated and only in VAL-treated cells is FAS significantly downregulated (Figure 4f, 4g). This data can therefore not explain the overall switch in lipid metabolism. Moreover, how should a change in mRNA levels but not protein levels affect enzymatic activity? One possibility the authors explore is the change in ACC phosphorylation state. The authors claim that the altered levels of malonyl-CoA that they clearly see (Figure 4i) is due to ACC inhibition by phosphorylation of ACC (p-ACC) which is reportedly carried out by AMPK, but the blots in Supplementary Figure 11 don't support this conclusion. After 48h p-ACC is most strikingly found in the untreated cells and TNF α (Supplementary Figure 11a). Overall, the p-ACC is quite different depending on which stimulus was used and never increased compared to untreated cells. Therefore this cannot explain the decrease in fatty acid biosynthesis. In summary the authors conclude that "fatty acid biosynthesis was strongly reduced for many apoptotic settings with varying kinetics and mechanisms that include

- i) the decrease of ACC and FAS expression
- ii) ACC inhibition by posttranslational modification
- iii) limitations in ACC substrate supply
- iv) energy depletion

However the data presented show

- i) ACC and FAS protein levels are only downregulated in one treatment each
- ii) ACC is not inhibited by phosphorylation (it is not more phosphorylated/inhibited than in the untreated control)
- iii) the ACC substrate acetyl-coA is not significantly reduced, only in TPG treated cells (however, the ACC product malonyl-CoA is reduced in 6 out of 8 cases)
- iv) the ATP/ADP levels are the same in almost all cells (treated/untreated), however the NADPH levels seem to be increased (Figure 5)

Fig. 5

The authors report a change in ADP/ATP levels when fibroblasts are treated with apoptotic agents. However, only 2 out of the 8 treated fibroblasts have an elevated ADP/ATP ratio (Figure 5a). More worrying though, is that the ADP/ATP ratio in cells that are untreated is reported to be 1, which is not considered physiological for healthy cells. The authors cite a paper (reference 36) that indicates that in normal proliferating cells the ADP/ATP ratio is below 0.11, whereas in apoptotic cells the levels increase to 0.1-1.0. This paper also reports a ratio of ADP/ATP of 15 for heat-shocked necrotic cells. If the authors agree with this study, they must conclude that all their ADP/ATP ratios indicate apoptosis (also the untreated and TNF α treated cells) and that treatment only affects ADP/ATP ratios for 2 out of 8/9 treated cells. Perhaps the authors have artificially set the ADP/ATP ratio w/o treatment to 1, in which case all other values are reported relative to the control cells, but this is extremely confusing and misleading.

Fig. 6

Figure 6h,i,k reports the change in PUFA-PC/total PC, p-AKT density, and cl.PARP density for cells that are untreated or "pre-treated" with TOFA, before treatment with apoptosis inducing agents. It seems that the grey and green bars (non-pretreated) cells were normalized to the w/o cells without TOFA, and that all the white bars represent normalized values to TOFA-pretreated cells. In my opinion it is not valid to statistically compare these two quantities (between green and white bars) that are normalized to different values. Figure 6b shows that TOFA treatment changes the PUFA-PC ratio (Figure 6b), but the values for TOFA-treated and untreated cells are shown to be equal in Figure 6h. Similarly, p-AKT density with TOFA treatment decreases in Figure 6c, but is again shown to be equal in Figure 6i. Finally, Figure 6j shows a big difference in cleaved PARP levels between TOFA-treated cells, but the values of treated and untreated cells are again set to be 100% in Figure 6k. The conclusion that TOFA does not have an additional effect to the apoptosis stimulating agents is flawed, at least on the basis of how the data has been treated/presented.

Fig. 7

Figure 7a shows that the pan-caspase inhibitor (Q-VD) either does not affect the western blot density of caspase 3 or even increases the level of uncleaved caspase 3. This is in striking contrast to Supplementary Figure 14a, representative for the data presented in Fig. 7a, which displays a marked decrease in caspase 3 signal in response to Q-VD. It is therefore questionable whether Q-VD is actually working in the authors' experiments. Again, it is not exactly clear what the values in Figure 7b-d have been normalized to but it seems that all the white bars were obtained by normalization to Q-VD pre-treated samples whereas the green bars are normalized to the untreated control (black bar). The p-tests are therefore not valid and a general conclusion cannot be inferred from these data. Of particular concern is that the authors do not show the effect of the caspase inhibitor on PUFA-PC levels (the values in Fig. 7c have been artificially set to 100% with and without the inhibitor), despite claiming that it has no effect (pg. In 6-9). If the authors' claim of a feed-forward loop is to be believed, the inhibition of apoptosis should lead to a reduction of PUFA-PC (as the authors graphically illustrate in their model, Supplementary Fig. 16).

Fig. 8

Supplementary Figure 15 appears to be the basis for the quantification in Figure 8. However, it is not clear that any normalisation of the p-Akt signal to total Akt signal has taken place. Visual inspection of the raw Western blots suggests that the decrease in p-AKT would be only significant for MCF-7 cells (when treated with MC). So whilst the decrease in p-AKT might hold true for some apoptotic treatments, it does not appear to be the case for most of the cell lines investigated. To derive a common mechanism based on this data is misleading.